
# Higher-form subsystem symmetry breaking: Subdimensional criticality and fracton phase transitions

**Brandon C. Rayhaun[1][*] and Dominic J. Williamson[2]**

**1** Stanford Institute for Theoretical Physics, Stanford University, CA 94305, USA
**2** Centre for Engineered Quantum Systems, School of Physics, University of Sydney, Sydney, New South Wales 2006, Australia

[*] brandonrayhaun@gmail.com

## Abstract

Subsystem symmetry has emerged as a powerful organizing principle for unconventional quantum phases of matter, most prominently fracton topological orders. Here, we focus on a special subclass of such symmetries, known as higher-form subsystem symmetries, which allow us to adapt tools from the study of conventional topological phases to the fracton setting. We demonstrate that certain transitions out of familiar fracton phases, including the X-cube model, can be understood in terms of the spontaneous breaking of higher-form subsystem symmetries. We find simple pictures for these seemingly complicated fracton topological phase transitions by relating them in an exact manner, via gauging, to spontaneous higher-form subsystem symmetry breaking phase transitions of decoupled stacks of lower-dimensional models. We harness this perspective to construct a sequence of unconventional subdimensional critical points in two and three spatial dimensions based on the stacking and gauging of canonical models with higher-form symmetry. Through numerous examples, we illustrate the ubiquity of coupled layer constructions in theories with higher-form subsystem symmetries.



# 1 Introduction

The study of phases of matter and the transitions between them lies at the heart of condensed matter physics. Landau's theory [1] attempts to paint a picture of their classification using spontaneous symmetry breaking as measured by local order parameters, but for quite some time it has been appreciated that this perspective requires extension: for example, there are topological phases that do not break any conventional symmetry, and transitions into these phases that cannot be diagnosed by any local order parameter [2–6] (see e.g. [7] for a review). However, by considering higher-form symmetries [8] and non-local order parameters, the scope of the Landau paradigm has been successfully broadened to include many of the phases it was previously thought not to accommodate; more general "categorical" notions of symmetry are anticipated to extend its reach even further [9–18] (see [19] for a nice review). Far from being mere theoretical curiosities, the non-local physical characteristics of such topological systems are of particular importance for applications, e.g. in quantum error correction [20, 21].

One ingredient which has been powerful in the classification and understanding of (2+1)D topological phases in particular is the picture of string-net condensation [22]. Among other things, it has led to explicit, representative wavefunctions for a large class of topological phases (those with gappable boundaries), as well as exactly soluble Hamiltonians that realize these wavefunctions as their ground states. It also makes manifest the powerful role of tensor category theory in encoding the universal properties of topological order in (2+1)D [23–25].

Recently [26–34], new classes of fracton "topological" phases (see Refs. [35, 36] for reviews) have inspired a surge of activity due their novel symmetry structures, and the associated mobility restrictions on their topological charges, which land such models beyond a straightforward description by the conventional framework of topological quantum field theory [37, 38]. These physical properties again lead to interesting and useful applications in quantum computation [27, 28, 39–42], connect to other branches of condensed matter physics [43–47], and raise intriguing challenges to the program of capturing universal properties of phases of matter using continuum quantum field theory [48–58].

There are several reoccurring ideas and themes that underlie investigations of fractonic physics. The concept of a zero-form subsystem symmetry [33, 59, 60], which generalizes conventional global symmetries to allow symmetry transformations that are supported on rigid submanifolds of space, has proven immensely valuable in understanding the properties of fracton phases and their characteristic mobility restrictions: indeed, if a symmetry acts only along a submanifold of space, then its corresponding charged excitations cannot move out of the submanifold without violating their associated conservation laws. The related (as explained below) approach of layering lower dimensional topological orders and condensing composite excitations [52, 61–65] (see also [66]) has also proven extremely useful: in particular, it has led to constructions like the cage-net models [67], which form a particular fractonic analog of the string-net approach to topological phases. This approach culminated in the topological defect network construction [68], which aims to describe all gapped fracton topological phases.

While the ideas of the previous paragraph have dramatically increased our understanding of fracton-adjacent physics, this subject is still in its infancy. For example, the structure of a *higher-form* subsystem symmetry (HFSS), while implicit as an important ingredient in many examples (particularly the X-cube model [33]), has not to our knowledge been emphasized to the same extent as have zero-form subsystem symmetries (although it has appeared, such as recently in Ref. [69]). Moreover, many of the general structural results on ordinary symmetries and their higher-form generalizations should admit extensions to the subsystem setting; while many of these extensions have been anticipated in examples, a systematic theory is still largely

lacking. Furthermore, the majority of attention has been on gapped fracton topological phases, while relatively little has been understood about their transitions into neighboring phases and the associated critical points, and whether or not such transitions can be incorporated into the Landau paradigm (we highlight the relevant recent works Refs. [70–76] in this direction). Finally, while e.g. cage-nets do constitute one fractonic analog of string-nets, a mathematical structure as powerful as tensor category theory in the case of topological phases has yet to be fully elucidated (the proposal of topological defect networks aims to establish a framework for the development of such a theory [68]).

## 1.1 Summary of results

In this work, we explore some of the issues introduced above in the context of translation invariant *lattice network models*. Heuristically, we define these to be systems with $D$ spatial dimensions which are obtained by foliating space with models of $d \leq D$ spatial dimensions— possibly for multiple values of $d$—and then coupling the various leaves of these foliations together. Models of this type have of course appeared in the fracton literature before [52, 61, 62, 64, 65]; we revisit and extend some of these known constructions, and also introduce new ones, obtaining a unified perspective by focusing particularly on aspects of their symmetry. Table 1 contains an overview of the network constructions we consider. In order to explain its contents, we start by sketching an impressionistic definition of $q$-form subsystem symmetry.

First, recall that a theory with $d$ spatial dimensions is said to have an ordinary $q$-form symmetry $G^{(q)}$ if it admits symmetry operators $U_g(M^{(d-q)})$ for every $(d-q)$-dimensional submanifold $M^{(d-q)}$ of spacetime, and every $g \in G$; by definition, these operators are required to furnish a representation of the group,

$$U_g\left(M^{(d-q)}\right) U_h\left(M^{(d-q)}\right) = U_{gh}\left(M^{(d-q)}\right), \quad \text{for all} \quad g, h \in G, \tag{1}$$

and to be topological in the sense that correlation functions that include them as insertions should be insensitive to deformations of the submanifold $M^{(d-q)}$, provided that those deformations do not move the submanifold through a charged operator. In order for this to make sense, the charged operators of a $q$-form symmetry must necessarily have dimension $q$.

A *foliated $q$-form subsystem symmetry* is a modest generalization of this notion; as an additional piece of input data, we must specify a foliation $\mathscr{F} = \{L^{(d-k)}\}$ of space, say by leaves $L^{(d-k)}$ of codimension $k$.[1] Informally, we say that a theory with $d$ spatial dimensions has a foliated $q$-form subsystem symmetry $G^{(q,k)}(\mathscr{F})/\sim$ of codimension $k$ (or a $(q,k)$-symmetry for short) if it has an ordinary $q$-form symmetry within each leaf $L^{(d-k)} \in \mathscr{F}$. More precisely, to each $L^{(d-k)} \in \mathscr{F}$, each submanifold $M^{(d-k-q)} \subset L^{(d-k)}$, and each $g \in G$, we require symmetry operators $U_g(M^{(d-q-k)})$ which are topological if $M^{(d-k-q)}$ is deformed *within* the leaf $L^{(d-k)}$ in which it is contained, provided again that we do not deform it through a charged operator. In order for an operator to be charged with respect to a subsystem symmetry associated to a leaf $L^{(d-k)}$, its intersection with $L^{(d-k)}$ must have dimension $q$. The case of an ordinary higher-form symmetry corresponds to $k=0$. When the choice of foliation is clear, we often omit it from the notation, $G^{(q,k)}/\sim \equiv G^{(q,k)}(\mathscr{F})/\sim$; if there are multiple foliations, we express this by writing $G^{(q,k)}(\mathscr{F}_1, \mathscr{F}_2, \dots)/\sim$. The quotient by $\sim$ throughout these definitions is meant to reflect the fact that the symmetry structure may have relations, of which we will see examples shortly. Although we have used continuum language in this definition, we work almost exclusively on the lattice, where the symmetry groups are represented by unitary operators that commute with the Hamiltonian.

---

[1] We restrict ourselves in this paper to smooth foliations, see e.g. [51] for a study of fracton models on spaces equipped with singular foliations.

Table 1: A glossary of the constructions considered in the main text (further examples appear in Appendix A). The first column provides the number of the relevant section. In the second column, DNM stands for the "decoupled network model" which forms the starting point of the construction. In the third column, the subgroup $\mathcal{H}$ of the global symmetry group of the DNM that is being gauged is listed. The symbol $\mathcal{F}^{\|}_{\mu_1\mu_2\dots}$ ($\mathcal{F}^{\perp}_{\mu_1\mu_2\dots}$) denotes the foliation of space by hyperplanes that are parallel (perpendicular) to the $\mu_1\mu_2\dots$ directions. In the fourth and fifth columns, Phase A (B) refers to the phase of DNM/$\mathcal{H}$ obtained by starting with the DNM in its trivial (higher-form subsystem symmetry breaking) phase. In the sixth and seventh columns, we list the topological sectors that are condensed across the phase transition from Phase A to Phase B, and vice versa; the submanifolds along which they're condensed are given in parentheses (see the main body of the text for more details). We have used the abbreviations ferro. and plaq. for ferromagnetic and plaquette, respectively.

| § | DNM | Gauged subgroup $\mathcal{H}$ | Phase A | Phase B | A→B | B→A |
|---|---|---|---|---|---|---|
| §3.1 | $^1H^{(0)}_{\mathbb{Z}_2}(\mathcal{F}^{\|}_x, \mathcal{F}^{\|}_y)$ Ising wires | $^D\mathbb{Z}^{(0)}_2$ diagonal 0-form | toric code | ferro. plaq. Ising | e charges (along lines) | corner kink dipoles |
| §3.2 | $^2H^{(1)}_{\mathbb{Z}_2}$ $\mathbb{Z}_2$ gauge theory | $\mathbb{Z}^{(0,1)}_2(\mathcal{F}^{\|}_x, \mathcal{F}^{\|}_y)$ linear subsystem | ferro. Ising wire stack | ferro. plaq. Ising | k-strings | – |
| §4.1 | $^1H^{(0)}_{\mathbb{Z}_2}(\mathcal{F}^{\|}_x, \mathcal{F}^{\|}_y, \mathcal{F}^{\|}_z)$ Ising wires | $\mathbb{Z}^{(0,1)}_2(\mathcal{F}^{\|}_{xy}, \mathcal{F}^{\|}_{yz}, \mathcal{F}^{\|}_{xz})/^D\mathbb{Z}^{(0)}_2$ planar subsystem | X-cube | ferro. cubic Ising | lineons | corner kink quadrupoles |
| §5.1 | $^2H^{(1)}_{\mathbb{Z}_2}(\mathcal{F}^{\|}_{xy})$ single $\mathbb{Z}_2$ gauge stack | $\mathbb{Z}^{(0,1)}_2(\mathcal{F}^{\perp}_x, \mathcal{F}^{\perp}_y)$ planar subsystem | double toric code stack | X-cube | p-strings (in 1x planes) | lineon dipoles (in 1x planes) |
| §5.2 | $^2H^{(1)}_{\mathbb{Z}_2}(\mathcal{F}^{\perp}_x, \mathcal{F}^{\perp}_y)$ double $\mathbb{Z}_2$ gauge stack | $\mathbb{Z}^{(0,1)}_2(\mathcal{F}^{\|}_{xy})$ planar subsystem | single toric code stack | X-cube | p-strings (in 2x planes) | fracton dipoles (in 2x planes) |
| §6.1 | $^2H^{(1)}_{\mathbb{Z}_2}(\mathcal{F}^{\perp}_x, \mathcal{F}^{\perp}_y, \mathcal{F}^{\perp}_z)$ triple $\mathbb{Z}_2$ gauge stack | $^D\mathbb{Z}^{(1)}_2$ diagonal 1-form | deconfined $\mathbb{Z}_2$ gauge theory | X-cube | flux loops (in 3x planes) | fracton dipole planons |
| §6.2 | $^3H^{(1)}_{\mathbb{Z}_2}$ $\mathbb{Z}_2$ gauge theory | $\mathbb{Z}^{(0,1)}_2(\mathcal{F}^{\perp}_x, \mathcal{F}^{\perp}_y, \mathcal{F}^{\perp}_z)$ planar subsystem | triple toric code stack | X-cube | p-strings | lineons (on string junctions) |

Higher-form subsystem symmetries are expected to behave similarly to ordinary higher-form symmetries: they admit subgroups, they can be spontaneously broken, they can be gauged, and so on and so forth. We anticipate and sketch some of these generalities in §2.1. One useful fact which underlies much of our analysis is that when a discrete, Abelian HFSS $A^{(q,k)}(\mathscr{F}_1, \mathscr{F}_2, \dots)\big/\sim$ is gauged in $d$ spatial dimensions, the gauged theory, while losing its original $A^{(q,k)}(\mathscr{F}_1, \mathscr{F}_2, \dots)\big/\sim$ global symmetry, is expected to gain an emergent "quantum" HFSS $\widehat{A}^{(d-q-k-1,k)}(\mathscr{F}_1, \mathscr{F}_2, \dots)\big/\sim$, where $\widehat{A} = \mathrm{hom}(A, U(1))$ is the Pontryagin dual of $A$. This is a modest generalization of a well-known result about ordinary higher-form symmetries [8, 77–79]; in lieu of presenting a fully general proof (see §2.1 for relevant comments), we explore how it is borne out in numerous examples.

It is relatively simple to obtain a lattice network model with a $G^{(q,k)}(\mathscr{F})$ symmetry. If one places a $d = D - k$ spatial-dimensional theory with a symmetry group $G^{(q)}$ on each leaf $L^{(D-k)}$ of $\mathscr{F}$, and does not couple the leaves together in any way, then the full $D$ spatial-dimensional network model can be thought of as having a $q$-form subsystem symmetry. Because the leaves are decoupled, the symmetry structure is free of relations, and we do not have to quotient by a nontrivial $\sim$ when we describe its symmetry group. However, models of interest typically *do* have relations. Take as an example the (2+1)D plaquette Ising model (PIM). Its Hilbert space consists of a qubit on each vertex of a 2d square lattice, and its Hamiltonian takes the form

$$^2 H_{\mathbb{Z}_2}^{(0,1)} = -J \sum_{i,j} \mathbf{Z}_{i,j} \mathbf{Z}_{i+1,j} \mathbf{Z}_{i,j+1} \mathbf{Z}_{i+1,j+1} - h \sum_{i,j} \mathbf{X}_{i,j} \,, \tag{2}$$

where the sums over $(i, j)$ run over pairs of integers which coordinatize the vertices of the lattice, and $\mathbf{X}_{i,j}$, $\mathbf{Z}_{i,j}$ are Pauli operators supported at the site $(i, j)$. Call $\mathscr{F}_{\mathrm{x}}^{\parallel}$ and $\mathscr{F}_{\mathrm{y}}^{\parallel}$ the foliations of the lattice by its rows and columns, respectively. This model enjoys a $\mathbb{Z}_2^{(0,1)}(\mathscr{F}_{\mathrm{x}}^{\parallel}, \mathscr{F}_{\mathrm{y}}^{\parallel})\big/\sim$ zero-form subsystem symmetry, in the sense that it admits unitary operators

$$U_j^{\mathrm{x}} = \prod_i \mathbf{X}_{i,j} \,, \qquad U_i^{\mathrm{y}} = \prod_j \mathbf{X}_{i,j} \,, \tag{3}$$

which are supported on the rows and columns of the lattice, respectively, and commute with the Hamiltonian.[2] This symmetry satisfies a relation of the form

$$\prod_j U_j^{\mathrm{x}} \prod_i U_i^{\mathrm{y}} = 1 \,, \tag{4}$$

which is simply a reflection of the fact that performing spin flips on all of the columns is identical to performing spin flips on all of the rows. One can describe this relation by saying that the true symmetry group of the plaquette Ising model involves a quotient by the subgroup generated by the left-hand side of Eq. (4); this is a conventional zero-form symmetry group, which we term the *diagonal subgroup* and label as $^{\mathrm{D}}\mathbb{Z}_2^{(0)}$. In other words, we can replace the quotient by $\sim$ with a quotient by $^{\mathrm{D}}\mathbb{Z}_2^{(0)}$ and write $\mathbb{Z}_2^{(0,1)}(\mathscr{F}_{\mathrm{x}}^{\parallel}, \mathscr{F}_{\mathrm{y}}^{\parallel})\big/^{\mathrm{D}}\mathbb{Z}_2^{(0)}$ for the global symmetry group of the PIM.

Contrast this with a more trivial example, say, a decoupled grid of transverse field Ising wires,[3]

$$^1 H_{\mathbb{Z}_2}^{(0)}\left(\mathscr{F}_{\mathrm{x}}^{\parallel}, \mathscr{F}_{\mathrm{y}}^{\parallel}\right) = -J \sum_{i,j}\left(\mathbf{Z}_{i,j}^{\mathrm{x}} \mathbf{Z}_{i+1,j}^{\mathrm{x}} + \mathbf{Z}_{i,j}^{\mathrm{y}} \mathbf{Z}_{i,j+1}^{\mathrm{y}}\right) - h \sum_{i,j}\left(\mathbf{X}_{i,j}^{\mathrm{x}} + \mathbf{X}_{i,j}^{\mathrm{y}}\right). \tag{5}$$

---

[2]This symmetry motivates our choice of notation $^2 H_{\mathbb{Z}_2}^{(0,1)}$ for the Hamiltonian of the plaquette Ising model: the $(0,1)$ in the superscript indicates that the model has a $\mathbb{Z}_2^{(0,1)}(\mathscr{F}_{\mathrm{x}}^{\parallel}, \mathscr{F}_{\mathrm{y}}^{\parallel})\big/\sim$ symmetry, and the 2 denotes its spatial dimension. We define an infinite family $^d H_{\mathbb{Z}_2}^{(q,k)}$ of similar models in §2.1.

[3]As we explain in more detail below, the notation $^1 H_{\mathbb{Z}_2}^{(0)}$ is meant to stand for the (1+1)D transverse field Ising model, and the apperance of $\mathscr{F}_{\mathrm{x}}^{\parallel}, \mathscr{F}_{\mathrm{y}}^{\parallel}$ indicates that we are foliating the TFIM along the rows and columns of space.

Here, we have placed two qubits on each site of the square lattice, one for each of the two wires that intersect it, and labeled the Pauli operators that act on the qubit coming from the wire pointing in the x direction as $\mathbf{X}_{i,j}^{\mathrm{x}}$, $\mathbf{Z}_{i,j}^{\mathrm{x}}$, and similarly for the qubit coming from the wire pointing in the y direction (cf. Figure 3). This network model also enjoys a linear zero-form subsystem symmetry which simply comes from the fact that each wire has a $\mathbb{Z}_2$ spin-flip symmetry,

$$\widetilde{U}_j^{\mathrm{x}} = \prod_i \mathbf{X}_{i,j}^{\mathrm{x}}, \quad \widetilde{U}_i^{\mathrm{y}} = \prod_j \mathbf{X}_{i,j}^{\mathrm{y}}. \tag{6}$$

However here, the subsystem symmetry is relation-free, and we can simply write it as $\mathbb{Z}_2^{(0,1)}(\mathscr{F}_{\mathrm{x}}^{\parallel}, \mathscr{F}_{\mathrm{y}}^{\parallel})$.

The differing symmetry structures between the plaquette Ising model and decoupled Ising wires would seem to preclude a "network" description of the former theory. However, if we insist on a coupled-wire description of $^2H_{\mathbb{Z}_2}^{(0,1)}$, there are two ways that we can proceed.

1) One option is to couple the wires together

$$^1H_{\mathbb{Z}_2}^{(0)}\left(\mathscr{F}_{\mathrm{x}}^{\parallel}, \mathscr{F}_{\mathrm{y}}^{\parallel}\right) \to {}^1H_{\mathbb{Z}_2}^{(0)}\left(\mathscr{F}_{\mathrm{x}}^{\parallel}, \mathscr{F}_{\mathrm{y}}^{\parallel}\right) + H_{\mathrm{C}}, \tag{7}$$

by adding a term $H_{\mathrm{C}}$ in the Hamiltonian that drives the model through a transition into a phase in which the diagonal subgroup $^{\mathrm{D}}\mathbb{Z}_2^{(0)}$ is spontaneously unbroken. This subgroup therefore acts trivially on the low energy Hilbert space of the model in this phase (as if it were imposed as a relation), and there is a chance that the low energy effective Hamiltonian coincides with the plaquette Ising model $^2H_{\mathbb{Z}_2}^{(0,1)}$. We argue that this is indeed the case in §3.1.1.

2) Another option is to simply gauge the diagonal subgroup,

$$^1H_{\mathbb{Z}_2}^{(0)}\left(\mathscr{F}_{\mathrm{x}}^{\parallel}, \mathscr{F}_{\mathrm{y}}^{\parallel}\right) \to {}^1H_{\mathbb{Z}_2}^{(0)}\left(\mathscr{F}_{\mathrm{x}}^{\parallel}, \mathscr{F}_{\mathrm{y}}^{\parallel}\right)\Big/{}^{\mathrm{D}}\mathbb{Z}_2^{(0)}, \tag{8}$$

which trivializes its action and thus imposes it as a relation by fiat. This can more vividly be thought of as immersing the decoupled network model in a bulk gauge theory, which in turn mediates interactions between its leaves.

One subtlety with this approach is that, as we have mentioned in a previous paragraph, a gauged theory gains its own emergent "quantum" global symmetry group; in the present case, in addition to inheriting a $\mathbb{Z}_2^{(0,1)}(\mathscr{F}_{\mathrm{x}}^{\parallel}, \mathscr{F}_{\mathrm{y}}^{\parallel})\Big/{}^{\mathrm{D}}\mathbb{Z}_2^{(0)}$ symmetry from the ungauged network model, the gauged theory grows an emergent one-form symmetry $^{\mathrm{D}}\widehat{\mathbb{Z}}_2^{(1)}$ which does not appear in the plaquette Ising model. Fortunately, we show in §3.1.2 that if one tunes the wires to be deep in their ferromagnetic phase, then in the gauged model the emergent symmetry $^{\mathrm{D}}\widehat{\mathbb{Z}}_2^{(1)}$ is unbroken and thus acts trivially on the low energy Hilbert space; that is, it is not a part of the global symmetry of the IR theory. On the other hand, the subsystem symmetry group $\mathbb{Z}_2^{(0,1)}(\mathscr{F}_{\mathrm{x}}^{\parallel}, \mathscr{F}_{\mathrm{y}}^{\parallel})\Big/{}^{\mathrm{D}}\mathbb{Z}_2^{(0)}$ is spontaneously broken, and so it *is* a non-trivial symmetry of the low energy theory. This matches the structure of the symmetries deep in the ferromagnetic phase of the plaquette Ising model and, correspondingly, we will show that the model is in a plaquette Ising phase.

It is also interesting to ask what happens in this construction if one tunes the un-gauged Ising wires in the other direction, so that they are in the trivial phase. In this case, it is possible to show that the roles of the two symmetries of the gauged theory are reversed: namely the subsystem symmetry group $\mathbb{Z}_2^{(0,1)}(\mathscr{F}_{\mathrm{x}}^{\parallel}, \mathscr{F}_{\mathrm{y}}^{\parallel})\Big/{}^{\mathrm{D}}\mathbb{Z}_2^{(0)}$ is unbroken, but the emergent one-form symmetry group $^{\mathrm{D}}\widehat{\mathbb{Z}}_2^{(1)}$ is broken. It turns out that this lands the

gauged model in a toric code phase. Thus, as the strength of the nearest-neighbor interactions on the Ising wires are tuned, the model undergoes a Landau transition between a ferromagnetic plaquette Ising model phase and a toric code phase. In between these two phases lies an interesting critical point, which can be described as a grid of Ising RCFT wires coupled to a bulk (2+1)D $\mathbb{Z}_2$ gauge theory. We say that this theory enjoys *subdimensional criticality* [74].

The first of these protocols is essentially the same mechanism that was used to uncover a coupled layer construction of the X-cube model in Ref. [61, 62], though it was described in terms of "p-string condensation" rather than spontaneous symmetry breaking there. The second method coincides with the one used to obtain the string-membrane-net picture of the X-cube model in Ref. [52], though again, symmetry considerations were not the focus there. We revisit both of these construction armed with our present perspective in §6.1.1 and §6.1.2, respectively. One new feature that our analysis reveals is that the coupled wire construction of the plaquette Ising model from Eq. (8) arises on the boundary of the analogous gauged coupled layer construction of the X-cube model.

 With these motivating examples of coupled wire constructions for the plaquette Ising model in mind, we are now in a position to state our general construction. For simplicity, we focus on network models for which the $d$ spatial-dimensional leaves support either

 a) the transverse field Ising model ${}^{d}H_{\mathbb{Z}_2}^{(0)}$,

 b) $\mathbb{Z}_2$ lattice gauge theory ${}^{d}H_{\mathbb{Z}_2}^{(1)}$, or more generally,

 c) $q$-form lattice gauge theory ${}^{d}H_{\mathbb{Z}_2}^{(q)}$ for $q > 1$,

however many aspects of our analysis should extend to the case that the leaves are kept more general. (See §2.2 for a review of the models ${}^{d}H_{\mathbb{Z}_2}^{(q)}$ in low dimensions.) The Hamiltonian ${}^{d}H_{\mathbb{Z}_2}^{(q)}$ admits a $q$-form symmetry, and it contains a free parameter which tunes the model between a $q$-form symmetry breaking phase and a trivial phase; for example, in the case of the transverse field Ising model, the parameter is simply the ratio between the strength of the nearest neighbor interaction and the strength of the transverse field. Therefore, the lattice network model obtained by taking stacks of trivially decoupled leaves of ${}^{d}H_{\mathbb{Z}_2}^{(q)}$ along foliations $\mathscr{F}_1, \mathscr{F}_2, \dots$ of space has a relation-free $\mathbb{Z}_2^{(q,k)}(\mathscr{F}_1, \mathscr{F}_2, \dots)$ HFSS, where $k = D - d$. It also possesses a parameter which can be thought of as passing the model between a trivial phase and a $q$-form subsystem symmetry breaking phase. We label the network model so-obtained as ${}^{d}H_{\mathbb{Z}_2}^{(q)}(\mathscr{F}_1, \mathscr{F}_2, \dots)$.

 Each subsection §A.B of this paper for A = 3, 4, 5, 6 (except for the last subsection of of each §A) involves the choice of two pieces of data:

 1) a decoupled network model ${}^{d}H_{\mathbb{Z}_2}^{(q)}(\mathscr{F}_1, \mathscr{F}_2, \dots)$, and

 2) an HFSS subgroup

$$\mathscr{H} := \mathbb{Z}_2^{(q',k')}(\mathscr{F}_1', \mathscr{F}_2', \dots)\big/\sim \, \subset \mathbb{Z}_2^{(q,k)}(\mathscr{F}_1, \mathscr{F}_2, \dots) =: \mathscr{G}, \tag{9}$$

 of its global symmetry group.

The choice of this subgroup is typically motivated by having a target model in mind which has a $(q,k)$-symmetry with relations described by $\mathscr{H}$; in other words, the target model admits an action of $\mathscr{G}\big/\mathscr{H}$. In the plaquette Ising model example described above, we choose $\mathscr{H} = {}^{\mathrm{D}}\mathbb{Z}_2^{(0)} \subset \mathbb{Z}_2^{(0,1)}(\mathscr{F}_{\mathrm{x}}^{\|}, \mathscr{F}_{\mathrm{y}}^{\|}) = \mathscr{G}$.

In §A.B.1, we couple the leaves of the network model together

$$^{d}H_{\mathbb{Z}_2}^{(q)}(\mathcal{F}_1,\mathcal{F}_2,\dots) \to {}^{d}H_{\mathbb{Z}_2}^{(q)}(\mathcal{F}_1,\mathcal{F}_2,\dots) + H_{\mathrm{C}}, \tag{10}$$

with terms in $H_{\mathrm{C}}$ chosen to respect the subgroup $\mathcal{H}$. Thus, one might say that we are probing the neighboring phase diagram of the network model "enriched" by its $\mathcal{H}$ symmetry. In most cases, $H_{\mathrm{C}}$ includes in it a parameter that actually preserves the full $\mathcal{G}$ symmetry and, when driven to large values (i.e. strong coupling), induces a Landau transition into a phase in which the $\mathcal{H}$ subgroup becomes unbroken. In such cases, we find that the low energy effective Hamiltonian of the theory in these strongly coupled phases becomes a model with a global symmetry of the form $\mathcal{G}/\mathcal{H}$. Our results show that several familiar models can be obtained in this way, including the plaquette Ising model, the cubic Ising model, and the X-cube model.

In §A.B.2, we instead gauge the subgroup $\mathcal{H}$,

$$^{d}H_{\mathbb{Z}_2}^{(q)}(\mathcal{F}_1,\mathcal{F}_2,\dots) \to {}^{d}H_{\mathbb{Z}_2}^{(q)}(\mathcal{F}_1,\mathcal{F}_2,\dots)\Big/\mathcal{H}. \tag{11}$$

This produces a model whose global symmetry group has two parts. On the one hand, it is expected to inherit a $(q,k)$-symmetry $\mathcal{G}/\mathcal{H}$ from the ungauged model, where now the subgroup is imposed as a relation as it has been gauged.[4] On the other hand, while the subgroup itself has disappeared, it is replaced by an emergent quantum HFSS group

$$\widehat{\mathcal{H}} := \widehat{\mathbb{Z}}_2^{(D-q'-k'-1,k')}(\mathcal{F}_1',\mathcal{F}_2',\dots)\Big/\sim. \tag{12}$$

The $q$-form subsystem symmetry breaking phase transition of the un-gauged network model is mapped after gauging to a phase transition in which the groups $\mathcal{G}/\mathcal{H}$ and $\widehat{\mathcal{H}}$ experience some pattern of partial symmetry breaking and un-breaking. To round this picture out, in each of our examples, we study how order/disorder parameters of the ungauged model descend to parameters that diagnose the transition of the gauged model; we also track how excitations condense across the transition. A glossary of the various constructions we obtain in this way is presented in Table 1. These examples aim to exhaust all possible constructions in $D \leq 3$ spatial dimensions that are based on stacking Ising-like layers with $q$-form symmetry groups $\mathbb{Z}_2^{(q)}$ (i.e. that are based on stacking the model $^{d}H_{\mathbb{Z}_2}^{(q)}$) on the cubic lattice.

A notable upshot of these constructions is that they recover a range of fractonic analogs of string-nets, including the string-membrane-net models of Ref. [52] and additional models that have not previously appeared to the best of our knowledge. Specifically, they naturally suggest new pictures for the groundstate wavefunctions of existing fracton phases characterized by the condensation of networks of objects of various dimensions: points, strings, membranes, and so on. By making these various objects explicit in the condensate picture, our models facilitate the study of phase transitions driven by the corresponding topological excitations. We harness this to uncover a variety of subdimensional phase transitions out of the plaquette Ising and X-cube models that have an exact description in terms of well known higher-form symmetry breaking transitions coupled to higher-form subsystem gauge fields. Although we have only worked out these pictures in detail for the $\mathbb{Z}_2$ case, we expect that they are readily generalized to $\mathbb{Z}_N$ and beyond, see for instance Ref. [74].

---

[4]This is at least true in our simple constructions. In more complicated setups, the quotient $\mathcal{G}/\mathcal{H}$ may be more involved, not simply a subgroup of the symmetry group [79].

## 1.2 Section outline

The structure of the remainder of this article is as follows.

In §1.3 we document basic notation and conventions.

In §2.1, we exposit various general ideas that are used throughout the rest of the paper. We start by reviewing ordinary higher-form symmetries, and then defining what we mean by a higher-form subsystem symmetry. We then describe several natural subgroups that a higher-form subsystem symmetry group possesses. We explain the notion of a spontaneously broken higher-form subsystem symmetry, sketch how to gauge a higher-form subsystem symmetry, and describe how these two ideas play with one another using the idea of an emergent "quantum" symmetry.

In §2.2.1–§2.2.3, we provide a lightning review of Ising-like prototype models ${}^{d}H_{\mathbb{Z}_2}^{(q)}$ (in spatial dimensions $d = 2, 3, 4$) which possess ordinary $q$-form symmetries: these are the transverse field Ising models and $q$-form gauge theories. Reviewing these models serves two purposes. First, it allows us to demonstrate the general perspective we take throughout this paper in a familiar setting. Second, it allows us to establish notations and conventions for these models, which we make heavy use of as ingredients in all of our constructions. In §2.2.4, we define more general Ising-like prototype models ${}^{d}H_{\mathbb{Z}_2}^{(q,k)}$, in arbitrary spatial dimensions, which possess $\mathbb{Z}_2^{(q,k)}/{\sim}$ symmetries: these include transverse field Ising models, $q$-form lattice gauge theories, plaquette Ising models, cubic Ising models, and tensor gauge theories with an X-cube phase.

Sections 3–6 can be read more or less independently from one another, however we are the most explicit and pedagogical in §3, and become briefer in later sections to avoid repetition. Each section is organized around a prototype model ${}^{d}H_{\mathbb{Z}_2}^{(q,k)}$. Each subsection, except for the last subsection within each of the sections, contains a different network model description of ${}^{d}H_{\mathbb{Z}_2}^{(q,k)}$, and explores different phase transitions out of the ${}^{d}H_{\mathbb{Z}_2}^{(q,k)}$ phase. The last subsection contains a "parent model" which recovers each of the network model descriptions of the previous subsections, as well as the associated phase transitions, in various special limits. There are additional degrees of freedom in the parent model that facilitate the study of a wide range of condensation driven phase transitions via single site perturbations to the Hamiltoinan. This parent model realizes a fracton analog of the string-net picture for topological phases.

In §3, we explore phase diagrams proximate to the (2+1)D plaquette Ising model, ${}^{2}H_{\mathbb{Z}_2}^{(0,1)}$. We start in §3.1 by studying coupled wire constructions. In §3.2, we explain how to recover the PIM by coupling a subsystem gauge theory to the toric code. In §3.3, we unify these two constructions into a single *point-string-net model*.

In §4, we study transitions out of the (3+1)D cubic Ising model, ${}^{3}H_{\mathbb{Z}_2}^{(0,2)}$. In §4.1, we offer a coupled wire construction. In §4.2, we show how this coupled wire construction can be combined with the X-cube model to give rise to a single *point-cage-net model*.

In §5 we study anisotropic constructions of the X-cube model, ${}^{3}H_{\mathbb{Z}_2}^{(1,1)}$. In §5.1, we show how the X-cube model can be obtained by gauging a subsystem subgroup of a single stack of ordinary gauge theory layers. In §5.2, we do something similar, but using two orthogonal stacks of gauge theory layers. In §5.3, we show that these two anisotropic constructions can be combined into a single *string-string-net model*.

In §6, we study isotropic constructions of the X-cube model. In §6.1, we review and slightly extend two known coupled layer constructions of the X-cube model. One novelty of this subsection over others is that we comment on properties of the boundary theory, and relate this back to the constructions we proposed in §3. In §6.2, we demonstrate how the X-cube model can be obtained from the (3+1)D toric code by coupling it to a subsystem gauge theory. We then show how these two viewpoints on the X-cube model are unified by the *string-membrane-net model*.

Finally, in §7 we conclude, and summarize numerous directions for future research.

Appendix A sketches how the ideas of the main text can be applied to obtain network constructions and phase diagrams out of the (3+1)D toric code. Here we are much more brief, and only summarize the parent models such network constructions lead to.

## 1.3 Notation and conventions

All of our models are defined on cubic lattices $\Lambda$ in varying dimensions. Throughout this paper, $v$ denotes a vertex of such a lattice, $e$ an edge, $p$ a plaquette, and $c$ a cube. Alternatively, when it is more convenient, we label objects by the coordinates of their center of mass, e.g. in a 2d square lattice, we label vertices $v$ by a pair of integers $(i, j)$, vertical edges by a pair $(i, j + \frac{1}{2})$, horizontal edges by $(i + \frac{1}{2}, j)$, and plaquettes by $(i + \frac{1}{2}, j + \frac{1}{2})$. We also generically use the symbol $\gamma$ to refer to a path through the lattice (i.e. a sequence of edges) and $m$ to refer to a membrane (i.e. a set of plaquettes). We use $\widetilde{\Lambda}$ to denote the dual lattice, and refer to the vertices, edges, paths, etc. of the dual lattice as $\tilde{v}$, $\tilde{e}$, $\tilde{\gamma}$, and so on. We often use the correspondence between objects of $\Lambda$ and dual objects of $\widetilde{\Lambda}$. For example, if $\Lambda$ is a 2d square lattice, then vertices $v$ can be equivalently thought of as dual plaquettes $\tilde{p}$, edges $e$ can be equivalently thought of as dual edges $\tilde{e}$, and so on.

We employ notation of the form $\sum_{v \in c}$ and $\sum_{c \ni v}$; the former indicates that one should sum over all the vertices which have non-trivial intersection with a given cube $c$ (i.e. that one should sum over the corners of the cube $c$) whereas the latter indicates a sum over all the cubes which have non-trivial intersection with $v$. More general symbols of a similar form, such as $\sum_{\sigma \in \rho}$ or $\sum_{\rho \ni \sigma}$, should be read similarly, although we often abuse the symbol "$\in$" and write e.g. $\sum_{e \in \tilde{\gamma}}$ to indicate a sum over edges $e$ that are intersected by the dual path $\tilde{\gamma}$, in spite of the fact that they are not "contained inside" $\tilde{\gamma}$.

We use $\mathscr{F} = \{L\}$ to denote a "foliation" of the lattice, by which we mean a discrete family of sublattices $L$. The notation $\mathscr{F}^{\parallel}_{\mu_1 \dots \mu_n}$ refers to the obvious foliation of a $d$-dimensional cubic lattice by $n$-dimensional sublattices which each span the $\mu_1 \dots \mu_n$ directions, whereas $\mathscr{F}^{\perp}_{\mu_1 \dots \mu_n}$ refers to the obvious foliation by $(d-n)$-dimensional sublattices which are each orthogonal to the $\mu_1 \dots \mu_n$ directions.

Throughout, $\mathbf{X}_\sigma$, $\mathbf{Z}_\sigma$ denotes a Pauli-$\mathbf{X}$/Pauli-$\mathbf{Z}$ operator acting on a qubit that is supported on the simplex $\sigma$. Occasionally, there is more than one qubit supported on each simplex $\sigma$, in which case we distinguish between them by decorating the superscripts of the Pauli operators with additional information. We typically label the eigenstates of a Pauli-$\mathbf{Z}$ operator as $|\uparrow\rangle$ and $|\downarrow\rangle$, and the eigenstates of Pauli-$\mathbf{X}$ as $|+\rangle$ and $|-\rangle$, i.e.

$$
\begin{aligned}
\mathbf{Z}|\uparrow\rangle &= +|\uparrow\rangle, & \mathbf{Z}|\downarrow\rangle &= -|\downarrow\rangle, \\
\mathbf{X}|+\rangle &= +|+\rangle, & \mathbf{X}|-\rangle &= -|-\rangle.
\end{aligned}
\tag{13}
$$

Finally, we also frequently make use of CNOT gates acting on pairs of qubits, which can be defined by their action on Pauli operators as

$$
\begin{aligned}
(C_1 X_2)\mathbf{X}_2(C_1 X_2)^\dagger &= \mathbf{X}_2, & (C_1 X_2)\mathbf{X}_1(C_1 X_2)^\dagger &= \mathbf{X}_1 \mathbf{X}_2, \\
(C_1 X_2)\mathbf{Z}_2(C_1 X_2)^\dagger &= \mathbf{Z}_1 \mathbf{Z}_2, & (C_1 X_2)\mathbf{Z}_1(C_1 X_2)^\dagger &= \mathbf{Z}_1.
\end{aligned}
\tag{14}
$$

## 2 Summary of Techniques

In this section, we explain some basic ideas which underlie the rest of the paper. In §2.1, we (informally and briefly) exposit some general principles related to global higher-form subsystem symmetries and their gauging. In §2.2, we review how these principles play out in simple

examples. Many of these examples form the building blocks out of which we construct the rest of the models that we consider in later sections; in §2.3, we explain the protocol with which we assemble them.

## 2.1 Generalities

As stated in the introduction, one of our goals is to study lattice network models with symmetry considerations in mind. The relevant kind of symmetry here is a *higher-form subsystem symmetry* (HFSS), a notion which we now explain. For the moment we are impressionistic and phrase things in continuum language, though everything we say is intended to most directly apply to lattice models. Throughout this entire paper, we restrict our attention to discrete symmetries.

**Higher-form global symmetries**
The idea of a higher-form global symmetry is by now standard, see e.g. Ref. [8] for a careful discussion and background references. In the continuum, the rough intuition is that a $(d+1)$-dimensional theory has a $q$-form symmetry group $G^{(q)}$ if it admits topological symmetry operators $U_g(M^{(d-q)})$ supported on $M^{(d-q)}$ for each $g \in G$ and each $(d-q)$-dimensional submanifold $M^{(d-q)}$ of spacetime, and if those operators further furnish a representation of $G$ in the sense that

$$U_g\left(M^{(d-q)}\right) U_h\left(M^{(d-q)}\right) = U_{gh}\left(M^{(d-q)}\right),\tag{15}$$

for each $(d-q)$-dimensional submanifold $M^{(d-q)}$. What is meant when one says that the $U_g(M^{(d-q)})$ are topological is that their correlation functions should not be sensitive to deformations of $M^{(d-q)}$, provided that one is careful not to deform $M^{(d-q)}$ through an operator which is charged under the symmetry. As the name suggests, the charged operators of a $q$-form symmetry are supported on $q$-dimensional submanifolds of spacetime. Furthermore, we only consider a topological operator to contribute to a $q$-form symmetry group if there exists a $q$-dimensional operator which is charged under it; thus we exclude e.g. the "porous" 0-form symmetries of [80] which act on lines but are blind to local operators.

If we take $M^{(d-q)}$ to be defined at a fixed instant of time, then we obtain an extended operator which acts on the Hilbert space of the theory; on the lattice such $U_g(M^{(d-q)})$ manifest themselves as extended unitary operators which commute with the Hamiltonian. A notable example which we review shortly is Kitaev's toric code, which admits a $\mathbb{Z}_2 \times \mathbb{Z}_2$ one-form symmetry group, and whose associated symmetry operators are simply the string operators of Ref. [20].

**Global subsystem symmetries**
Another notion is that of a (zero-form) subsystem symmetry. Here we focus on foliated subsystem symmetry (sometimes referred to as type-I, due to its connection to a subset of type-I fracton phases [33]). To define it, consider a foliation $\mathscr{F} = \{L^{(d-k)}\}$ of *space* by codimension $k$ leaves, i.e. leaves $L^{(d-k)}$ of spatial dimension $d-k$.[5] Roughly speaking, a Hamiltonian is said to have a foliated zero-form subsystem symmetry $G^{(0,k)}(\mathscr{F})/\sim$ of codimension $k$ if there are operators $U_g(L^{(d-k)})$ supported on $L^{(d-k)}$ for each $g \in G$ and each leaf $L^{(d-k)} \in \mathscr{F}$, and if $U_g(L^{(d-k)})U_h(L^{(d-k)}) = U_{gh}(L^{(d-k)})$. (Here and throughout the rest of the paper, since we are working with lattice models, we focus on operators that act at a fixed instant in time.) We

---

[5]Throughout this paper we essentially only consider the natural smooth foliations of flat space (either $\mathbb{R}^d$ or a $d$-torus $T^d$). Extra care is likely needed in the more general case, e.g. to accommodate singular foliations, as considered in Ref. [51].

generally deal with systems that admit subsystem symmetries with respect to several different foliations $\mathscr{F}_1, \mathscr{F}_2, \ldots$; when this is the case, we use the notation $G^{(0,k)}(\mathscr{F}_1, \mathscr{F}_2, \ldots)\big/\sim$. When there are $\binom{d}{d-k}$ foliations whose leaves all intersect orthogonally, we refer to such a symmetry as isotropic. When $k = d - 1$, we call the symmetry a linear subsystem symmetry; when $k = d - 2$, we refer to it as a planar subsystem symmetry.

There are a few remarks we can make. First, we emphasize that although the definition of a subsystem symmetry appears similar to the definition of a higher-form symmetry, they are distinct notions. For example, the symmetry operators of a subsystem symmetry are only defined on the leaves of the foliations, whereas the symmetry operators of a higher-form symmetry are defined for any submanifold of the right dimension. Moreover, the symmetry operators of a subsystem symmetry are rigid, whereas those of a higher-form symmetry are topological, i.e. deformable.

Second, we remark that it is often the case that various combinations of the symmetry operators act trivially. For example, a typical relation one encounters (and that we ourselves find when we come to the plaquette Ising model in §3) is

$$\prod_{L^{(d-k)} \in \mathscr{F}_1} U_g(L^{(d-k)}) = \prod_{L^{(d-k)} \in \mathscr{F}_2} U_g(L^{(d-k)}). \tag{16}$$

Technically speaking, this means that the subsystem symmetry group is acting non-faithfully, and that the "true" symmetry group should involve a quotient by this relation. The quotient by $\sim$ in the notation $G^{(0,k)}(\mathscr{F}_1, \mathscr{F}_2, \ldots)\big/\sim$ is meant to indicate the possible existence of relations of this kind, though without specifying what they are. In the cases that we *can* be more specific, we will write $G^{(0,k)}(\mathscr{F}_1, \mathscr{F}_2, \ldots)\big/R$, where $R$ is typically some subgroup of $G^{(0,k)}(\mathscr{F}_1, \mathscr{F}_2, \ldots)$ (we define a set of distinguished subgroups shortly). In the case that the symmetry is relation-free, we simply write $G^{(0,k)}(\mathscr{F}_1, \mathscr{F}_2, \ldots)$. In either case, we refer to both symmetry structures, with and without relations, as subsystem symmetries.

## Higher-form global subsystem symmetries

It is fruitful to combine these two ideas into a single notion, that of a *higher-form subsystem symmetry*. We say that a $(d+1)$-dimensional system has a $q$-form foliated subsystem symmetry of codimension $k$ for a group $G$ if, for each $g \in G$, each leaf $L^{(d-k)}$ of the foliation, and each $(d - k - q)$-dimensional submanifold $M^{(d-k-q)} \subset L^{(d-k)}$, there are symmetry operators $U_g(M^{(d-k-q)})$ supported on $M^{(d-k-q)}$ that are topological *within* $L^{(d-k)}$. That is, the operators $U_g(M^{(d-k-q)})$ are insensitive to deformations of $M^{(d-k-q)}$ provided that $M^{(d-k-q)}$ never leaves the leaf in which it is contained. We call such a symmetry a $(q, k)$-symmetry for short, and use the notation $G^{(q,k)}(\mathscr{F}_1, \mathscr{F}_2, \ldots)\big/\sim$ to specify this structure. We abbreviate to $G^{(q,k)}\big/\sim$ when we would like to suppress the choice of foliations, to $G^{(q)}$ when $k = 0$, and to simply $G$ when $q = k = 0$. Again, the quotient by $\sim$ is a shorthand to indicate that there may (or may not) be relations, without specifying precisely what they are. When we are being more specific about what the relations are, we write this as $G^{(q,k)}(\mathscr{F}_1, \mathscr{F}_2, \ldots)\big/R$; when the higher-form subsystem symmetry is relation-free, we simply write it as $G^{(q,k)}(\mathscr{F}_1, \mathscr{F}_2, \ldots)$. The prototypical example of a theory with a higher-form subsystem symmetry is the X-cube model [33], which admits a $\mathbb{Z}_2^{(1,1)}\big/\sim$ symmetry, this is reviewed below.

Whenever there is a higher-form global subsystem symmetry, there are natural "subgroups" one can consider. Describing these subgroups is useful e.g. so that we can more explicitly specify relations.

*Subgroups by refinement*

For example, let $n$ be a non-negative integer. Say that a foliation $\mathcal{F}_{\mathrm{R}}$ of codimension $(k+n)$ is a *refinement* of a foliation $\mathcal{F}$ of codimension $k$ if each leaf $L_{\mathrm{R}}^{(d-k-n)}$ of $\mathcal{F}_{\mathrm{R}}$ is contained in some leaf $L^{(d-k)}$ of $\mathcal{F}$, and if, for each $L^{(d-k)} \in \mathcal{F}$, the set $\{L_{\mathrm{R}}^{(d-k-n)} \mid L_{\mathrm{R}}^{(d-k-n)} \subset L^{(d-k)}\}$ is a foliation of $L^{(d-k)}$. Then a $G^{(q,k)}(\mathcal{F})\big/\sim$ symmetry admits a "subgroup" of the form

$$G^{(q-n,k+n)}\left(\mathcal{F}_{\mathrm{R}}^{(1)}, \mathcal{F}_{\mathrm{R}}^{(2)}, \dots\right)\big/\sim \, \subset G^{(q,k)}(\mathcal{F})\big/\sim, \tag{17}$$

for any set of codimension $(k+n)$ refinements $\mathcal{F}_{\mathrm{R}}^{(1)}, \mathcal{F}_{\mathrm{R}}^{(2)}, \dots$ of the foliation $\mathcal{F}$. The reason this makes sense is that the symmetry operators of a $(q,k)$-symmetry are supported on $(d-q-k)$-dimensional submanifolds, as are the symmetry operators of a $(q-n, k+n)$-symmetry, and so we can define the symmetry operators of the latter to be equal to the symmetry operators of the former.

As an example, consider a $G^{(1)}$ one-form symmetry in (2+1)D, whose symmetry operators are topological line operators. By picking a foliation $\mathcal{F}$ of space by lines, and restricting our attention to just the line operators of $G^{(1)}$ that are supported on the leaves of this foliation, we obtain a $G^{(0,1)}(\mathcal{F})\big/\sim$ subgroup of $G^{(1)}$. We make use of this precise example in later sections, e.g. in §3.2 and §6.2.

*Subgroups by coarsening*

In the other direction, say that a foliation $\mathcal{F}_{\mathrm{C}}$ of codimension $(k-n)$ is a *coarsening* of a foliation $\mathcal{F}$ of codimension $k$ if $\mathcal{F}$ is a refinement of $\mathcal{F}_{\mathrm{C}}$. A symmetry $G^{(q,k)}(\mathcal{F})\big/\sim$ admits various "diagonal" subgroups of the shape

$$G^{(q,k-n)}(\mathcal{F}_{\mathrm{C}})\big/\sim \, \subset G^{(q,k)}(\mathcal{F})\big/\sim. \tag{18}$$

The symmetry operators $U_g^{\mathrm{C}}(M^{(d-q-k+n)})$ of $G^{(q,k-n)}(\mathcal{F}_{\mathrm{C}})$ are defined in terms of the symmetry operators $U_g(N^{(d-q-k)})$ of $G^{(q,k)}(\mathcal{F})$ as

$$U_g^{\mathrm{C}}(M^{(d-q-k+n)}) \sim \prod_{L^{(d-k)} \in \mathcal{F}} U_g\left(M^{(d-q-k+n)} \cap L^{(d-k)}\right). \tag{19}$$

See Figure 1 for a visualization.

The reason it makes sense to plug $M^{(d-q-k+n)} \cap L^{(d-k)}$ into $U_g$ is that the submanifold $M^{(d-q-k+n)}$ is $(d-q-k+n)$-dimensional, and the leaves of $\mathcal{F}$ are $(d-k)$-dimensional; they both are living inside a leaf of $\mathcal{F}_{\mathrm{C}}$ which has dimension $d-k+n$, and therefore they generically[6] have an intersection (if it is not empty) of dimension $d-q-k$, which is precisely the right dimension for the symmetry operators of a $(q,k)$-symmetry in $d$ spatial dimensions. For example, when $q=0$ and $n=k$, the symmetry operators of the diagonal zero-form group $^{\mathrm{D}}G^{(0)}$ simply consist of the product of the symmetry operators of $G^{(0,k)}$ over all the leaves of the foliation.

One can also take diagonal subgroups when there are multiple foliations in the natural way. That is, if $\mathcal{F}_1, \dots, \mathcal{F}_n$ admit a common coarsening $\mathcal{F}_{\mathrm{C}}$, then one can identify a mutual diagonal subgroup

$$U_g^{\mathrm{C}}(M^{(d-q-k+n)}) = \prod_{i=1}^{n} \prod_{L^{(d-k)} \in \mathcal{F}_i} U_g^{(i)}\left(M^{(d-q-k+n)} \cap L^{(d-k)}\right). \tag{20}$$

---

[6]On the lattice, this assertion appears to be a safe one, however more care may be required in the continuum.

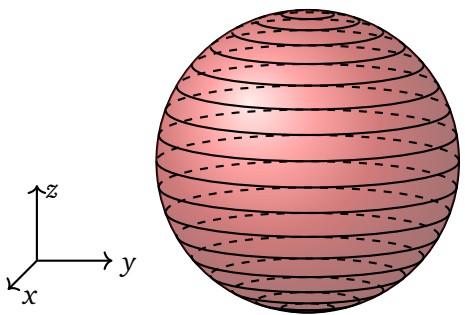

Figure 1: A relation-free $G^{(1,1)}$ symmetry in (3+1)D (i.e. a planar one-form subsystem symmetry group) admits an ordinary one-form subgroup $^{\mathrm{D}}G^{(1)}$ called the diagonal subgroup. The symmetry operators of $G^{(1,1)}$ live on lines supported on the 2 spatial-dimensional leaves of the foliation. The symmetry operators of $^{\mathrm{D}}G^{(1)}$ live on membranes. The membrane operators are built up by taking a product of line operators over the leaves of the foliation, according to Eq. (19). In this case, the foliation $\mathscr{F}_{\mathrm{z}}^{\perp} = \{L^{(2)}\}$ is by planes perpendicular to the z direction, and a symmetry operator supported on a 2-sphere is built out of line operators on the circles $S^2 \cap L^{(2)}$ which foliate it.

*Symmetries of boundaries*

It is also useful to note what happens when one places a theory with a higher-form global subsystem symmetry $G^{(q,k)}(\mathscr{F})\big/\sim$ on a manifold with boundary. In particular, let $q > 0$ and take the spatial manifold $M^{(d)}$ to be such that $\partial M^{(d-1)}$ is transverse to every leaf of $\mathscr{F}$. Furthermore, call $\mathscr{F}_{\mathrm{B}} = \{L^{(d-k)} \cap \partial M^{(d-1)} \mid L^{(d-k)} \in \mathscr{F}\}$ the induced foliation of the boundary. Then the boundary theory enjoys an action of $G^{(q-1,k)}(\mathscr{F}_{\mathrm{B}})\big/\sim$ (though on the low energy subspace, this action may potentially have non-trivial kernel). The reason for this is that one can simply push the symmetry operators of $G^{(q,k)}(\mathscr{F})\big/\sim$ along the leaves of the foliation until they are supported entirely on the boundary, where they can then be interpreted as $G^{(q-1,k)}(\mathscr{F}_{\mathrm{B}})\big/\sim$ symmetry operators of the boundary theory.

**Spontaneous breaking of higher-form global subsystem symmetries**

Our focus throughout this work is on quantum phases and phase transitions that are characterized by the spontaneous breaking of discrete higher-form subsystem symmetries. Here, we make explicit an assumption that is used in various places throughout this work: heuristically, the assumption says that spontaneous symmetry breaking phase transitions can always be diagnosed by the scaling behavior of order/disorder parameters. Accordingly, we call this assumption the *Landau assumption*. A pleasant feature of such a characterization by order/disorder parameters is that, as we explain below, it plays well with gauging. We also comment on the relation of this characterization to a characterization by the superselection sector of the excitation created at the boundary of a truncated symmetry operator; for lack of a better name, we refer to such objects as *truncated symmetry excitations* (TSEs). We do not claim that the notions we introduce below are necessarily fully general, but they are suitable for the purposes of what we set out to do in this paper.

*Order parameters*

Let us begin with a discussion of order parameters in the context of ordinary $q$-form symmetries $G^{(q)}$. An operator $\mathcal{O}$ is a *candidate order parameter* for an element $g \in G$ if it is a "large" $q$-dimensional operator that is globally $g$-symmetric, yet locally looks charged. For example,

when $q = 0$, we usually consider operators of the form

$$\mathcal{O}_{\vec{x},\vec{x}'} = \mathcal{O}_{\vec{x}}^{\dagger} \mathcal{O}_{\vec{x}'}, \tag{21}$$

as $|\vec{x} - \vec{x}'| \to \infty$, where $\mathcal{O}_{\vec{x}}$ is a local operator that has a non-trivial transformation under the action of $g$. When $q = 1$, we consider large, contractible closed loop operators $\mathcal{O}_{\gamma}$. And so on and so forth.

Let $\mathcal{T}$ be a gapped theory with a $G^{(q)}$ symmetry. The *Landau assumption for order parameters* says the following.

1) If an element $g \in G$ is spontaneously broken, then there exists a candidate order parameter $\mathcal{O}$ for $g$ whose vacuum expectation value $\langle \mathcal{O} \rangle$ decays with its $q$-dimensional size. In this case, we say that $\mathcal{O}$, or $\langle \mathcal{O} \rangle$, is an order parameter for the spontaneous breaking of $g$.

2) If an element $g \in G$ is unbroken, then all candidate order parameters for $g$ have vacuum expectation values that decay with the $(q + 1)$-dimensional size of their bulk.

A few remarks are in order. The first is that, for ordinary zero-form symmetries, often $\mathcal{O}_{\vec{x}}$ is the object that is referred to as the order parameter, as opposed to $\mathcal{O}_{\vec{x},\vec{x}'}$, and one correspondingly uses $\langle \mathcal{O}_{\vec{x}} \rangle$ to diagnose spontaneous symmetry breaking. One reason we have opted to consider the two-point function as opposed to the one-point function is that the latter always vanishes for reasons of symmetry; this forces one to introduce symmetry-breaking perturbations, and study the behavior of $\langle \mathcal{O}_{\vec{x}} \rangle$ as they are turned off. We find it more straightforward to simply consider the two-point function.

Second, it might strike the reader as surprising that we have introduced this as an assumption, rather than as a fact. To the best of our knowledge, the "Landau assumption" as stated has not been proven in full generality, although see e.g. [81] for work in this direction in the simplest case of $\mathbb{Z}_2^{(0)}$ symmetries in (1+1)D.

We make a similar assumption in the case of higher-form subsystem symmetries. Here, if we are interested in an element $g$ that implements the symmetry on a particular leaf $L^{(d-k)}$, then a candidate order parameter for $g$ should be an operator that, when truncated to the leaf $L^{(d-k)}$, looks like a candidate order parameter for an ordinary $q$-form symmetry on the leaf. When we refer to the size of the order parameter, we will typically mean the size as measured in this leaf-truncated operator. Similar comments apply to the case that we are considering an element $g$ that acts on multiple leaves simultaneously.

*Disorder parameters*

Disorder parameters play a dual role to order parameters in diagnosing the spontaneous breaking of a symmetry. (See e.g. Ref. [82] for a discussion.) Again, we warm up with the case of discrete ordinary higher-form symmetries $G^{(q)}$. An operator $\mathcal{D}$ is a *candidate disorder parameter* for an element $g \in G$ if it is a $g$-symmetric operator that is obtained as a truncated symmetry operator $U_g(M^{(d-q)})$, decorated by an operator on its boundary $\partial M^{(d-q-1)}$. The *Landau assumption for disorder parameters* says the following.

1) If an element $g \in G$ is unbroken, then there exists a candidate disorder parameter $\mathcal{D}$ for $g$ whose vacuum expectation value $\langle \mathcal{D} \rangle$ decays with the $(d-q-1)$-dimensional size of the boundary. In this case, we say that $\mathcal{D}$, or $\langle \mathcal{D} \rangle$, is a disorder parameter for $g$.

2) If an element $g \in G$ is broken, then all candidate disorder parameters for $g$ have a vacuum expectation value that decays with the $(d-q)$-dimensional size of the bulk of the operator.

The scaling behavior of disorder parameters is closely related to the superselection sector of the associated TSEs. If the superselection sector of the TSE is trivial, i.e. if there exists some operator acting within a local neighborhood of the TSE (taken as the size of the truncated symmetry operator diverges) that removes the TSE from the ground state, then the disorder parameter is expected to decay exponentially with the size of the boundary. This is because the action of the truncated symmetry within the ground space is equivalent to that of an operator within a neighborhood of the boundary. In contrast, for TSEs which belong to nontrivial superselection sectors, no operator acting within a neighborhood of the boundary of the truncated symmetry operator can remove the TSE. Thus, the disorder parameter should decay faster than the size of the boundary region, and is generically expected to decay with the size of the bulk of the truncated symmetry. This point of view can be used to deduce which TSEs are condensed across a spontaneous symmetry breaking phase transition. That is, for each set of disorder parameters that exhibit a jump from boundary to bulk scaling across a phase transition, the associated TSEs are condensed as they switch from a nontrivial to the trivial superselection sector.

Again, this discussion generalizes to discrete higher-form subsystem symmetries. If we are interested in an element $g$ that implements the symmetry on a particular leaf $L^{(d-k)}$, then a candidate disorder parameter for $g$ should be a truncated version of the symmetry operator that implements $g$, decorated by an operator along its boundary.

**Gauging $\mathbb{Z}_2^{(q,k)}/\sim$ symmetries**

The examples we treat are sufficiently simple that we do not attempt to give a fully general prescription for how to carry out the gauging of a $(q,k)$-symmetry group (see Ref. [33, 34, 83] for some work in this direction). However there are a few steps that we carry out in all the models we consider, and it is useful to summarize them in words. We distinguish between two slightly different kinds of gauging, which we call *strict gauging* and *energetic gauging*.

1) First, one enlarges the Hilbert space to include degrees of freedom describing the gauge field.

2) Next, one imposes a Gauss's law constraint on the enlarged Hilbert space so that only gauge invariant states are considered. In this paper, we exclusively work with $\mathbb{Z}_2^{(q,k)}/\sim$ symmetries, and our convention is that if the symmetry generators are made up of Pauli-**X** (Pauli-**Z**) operators, then the associated Gauss's law operators are made up entirely out of Pauli-**X** (Pauli-**Z**) operators as well.

3) One should then modify the terms of the ungauged Hamiltonian so that they commute with the Gauss's law constraint. We refer to this as "covariantizing" the Hamiltonian, and it is this process which couples the "matter" fields of the original model to the gauge field.

4) Finally, one can sometimes define local "flux operators" which for our purposes are operators that commute with the Gauss's law constraint, act non-trivially only on the degrees of freedom of the gauge field, and by convention—in the case of $\mathbb{Z}_2^{(q,k)}/\sim$ symmetries generated by Pauli-**X**s (Pauli-**Z**s)—are formed entirely out of Pauli-**Z**s (Pauli-**X**s). The difference between strict gauging versus energetic gauging lies in how we incorporate these flux operators.

   (a) In strict gauging, we impose that the flux operators all have eigenvalue 1 as a *constraint* on the Hilbert space, which can be thought of as a kind of flatness condition for the gauge field. One reason for doing this is that, when they exist, they can

often be interpreted as implementing a Gauss's law constraint of the *gauged* theory. Furthermore, imposing them as a constraint turns out to guarantee certain emergent "quantum" global symmetries of the gauged theory (which are commented on shortly). With this protocol, the number of parameters of the Hamiltonian is the same before and after gauging, and one can track how different phases and phase transitions are mapped under the gauging procedure. This procedure also more closely mirrors what happens when one gauges a discrete symmetry in the continuum.

(b) In energetic gauging, we add the flux operators as terms in the gauged Hamiltonian, i.e. we impose them only energetically. This way, we are free to add any other terms to the Hamiltonian to control the energetics of the gauge fields, so long as they commute with the Gauss's law constraint, but even if they do not commute with the flux operators. (This is in contrast with the case of strict gauging, where we can only add terms which commute with the flux operators.) Under this protocol, the number of parameters of the gauged model is greater than the number of parameters of the ungauged model by at least 1 (the strength of the flux term). This protocol is more useful if we are interested in recovering a more general gauged Hamiltonian with more knobs to tune.

Unless we explicitly say otherwise, it should be assumed that the strict gauging procedure is being used.

It will be useful to keep track of the fate of certain classes of excitations under the gauging map, as we make use of these properties in the examples throughout the text. TSEs become topologically trivial in the gauged theory. Symmetry twists, i.e. defects created at the end of a truncated symmetry excitation, are locally equivalent to standalone flux excitations after gauging (i.e. violations of the flux operators) because any TSE they were attached to is trivialized during the gauging procedure. Similarly, excitations charged under the symmetry are mapped to gauge charges after gauging. These gauge charges are by definition TSEs of the dual emergent quantum symmetry operators, described below.

We now establish some expectations about the properties of the theory $\mathcal{T}\big/(G^{(q,k)}\big/{\sim})$ obtained by gauging a non-anomalous $G^{(q,k)}\big/{\sim}$ symmetry of a theory $\mathcal{T}$ defined in $(d+1)$ spacetime dimensions. We specialize throughout to the case that $G$ is a finite Abelian group.

**Expectation 1:** The gauged theory $\mathcal{T}\big/\big(G^{(q,k)}\big/{\sim}\big)$ enjoys (at least) an emergent "quantum" global symmetry group $\widehat{G}^{(d-k-q-1,k)}\big/{\sim}$, where $\widehat{G} = \mathrm{hom}(G, U(1))$ is the Pontryagin dual of $G$. Moreover, re-gauging this emergent symmetry should recover the original theory, i.e.

$$\mathcal{T}\big/(G^{(q,k)}\big/{\sim})\big/(\widehat{G}^{(d-k-q-1,k)}\big/{\sim}) \cong \mathcal{T}. \tag{22}$$

Both of these claims generalize familiar phenomena in the context of ordinary higher-form symmetries $G^{(q)}$ (cf. e.g. [8, 77–79]). See also [83] for examples in the setting of subsystem symmetries.

*Sketch of the idea:* We briefly and roughly remind the reader of the intuition behind this emergent global symmetry group in the case of ordinary higher-form global symmetries $G^{(q)}$. We want to argue that $\mathcal{T}\big/G^{(q)}$ enjoys a $\widehat{G}^{(d-q-1)}$ global symmetry; since $q$-form symmetries act on $q$-dimensional operators, we should understand the space of $(d-q-1)$-dimensional operators in $\mathcal{T}\big/G^{(q)}$. These can be described as follows. Let $^{(d-q-1)}\mathcal{T}$ be the space of $(d-q-1)$-dimensional operators of the theory $\mathcal{T}$, let $^{(d-q-1)}\mathcal{T}_g$ be the space of $(d-q-1)$-dimensional operators which can live on the boundary $\partial M^{(d-q-1)}$ of an open symmetry operator $U_g(M^{(d-q)})$,

and call $^{(d-q-1)}\mathcal{T}_g^+$ its $G^{(q)}$-invariant subspace. (There is only potentially a difference between $^{(d-q-1)}\mathcal{T}_g^+$ and $^{(d-q-1)}\mathcal{T}_g$ in the case that $2q = d-1$, e.g. zero-form symmetries in (1+1)D, one-form symmetries in (3+1)D, and so on.) By definition, the genuinely $(d-q-1)$-dimensional operators of $\mathcal{T}/G^{(q)}$ can be identified as

$$^{(d-q-1)}\left(\mathcal{T}/G^{(q)}\right) \simeq \bigoplus_{g \in G} {}^{(d-q-1)}\mathcal{T}_g^+ \, . \tag{23}$$

The statement then is that there is a symmetry action of $\widehat{G} = \hom(G, U(1))$ on this space of $(d-q-1)$-dimensional operators of the gauged theory: an element $\chi \in \widehat{G}$ simply acts on all operators in the summand $^{(d-q-1)}\mathcal{T}_g^+$ by multiplication by $\chi(g)$. Since the Pontryagin dual acts on $(d-q-1)$-dimensional operators, we interpret it as a $\widehat{G}^{(d-q-1)}$ symmetry of $\mathcal{T}/G^{(q)}$.

A similar idea generalizes easily to relation-free higher-form subsystem symmetries $G^{(q,k)}(\mathcal{F}_1, \mathcal{F}_2, \dots)$, since their gauging can essentially be thought of as simply gauging ordinary $q$-form symmetries leaf-by-leaf along the foliations. This is the main case that is needed in this paper.

In the case that there are relations, more care is needed in general to make sense of this proposal. We leave this interesting question to future work. For an infinite family of examples where Expectation 1 holds, see §2.2.4. The network construction presented in §4.1.2 also involves gauging an HFSS with relations.

Our second expectation concerns how spontaneous symmetry breaking plays with gauging. For simplicity, we only state how this works for ordinary $q$-form symmetries, and content ourselves with seeing how a similar idea plays out in examples of subsystem gauging throughout the rest of the paper. Even in the simpler setting of $q$-form symmetries, we were not able to find the following claim in the literature.

**Expectation 2:** Consider the case of a gapped theory $\mathcal{T}$ with a discrete $q$-form symmetry $G^{(q)}$ which is spontaneously broken down to a subgroup $H^{(q)}$. We further assume that the theory has trivial order under the unbroken subgroup $H^{(q)}$. The claim is then that, in the gauged theory $\mathcal{T}/G^{(q)}$, the emergent quantum symmetry group $\widehat{G}^{(d-q-1)}$ is broken down to the subgroup $K^{(d-q-1)}$ defined by

$$K = \left\{ \chi \in \widehat{G} \,|\, H \subset \ker \chi \right\} \, . \tag{24}$$

In particular, if $G^{(q)}$ is completely spontaneously broken, then $\widehat{G}^{(d-q-1)}$ is completely unbroken; if $G^{(q)}$ is unbroken, then $\widehat{G}^{(d-q-1)}$ is completely broken.

*Remark:* In the above statement, it is important that the theory has trivial order under the unbroken subgroup $H^{(q)}$ rather than nontrivial symmetry-preserving order such as symmetry-protected or symmetry-enriched topological order.

An example of this which is familiar to high-energy theorists is Yang-Mills theory in (3+1)D: it is believed that the pure $SU(N)$ theory enjoys an area law for its fundamental Wilson loop, suggesting that its electric $\mathbb{Z}_N^{(1)}$ one-form symmetry is unbroken, whereas the $SU(N)/\mathbb{Z}_N$ Yang-Mills theory (which can be thought of as being obtained by gauging the $\mathbb{Z}_N^{(1)}$ symmetry of $SU(N)$ YM) should have a 't Hooft line with a perimeter law, signaling that its magnetic $\widehat{\mathbb{Z}}_N^{(1)}$ symmetry (i.e. the emergent "quantum" global symmetry guaranteed by the gauging of $\mathbb{Z}_N^{(1)}$) is spontaneously broken. We see how a similar idea holds for subsystem symmetries in examples throughout the rest of the paper.

*Sketch of idea:* Let us offer one potential proof strategy. It rests on the Landau assumption explained earlier in this section.

First let us show that, in the gauged theory $\mathcal{T}/G^{(q)}$, symmetry elements $\chi$ belonging to $K$ are unbroken. We can do this by showing that all candidate order parameters have vacuum expectation values which decay with the size of the $(d-q)$-dimensional region which they bound. In the ungauged theory $\mathcal{T}$, consider a candidate disorder parameter $\mathcal{D}_g$ for some $g \in G$. Since it is by definition $G^{(q)}$-even, it descends to an operator of the gauged theory $\mathcal{T}/G^{(q)}$. Since the bulk of the operator is by definition implementing a symmetry that is being gauged, the bulk acts trivially in the gauged theory, and the operator becomes a genuinely $(d-q-1)$-dimensional operator $\widehat{\mathcal{O}}_g$. Such an operator can be used to define a candidate order parameter in $\mathcal{T}/G^{(q)}$ for any $\chi$ for which $\chi(g) \neq 1$. Thus, candidate order parameters $\widehat{\mathcal{O}}_g$ in $\mathcal{T}/G^{(q)}$ for elements $\chi \in K$ come from candidate disorder parameters $\mathcal{D}_g$ in $\mathcal{T}$ for elements $g \notin H$. By assumption, such $g$ are spontaneously broken in $\mathcal{T}$, and so candidate disorder parameters for such $g$ all have vacuum expectation values that decay with the $(d-q)$-dimensional size of the bulk of the operator. Moreover, since we are gauging a discrete symmetry, we have that

$$\langle \widehat{\mathcal{O}}_g \rangle_{\mathcal{T}/G^{(q)}} = \langle \mathcal{D}_g \rangle_{\mathcal{T}}. \tag{25}$$

Thus, all candidate order parameters satisfy that $\langle \widehat{\mathcal{O}}_g \rangle$ decays with the $(d-q)$-dimensional size of the bulk of the operator. According to the Landau assumption, this means that $K$ is unbroken.

On the other hand, let us show that symmetry elements $\chi$ that are not inside $K$ are broken. We can do this by showing that there exists a candidate order parameter whose vacuum expectation value decays with its $(d-q-1)$-dimensional size. If $\chi$ is not in $K$, then it means there is some $g \in H$ such that $\chi(g) \neq 1$. In the ungauged theory $\mathcal{T}$, such a $g$ is unbroken, and so by the Landau assumption, there exists a disorder parameter $\mathcal{D}_g$ whose vacuum expectation value decays with its $(d-q-1)$-dimensional boundary. In the gauged theory, this descends to an operator $\widehat{\mathcal{O}}_g$ that is locally charged with respect to $\chi$, and has a VeV that decays with its $(d-q-1)$-dimensional size. This means that $\widehat{\mathcal{O}}_g$ is an order parameter that diagnoses the spontaneous breaking of $\chi$.

**Local equivalence and condensation of topological excitations**

In many of the examples throughout this work we deal with topologically nontrivial phases and their excitations. For our purposes a gapped quantum phase of matter is considered topological if it is disconnected from the trivial phase under gap preserving adiabatic deformations and additionally admits no local order parameters. The trivial phase is defined to be the phase that contains a totally decoupled paramagnetic Hamiltonian. Within topological phases, excitations are often grouped into *superselection sectors*. That is, they are considered only up to equivalence under local operations in a neighborhood of the region on which they are supported (i.e. where their associated energy density is supported). Excitations that can be created within a neighborhood of their support are considered topologically trivial. Excitations that can only be created by an operator on a much larger region than the support of the excitation itself are *topologically nontrivial,* often referred to simply as *topological*. Similarly, a pair of excitations supported on some region that differ only by the application of an operator on a neighborhood of that region are considered topologically equivalent. A familiar example of a topologically trivial excitation is a spinon in the trivial phase of the Ising model. An example of a topologically nontrivial excitation is an anyon in the (2+1)D toric code.

In many of the examples throughout the text we consider phase transitions that are driven by the fluctuation and condensation of topological excitations. We say that the addition of a

local term to a Hamiltonian supporting a topological phase *fluctuates* some topological excitation if the term introduces a matrix element for that excitation to change its position. As the strength of such terms are increased sufficiently we expect the topological excitations that are fluctuating to condense [84]. Here we are assuming that all the fluctuating excitations have trivial mutual topological braiding processes, otherwise there may be some inconsistency that prevents condensation. We consider an excitation, defined by the violation of a certain set of local terms in a Hamiltonian whose interaction strengths are being tuned, to be *condensed* if it switches from being topologically nontrivial to topologically trivial as a phase transition point is crossed. This occurs for example to the electric anyons of the toric code as it is driven into the trivial phase by their condensation.

Many of the examples we consider contain fractonic topological excitations that obey some mobility restrictions, i.e. they can only be moved within a subdimensional manifold via local operators. In some of the examples we also encounter interesting related phenomena where particular representative topological excitations have restricted mobility, even though there are no mobility restrictions on the associated topological superselection sector. That is, such excitations have mobility restrictions unless they are transformed via the application of local operators into some different representative excitations for the same topological equivalence class.[7] In both of the situations outlined in this paragraph, when terms are added to a Hamiltonian that correspond to hopping operators for some topological excitation within a subdimensional manifold, we say that such terms cause the associated excitations to fluctuate within that manifold. When the strength of those Hamiltonian terms is increased sufficiently to induce a phase transition, we similarly say the corresponding excitations are condensed within the subdimensional manifold.

To orient the reader, and to establish notations and conventions, we now review how all the principles that we have laid out in this section play out in the more familiar theories ${}^d H_{\mathbb{Z}_2}^{(q)}$ with ordinary higher-form symmetries. For the rest of the paper, we specialize to the case that $G = \mathbb{Z}_2$.

## 2.2 Review of building blocks

As we have already mentioned, our constructions involve taking simple building blocks (which in the present context, are the theories ${}^d H_{\mathbb{Z}_2}^{(q)}$ with ordinary higher-form symmetries), foliating space with them, and applying various gauging procedures to obtain new theories. Therefore we start by establishing the basic properties of these ${}^d H_{\mathbb{Z}_2}^{(q)}$ Hamiltonians: symmetries, gauging, duality, order/disorder parameters, boundaries, and so on.

In (1+1)D, only ordinary zero-form symmetries are possible. The standard choice of a model with such a symmetry is the transverse field Ising model (TFIM) ${}^1 H_{\mathbb{Z}_2}^{(0)}$ which we review in §2.2.1. In (2+1)D, it is possible to have theories with zero-form or one-form symmetries. Here, the standard choices are again the (2+1)D TFIM ${}^2 H_{\mathbb{Z}_2}^{(0)}$ and $\mathbb{Z}_2$ lattice gauge theory ${}^2 H_{\mathbb{Z}_2}^{(1)}$, respectively. We exposit both these models and the relations between them in §2.2.2. In (3+1)D, it is possible to have zero-form, one-form, or two-form symmetries. We review the various $\mathbb{Z}_2$ higher-form gauge theories that realize these symmetries in §2.2.3. Finally, we show in §2.2.4 how all of these considerations can be generalized to the higher-form subsystem setting by sketching how they play out in an infinite family of models ${}^d H_{\mathbb{Z}_2}^{(q,k)}$ in $d$ spatial dimensions with $\mathbb{Z}_2^{(q,k)} / \sim$ symmetry.

Readers who are familiar with this material can safely skip this subsection on a first read through.

---

[7]A similar property, and its association with subsystem symmetry, is explored in Ref. [85].

### 2.2.1 (1+1)D transverse field Ising model

Consider a one-dimensional lattice with $N$ sites on a circle, and place a qubit on each site so that the Hilbert space is $\mathcal{H} = (\mathbb{C}^2)^{\otimes N}$. We take the Hamiltonian of the transverse field Ising model to be

$$^1 H^{(0)}_{\mathbb{Z}_2}(J,h) = -\sum_{i=0}^{N-1}(J\mathbf{Z}_i\mathbf{Z}_{i+1} + h\mathbf{X}_i),\tag{26}$$

where $\mathbf{X}_i, \mathbf{Z}_i$ are Pauli operators supported at site $i$, and the boundary conditions are $\mathbf{Z}_{i+L} = Q\mathbf{Z}_i$, with $Q = +1$ corresponding to periodic boundary conditions, and $Q = -1$ corresponding to anti-periodic boundary conditions. This model enjoys an ordinary zero-form global $\mathbb{Z}_2^{(0)}$ symmetry that is implemented by the unitary operator

$$U = \prod_{i=0}^{N-1}\mathbf{X}_i.\tag{27}$$

An example of a local operator which is charged under this $\mathbb{Z}_2$ is simply $\mathbf{Z}_i$. This symmetry allows us to split the Hilbert space into even/odd sectors,

$$\mathcal{H} = \bigoplus_{\widetilde{Q}=\pm 1}\mathcal{H}^{(\widetilde{Q})},\tag{28}$$

depending on the eigenvalue of $U$.

In the thermodynamic limit, $N \to \infty$, the TFIM famously enjoys a phase transition at the critical point $J/h = 1$ that is associated with the spontaneous breaking of the global $\mathbb{Z}_2^{(0)}$ symmetry. When $J/h < 1$, the model has a unique, gapped, ground state $|\Omega\rangle$ that is invariant under the action of $U$, and the system is said to be in the disordered, or symmetry-preserving phase. The single quasi-particle excitations above the vacuum in this phase are spinons which, in the limit that $J/h \ll 1$, can be obtained by acting with a spin flip operator on the vacuum,

$$|i\rangle \sim \mathbf{Z}_i|\Omega\rangle + \cdots\tag{29}$$

Corrections to this expression can be computed using standard perturbation theory techniques.

On the other hand, when $J/h > 1$, the model has a pair of degenerate ground states that are mixed into each other by the action of $U$, and the theory is said to be in its ordered, or symmetry breaking phase. In this case, the low-lying quasi-particle excitations are pairs of domain walls (a.k.a. kinks) that live on the links of the lattice, and which separate islands of up spins from down spins. Deep in the ordered phase, $J/h \gg 1$, they can be obtained from either of the two vacua as

$$|i - \tfrac{1}{2}, j + \tfrac{1}{2}\rangle \sim \prod_{k=i}^{j}\mathbf{X}_k|\Omega\rangle + \cdots,\tag{30}$$

where in the above expression, we have denoted the links of the lattice by half integers.

As we have explained in §2.1, the philosophy of the Landau program is to identify operators that diagnose which phase one is in through their ground state expectation values. Roughly, for zero-form symmetries in (1+1)D, if the operator has a non-zero expectation value in the ordered phase while having zero expectation value in the disordered phase, it is said to be an *order parameter*; if its expectation values are the other way around, it is called a *disorder parameter*. It is known that every gapped, translationally invariant, $\mathbb{Z}_2$-symmetric spin chain

admits order and disorder parameters [81]. For the Hamiltonian in Eq. (26), it is standard to consider the two-point function

$$\mathscr{O} = \lim_{|i-j|\to\infty} \langle \mathscr{O}_{i,j} \rangle, \qquad \mathscr{O}_{i,j} = \mathbf{Z}_i \mathbf{Z}_j, \tag{31}$$

as an order parameter, and to take a truncated symmetry operator

$$\mathscr{D} = \lim_{|i-j|\to\infty} \langle \mathscr{D}_{i,j} \rangle, \qquad \mathscr{D}_{i,j} = \prod_{k=i}^{j} \mathbf{X}_k, \tag{32}$$

as a disorder parameter. Notice that $\mathscr{O}_{ij}$ creates a pair of spinons when $J/h \ll 1$, while $\mathscr{D}_{ij}$ creates a pair of domain-walls when $J/h \gg 1$. When one is away from these extreme limits of $J/h$, the true spinon/domain-wall creation operators still have non-zero overlap with $\mathscr{O}_{ij}$ and $\mathscr{D}_{ij}$, respectively. For this reason, the pattern of expectation values enjoyed by the order/disorder parameters can be interpreted as the statement that the vacuum in the symmetry preserving phase is a condensate of domain-walls, while spinons are condensed in the vacua of the symmetry-breaking phase.

Now, we can consider gauging the global symmetry. According to the discussion of the previous subsection, this should produce a theory with an emergent quantum $\widehat{\mathbb{Z}}_2^{(0)}$ symmetry. Let us see how this comes about in practice. In the present case, the first step of the gauging protocol outlined in §2.1 can be accomplished by placing an additional qubit on each edge of the lattice; we write $\mathbf{X}_{i+\frac{1}{2}}$, $\mathbf{Z}_{i+\frac{1}{2}}$ to denote Paulis acting on the edge between sites $i$ and $i+1$. The Gauss's law constraint is then that all states should satisfy

$$G_i := \mathbf{X}_{i-\frac{1}{2}} \mathbf{X}_i \mathbf{X}_{i+\frac{1}{2}} = 1. \tag{33}$$

The transverse field terms of the Ising model already commute with this constraint, however the other terms do not. To covariantize them, we couple them to the gauge field by replacing $\mathbf{Z}_i \mathbf{Z}_{i+1} \to \mathbf{Z}_i \mathbf{Z}_{i+\frac{1}{2}} \mathbf{Z}_{i+1}$. As for the last step of the gauging protocol, there are no local flux terms for us to consider, and so the difference between strict gauging and energetic gauging is somewhat immaterial. All in all, the gauged Hamiltonian becomes

$$^1H_{\mathbb{Z}_2}^{(0)} \big/ \mathbb{Z}_2^{(0)} = -\sum_{i=0}^{N-1} \left( J \mathbf{Z}_i \mathbf{Z}_{i+\frac{1}{2}} \mathbf{Z}_{i+1} + h \mathbf{X}_i \right)$$

$$G_i = \mathbf{X}_{i-\frac{1}{2}} \mathbf{X}_i \mathbf{X}_{i+\frac{1}{2}} = 1. \tag{34}$$

In its current form, solving the Gauss's law constraint is somewhat cumbersome because it is a three-body term. However, we can apply a local unitary circuit which localizes it to a single site. Indeed, consider the circuit

$$V = H^{\otimes 2N} \prod_{i=0}^{N-1} \left( C_i \mathbf{X}_{i-\frac{1}{2}} \right) \left( C_i \mathbf{X}_{i+\frac{1}{2}} \right), \tag{35}$$

where $C_i \mathbf{X}_{i\pm\frac{1}{2}}$ are CNOT gates, and $H^{\otimes 2N}$ is a tensor product of Hadamard rotations which swaps all Pauli-$\mathbf{X}$ operators with Pauli-$\mathbf{Z}$ operators. Using the identities in Eq. (14), one finds that conjugating by $V$ leads to the Hamiltonian

$$V \left( {}^1H_{\mathbb{Z}_2}^{(0)} \big/ \mathbb{Z}_2^{(0)} \right) V^\dagger = -\sum_{i=0}^{N-1} \left( J \mathbf{X}_{i+\frac{1}{2}} + h \mathbf{Z}_{i-\frac{1}{2}} \mathbf{Z}_i \mathbf{Z}_{i+\frac{1}{2}} \right)$$

$$V G_i V^\dagger = \mathbf{Z}_i = 1. \tag{36}$$

Now, imposing $V G_i V^\dagger = 1$ simply freezes the qubits on the lattice sites to the $+1$ eigenstate of Pauli-$\mathbf{Z}$. Therefore, the gauged Hamiltonian simply goes over to

$$V\left(^1 H_{\mathbb{Z}_2}^{(0)}(J,h)\Big/\mathbb{Z}_2^{(0)}\right)V^\dagger \xrightarrow{V G_i V^\dagger = 1} -\sum_{i=0}^{N-1}\left(J\mathbf{X}_{i+\frac{1}{2}} + h\mathbf{Z}_{i-\frac{1}{2}}\mathbf{Z}_{i+\frac{1}{2}}\right) \cong \widetilde{H}_{\mathbb{Z}_2}(h,J). \qquad (37)$$

We see that we have recovered the Kramers–Wannier dual of the Ising model.[8] The tilde in $\widetilde{H}_{\mathbb{Z}_2}$ is meant to emphasize that the model is defined on the dual lattice. We see that we have gained an emergent $\widehat{\mathbb{Z}}_2^{(0)}$ symmetry implemented by

$$\widehat{U} = \prod_i \mathbf{X}_{i+\frac{1}{2}}. \qquad (38)$$

Because $J$ and $h$ have been exchanged, the spontaneous symmetry breaking phase of $^1 H_{\mathbb{Z}_2}^{(0)}$ maps after gauging to the symmetry preserving phase of $^1\widetilde{H}_{\mathbb{Z}_2}^{(0)} \cong {}^1 H_{\mathbb{Z}_2}^{(0)}\big/\mathbb{Z}_2^{(0)}$, and vice versa. In connection with this, one can check that the order/disorder parameters are exchanged by the gauging procedure. For example, in the thermodynamic limit, there is essentially a unique way to covariantize the order parameter (at least if one would like to maintain finite support), and after conjugating by $V$, it leads to

$$\mathscr{O}_{ij} = \mathbf{Z}_i \mathbf{Z}_j \xrightarrow{\text{cov.}} \mathbf{Z}_i\left(\prod_{k=i}^{j-1}\mathbf{Z}_{k+\frac{1}{2}}\right)\mathbf{Z}_j \xrightarrow{V} \prod_{k=1}^{j-1}\mathbf{X}_{k+\frac{1}{2}} = \widetilde{\mathscr{D}}_{i+\frac{1}{2},\,j-\frac{1}{2}}, \qquad (39)$$

which is the disorder parameter of $^1\widetilde{H}_{\mathbb{Z}_2}^{(0)}$. One can similarly track the fate of the disorder parameter of $^1 H_{\mathbb{Z}_2}^{(0)}$, and one finds that it is mapped to the order parameter of $^1\widetilde{H}_{\mathbb{Z}_2}^{(0)}$. Deep in the symmetry-breaking/symmetry-preserving phases, these manipulations can also be interpreted as describing how spinons are converted into domain-walls after gauging, and vice versa.

Finally, it is straightforward to see that re-gauging this emergent symmetry results back in the original TFIM,

$$^1\widetilde{H}_{\mathbb{Z}_2}^{(0)}(h,J)\Big/\widehat{\mathbb{Z}}_2^{(0)} \cong H_{\mathbb{Z}_2}^{(0)}(J,h). \qquad (40)$$

The entire discussion of this subsection is summarized in Table 2.

### 2.2.2 (2+1)D transverse field Ising model and $\mathbb{Z}_2$ gauge theory

We now repeat this discussion in one higher dimension. Consider an $N_x \times N_y$ square lattice on a torus, and place a qubit on each site $v = (i,j) \in \mathbb{Z}/N_x\mathbb{Z} \times \mathbb{Z}/N_y\mathbb{Z}$. The Hamiltonian of the transverse field Ising model in (2+1)D is

$$^2 H_{\mathbb{Z}_2}^{(0)} = -J\sum_{\langle v,v'\rangle}\mathbf{Z}_v \mathbf{Z}_{v'} - h\sum_v \mathbf{X}_v, \qquad (41)$$

where the first sum is over nearest neighbors on the lattice. This model again enjoys an ordinary global $\mathbb{Z}_2^{(0)}$ symmetry generated by

$$U = \prod_v \mathbf{X}_v. \qquad (42)$$

---

[8]Technically, $^1\widetilde{H}_{\mathbb{Z}_2}^{(0)}(h,J)$ is only dual to $^1 H_{\mathbb{Z}_2}^{(0)}(J,h)$ when the two models are restricted to their $\mathbb{Z}_2$-even sectors, while the charged sectors are dual to twisted sectors and vice versa; however, in keeping with tradition, we often abuse the word "dual" by ignoring this subtlety. See e.g. Ref. [86] for a careful treatment of Kramers-Wannier duality.

Table 2: A summary of the (1+1)D transverse field Ising model before and after gauging its global symmetry. The model after gauging is the (1+1)D TFIM again, but on the dual lattice. We have ommitted the second half of the table because it is essentially a mirror reflection of the first half.

| ${}^1H_{\mathbb{Z}_2}^{(0)}$ | ${}^1H_{\mathbb{Z}_2}^{(0)}\big/\mathbb{Z}_2^{(0)} \cong {}^1\widetilde{H}_{\mathbb{Z}_2}^{(0)}$ |
|---|---|
| $\mathbb{Z}_2^{(0)}$ symmetry: $U = \prod_i \mathbf{X}_i$ | $\widehat{\mathbb{Z}}_2^{(0)}$ symmetry: $\widehat{U} = \prod_i \mathbf{X}_{i+\frac{1}{2}}$ |
| ordered/sym. breaking phase: $J/h \gg 1$<br>order parameter: $\langle \mathbf{Z}_i \mathbf{Z}_j \rangle \sim$ const.<br>disorder parameter: $\langle \prod_{k=i}^{j} \mathbf{X}_i \rangle \sim$ exp. decay<br>excitations: domain-walls $\prod_{k=i}^{j} \mathbf{X}_k \lvert \Omega \rangle + \cdots$<br>condensed: spinons | disordered/sym. preserving phase: $J/h \gg 1$<br>disorder parameter: $\langle \prod_{k=i}^{j-1} \mathbf{X}_{k+\frac{1}{2}} \rangle \sim$ const.<br>order parameter: $\langle \mathbf{Z}_{i+\frac{1}{2}} \mathbf{Z}_{j+\frac{1}{2}} \rangle \sim$ exp. decay<br>excitations: spinons $\mathbf{Z}_i \lvert \Omega \rangle + \cdots$<br>condensed: domain-walls |
| disordered/sym. preserving phase: $J/h \ll 1$<br>disorder parameter: $\langle \prod_{k=i}^{j} \mathbf{X}_k \rangle \sim$ const.<br>$\vdots$ | ordered sym. breaking phase: $J/h \ll 1$<br>order parameter: $\langle \mathbf{Z}_{i+\frac{1}{2}} \mathbf{Z}_{j+\frac{1}{2}} \rangle \sim$ const.<br>$\vdots$ |

As before, the basic charged operators are $\mathbf{Z}_v$, and we can decompose the Hilbert space into even and odd sectors with respect to the action of $U$. We now must specify whether the boundary conditions are periodic or antiperiodic around each cycle of the torus; that is, we take $\mathbf{Z}_{i+L,j} = Q_A \mathbf{Z}_{i,j}$ and $\mathbf{Z}_{i,j+L} = Q_B \mathbf{Z}_{i,j}$, where $Q_A, Q_B = \pm 1$ indicate whether the boundary conditions are periodic or anti-periodic around the A and B cycles of the torus, respectively.

Just as in (1+1)D, the (2+1)D TFIM enjoys a spontaneous symmetry breaking phase transition at some critical value of $J/h$. We can again ask for order and disorder parameters that diagnose the phase. The order parameter for ${}^2H_{\mathbb{Z}_2}^{(0)}$ is basically the same as for ${}^1H_{\mathbb{Z}_2}^{(0)}$,

$$\mathcal{O} = \lim_{|v-v'|\to\infty} \langle \mathcal{O}_{v,v'} \rangle, \qquad \mathcal{O}_{v,v'} = \mathbf{Z}_v \mathbf{Z}_{v'}. \tag{43}$$

As before, we diagnose the phase depending on whether it limits to a constant or is exponentially decaying with the separation $|v-v'|$. Accordingly, we say that the spinon quasi-particles are either condensed or not, respectively. The disorder parameter can again be taken to be a truncated symmetry operator. That is, we choose a large connected domain wall (i.e. a loop $\tilde{\gamma}$ on the dual lattice), and we take a product over sites in the interior of this domain wall,

$$\mathcal{D} = \lim_{A(\tilde{\gamma})\to\infty} \langle \mathcal{D}_{\tilde{\gamma}} \rangle, \qquad \mathcal{D}_{\tilde{\gamma}} = \prod_{v\in\tilde{\gamma}^\circ} \mathbf{X}_v, \tag{44}$$

where $A(\tilde{\gamma})$ is the area of the loop $\tilde{\gamma}$, and $\tilde{\gamma}^\circ$ is the set of vertices in the interior of $\tilde{\gamma}$. We say that we are in the disordered phase if $\mathcal{D}$ decays with the perimeter of $\tilde{\gamma}$, and that we are in the ordered phase if it decays with the area. Accordingly, the domain-walls are either condensed or not, respectively.

Let us again consider gauging the global symmetry, at first according to the "strict" gauging procedure outlined in §2.1. As before, we enlarge the Hilbert space by placing qubits on each edge of the lattice. The model one then obtains is structurally similar to the one obtained in one lower dimension; the new feature is that the gauge field now admits non-trivial local flux terms $F_p$,

$$
\begin{aligned}
{}^2H_{\mathbb{Z}_2}^{(0)}\big/\mathbb{Z}_2^{(0)} &= -J\sum_{\langle v,v' \rangle} \mathbf{Z}_v \mathbf{Z}_{\langle v,v' \rangle} \mathbf{Z}_{v'} - h\sum_v \mathbf{X}_v, \\
F_p &= \prod_{e\in p} \mathbf{Z}_e = 1, \qquad G_v = \mathbf{X}_v \prod_{e\ni v} \mathbf{X}_e = 1.
\end{aligned}
\tag{45}
$$

In the above, we are labeling edges using their boundary vertices, $e = \langle v, v' \rangle$. We impose both $F_p = 1$ and $G_v = 1$ as constraints on the Hilbert space.

Again, we apply a local unitary circuit to make imposing $G_v = 1$ more straightforward,

$$V = H^{\otimes N} \prod_{\langle v,v' \rangle} (C_v \mathbf{X}_{\langle v,v' \rangle})(C_{v'} \mathbf{X}_{\langle v,v' \rangle}), \tag{46}$$

where $N$ is the total number of qubits of the gauged model. Conjugating by this unitary leads to

$$V \left( {}^2 H^{(0)}_{\mathbb{Z}_2} \Big/ \mathbb{Z}^{(0)}_2 \right) V^\dagger = -J \sum_e \mathbf{X}_e - h \sum_v \mathbf{Z}_v \prod_{e \ni v} \mathbf{Z}_e$$

$$V F_p V^\dagger = \prod_{e \in p} \mathbf{X}_e = 1, \quad V G_v V^\dagger = \mathbf{Z}_v = 1. \tag{47}$$

If we now shift perspectives to the dual lattice (so that vertices $v$ become plaquettes $\tilde{p}$, plaquettes $p$ become vertices $\tilde{v}$, and edges $e$ get sent to edges $\tilde{e}$) and impose $V G_v V^\dagger = 1$, we find

$$V \left( {}^2 H^{(0)}_{\mathbb{Z}_2} \Big/ \mathbb{Z}^{(0)}_2 \right) V^\dagger \xrightarrow{V G_v V^\dagger = 1} -J \sum_{\tilde{e}} \mathbf{X}_{\tilde{e}} - h \sum_{\tilde{p}} \prod_{\tilde{e} \in \tilde{p}} \mathbf{Z}_{\tilde{e}} \cong {}^2 \widetilde{H}^{(1)}_{\mathbb{Z}_2}$$

$$V F_{\tilde{v}} V^\dagger = \prod_{\tilde{e} \ni \tilde{v}} \mathbf{X}_{\tilde{e}} =: \widetilde{G}_{\tilde{v}}, \tag{48}$$

which is the Hamiltonian of $\mathbb{Z}_2$ lattice gauge theory on the dual lattice (see Ref. [87] for a standard reference on lattice gauge theories). We have interpreted the flux term $V F_{\tilde{v}} V^\dagger$ as the Gauss's law constraint $\widetilde{G}_{\tilde{v}}$ of ${}^2 \widetilde{H}^{(1)}_{\mathbb{Z}_2}$, so we demand that $\widetilde{G}_{\tilde{v}} = 1$. Thus, gauging again recovers the Kramers–Wannier dual (or rather, the Wegner dual [2]) of the (2+1)D TFIM.

We note in passing that if we were to instead use the "energetic gauging" protocol, our gauged theory would take the form

$$^2 H^{(0)}_{\mathbb{Z}_2} \Big|_E \mathbb{Z}^{(0)}_2 = -J \sum_{\langle v,v' \rangle} \mathbf{Z}_v \mathbf{Z}_{\langle v,v' \rangle} \mathbf{Z}_{v'} - h \sum_v \mathbf{X}_v - t \sum_p \prod_{e \in p} \mathbf{Z}_e - U \sum_e \mathbf{X}_e,$$

$$G_v = \mathbf{X}_v \prod_{e \ni v} \mathbf{X}_e, \tag{49}$$

where now, we have incorporated the flux operators into a term in the Hamiltonian that imposes them energetically, and we have further added an electric tension term proportional to $U$. After performing the same manipulations as above, we are lead to

$$V \left( {}^2 H^{(0)}_{\mathbb{Z}_2} \Big|_E \mathbb{Z}^{(0)}_2 \right) V^\dagger \xrightarrow{V G_v V^\dagger = 1} -h \sum_{\tilde{p}} \prod_{\tilde{e} \in \tilde{p}} \mathbf{Z}_{\tilde{e}} - t \sum_{\tilde{v}} \prod_{\tilde{e} \ni \tilde{v}} \mathbf{X}_{\tilde{e}} - U \sum_{\tilde{e}} \mathbf{Z}_{\tilde{e}} - J \sum_{\tilde{e}} \mathbf{X}_{\tilde{e}}. \tag{50}$$

This is essentially a toric code model perturbed by uniform X and Z fields. We return to the toric code shortly.

Note that ${}^2 \widetilde{H}^{(1)}_{\mathbb{Z}_2}$ on the dual lattice has an emergent one-form symmetry $\widehat{\mathbb{Z}}^{(1)}_2$, whose associated symmetry operators are supported on paths through the *original* lattice,

$$\widehat{U}_\gamma = \prod_{\tilde{e} \in \gamma} \mathbf{X}_{\tilde{e}}, \tag{51}$$

where the product is over edges in the dual lattice which are perpendicularly bisected by the path $\gamma$. As is standard, the symmetry operators of the emergent quantum one-form symmetry

associated to contractible paths are generated by the flux operators/emergent Gauss's law operators. The charged operators of a one-form symmetry are line operators; in this case, the basic charged operator is the Wilson-Wegner loop

$$\widetilde{\mathscr{O}}_{\tilde{\gamma}} = \prod_{\tilde{e}\in\tilde{\gamma}} \mathbf{Z}_{\tilde{e}}\,, \tag{52}$$

which we see momentarily is an order parameter for the spontaneous breaking of $\widehat{\mathbb{Z}}_2^{(1)}$. We also note in passing that, in accordance with the discussion in §2.1, this one-form symmetry group admits an isotropic $\mathbb{Z}_2^{(0,1)}(\mathscr{F}_x^{\parallel},\mathscr{F}_y^{\parallel})$ subgroup generated by the $\widehat{U}_\gamma$ associated to the paths $\gamma$ that lie entirely on a single row or column of the original lattice; we make use of this fact below, in §3.2. If one were to instead use the energetic gauging procedure, the term proportional to $U$ in Eq. (50) would break the symmetry from Eq. (51). It is only when one uses the strict gauging procedure, where this term is forbidden because it does not commute with the flux operators $F_p$, that we are guaranteed an emergent symmetry of the gauged theory. This is related to the statement that the one-form symmetry of $\mathbb{Z}_2$ lattice gauge theory is a consequence of Gauss's law (imposed as a constraint).

It is well known that as $h$ is cranked up from zero to infinity, this model undergoes a confinement to deconfinement phase transition at some critical value of $h/J$ (see e.g. Ref. [88–90] for more information) that coincides with the spontaneous breaking of $\widehat{\mathbb{Z}}_2^{(1)}$. Thus, the spontaneous symmetry breaking phase of the TFIM maps after gauging to the phase of $\mathbb{Z}_2$ lattice gauge theory in which the one-form symmetry is preserved, while the symmetry preserving phase of the TFIM maps to the phase where the one-form symmetry is broken. We can obtain order and disorder parameters for this phase diagram by studying how those of the TFIM map under gauging. Starting with the order parameter, one finds after covariantizing and conjugating by $V$ that it gets mapped to a truncated symmetry operator,

$$\mathscr{O}_{v,v'} = \mathbf{Z}_v \mathbf{Z}_{v'} \xrightarrow{\text{cov.}} \mathbf{Z}_v \left(\prod_{e\in\gamma} \mathbf{Z}_e\right) \mathbf{Z}_{v'} \xrightarrow{V} \prod_{\tilde{e}\in\gamma} \mathbf{X}_{\tilde{e}} =: \widetilde{\mathscr{D}}_{\tilde{p},\tilde{p}'}\,, \tag{53}$$

which can hence serve as a disorder parameter for $^2\widetilde{H}_{\mathbb{Z}_2}^{(1)}$. In the above, we have chosen a path $\gamma$ on the original lattice whose end points are $v$ and $v'$; it might seem that this is an arbitrary choice, however matrix elements of $\widetilde{\mathscr{D}}_{\tilde{p},\tilde{p}'}$ taken between gauge invariant states are insensitive to $\tilde{\gamma}$ because of Gauss's law. Note that deep in the deconfined phase, this disorder operator is a creation operator for a pair of quasi-particle excitations (called e.g. visons, magnetic monopoles, $\pi$-fluxes, etc.) that can be thought of as living on the plaquettes $\tilde{p}$, $\tilde{p}'$. Because this operator has a non-decaying expectation value in the confined phase, one says that monopoles are condensed in this phase. Here the $\mathbb{Z}_2$ charges of the Ising model are mapped to gauge fluxes rather than gauge charges as we have changed basis after gauging such that the dual symmetry is a product of Pauli-$\mathbf{X}$ operators.

The disorder operator of the TFIM does not need to be covariantized; conjugating it by $V$ leads to an order parameter for $\mathbb{Z}_2$ LGT,

$$\mathscr{D}_{\tilde{\gamma}} = \prod_{v\in\tilde{\gamma}^\circ} \mathbf{X}_v \xrightarrow{V} \prod_{\tilde{e}\in\tilde{\gamma}} \mathbf{Z}_{\tilde{e}} =: \widetilde{\mathscr{O}}_{\tilde{\gamma}}\,, \tag{54}$$

which is none other than the Wegner–Wilson loop. This loop is charged under the one-form symmetry, so depending on whether it satisfies an area law or a perimeter law, the Wilson loop diagnoses the phase to be either confined or deconfined (symmetry preserving or symmetry breaking) respectively. Deep in the confined phase, this operator can be thought of as a creation operator for electric flux line excitations.

Table 3: A summary of the properties of the (2+1)D transverse field Ising model before and after gauging. After gauging, the model can be identified with $\mathbb{Z}_2$ gauge theory on the dual lattice. The notation $\gamma(\tilde{p}, \tilde{p}')$ indicates that it is a path through the lattice with endpoints $\tilde{p}, \tilde{p}'$, and $\tilde{\gamma}^\circ$ denotes the set of vertices in the interior of the path $\tilde{\gamma}$.

| ${}^2H_{\mathbb{Z}_2}^{(0)}$ | ${}^2H_{\mathbb{Z}_2}^{(0)}\big/\mathbb{Z}_2^{(0)} \cong {}^2\widetilde{H}_{\mathbb{Z}_2}^{(1)}$ |
|---|---|
| $\mathbb{Z}_2^{(0)}$ symmetry: $U = \prod_v \mathbf{X}_v$ | $\widehat{\mathbb{Z}}_2^{(1)}$ symmetry: $\widehat{U}_\gamma = \prod_{\tilde{e} \in \gamma} \mathbf{X}_{\tilde{e}}$ |
| ordered/sym. breaking: $J/h \gg 1$ <br> order parameter: $\langle \mathbf{Z}_v \mathbf{Z}_{v'} \rangle \sim$ const. <br> disorder parameter: $\langle \prod_{v \in \tilde{\gamma}^\circ} \mathbf{X}_v \rangle \sim$ area law <br> excitations: domain-walls $\prod_{v \in \tilde{\gamma}^\circ} \mathbf{X}_v \lvert\Omega\rangle + \cdots$ <br> condensed: spinons | disordered/sym. preserving/confined: $J/h \gg 1$ <br> disorder parameter: $\langle \prod_{\tilde{e} \in \gamma(\tilde{p}, \tilde{p}')} \mathbf{X}_{\tilde{e}} \rangle \sim$ const. <br> order parameter: $\langle \prod_{\tilde{e} \in \tilde{\gamma}} \mathbf{Z}_{\tilde{e}} \rangle \sim$ area law <br> excitations: electric flux line $\prod_{\tilde{e} \in \tilde{\gamma}} \mathbf{Z}_{\tilde{e}} \lvert\Omega\rangle + \cdots$ <br> condensed: magnetic monopoles/visons |
| disordered/sym. preserving: $J/h \ll 1$ <br> disorder parameter: $\langle \prod_{v \in \tilde{\gamma}^\circ} \mathbf{X}_v \rangle \sim$ perim. law <br> order parameter: $\langle \mathbf{Z}_v \mathbf{Z}_{v'} \rangle \sim$ exp. decay <br> excitations: spinons $\mathbf{Z}_v \lvert\Omega\rangle + \cdots$ <br> condensed: domain-walls | ordered/sym. breaking/deconfined: $J/h \ll 1$ <br> order parameter: $\langle \prod_{\tilde{e} \in \tilde{\gamma}} \mathbf{Z}_{\tilde{e}} \rangle \sim$ perim. law <br> disorder parameter: $\langle \prod_{\tilde{e} \in \gamma(\tilde{p}, \tilde{p}')} \mathbf{X}_{\tilde{e}} \rangle \sim$ exp. decay <br> excitations: monopoles $\prod_{\tilde{e} \in \gamma(\tilde{p}, \infty)} \mathbf{X}_{\tilde{e}} \lvert\Omega\rangle + \cdots$ <br> condensed: electric flux lines |

Re-gauging the one-form symmetry $\widehat{\mathbb{Z}}_2^{(1)}$ re-obtains the original TFIM. To see this, this time we place the gauge degrees of freedom on the plaquettes $\tilde{p}$, impose Gauss's law, and covariantize to find

$$
{}^2\widetilde{H}_{\mathbb{Z}_2}^{(1)}\big/\widehat{\mathbb{Z}}_2^{(1)} = -J \sum_{\tilde{e}} \mathbf{X}_{\tilde{e}} - h \sum_{\tilde{p}} \mathbf{Z}_{\tilde{p}} \prod_{\tilde{e} \in \tilde{p}} \mathbf{Z}_{\tilde{e}}
$$
$$
\widetilde{G}_{\tilde{v}} = \prod_{\tilde{e} \ni \tilde{v}} \mathbf{X}_{\tilde{e}}, \qquad \widetilde{G}'_{\langle \tilde{p}, \tilde{p}' \rangle} = \mathbf{X}_{\tilde{p}} \mathbf{X}_{\langle \tilde{p}, \tilde{p}' \rangle} \mathbf{X}_{\tilde{p}'}.
$$

(55)

In the above, $\widetilde{G}_{\tilde{v}}$ is the Gauss's law constraint of ${}^2\widetilde{H}_{\mathbb{Z}_2}^{(1)}$ (which was present even before gauging) while $\widetilde{G}'_{\langle \tilde{p}, \tilde{p}' \rangle}$ is the Gauss's law constraint that one imposes to gauge the one-form symmetry. We can solve this latter constraint easily after conjugating by the circuit

$$
\widetilde{V} = H^{\otimes N} \prod_{\tilde{p}} \prod_{\tilde{e} \in \tilde{p}} C_{\tilde{e}} \mathbf{X}_{\tilde{p}},
$$

(56)

where $N$ is the total number of qubits, which yields

$$
\widetilde{V} \left( \widetilde{H}_{\mathbb{Z}_2}^{(1)} \big/ \widehat{\mathbb{Z}}_2^{(1)} \right) \widetilde{V}^\dagger = -J \sum_{\langle \tilde{p}, \tilde{p}' \rangle} \mathbf{Z}_{\tilde{p}} \mathbf{Z}_{\langle \tilde{p}, \tilde{p}' \rangle} \mathbf{Z}_{\tilde{p}'} - h \sum_{\tilde{p}} \mathbf{X}_{\tilde{p}}
$$
$$
\widetilde{V} \widetilde{G}_{\tilde{v}} \widetilde{V}^\dagger = \prod_{\tilde{e} \ni \tilde{v}} \mathbf{Z}_{\tilde{e}}, \quad \widetilde{V} \widetilde{G}'_{\tilde{e}} \widetilde{V}^\dagger = \mathbf{Z}_{\tilde{e}}.
$$

(57)

Imposing $\widetilde{V} \widetilde{G}'_{\tilde{e}} \widetilde{V}^\dagger = 1$ then freezes $\mathbf{X}_{\tilde{e}} = 1$, so that $\widetilde{V} \widetilde{G}_{\tilde{v}} \widetilde{V}^\dagger = 1$ is automatically satisfied as well. Switching perspective back to the original lattice then simply recovers the Hamiltonian of the TFIM. The entire discussion thus far is summarized in Table 3.

Before moving on, we briefly comment on how the relationship between the TFIM and LGT is modified when defects are introduced. First, imagine flipping the nearest neighbor couplings in the TFIM from ferromagnetic to antiferromagnetic on some network of bonds. This can be described by taking the Hamiltonian to be

$$
{}^2H_{\mathbb{Z}_2}^{(0)}(Q) = -J \sum_{\langle v, v' \rangle} \mathbf{Z}_v Q_{\langle v, v' \rangle} \mathbf{Z}_{v'} - h \sum_v \mathbf{X}_v,
$$

(58)

where $Q = \{Q_e = \pm 1\}$ is a defect network which determines whether each coupling is ferromagnetic or anti-ferromagnetic; when all the $Q_e = 1$, this simply describes the TFIM without any defects present. If one traces through the steps we have taken in this section to gauge the global $\mathbb{Z}_2^{(0)}$ symmetry, one finds that one ends up with

$$
{}^2H_{\mathbb{Z}_2}^{(0)}(Q)\big/\mathbb{Z}_2^{(0)} \simeq -J\sum_{\tilde{e}}\mathbf{X}_{\tilde{e}} - h\sum_{\tilde{p}}\prod_{\tilde{e}\in\tilde{p}}\mathbf{Z}_{\tilde{e}}
$$
$$
\widetilde{G}_{\tilde{v}}(Q) = \prod_{\tilde{e}\ni\tilde{v}}Q_{\tilde{e}}\mathbf{X}_{\tilde{e}}\,. \tag{59}
$$

We see that the defects do not appear explicitly in the Hamiltonian itself, but rather only in the definition of Gauss's law. One of course typically restricts to the gauge invariant subspace of the extended Hilbert space corresponding to $\widetilde{G}_{\tilde{v}}(Q) = 1$; in the present case, this is equivalent to eliminating the defects entirely, and working in the subspace of ordinary $\mathbb{Z}_2$ lattice gauge theory defined by $\widetilde{G}_{\tilde{v}} = \pm 1$, where $\widetilde{G}_{\tilde{v}} = -1$ if $\tilde{v}$ is the *endpoint* of one of the defect lines in $Q$ (now thought of as a network of lines on the dual lattice), and $+1$ otherwise. Thus, after gauging, a network of antiferromagnetic defect lines in the (2+1)D TFIM becomes equivalent to an insertion of probe electric particles (i.e. local violations of Gauss's law) in $\mathbb{Z}_2$ lattice gauge theory.

**Relationship to toric code and boundaries**
Let us now revert from the dual lattice back to the regular lattice. When the on-site Pauli-**X** term is turned off and Gauss's law is imposed energetically in $\mathbb{Z}_2$ lattice gauge theory, the resulting model is called the (2+1)D toric code,

$$
H_{\text{TC}} = -\sum_{v}A_v - \sum_{p}B_p\,,
$$
$$
A_v = \prod_{e\ni v}\mathbf{X}_e\,, \qquad B_p = \prod_{e\in p}\mathbf{Z}_e\,. \tag{60}
$$

In addition to the one-form symmetry from before, Eq. (51), the toric code has an additional $\mathbb{Z}_2^{(1)}$ one-form symmetry. These two one-form symmetries are typically referred to as "electric" and "magnetic". In the context of the toric code, the discussion is typically phrased in terms of "string operators"

$$
S_e(\gamma) = \prod_{e\in\gamma}\mathbf{Z}_e\,, \qquad S_m(\tilde{\gamma}) = \prod_{e\in\tilde{\gamma}}\mathbf{X}_e\,, \tag{61}
$$

where again, $\gamma, \tilde{\gamma}$ are paths through the lattice and dual lattice respectively. These string operators perform double duty. For example, $S_e(\gamma)$ is the symmetry operator for one of the $\mathbb{Z}_2^{(1)}$ one-form symmetries, while being the charged line operator with respect to the other, and similarly for $S_m(\tilde{\gamma})$. In particular, these operators are topological in the sense that they only depend on the homotopy class of the defining path (at least after one is careful to introduce punctures in the underlying manifold where there exist quasiparticles). Now, there are also two types of quasiparticles: electric $e$ particles and magnetic $m$ particles, which are violations of the vertex terms and the plaquette terms, respectively. They can be created at the end points of string operators (see Figure 2).

We now move on to a discussion of the different kinds of boundaries of the toric code. See e.g. Refs. [91–94] for related discussions.

**Rough boundary**  Let us consider placing the toric code on a semi-infinite cylinder $M = (-\infty, 0] \times S^1$, so that the manifold on which it is defined has a circle boundary, $\partial M = S^1$. We label the number of lattice sites along the circle direction as $L$.

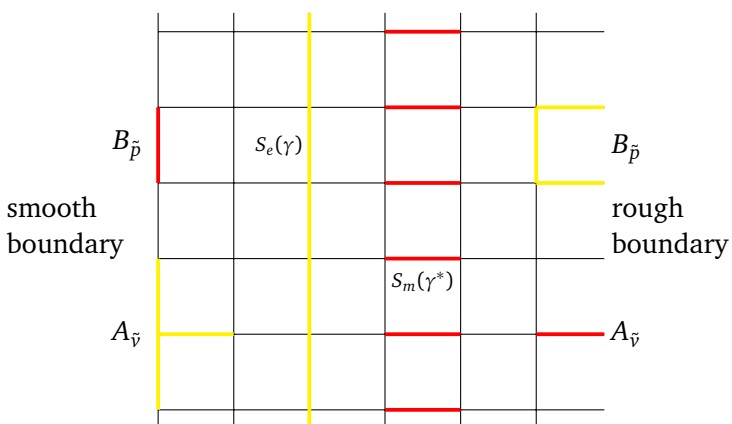

Figure 2: The rough and smooth boundaries of the toric code, along with their associated boundary vertex and plaquette operators, $A_{v_\partial}$ and $B_{p_\partial}$. An edge colored yellow indicates a Pauli-**Z** operator acting on that edge, while an edge colored red indicates a Pauli-**X**.

There are two common choices for how to give the underlying lattice a boundary, which are referred to as "rough" and "smooth", depicted in Figure 2. Let us first choose the rough boundary for concreteness, and return to the smooth boundary later. We assume that the Hamiltonian in Eq. (60) contains only operators associated to *bulk* vertices and plaquettes, which we define to be vertices and plaquettes which intersect exactly four edges. We must decide what form the Hamiltonian takes near the boundary. Let us use the notation $v_\partial$, $p_\partial$ for boundary vertices and boundary plaquettes. It is common to include either boundary vertex terms $A_{v_\partial}$ (i.e. the single body X term on the only edge emanating from the boundary vertex $v_\partial$) or boundary plaquette terms $B_{p_\partial}$ (i.e. the three body Z term on the three edges which border the boundary plaquette $p_\partial$) because they both commute with all the terms in $H_{\text{TC}}$, but not both if one would like to preserve the commuting-projector property of the toric code. For now, we include neither, and instead work with "free" boundary conditions, taking Eq. (60) as the complete specification of the Hamiltonian, with the sums over $v$ and $p$ understood to be sums over only bulk vertices and bulk plaquettes. We add in boundary vertex/plaquette terms in a moment.

Let us now determine the ground space. First, we define $W$ and $\widetilde{W}$ to be the electric and magnetic string operators from Eq. (61) respectively, with the paths taken to be ones which wrap the circle. These two operators commute with one another, and of course with the toric code Hamiltonian, and so we can use their eigenvalues ($Q$ and $\widetilde{Q}$ respectively) to label different topological sectors of the ground space,

$$\mathcal{H}_{\text{gs}} = \bigoplus_{Q=\pm 1} \bigoplus_{\widetilde{Q}=\pm 1} \mathcal{H}_{\text{gs}}^{(Q,\widetilde{Q})}. \tag{62}$$

Now, because the boundary vertex and plaquette operators $A_{v_\partial}$ and $B_{p_\partial}$ also commute with $H_{\text{TC}}$, $W$, and $\widetilde{W}$, they map each topological sector $\mathcal{H}_{\text{gs}}^{(Q,\widetilde{Q})}$ to itself. In fact, it is straightforward to convince oneself that a state in $\mathcal{H}_{\text{gs}}$ is determined completely by its eigenvalues under the following mutually commuting operators:

$$\left\{ W, \widetilde{W}, A_{v_\partial} \right\}. \tag{63}$$

Thus, we can think of the degrees of freedom of $\mathcal{H}_{\text{gs}}^{(Q,\widetilde{Q})}$ as those of a (1+1)D spin chain whose $L$ effective qubits live on the boundary vertices. The algebra furnished by $\{A_{v_\partial}, B_{p_\partial}\}$ within

$\mathcal{H}_{\text{gs}}^{(Q,\widetilde{Q})}$ is the same as that of the Paulis $\{\hat{\mathbf{X}}_{v_\partial}, \hat{\mathbf{Z}}_{v_\partial}\hat{\mathbf{Z}}_{v_\partial+1}\}$ acting on the effective boundary spin chain in the $\prod_{v_\partial} \hat{\mathbf{X}}_{v_\partial} = \widetilde{Q}$ subspace (i.e. the $\mathbb{Z}_2$ even/odd subspace), subject to the boundary condition $\hat{\mathbf{Z}}_0 = Q\hat{\mathbf{Z}}_L$ (i.e. periodic or antiperiodic boundary conditions).

Because these edge states are exactly degenerate, we can take the effective edge Hamiltonian which governs them to be identically zero, $H_{\text{edge}} = 0$. In order to obtain more interesting boundary dynamics, we may consider perturbing the toric code as follows:

$$H_{\text{TC}} \to H_{\text{TC}} + H_\partial \,, \qquad H_\partial = -h_X \sum_{\substack{\text{boundary} \\ v_\partial}} A_{v_\partial} - h_Z \sum_{\substack{\text{boundary} \\ p_\partial}} B_{p_\partial} \,. \tag{64}$$

We take $h_X$ and $h_Z$ to be sufficiently small so that the effect of these fields is to split the energy levels within $\mathcal{H}_{\text{gs}}^{(Q,\widetilde{Q})}$, but otherwise keep them well separated from excited states. We may then encode this splitting in an effective edge Hamiltonian $H_{\text{edge}}$ which acts on $\mathcal{H}_{\text{gs}}^{(Q,\widetilde{Q})}$. Because the boundary terms commute with the bulk terms, the edge Hamiltonian can be represented as

$$H_{\text{edge}} = -\sum_{v_\partial} \left( h_X \hat{\mathbf{X}}_{v_\partial} + h_Z \hat{\mathbf{Z}}_{v_\partial}\hat{\mathbf{Z}}_{v_\partial+1} \right) \,, \tag{65}$$

where the sum is over boundary vertices. In other words, the (1+1)D transverse field Ising model controls the edge modes of the perturbed toric code model, Eq. (64). Again, $H_{\text{edge}}$ acts in the $\mathbb{Z}_2$ even/odd subspace depending on the value of $\widetilde{Q}$, and is taken to have periodic/antiperiodic boundary conditions depending on the value of $Q$.

The fact that we obtained an edge Hamiltonian with a $\mathbb{Z}_2^{(0)}$ global symmetry can be understood as a consequence of the $\mathbb{Z}_2^{(1)}$ one-form symmetry of the bulk. Indeed, the $\mathbb{Z}_2^{(0)}$ global symmetry is generated by the unitary $\prod_{v_\partial} \hat{\mathbf{X}}_{v_\partial} = \prod_{e_\partial} \mathbf{X}_{e_\partial}$, which is none other than a one-form symmetry generator of the bulk toric code brought to the boundary of the half space (which we have called $\widetilde{W}$). At a technical level, the $\mathbb{Z}_2^{(0)}$ symmetry is imposed by the fact that it is impossible to construct a local operator in the toric code that acts as $\hat{\mathbf{Z}}_{v_\partial}$ on the effective edge spin chain; therefore, if we take $H_\partial$ to be general and compute an effective edge Hamiltonian in perturbation theory, the perturbation series only produces terms which are products of $\hat{\mathbf{X}}_{v_\partial}$ and $\hat{\mathbf{Z}}_{v_\partial}\hat{\mathbf{Z}}_{v_\partial+1}$. That is, for generic local perturbations, we expect

$$H_{\text{edge}} = H_{\text{edge}} \left( \{\hat{\mathbf{X}}_{v_\partial}\}, \{\hat{\mathbf{Z}}_{v_\partial}\hat{\mathbf{Z}}_{v_\partial+1}\} \right) \,, \tag{66}$$

to be a function just of single body Pauli-$\mathbf{X}$ terms and two-body Pauli-$\mathbf{Z}$ terms, which are always $\mathbb{Z}_2$-symmetric.

We may also identify boundary excitations of $H_{\text{edge}}$ with bulk excitations of the toric code. In the unperturbed toric code, the string operators in Eq. (61) produce the toric code $e$ and $m$ particles at their endpoints. When perturbations are introduced, the operators that create these excitations generically require modification. However, if we turn on $h_X \neq 0$ while keeping $h_Z = 0$, for example, then the string operators $S_e(\gamma)$ that create electric particles are not corrected. From the perspective of the edge degrees of freedom, the effective Hamiltonian simplifies to $H_{\text{edge}} \sim -h_X \sum_{v_\partial} \hat{\mathbf{X}}_{v_\partial}$. If we push the string operator which creates electric particles to the boundary, we see that it can be expressed simply in terms of operators acting on the virtual spin chain as

$$S_e(\gamma) \sim \hat{\mathbf{Z}}_{v_\partial} \hat{\mathbf{Z}}_{v_\partial'} \,, \tag{67}$$

where $v_\partial, v_\partial'$ are the endpoints of $\gamma$. Since this operator creates two spin excitations at $v_\partial$ and $v_\partial'$ in the boundary transverse field Ising model, we see that spin excitations in the edge Hamiltonian are simply bulk $e$ particles which have been pushed to the boundary. We expect the takeaway of this discussion to continue to hold once one dials $h_Z$ away from zero.

Similarly, if one takes $h_Z \neq 0$, $h_X = 0$, then the string operators which create magnetic particles are the same as the operators $S_m(\tilde{\gamma})$ which create them in the unperturbed toric code, and the edge Hamiltonian simplifies to $H_{\text{edge}} \sim -h_Z \sum_{v_\partial} \hat{\mathbf{Z}}_{v_\partial} \hat{\mathbf{Z}}_{v_\partial+1}$. If one pushes the magnetic string operators to the boundary, they take the form

$$S_m(\tilde{\gamma}) \sim \prod_{v'_\partial \leq v_\partial \leq v''_\partial} \hat{\mathbf{X}}_{v_\partial} \, . \tag{68}$$

The operator on the right hand side creates domain walls in the virtual spin chain, and so we are lead to identify domain wall excitations in the edge model with bulk magnetic particles which have been pushed to the boundary. Again, nothing of importance changes if one deforms away from $h_X = 0$.

**Smooth boundary** Before moving on, let us comment briefly on what happens in the case of smooth boundaries. Here again, we only include bulk vertex/plaquette terms in the Hamiltonian, and in particular, exclude three-body $A_{v_\partial}$ and single-body $B_{p_\partial}$ terms which can be associated to boundary vertices/plaquettes.

We start by noting that the toric code with these boundary conditions is dual (under electric-magnetic duality) to the toric code with rough boundary conditions discussed in the previous subsection. To see this, first perform a rotation $\mathbf{X}_e \to \mathbf{Z}_e$, $\mathbf{Z}_e \to -\mathbf{X}_e$ on all edges, and then interpret the qubits as living on the edges of the dual lattice. The combination of these two actions maps vertex terms to plaquette terms (hence the name electric-magnetic duality) and swaps the rough and smooth boundary conditions. Therefore, our expectation is that the physics of the edge is equivalent; actually, it turns out that the edge Hamiltonian in the case of smooth boundary conditions is Kramers-Wannier dual to the edge Hamiltonian with with rough boundary conditions. Because the logic of the argument that shows this is identical to the one employed in the case of rough boundary conditions, we do not go through it explicitly.

### 2.2.3 (3+1)D transverse field Ising model and higher-form $\mathbb{Z}_2$ gauge theories

We conclude by summarizing some basic facts about the theories ${}^3H_{\mathbb{Z}_2}^{(q)}$ for $q = 0, 1, 2$. Because there is nothing fundamentally new in $d = 3$, we are brief.

The model with $q = 0$ is again the transverse field Ising model

$$ {}^3H_{\mathbb{Z}_2}^{(0)} = -J \sum_{\langle v, v' \rangle} \mathbf{Z}_v \mathbf{Z}_{v'} - h \sum_v \mathbf{X}_v \, , \tag{69}$$

with qubits on the sites $v$ of the 3d cubic lattice, and with its global symmetry implemented by the operator

$$U = \prod_v \mathbf{X}_v \, . \tag{70}$$

The order and disorder parameters that diagnose whether this symmetry is spontaneously broken or not are

$$\begin{aligned} \mathcal{O} &= \lim_{|v-v'| \to \infty} \langle \mathcal{O}_{v,v'} \rangle \, , \quad \mathcal{O}_{v,v'} = \mathbf{Z}_v \mathbf{Z}_{v'} \\ \mathcal{D} &= \lim_{\text{Vol}(\tilde{m}) \to \infty} \langle \mathcal{D}_{\tilde{m}} \rangle, \quad \mathcal{D}_{\tilde{m}} = \prod_{v \in \tilde{m}^\circ} \mathbf{X}_v \, , \end{aligned} \tag{71}$$

where $\tilde{m}$ is a connected domain wall, or a closed membrane on the dual lattice, and $\tilde{m}^\circ$ is the set of vertices in the interior of the domain wall.

If one gauges this global symmetry, one obtains

$$^3H_{\mathbb{Z}_2}^{(0)}\Big/\mathbb{Z}_2^{(0)} = -J\sum_{\langle v,v'\rangle}\mathbf{Z}_v\mathbf{Z}_{\langle v,v'\rangle}\mathbf{Z}_{v'} - h\sum_v \mathbf{X}_v\,,$$

$$F_p = \prod_{e\in p}\mathbf{Z}_e\,,\qquad G_v = \mathbf{X}_v\prod_{e\ni v}\mathbf{X}_e\,. \tag{72}$$

One can again follow the same kind of protocol as in the previous subsections to simplify this theory. After applying the same unitary circuit as in Eq. (46) (extended in the obvious way to three dimensions), imposing $VG_vV^\dagger = 1$, and switching to the dual lattice, the model one obtains is

$$V\left(^3H_{\mathbb{Z}_2}^{(0)}\Big/\mathbb{Z}_2^{(0)}\right)V^\dagger \xrightarrow{VG_vV^\dagger=1} -J\sum_{\tilde{p}}\mathbf{X}_{\tilde{p}} - h\sum_{\tilde{c}}\prod_{\tilde{p}\in\tilde{c}}\mathbf{Z}_{\tilde{p}} = {}^3\widetilde{H}_{\mathbb{Z}_2}^{(2)}$$

$$VF_{\tilde{e}}V^\dagger = \prod_{\tilde{p}\ni\tilde{e}}\mathbf{X}_{\tilde{p}} = \widetilde{G}_{\tilde{e}}\,, \tag{73}$$

where $\tilde{p}$, $\tilde{c}$, and $\tilde{e}$ are plaquettes, cubes, and edges of the dual lattice, respectively. That is, we recover two-form lattice gauge theory, where the flux term $VF_{\tilde{e}}V^\dagger$ now plays the role of the Gauss's law term of $^3\widetilde{H}_{\mathbb{Z}_2}^{(2)}$. When $J = 0$ and the Gauss's law term is energetically imposed, the model is also known as the (3+1)D toric code.[9]

This theory has an emergent two-form symmetry group $\widehat{\mathbb{Z}}_2^{(2)}$. The symmetry operators are implemented by string operators associated to closed paths $\gamma$ through the *original* lattice

$$\widehat{U}_\gamma = \prod_{\tilde{p}\in\gamma}\mathbf{X}_{\tilde{p}}\,, \tag{74}$$

where the product is over the plaquettes $\tilde{p}$ of the dual lattice which are pierced by $\gamma$. As an aside, we note that a two-form symmetry group in (3+1)D admits at least two subsystem subgroups. Namely, there is a $\mathbb{Z}_2^{(1,1)}(\mathscr{F}_x^\perp,\mathscr{F}_y^\perp,\mathscr{F}_z^\perp)$ and a $\mathbb{Z}_2^{(0,2)}(\mathscr{F}_x^\parallel,\mathscr{F}_y^\parallel,\mathscr{F}_z^\parallel)$ subgroup.

As one varies $J/h$, there is a phase transition at some critical value of $J/h$ that corresponds to the spontaneous breaking of this two-form symmetry. When $J \gg h$, the theory preserves the two-form symmetry, and when $h \gg J$, the theory spontaneously breaks the $\widehat{\mathbb{Z}}_2^{(2)}$. Thus, we confirm that the spontaneous breaking of the original $\mathbb{Z}_2^{(0)}$ of the TFIM corresponds after gauging to the symmetry preserving phase of the two-form gauge theory and vice versa. The order and disorder parameters of $^3\widetilde{H}_{\mathbb{Z}_2}^{(2)}$ can be found by studying how those of the (3+1)D TFIM are mapped under gauging, as we have done in (1+1)D and (2+1)D. If one does this, one finds

$$\widetilde{\mathscr{O}} = \lim_{\mathrm{Vol}(\tilde{m})\to\infty}\langle\widetilde{\mathscr{O}}_{\tilde{m}}\rangle\,,\qquad \widetilde{\mathscr{O}}_{\tilde{m}} = \prod_{\tilde{p}\in\tilde{m}}\mathbf{Z}_{\tilde{p}}\,,$$

$$\widetilde{\mathscr{D}} = \lim_{|\tilde{c}-\tilde{c}'|\to\infty}\langle\widetilde{\mathscr{D}}_{\tilde{c},\tilde{c}'}\rangle\,,\qquad \widetilde{\mathscr{D}}_{\tilde{c},\tilde{c}'} = \prod_{\tilde{p}\in\gamma}\mathbf{X}_{\tilde{p}}\,. \tag{75}$$

In the above, the order parameter is a Wilson–Wegner membrane supported on $\tilde{m}$, that is charged under the two-form symmetry. For the disorder parameter, $\gamma$ is an arbitrary path of the original lattice whose endpoints are the cubes $\tilde{c}$, $\tilde{c}'$ of the dual lattice (again, $\widetilde{\mathscr{D}}$ doesn't depend on this choice of path), and the product in the definition of $\widetilde{\mathscr{D}}_{\tilde{c},\tilde{c}'}$ is over plaquettes of the dual lattice that are pierced by $\gamma$.

---

[9]We thank the authors of Ref. [95] for sharing their work which investigates natural phase transitions out of the (3+1)D toric code phase.

Table 4: A summary of the properties of the (3+1)D transverse field Ising model before and after gauging. After gauging, the model can be identified with $\mathbb{Z}_2$ two-form gauge theory on the dual lattice. The notation $\gamma(\tilde{c}, \tilde{c}')$ denotes a path through the lattice with endpoints $\tilde{c}, \tilde{c}'$.

| $^3H^{(0)}_{\mathbb{Z}_2}$ | $^3H^{(0)}_{\mathbb{Z}_2}\big/\mathbb{Z}^{(0)}_2 \cong {}^3\widetilde{H}^{(2)}_{\mathbb{Z}_2}$ |
|---|---|
| $\mathbb{Z}^{(0)}_2$ symmetry: $U = \prod_v \mathbf{X}_v$ | $\widehat{\mathbb{Z}}^{(2)}_2$ symmetry: $\widehat{U}_\gamma = \prod_{\tilde{p} \in \gamma} \mathbf{X}_{\tilde{p}}$ |
| ordered/sym. breaking: $J/h \gg 1$ <br> order parameter: $\langle \mathbf{Z}_v \mathbf{Z}_{v'} \rangle \sim$ const. <br> disorder parameter: $\langle \prod_{v \in \tilde{m}^\circ} \mathbf{X}_v \rangle \sim$ volume law <br> excitations: domain-walls $\prod_{v \in \tilde{m}^\circ} \mathbf{X}_v |\Omega\rangle + \cdots$ <br> condensed: spinons | disordered/sym. preserving/confined: $J/h \gg 1$ <br> disorder parameter: $\langle \prod_{\tilde{p} \in \gamma(\tilde{c},\tilde{c}')} \mathbf{X}_{\tilde{p}} \rangle \sim$ const. <br> order parameter: $\langle \prod_{\tilde{p} \in \tilde{m}} \mathbf{Z}_{\tilde{p}} \rangle \sim$ volume law <br> excitations: membrane-nets $\prod_{\tilde{p} \in \tilde{m}} \mathbf{Z}_{\tilde{p}} |\Omega\rangle + \cdots$ <br> condensed: magnetic monopoles/visons |
| disordered/sym. preserving: $J/h \ll 1$ <br> disorder parameter: $\langle \prod_{v \in \tilde{m}} \mathbf{X}_v \rangle \sim$ area law <br> order parameter: $\langle \mathbf{Z}_v \mathbf{Z}_{v'} \rangle \sim$ exp. decay <br> excitations: spinons $\mathbf{Z}_v |\Omega\rangle + \cdots$ <br> condensed: domain-walls | ordered/sym. breaking/deconfined: $J/h \ll 1$ <br> order parameter: $\langle \prod_{\tilde{p} \in \tilde{m}} \mathbf{Z}_{\tilde{p}} \rangle \sim$ area law <br> disorder parameter: $\langle \prod_{\tilde{p} \in \gamma(\tilde{c},\tilde{c}')} \mathbf{X}_{\tilde{p}} \rangle \sim$ exp. decay <br> excitations: monopoles $\prod_{\tilde{p} \in \gamma(\tilde{c},\infty)} \mathbf{X}_{\tilde{p}} |\Omega\rangle + \cdots$ <br> condensed: membrane-nets/electric flux membranes |

Regauging this emergent symmetry leads to the original TFIM. We can see this by placing qubits on the cubes of the dual lattice, which leads to

$$
{}^3\widetilde{H}^{(2)}_{\mathbb{Z}_2}\big/\widehat{\mathbb{Z}}^{(2)}_2 = -J \sum_{\tilde{p}} \mathbf{X}_{\tilde{p}} - h \sum_{\tilde{c}} \mathbf{Z}_{\tilde{c}} \prod_{\tilde{p} \in \tilde{c}} \mathbf{Z}_{\tilde{p}} ,
$$
$$
\widetilde{G}_{\tilde{e}} = \prod_{\tilde{p} \ni \tilde{e}} \mathbf{X}_{\tilde{p}}, \qquad \widetilde{G}'_{\langle \tilde{c}, \tilde{c}' \rangle} = \mathbf{X}_{\tilde{c}} \mathbf{X}_{\langle \tilde{c}, \tilde{c}' \rangle} \mathbf{X}_{\tilde{c}'} ,
$$
(76)

where in the above, $\langle \tilde{c}, \tilde{c}' \rangle$ specifies a plaquette $\tilde{p}$ of the dual lattice through its two neighboring cubes. Defining

$$
V = H^{\otimes N} \prod_{\tilde{c}} \prod_{\tilde{p} \in \tilde{c}} C_{\tilde{p}} \mathbf{X}_{\tilde{c}} ,
$$
(77)

we find, after switching back to the original lattice, that

$$
V \left( {}^3\widetilde{H}^{(2)}_{\mathbb{Z}_2}\big/\widehat{\mathbb{Z}}^{(2)}_2 \right) V^\dagger \xrightarrow{V\widetilde{G}'_{\langle \tilde{c}, \tilde{c}' \rangle} V^\dagger = 1} {}^3H^{(0)}_{\mathbb{Z}_2} .
$$
(78)

The salient elements of this discussion are summarized in Table 4.

Similar manipulations apply to ordinary $\mathbb{Z}_2$ lattice gauge theory in three dimensions,

$$
{}^3H^{(1)}_{\mathbb{Z}_2} = -U \sum_e \mathbf{X}_e - t \sum_p \prod_{e \in p} \mathbf{Z}_e ,
$$
$$
G_v = \prod_{e \ni v} \mathbf{X}_e .
$$
(79)

The theory has a $\mathbb{Z}^{(1)}_2$ one-form symmetry, whose symmetry operators are supported on surfaces. More specifically, let $\tilde{m}$ be a closed membrane of the dual lattice, composed of a series of plaquettes of the dual lattice. Then the symmetry operators are

$$
U_{\tilde{m}} = \prod_{e \in \tilde{m}} \mathbf{X}_e ,
$$
(80)

Table 5: A summary of the properties of (3+1)D $\mathbb{Z}_2$ one-form lattice gauge theory, before and after gauging. The objects $m(\gamma)$ and $\tilde{m}(\tilde{\gamma})$ are open membranes on the lattice and dual lattice with boundaries $\gamma$ and $\tilde{\gamma}$, respectively.

| $^3H_{\mathbb{Z}_2}^{(1)}$ | $^3H_{\mathbb{Z}_2}^{(1)}\big/\mathbb{Z}_2^{(1)} \cong {}^3\widetilde{H}_{\mathbb{Z}_2}^{(1)}$ |
|---|---|
| $\mathbb{Z}_2^{(1)}$ symmetry: $U_{\tilde{m}} = \prod_{e\in\tilde{m}} \mathbf{X}_e$ | $\widehat{\mathbb{Z}}_2^{(1)}$ symmetry: $\widehat{U}_m = \prod_{\tilde{e}\in m} \mathbf{X}_{\tilde{e}}$ |
| ordered/sym. breaking: $t/U \gg 1$<br>order param: $\langle\prod_{e\in\gamma} \mathbf{Z}_e\rangle \sim$ perim. law<br>disorder param: $\langle\prod_{e\in\tilde{m}(\tilde{\gamma})} \mathbf{X}_e\rangle \sim$ area law<br>excitations: magnetic strings $\prod_{e\in\tilde{m}(\tilde{\gamma})} \mathbf{X}_e\lvert\Omega\rangle + \cdots$<br>condensed: electric flux lines | disordered/sym. preserving: $t/U \gg 1$<br>disorder param: $\langle\prod_{\tilde{e}\in m(\gamma)} \mathbf{X}_{\tilde{e}}\rangle \sim$ perim. law<br>order param: $\langle\prod_{\tilde{e}\in\tilde{\gamma}} \mathbf{Z}_{\tilde{e}}\rangle \sim$ area law<br>excitations: electric flux lines $\prod_{\tilde{e}\in\tilde{\gamma}} \mathbf{Z}_{\tilde{e}}\lvert\Omega\rangle + \cdots$<br>condensed: magnetic strings |
| disordered/sym. preserving: $t/U \ll 1$<br>disorder param: $\langle\prod_{e\in\tilde{m}(\tilde{\gamma})} \mathbf{X}_e\rangle \sim$ perim. law<br>$\vdots$ | ordered/sym. breaking: $t/U \ll 1$<br>order parameter: $\langle\prod_{\tilde{e}\in\tilde{\gamma}} \mathbf{Z}_{\tilde{e}}\rangle \sim$ perim. law<br>$\vdots$ |

where the product is over the dual plaquettes that make up the membrane, thought of as edges of the original lattice. This symmetry group admits a $\mathbb{Z}_2^{(0,1)}(\mathscr{F}_{\mathrm{x}}^{\perp}, \mathscr{F}_{\mathrm{y}}^{\perp}, \mathscr{F}_{\mathrm{z}}^{\perp})$ subgroup.

There is a confinement/deconfinement phase transition which occurs at the critical value of $U/t = 1$ because $\mathbb{Z}_2$ lattice gauge theory is self-dual in (3+1)D. The charged operators of a one-form symmetry in (3+1)D are strings, and so we expect the order parameter to be a line operator. In fact, it is the same as in (2+1)D, namely the Wegner–Wilson loop,

$$\mathscr{O} = \lim_{A(\gamma)\to\infty} \langle\mathscr{O}_\gamma\rangle, \qquad \mathscr{O}_\gamma = \prod_{e\in\gamma} \mathbf{Z}_e, \tag{81}$$

where $\gamma$ is a closed loop of the lattice. The disorder parameter can again be taken to be a truncated symmetry operator. That is, we define

$$\mathscr{D} = \lim_{A(\tilde{\gamma})\to\infty} \langle\mathscr{D}_{\tilde{\gamma}}\rangle, \quad \mathscr{D}_{\tilde{\gamma}} = \prod_{e\in\tilde{m}} \mathbf{X}_e, \tag{82}$$

where $\tilde{m}$ is again a membrane of the dual lattice (this time open) whose boundary is $\tilde{\gamma}$, a path of the dual lattice. In (3+1)D, $\mathbb{Z}_2$ lattice gauge theory maps to itself under gauging of its one-form symmetry, except that its confined phase is mapped to its deconfined phase, its order parameter is mapped to its disorder parameter, and so on. See Table 5 for a summary of this discussion.

### 2.2.4  Prototype models $^dH_{\mathbb{Z}_2}^{(q,k)}$ with $\mathbb{Z}_2^{(q,k)}\big/\sim$ symmetry in all dimensions

Many of the features we have highlighted thus far in the context of models with ordinary $q$-form symmetries can be generalized to the setting of $q$-form subsystem symmetries. We anticipate some of these generalizations here, filling in further details as needed throughout the remainder of the text.

There is a standard choice of Ising-like Hamiltonians $^dH_{\mathbb{Z}_2}^{(q,k)}$ in $(d+1)$ spacetime dimensions (for $d > k+q$) that respect isotropic $\mathbb{Z}_2^{(q,k)}\big/\sim$ symmetry groups. To define them, let us consider a $d$-dimensional cubic lattice and consider it as a simplicial complex $\Lambda$, using the notation $\Lambda_p$ for the fundamental $p$-simplices. Thus, $\Lambda_0$ is the set of lattice sites, $\Lambda_1$ is the set of edges, $\Lambda_2$ is the set of square plaquettes, $\Lambda_3$ is the set of cubes, and so on. We also use the notation $\mu_1 \ldots \mu_n$ to denote a collection of directions in the lattice. For example, if $d = 3$ and $n = 2$, then the possibilities are $\mu_1\mu_2 = \mathrm{xy}, \mathrm{yz}, \mathrm{xz}$.

With these conventions in place, we place the qubits on the fundamental $q$-simplices, and define

$$^d H_{\mathbb{Z}_2}^{(q,k)} = -U \sum_{\sigma \in \Lambda_q} \mathbf{X}_\sigma - t \sum_{\omega \in \Lambda_{q+k+1}} \prod_{\sigma \in \omega} \mathbf{Z}_\sigma \,, \tag{83}$$

where $\mathbf{X}_\sigma$, $\mathbf{Z}_\sigma$ are the Pauli operators that act on the qubit associated to the $q$-simplex $\sigma$. In the case that $q > 0$, the theory is a generalized gauge theory, and we accordingly supplement the Hamiltonian with a Gauss's law constraint,

$$G_\rho^{\mu_1 \ldots \mu_{d-k}} = \prod_{\substack{\sigma \ni \rho \\ \sigma \| \mu_1 \ldots \mu_{d-k}}} \mathbf{X}_\sigma = 1 \,, \tag{84}$$

which we require to hold for every $\rho \in \Lambda_{q-1}$, and for every collection $\mu_1 \ldots \mu_{d-k}$ of $d - k$ directions in the lattice.

If one unpacks these definitions for small values of $q$ and $k$, one finds a venerable list of familiar models. For example, when $q = k = 0$, this recovers the transverse field Ising model; when $q = 1$, $k = 0$, one finds ordinary $\mathbb{Z}_2$ lattice gauge theory; when $q > 1$, $k = 0$, the Hamiltonian is that of $q$-form gauge theory; when $q = 0$, $k = 1$, one obtains the plaquette Ising models; when $q = 0$, $k = 2$, this leads to cubic Ising models; and finally when $q = k = 1$, this recovers the X–cube models. This is summarized in Table 6. Part of this paper involves studying the myriad interrelations between these $\mathbb{Z}_2$ models, for different values of $d, q$, and $k$.

The symmetry operators of these theories can be defined as follows. Consider the foliation $\mathscr{F}_{\mu_1 \ldots \mu_{d-k}}^{\|}$ of the cubic lattice. Each leaf $L^{(d-k)}$ is thought of as a $(d-k)$-dimensional cubic sublattice that is parallel to the $\mu_1 \ldots \mu_{d-k}$ directions. Call $L^{*(d-k)}$ the dual of this sublattice (thought of as a simplicial complex), and let $\tilde{m}^{(d-k-q)}$ be a closed $(d-k-q)$-dimensional membrane built out of fundamental $(d-k-q)$-simplices of $L^{*(d-k)}$. Thinking of $\tilde{m}^{(d-k-q)}$ as an object of the original cubic lattice $\Lambda$, the operators

$$U_{\tilde{m}^{(d-k-q)}}^{L^{(d-k)}} = \prod_{\sigma \in \tilde{m}^{(d-k-q)}} \mathbf{X}_\sigma \,, \qquad L^{(d-k)} \in \mathscr{F}_{\mu_1 \ldots \mu_{d-k}}^{\|} \,, \tag{85}$$

can be thought of as being supported on the leaf $L^{(d-k)}$. These operators generate the $\mathbb{Z}_2^{(q,k)} / \sim$ symmetry of the model, where the product is over $q$-simplices of $\Lambda$ that intersect $\tilde{m}^{(d-k-q)}$. We remark that these symmetry operators satisfy non-trivial relations when $k > 0$. To describe an example, note that the foliation $\mathscr{F}_{\mu_1 \ldots \mu_{d-k+1}}^{\|}$ can be obtained as a coarsening (cf. §2.1) of any of the foliations $\mathscr{F}_{\mu_1 \ldots \hat{\mu}_i \ldots \mu_{d-k+1}}^{\|}$, where $\hat{\mu}_i$ indicates that we omit $\mu_i$ from the list $\mu_1 \ldots \mu_{d-k+1}$. There are $d - k + 1$ such foliations; since $d > k + q$, we can pick $q + 2$ of them, and pass to the associated diagonal subgroup $\mathbb{Z}_2^{(q,k-1)} / \sim$. Each $q$-simplex that lives in a leaf $L^{(d-k+1)}$ of $\mathscr{F}_{\mu_1 \ldots \mu_{d-k+1}}^{\|}$ belongs to 2 of the $q + 2$ foliations that were used to construct the diagonal subgroup, and therefore the coarsened symmetry operator $(U^C)_{\tilde{m}^{(d-k-q+1)}}^{L^{(d-k+1)}}$ (cf. Eq. (20)) involves a product of two Pauli-X operators at each $q$-simplex that is intersected by $\tilde{m}^{(d-k-q+1)}$. It follows that the entire subgroup acts trivially. As an example, this construction recovers precisely the diagonal subgroup $^D\mathbb{Z}_2^{(0)}$ when applied to the (2+1)D plaquette Ising model $^2H_{\mathbb{Z}_2}^{(0,1)}$, described in §1.1 (cf. e.g. Eq. (4)).

It is interesting to compare the behavior of these models in the extreme limits of their parameters. In the case that $U \gg t$, the model is proximate to a trivial phase with a single gapped ground state in which every $q$-simplex is in a $+1$ eigenstate of Pauli-$\mathbf{X}$. On the other hand, when $t \gg U$, the model has a large ground state degeneracy, with the symmetry operators in Eq. (85) acting non-trivially on this ground space (when wrapped around non-trivial

Table 6: A glossary of the various Ising-like theories. The Hamiltonian ${}^d H_{\mathbb{Z}_2}^{(q,k)}$ is well-defined in any spatial dimension $d$ such that $d > k + q$, and has a global $(q,k)$-symmetry $\mathbb{Z}_2^{(q,k)} / \sim$. The qubits of ${}^d H_{\mathbb{Z}_2}^{(q,k)}$ are placed on $q$-simplices. The notation $\langle v, v' \rangle$ indicates nearest neighbor vertices.

| $(q,k)$ | Name | ${}^d H_{\mathbb{Z}_2}^{(q,k)}$ | $G_\rho^{\mu_1 \cdots \mu_{d-k}}$ |
|---------|------|---------------------------------|-----------------------------------|
| $(0,0)$ | transverse field Ising models (TFIM) | $-t \sum_{\langle v,v' \rangle} \mathbf{Z}_v \mathbf{Z}_{v'} - U \sum_v \mathbf{X}_v$ | |
| $(1,0)$ | lattice gauge theories (LGT) | $-t \sum_p \prod_{e \in p} \mathbf{Z}_e - U \sum_e \mathbf{X}_e$ | $\prod_{e \ni v} \mathbf{X}_e$ |
| $(2,0)$ | two-form gauge theories (LGT$_2$) | $-t \sum_c \prod_{p \in c} \mathbf{Z}_p - U \sum_p \mathbf{X}_p$ | $\prod_{p \ni e} \mathbf{X}_p$ |
| $(0,1)$ | plaquette Ising models (PIM) | $-t \sum_p \prod_{v \in p} \mathbf{Z}_v - U \sum_v \mathbf{X}_v$ | |
| $(1,1)$ | X-cube models (XC) | $-t \sum_c \prod_{e \in c} \mathbf{Z}_e - U \sum_e \mathbf{X}_e$ | $\prod_{\substack{e \ni v \\ e \| \mu_1 \cdots \mu_{d-k}}} \mathbf{X}_e$ |
| $(0,2)$ | cubic Ising models (CIM) | $-t \sum_c \prod_{v \in c} \mathbf{Z}_v - U \sum_v \mathbf{X}_v$ | |

cycles). In fact, the $\mathbb{Z}_2^{(q,k)} / \sim$ is completely spontaneously broken. Thus, as the competition between $U$ and $t$ is varied, these models should undergo at least one phase transition. For some choices of $(q,k)$ (e.g. $k = 0$ and $q = 0,1$), it is well-established that there is only one phase transition point, and that it happens at a finite value of $U/t$. It is natural to speculate that a similar story holds for all $(q,k)$, but to our knowledge this has not been established rigorously in the literature.

It is also interesting to ask what happens when one gauges the HFSS. One can follow a procedure copied almost mutatis mutandis from §2.2.1–§2.2.3, and obtain that

$$ {}^d H_{\mathbb{Z}_2}^{(q,k)} \Big/ \left( \mathbb{Z}_2^{(q,k)} \Big/ \sim \right) \cong {}^d \widetilde{H}_{\mathbb{Z}_2}^{(d-q-k-1,k)} , \tag{86} $$

where the tilde in the Hamiltonian on the right-hand side indicates that the model is defined on the dual lattice. We immediately find a result that is consistent with Expectation 1 from §2.1: upon gauging the $(q,k)$-symmetry, we find a theory with an emergent quantum $(d-k-q-1,k)$-symmetry. Furthermore, it turns out that, assuming the speculations of the previous paragraph are true, gauging maps the symmetry preserving phase of ${}^d H_{\mathbb{Z}_2}^{(q,k)}$ to the symmetry breaking phase of ${}^d \widetilde{H}_{\mathbb{Z}_2}^{(d-q-k-1,k)}$, and vice versa. This is a version of Expectation 2, suitably extended to the setting of higher-form subsystem symmetries.

The case that $G = \mathbb{Z}_2$ is straightforward enough that we can simply write down models with the desired symmetry groups, as we have done above. However, for the purpose of uncovering structure that generalizes, it is useful to imagine how one might obtain these models starting with simpler building blocks. We turn to this next.

## 2.3 Assembling the building blocks

Having established the basic properties of the building blocks ${}^d H_{\mathbb{Z}_2}^{(q)}$, we now explain the protocol that we use to assemble them throughout the rest of this paper, and also anticipate some

of the general structure that underlies most of our results.

As we described in the introduction, each section §A for A $= 3, 4, 5, 6$ is organized around one of the models ${}^D H_{\mathbb{Z}_2}^{(q,k)}$. Each subsection §A.B (except for the last subsection of §A) is dedicated to two related network constructions that recover phase diagrams containing (at least the symmetry breaking phase of) ${}^D H_{\mathbb{Z}_2}^{(q,k)}$ in some corner. These two network constructions involve two pieces of input data.

The first piece of data is a *decoupled* network model, which in the present paper simply involves a choice of building block theory ${}^d H_{\mathbb{Z}_2}^{(q)}$, and a choice of foliations $\mathscr{F}_1, \mathscr{F}_2, \dots$ of space. (Although we are using continuum language, in the case of lattice theories, the leaves of a foliation are thought of as a discrete family of $d$-dimensional sublattices of the $D$-dimensional lattice on which the $(D + 1)$-dimensional model ultimately lives.) The decoupled network model is then obtained by foliating space with the theories ${}^d H_{\mathbb{Z}_2}^{(q)}$ along the leaves of the foliations $\mathscr{F}_1, \mathscr{F}_2, \dots$, and its Hamiltonian can schematically be written as

$$
{}^d H_{\mathbb{Z}_2}^{(q)}(\mathscr{F}_1, \mathscr{F}_2, \dots) \sim \sum_i \sum_{L \in \mathscr{F}_i} {}^d H_{\mathbb{Z}_2}^{(q)}\Big|_L. \tag{87}
$$

This decoupled network model has an obvious (relation-free) $q$-form subsystem symmetry of codimension $D - d$, i.e. a $\mathbb{Z}_2^{(q,k)}(\mathscr{F}_1, \mathscr{F}_2, \dots)$ symmetry with $k \equiv D - d$. Tuning the free parameter of its ${}^d H_{\mathbb{Z}_2}^{(q)}$ leaves drives a phase transition that can be described as spontaneously breaking this higher-form subsystem symmetry.

The second piece of data is a subgroup

$$
\mathscr{H} \equiv \mathbb{Z}_2^{(q',k')}\big(\mathscr{F}_1', \mathscr{F}_2', \dots\big)\big/\!\sim \, \subset \mathbb{Z}_2^{(q,k)}(\mathscr{F}_1, \mathscr{F}_2, \dots) \tag{88}
$$

of the global symmetry group of the decoupled network model. This subgroup is generally chosen based on relations satisfied in the target model ${}^D H_{\mathbb{Z}_2}^{(q,k)}$, i.e. we want that

$$
\mathbb{Z}_2^{(q,k)}(\mathscr{F}_1, \mathscr{F}_2, \dots)\big/\mathscr{H} \subset \mathrm{Sym}\Big({}^D H_{\mathbb{Z}_2}^{(q,k)}\Big). \tag{89}
$$

The two network constructions of §A.B are then obtained as follows.

1) In §A.B.1, we deform the decoupled network model by terms that couple the leaves together and preserve the $\mathscr{H}$ symmetry,

$$
{}^d H_{\mathbb{Z}_2}^{(q)}(\mathscr{F}_1, \mathscr{F}_2, \dots) \to {}^d H_{\mathbb{Z}_2}^{(q)}(\mathscr{F}_1, \mathscr{F}_2, \dots) + H_{\mathrm{C}}, \tag{90}
$$

where $H_{\mathrm{C}}$ is a coupling Hamiltonian that features operators whose support spans multiple leaves. That is, we consider the decoupled network model enriched by its $\mathscr{H}$ symmetry. Generally, $H_{\mathrm{C}}$ includes a parameter that preserves the full subsystem symmetry group and drives the system into a phase where the subgroup $\mathscr{H}$ becomes spontaneously unbroken; thus, $\mathscr{H}$ acts trivially in the low energy effective Hilbert space in this phase, and we are be able to identify the residual group which acts faithfully, $\mathbb{Z}_2^{(q,k)}(\mathscr{F}_1, \mathscr{F}_2, \dots)\big/\mathscr{H}$, with (a subgroup of) the symmetry group of ${}^D H_{\mathbb{Z}_2}^{(q,k)}$. The transitions we obtain in this way are generalizations of the p-string condensation mechanism of Ref. [61].

2) In §A.B.2, we couple the leaves together by gauging the $\mathscr{H}$ symmetry,

$$
{}^d H_{\mathbb{Z}_2}^{(q)}(\mathscr{F}_1, \mathscr{F}_2, \dots) \to {}^d H_{\mathbb{Z}_2}^{(q)}(\mathscr{F}_1, \mathscr{F}_2, \dots)\big/\mathscr{H}. \tag{91}
$$

That is, we immerse the decoupled network model inside of a bulk gauge theory which mediates its interactions. Here, we find that the symmetry breaking phase of the decoupled network model maps after gauging to the symmetry breaking phase of ${}^D H_{\mathbb{Z}_2}^{(q,k)}$. The

symmetry preserving phase of the decoupled network model generally maps to a "pure gauge theory" of the global symmetry $\mathcal{H}$. Thus, the gauged network model has a phase diagram that interpolates between these two phases, and we may leverage our knowledge of the phase structure of the ungauged model (using various well-known results covered in §2.2) to understand the transition of the gauged model. For example, we find that we can obtain order/disorder parameters for the transition by simply studying how order/disorder parameters of the decoupled network model are mapped under the gauging of $\mathcal{H}$. Furthermore, the decoupled subdimensional critical point of the ungauged network model $^{d}H_{\mathbb{Z}_2}^{(q)}(\mathscr{F}_1, \mathscr{F}_2, \dots)$ maps to a coupled subdimensional critical point that sits in between the two phases described above.

Finally, in the last subsection §A.B of each section, we write down a parent model that recovers all of the network constructions above in special limits.

# 3 (2+1)D Plaquette Ising Transitions

Our goal in this section is to produce interesting phase diagrams involving $\mathbb{Z}_2^{(0,1)}/\sim$ zero-form subsystem symmetries in (2+1)D. The prototypical example of a model with such symmetries is the plaquette Ising model $^{2}H_{\mathbb{Z}_2}^{(0,1)}$ (also known as the Xu-Moore model [96–98]), and it features heavily in our analysis, emerging deep in different corners of the phase diagrams we study.

We start in §3.1 by studying coupled wire constructions (and gaugings thereof), and demonstrate that they recover phase diagrams that involve the plaquette Ising model. In §3.2, we take anyon models such as the toric code or $\mathbb{Z}_2$ gauge theory as our starting point, and show that gauging subsystem subgroups of their one-form symmetry leads to another perspective on the plaquette Ising model. Finally, in §3.3, we unify these two constructions within a "parent model" which we call the *point-string-net model* (PSN), in analogy with the string-membrane-net model of Ref. [52]. It has the property that it specializes to the models explored in §3.1-§3.2 in various limits, and therefore exhibits all the phase transitions of interest in a single setting.

## 3.1 Coupled wire constructions

In this subsection, inspired by analogous methods for producing $\mathbb{Z}_2$ foliated fracton theories in (3+1)D, we model the linear subsystem symmetries of the plaquette Ising model using $\mathbb{Z}_2^{(0)}$-symmetric wire arrays. In the language of §2.3, the two pieces of input data for this subsection are the decoupled network model $^{1}H_{\mathbb{Z}_2}^{(0)}(\mathscr{F}_{\mathrm{x}}^{\parallel}, \mathscr{F}_{\mathrm{y}}^{\parallel})$ and the subgroup $\mathcal{H} = {}^{\mathrm{D}}\mathbb{Z}_2^{(0)}$ of its global symmetry group that enacts a $\mathbb{Z}_2$ transformation simultaneously on all of the wires.

We start in §3.1.1 by strongly coupling together a crossed array of (1+1)D transverse field Ising wires, which drives a "k-string condensation" transition (k is for kink) from a decoupled wire phase to the plaquette Ising model; we focus on understanding this k-string condensation transition using ideas of spontaneous symmetry breaking. This idea generalizes an analogous construction of the (3+1)D X–cube model using strongly coupled (2+1)D toric code layers and p-string condensation, first presented in Refs. [61,62]. We call this model the *Ising quilt*.

In §3.1.2, we study the effect of gauging the diagonal $^{\mathrm{D}}\mathbb{Z}_2^{(0)}$ subgroup of the Ising quilt. This is inspired by the string-membrane-net picture of the X-cube model in one dimension higher, first studied in Ref. [52]. Here, we find that the subsystem symmetry breaking phase transition of the ungauged wires (i.e. the breaking of the ordinary $\mathbb{Z}_2^{(0)}$ symmetry simultaneously on each TFIM wire) maps after gauging to a subdimensional phase transition between

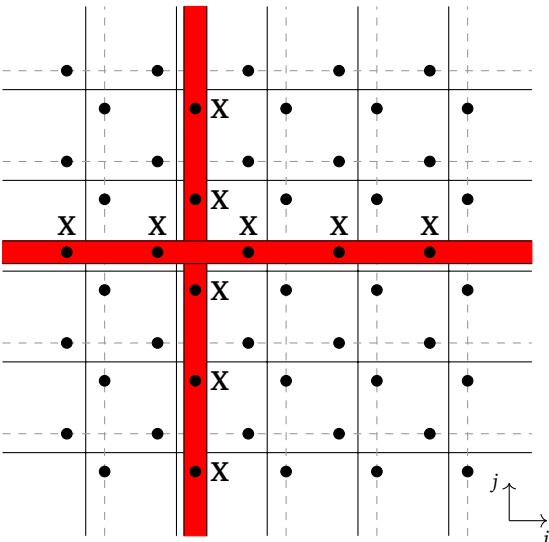

Figure 3: The Ising quilt (c.f. Eq. (93)) is defined on a two dimensional square lattice with two qubits per site. We represent the qubit $\mathcal{H}_{i,j}^{\mathrm{x}}$ with a black dot slightly above and to the left of site $(i,j)$, and the qubit $\mathcal{H}_{i,j}^{\mathrm{y}}$ with a black dot slightly below and to the right of site $(i,j)$; the wires from which these qubits originate are drawn with dashed gray lines. The subsystem symmetry operators $U_j^{\mathrm{x}}$ and $U_i^{\mathrm{y}}$ defined in Eq. (97) are indicated by the horizontal and vertical red bars, respectively.

the ferromagnetic plaquette Ising model and the (2+1)D toric code.

Finally, in §3.1.3 we study how well-known dualities, such as Kramers-Wannier duality, play with this coupled wire construction. As we see below, this is tantamount to studying the effects of gauging the full linear subsystem symmetry group of the Ising quilt.

### 3.1.1 Coupling an array of transverse field Ising wires

Following the discussion in §2.3, one natural approach for constructing a non-trivial theory with a $G^{(0,1)}/{\sim}$ subsystem symmetry is to assemble $(1+1)$D wires, each of which have a global symmetry group $G^{(0)}$, into an intersecting "quilt" of $N_{\mathrm{x}}$ rows and $N_{\mathrm{y}}$ columns, and then directly couple them together in a way that respects this symmetry. To accomplish this, we place a $G^{(0)}$-symmetric (1+1)D theory on each of the leaves of two foliations $\mathscr{F}_{\mathrm{x}}^{\parallel}$ and $\mathscr{F}_{\mathrm{y}}^{\parallel}$, and then couple these leaves together. The resulting model, thought of as a $(2+1)$D system, has a subextensively large symmetry group $G^{(0,1)}(\mathscr{F}_{\mathrm{x}}^{\parallel},\mathscr{F}_{\mathrm{y}}^{\parallel})$ of subsystem symmetries. We then crank up the coupling, aiming to pass through a phase transition into a more interesting phase than that of a collection of trivially decoupled wires. In our case, we find that this transition is from a phase in which the *diagonal* global symmetry subgroup $^{\mathrm{D}}G^{(0)}$ is spontaneously broken to one in which it acts trivially on the low energy subspace of the Hilbert space. Correspondingly, the low energy effective Hamiltonian has an effective symmetry group $G^{(0,1)}(\mathscr{F}_{\mathrm{x}}^{\parallel},\mathscr{F}_{\mathrm{y}}^{\parallel})/^{\mathrm{D}}G^{(0)}$; the quotient by $^{\mathrm{D}}G^{(0)}$ can be thought of as a non-trivial relation between the symmetries on the rows and on the columns of the quilt, and is the reason we are taken out of a decoupled wire phase.

### The Ising quilt

We illustrate the effectiveness of this idea by taking for simplicity $G = \mathbb{Z}_2$ and choosing each of the wires to be transverse field Ising models (TFIM); however, we have in mind future

applications of this idea to more general groups. To implement the proposal of the previous paragraph, we place two qubits at each site $v = (i, j)$ of a square lattice,

$$\mathcal{H} = \bigotimes_{i,j} \left( \mathcal{H}^{x}_{i,j} \otimes \mathcal{H}^{y}_{i,j} \right), \qquad \left( \mathcal{H}^{x}_{i,j} \cong \mathcal{H}^{y}_{i,j} \cong \mathbb{C}^2 \right), \tag{92}$$

and label the Pauli operators which act on the first qubit $\mathcal{H}^{x}_{i,j}$ at site $(i, j)$ as $\mathbf{X}^{x}_{i,j}$, $\mathbf{Z}^{x}_{i,j}$, and similarly for the Paulis which act on the second qubit. We think of the factors $\bigotimes_i \mathcal{H}^{x}_{i,j}$ and $\bigotimes_j \mathcal{H}^{y}_{i,j}$ as the degrees of freedom of spin chains that reside on the rows and columns respectively of an $N_x \times N_y$ grid, which we can take for simplicity to have periodic boundary conditions. We call $\mathscr{F}^{\parallel}_{x}$ the foliation of space by the rows of the lattice, and $\mathscr{F}^{\parallel}_{y}$ the foliation of space by the columns of the lattice. See Figure 3 for a visual summary of the setup.

The resulting *Ising quilt* has Hamiltonian[10]

$$H_{\text{IQ}} = {}^{1}H^{(0)}_{\mathbb{Z}_2} \left( \mathscr{F}^{\parallel}_{x}, \mathscr{F}^{\parallel}_{y} \right) + H_{\text{C}}, \tag{93}$$

where ${}^{1}H^{(0)}_{\mathbb{Z}_2}(\mathscr{F}^{\parallel}_{x}, \mathscr{F}^{\parallel}_{y})$ is the Hamiltonian of the decoupled TFIM wires,

$$ {}^{1}H^{(0)}_{\mathbb{Z}_2}(\mathscr{F}^{\parallel}_{x}, \mathscr{F}^{\parallel}_{y}) = -\sum_{i,j} \left( J\mathbf{Z}^{x}_{i,j}\mathbf{Z}^{x}_{i+1,j} + h\mathbf{X}^{x}_{i,j} + J\mathbf{Z}^{y}_{i,j}\mathbf{Z}^{y}_{i,j+1} + h\mathbf{X}^{y}_{i,j} \right) \tag{94}$$

$$ = -\sum_{i,j} \left( J \quad \text{[figure]} \quad + h \quad \text{[figure]} \quad + J \quad \text{[figure]} \quad + h \quad \text{[figure]} \quad \right) $$

and $H_{\text{C}}$ contains the inter-wire coupling terms,

$$ H_{\text{C}} = -\sum_{i,j} \left( K_X \mathbf{X}^{x}_{i,j}\mathbf{X}^{y}_{i,j} + K_Z \mathbf{Z}^{x}_{i,j}\mathbf{Z}^{y}_{i,j} \right) $$

$$ = -\sum_{i,j} \left( K_X \quad \text{[figure]} \quad + K_Z \quad \text{[figure]} \quad \right). \tag{95}$$

This model is spiritually similar to the coupled layer construction of the X-cube introduced in Refs. [61, 62]; we expound upon this in §6.

If we take $K_Z = 0$, then the Ising quilt respects the $\mathbb{Z}^{(0)}_2$ symmetry on each TFIM wire,

$$ \left[ H_{\text{IQ}} \Big|_{K_Z = 0}, U^{x}_j \right] = 0, \qquad \left[ H_{\text{IQ}} \Big|_{K_Z = 0}, U^{y}_i \right] = 0, \tag{96}$$

where $U^{x}_j$ and $U^{y}_i$ are the unitary symmetry operators acting on the $j$th row and $i$th column respectively,

$$ U^{x}_j = \prod_i \mathbf{X}^{x}_{i,j}, \qquad U^{y}_i = \prod_j \mathbf{X}^{y}_{i,j}. \tag{97}$$

We think of the group generated by these unitaries as a group of "subsystem symmetries" of the Ising quilt, $\mathbb{Z}^{(0,1)}_2(\mathscr{F}^{\parallel}_{x}, \mathscr{F}^{\parallel}_{y})$. If we allow $K_Z \neq 0$, then this subsystem symmetry group is explicitly broken down to the ${}^{D}\mathbb{Z}^{(0)}_2$ subgroup generated by

$$ U^{D} = \prod_{i,j} \mathbf{X}^{x}_{i,j}\mathbf{X}^{y}_{i,j} = \prod_j U^{x}_j \prod_i U^{y}_i, \tag{98}$$

---

[10]Throughout this section, we do not pay careful attention to boundary conditions.

which we refer to as the *diagonal subgroup*. Following the discussion of §2.1, this subgroup is the one obtained by coarsening both $\mathscr{F}_x^{\parallel}$ and $\mathscr{F}_y^{\parallel}$ to the trivial foliation $\mathscr{F}_{xy}^{\parallel}$.

**Phases and excitations**

We now analyze the corners of the phase diagram that one encounters as one tunes the various parameters of the Ising quilt to be very large or very small, as well as the excitations in these phases.

*Weakly interacting wires at $K_Z = 0$ and $K_X \ll 1$*

When $K_Z = K_X = 0$, the Ising quilt simply consists of decoupled (1+1)D TFIM wires. One can also turn on a small amount of $K_X$ and stay in this phase, however we strictly enforce $K_Z = 0$ in order to preserve the subsystem symmetries. In this regime, as one varies the competition between the terms proportional to $J$ and $h$, the individual wires each undergo a symmetry breaking phase transition, as reviewed in §2.2.1; we interpret this as a subsystem symmetry breaking phase transition of the full (2+1)D model. The quasi-particle excitations on either side of this transition can be characterized in terms of spinons/domain-walls on the individual TFIM wires. (Cf. §2.2.1 for a review of the relevant aspects of the (1+1)D TFIM.) Our philosophy is that because this transition is equivalent to subextensively many more familiar lower-dimensional phase transitions, we can leverage it to understand any (more interesting) phase transitions that are related to it by gauging. We turn to this shortly, in the next subsection.

But before we do so, we offer some explorations of the rest of the phase diagram of the Ising quilt, in particular at strong coupling. We expect (and verify shortly) that the model passes through a phase transition as we dial the strength of either $K_Z$ or $K_X$ (or both). Our strategy is to first perturbatively determine the low-energy effective Hamiltonians that describe the Ising quilt deep in these new phases. In doing so we run into a familiar cast of characters.

*(2+1)D plaquette Ising model at $K_Z = 0$ and $K_X \gg 1$*

First, let us take $K_Z = 0$ and study what happens at $K_X \gg 1$. To do this, we view $H_C$ as a soluble Hamiltonian, and consider adding ${}^1 H_{\mathbb{Z}_2}^{(0)}(\mathscr{F}_x^{\parallel}, \mathscr{F}_y^{\parallel})$ as a perturbation. The former has an extensively degenerate ground space, spanned by states that, on each site $(i,j)$, have both qubits in the same eigenstate of Pauli-**X**. More precisely, its ground space takes the shape

$$\mathcal{H}_{\mathrm{gs}} = \bigotimes_{i,j} \mathcal{H}_{i,j}, \tag{99}$$

where $\mathcal{H}_{i,j} \subset \mathcal{H}_{i,j}^x \otimes \mathcal{H}_{i,j}^y$ is the space spanned by $\{|{+}{+}\rangle, |{-}{-}\rangle\}$ with $|\pm\rangle$ the eigenstates of Pauli-**X**. The addition of the perturbation ${}^1 H_{\mathbb{Z}_2}^{(0)}(\mathscr{F}_x^{\parallel}, \mathscr{F}_y^{\parallel})$ splits the energy levels of the states in $\mathcal{H}_{\mathrm{gs}}$, but still keeps them well separated by a large gap from the rest of the excited states. Our goal is to write down a low energy effective Hamiltonian $H_{\mathrm{eff}}$ that acts on $\mathcal{H}_{\mathrm{gs}}$ and whose eigenvalues encode the energies of these states.

If we think of the states $\{|{+}{+}\rangle, |{-}{-}\rangle\}$ at the site $(i,j)$ as the $|\pm\rangle_{\mathrm{eff}}$ states of an effective qubit, the operators

$$\mathbf{Z}_{i,j} = \mathbf{Z}_{i,j}^x \mathbf{Z}_{i,j}^y, \qquad \mathbf{X}_{i,j} = \mathbf{X}_{i,j}^x = \mathbf{X}_{i,j}^y \tag{100}$$

act in the standard way as Pauli operators,

$$\mathbf{X}_{i,j}|{\pm}{\pm}\rangle = \pm|{\pm}{\pm}\rangle, \qquad \mathbf{Z}_{i,j}|{\pm}{\pm}\rangle = |{\mp}{\mp}\rangle. \tag{101}$$

The low energy Hilbert space $\mathcal{H}_{\mathrm{gs}}$ then consists of one qubit per lattice site, and $H_{\mathrm{eff}}$ resembles a (2+1)D lattice model built out of the operators $\mathbf{Z}_{i,j}, \mathbf{X}_{i,j}$. In fact, one can show that to fourth

order in perturbation theory, up to an overall constant

$$H_{\text{eff}} \sim -\sum_{i,j} \left( \tilde{J} \mathbf{Z}_{i,j} \mathbf{Z}_{i+1,j} \mathbf{Z}_{i+1,j+1} \mathbf{Z}_{i,j+1} + \tilde{h} \mathbf{X}_{i,j} \right)$$

$$= -\sum_{i,j} \left( \tilde{J} \quad \begin{array}{c} \end{array} + \tilde{h} \quad \begin{array}{c} \end{array} \right), \tag{102}$$

i.e. $H_{\text{eff}}$ is the plaquette Ising model. Here, $\tilde{h} \sim 2h(1 + O(1/K^2))$ and $\tilde{J} \sim \#\frac{J^4}{K^3}(1 + O(1/K))$. The precise values of $\tilde{J}, \tilde{h}$ are unimportant for what follows.

This plaquette Ising model inherits its subsystem symmetries from the Ising quilt. In particular, when restricted to $\mathcal{H}_{\text{gs}}$, the symmetries in Eq. (97) become

$$U_j^{\text{x}}|_{\mathcal{H}_{\text{gs}}} = \prod_i \mathbf{X}_{i,j}, \quad U_i^{\text{y}}|_{\mathcal{H}_{\text{gs}}} = \prod_j \mathbf{X}_{i,j}, \tag{103}$$

which one can verify commute with $H_{\text{eff}}$. In the IR, the diagonal ${}^{\text{D}}\mathbb{Z}_2^{(0)}$ subgroup acts trivially on the Hilbert space, so that the full symmetry group of the plaquette Ising model is $\mathbb{Z}_2^{(0,1)}(\mathscr{F}_{\text{x}}^{\parallel}, \mathscr{F}_{\text{y}}^{\parallel}) / {}^{\text{D}}\mathbb{Z}_2^{(0)}$.

Furthermore, as one might expect, as one varies the ratio $J/h$ in the Ising quilt (while keeping $K_X$ strong), the plaquette Ising model undergoes a subsystem symmetry breaking phase transition. However, we emphasize that this does *not* occur when $J/h = 1$, but rather when $\tilde{J}/\tilde{h} = 1$. In more detail, when $\tilde{J}/\tilde{h} \ll 1$, the subsystem symmetry group is preserved, and the excitations are spinons,

$$|v\rangle \sim \mathbf{Z}_v|\Omega\rangle + \cdots \tag{104}$$

When $\tilde{J}/\tilde{h} \gg 1$, the subsystem symmetry group is spontaneously broken; the spinons are condensed, and the excitations are fractons supported at the four corners of a membrane operator,

$$|p_1, p_2, p_3, p_4\rangle \sim \prod_{v \in R} \mathbf{X}_v|\Omega\rangle + \cdots, \tag{105}$$

where $p_1, \ldots, p_4$ are the four corner plaquettes of a rectangle $R$.

Now, in order to more directly connect the physics of the plaquette Ising model to that of decoupled TFIM wires, we study the transition between these two phases as one drives $K_X$ from small to large values. We see that it is similar to the "p-string condensation" mechanism of Ref. [61], and we follow some aspects of their discussion closely; however, at the end of this subsection we are also able to offer an alternative perspective which relates this transition to more familiar physics. See also Ref. [99] for a related discussion.

For simplicity, let us turn off the transverse fields on each of the wires, i.e. set $h = 0$, so that we are deep in the subsystem symmetry broken phase of the Ising quilt. When $K_X = 0$, the operator $\mathbf{X}_{i,j}^{\text{x}} \mathbf{X}_{i,j}^{\text{y}}$ creates two pairs of domain walls/kinks, one pair on each of the two TFIM wires which intersect the site $(i, j)$. If we represent a kink on a TFIM wire as a squiggly line segment which bisects the edge on which the kink lives, then the operator $\mathbf{X}_{i,j}^{\text{x}} \mathbf{X}_{i,j}^{\text{y}}$ creates a squiggly "k-string" loop (k for kink) formed out of the four squiggles which bisect the edges that emanate from the site $(i, j)$. Acting with this operator on several neighboring sites creates larger and larger k-string loops.

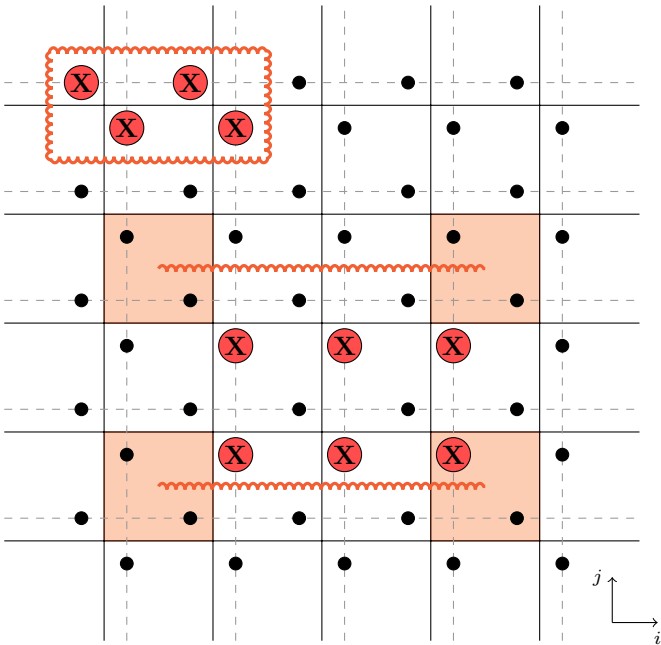

Figure 4: A closed k-string and a pair of open k-strings with excitations supported at their end points.

It is fruitful to consider open k-strings, i.e. k-strings with endpoints. These can be created for example by acting with $\prod_{(i,j)\in R}\mathbf{X}_{i,j}^{\mathrm{y}}$, where the product is over sites inside of some rectangle $R$. Since we are using Paulis which act only on the TFIM columns (and not the rows), this creates two horizontal k-strings whose four endpoints reside inside plaquettes which corner the rectangle $R$. We consider the endpoints of these open k-strings as excitations in their own right. See Figure 4.

As we drive $K_X$ to larger values, the k-strings proliferate throughout the system and eventually condense. Because application of $\mathbf{X}_{i,j}^{\mathrm{x}}\mathbf{X}_{i,j}^{\mathrm{y}}$ creates closed loops and therefore cannot move around the endpoint of an open k-string, the bulk of the k-strings fluctuate and become unobservable, while their endpoints remain well-defined objects that we can identify with the excitations of the PIM that violate the plaquette terms. In tandem with this condensation, the diagonal $^{\mathrm{D}}\mathbb{Z}_2^{(0)}$ subgroup of the subsystem symmetry group transitions from being spontaneously broken to being preserved.

This entire discussion is in complete analogy with the p-string condensation mechanism of Ref. [61], with kinks in the TFIM wires playing the role of magnetic $m$ particles in the toric code layers, and the endpoints of k-strings/the associated PIM excitations playing the role of the totally immobile fractons of the X–cube model. The perspective we emphasize presently is that the k-string condensation transition is a conventional phase transition of Landau type, in the sense that it coincides with the spontaneous unbreaking of a global symmetry.

*(2+1)D transverse field Ising model at $K_X = 0$ and $K_Z \gg 1$*

Before moving on, we briefly comment on the low energy effective Hamiltonian one obtains if one instead drives $K_Z \gg 1$ in the Ising quilt of Eq. (93), taking $K_X = 0$. In this case, the coupling breaks the full collection of subsystem symmetries down to the diagonal $^{\mathrm{D}}\mathbb{Z}_2^{(0)}$ subgroup, and consequently we expect to recover a more conventional model with ordinary $\mathbb{Z}_2$ symmetry. Indeed, one need only go to second order in perturbation theory in this case to find the (2+1)D transverse field Ising model as the effective Hamiltonian (up to an overall

constant),

$$H_{\text{eff}} \sim -\sum_{i,j} \left( J'(\mathbf{Z}_{i,j}\mathbf{Z}_{i+1,j} + \mathbf{Z}_{i,j}\mathbf{Z}_{i,j+1}) + h'\mathbf{X}_{i,j} \right), \tag{106}$$

where now, $J' \sim \#J^2/K(1 + O(1/K))$ and $h' \sim 2h(1 + O(1/K))$. The above Hamiltonian acts on the ground space of $H_{\text{C}}|_{K_X=0}$ in the $K_Z \to \infty$ limit, which consists of an effective qubit at each site,

$$\mathcal{H}_{\text{gs}} = \bigotimes_{i,j} \mathcal{H}_{i,j}, \tag{107}$$

where $\mathcal{H}_{i,j} \subset \mathcal{H}_{i,j}^{\text{x}} \otimes \mathcal{H}_{i,j}^{\text{y}}$ is spanned by $\{|\uparrow\uparrow\rangle, |\downarrow\downarrow\rangle\}$ with $|\uparrow\rangle/|\downarrow\rangle$ the eigenstates of Pauli-$\mathbf{Z}$. The operators acting on this effective qubit which appear in the effective Hamiltonian above are defined in terms of Paulis acting on the full Hilbert space as

$$\mathbf{Z}_{i,j} = \mathbf{Z}_{i,j}^{\text{x}} = \mathbf{Z}_{i,j}^{\text{y}}, \qquad \mathbf{X}_{i,j} = \mathbf{X}_{i,j}^{\text{x}}\mathbf{X}_{i,j}^{\text{y}}. \tag{108}$$

See §2.2.2 for further properties of the (2+1)D TFIM, including discussion of its spinon and domain wall excitations.

**Order and disorder parameters**

There are several useful order and disorder parameters that one can use to probe the phase diagram of the Ising quilt. Throughout this mini-section, we set $K_Z = 0$. We are primarily interested in understanding the subsystem symmetry breaking phase transition when $K_X$ is also set to 0, but the parameters we write down here can be used to understand the rest of the phase diagram as well, away from $K_X = 0$.

One possibility is to consider order/disorder parameters on the individual wires,

$$\begin{aligned}
\mathcal{O}_{(i,j),r}^{\text{x}} &= \mathbf{Z}_{i,j}^{\text{x}}\mathbf{Z}_{i+r,j}^{\text{x}}, & \mathcal{D}_{(i,j),r}^{\text{x}} &= \prod_{i \le k \le i+r} \mathbf{X}_{k,j}^{\text{x}}, \\
\mathcal{O}_{(i,j),r}^{\text{y}} &= \mathbf{Z}_{i,j}^{\text{y}}\mathbf{Z}_{i,j+r}^{\text{y}}, & \mathcal{D}_{(i,j),r}^{\text{y}} &= \prod_{j \le k \le j+r} \mathbf{X}_{i,k}^{\text{y}}.
\end{aligned} \tag{109}$$

Here, $\mathcal{O}_{(i,j),r}^{\text{x}}$ can be thought of equally well as a candidate order parameter for the subsystem symmetry supported on the $j$th row, or as a candidate order parameter for the diagonal zero-form symmetry $^{\text{D}}\mathbb{Z}_2^{(0)}$. On the other hand, one can only think of $\mathcal{D}_{(i,j),r}^{\text{x}}$ as a candidate disorder parameter for the subsystem symmetry on the $j$th row. On the $K_X = 0$ line these behave as

$$\left\langle \mathcal{O}_{(i,j),r}^{\mu} \right\rangle \xrightarrow[r \to \infty]{K_X=0} \begin{cases} \#e^{-r/\xi}, & J/h \ll 1, \\ \text{const.}, & J/h \gg 1, \end{cases} \qquad \left\langle \mathcal{D}_{(i,j),r}^{\mu} \right\rangle \xrightarrow[r \to \infty]{K_X=0} \begin{cases} \text{const.}, & J/h \ll 1, \\ \#e^{-r/\xi}, & J/h \gg 1, \end{cases} \tag{110}$$

and so these can be reliably used to diagnose the spontaneous symmetry breaking phase transition as $J/h$ is varied in the decoupled wire phase, depending on whether they have long range order or not.

The order parameters above can also be combined in various ways. For example, it is natural to consider the combinations

$$^{\text{SS}}\mathcal{O}_R = \prod_{v \in \text{corner}(R)} \mathbf{Z}_v^{\text{x}}\mathbf{Z}_v^{\text{y}}, \quad ^{\text{SS}}\mathcal{D}_R^{\text{x}} = \prod_{v \in R} \mathbf{X}_v^{\text{x}}, \quad ^{\text{SS}}\mathcal{D}_R^{\text{y}} = \prod_{v \in R} \mathbf{X}_v^{\text{y}}, \tag{111}$$

where here, $R$ is a rectangle, thought of as a set of vertices, and corner($R$) is the set of 4 vertices which form the corners of $R$. Note that $^{\text{SS}}\mathcal{O}_R$ and $^{\text{SS}}\mathcal{D}_R^{\mu}$ can be thought of as candidate

order/disorder parameters for the generators of the subsystem symmetry. In the decoupled wire phase ($K_X = K_Z = 0$), we have that

$$\langle {}^{\mathrm{SS}}\mathcal{O}_R \rangle \xrightarrow[\text{large } R]{K_X = 0} \begin{cases} \# e^{-P(R)/\xi}, & J/h \ll 1 \\ \text{const.,} & J/h \gg 1, \end{cases} \qquad \langle {}^{\mathrm{SS}}\mathcal{D}_R^{\mathrm{x}} \rangle \xrightarrow[\text{large } R]{K_X = 0} \begin{cases} \# e^{-\text{height}(R)/\tilde{\xi}}, & J/h \ll 1 \\ \# e^{-A(R)/A_0}, & J/h \gg 1, \end{cases} \tag{112}$$

where $P(R)$ is the perimeter of $R$ and $A(R)$ is the area. One reason to consider these parameters is that in the $K_X \gg 1$ limit (i.e. in the plaquette Ising model phase), they collapse to "natural" order parameters for the plaquette Ising model,

$$ {}^{\mathrm{SS}}\mathcal{O}_R \xrightarrow{K_X \gg 1} \prod_{v \in \text{corner}(R)} \mathbf{Z}_v, \qquad {}^{\mathrm{SS}}\mathcal{D}_R^{\mu} \xrightarrow{K_X \gg 1} \prod_{v \in R} \mathbf{X}_v. \tag{113}$$

We call these order/disorder parameters "natural" because they map to one another under the self-duality of the plaquette Ising model. It is possible to show [100] that in the PIM, Eq. (102), these satisfy

$$\langle {}^{\mathrm{SS}}\mathcal{O}_R \rangle \xrightarrow[\text{large } R]{K_X \gg 1} \begin{cases} \# e^{-A(R)/A_0'}, & \tilde{J}/\tilde{h} \ll 1 \\ \text{const.,} & \tilde{J}/\tilde{h} \gg 1, \end{cases} \qquad \langle {}^{\mathrm{SS}}\mathcal{D}_R^{\mu} \rangle \xrightarrow[\text{large } R]{K_X \gg 1} \begin{cases} \text{const.,} & \tilde{J}/\tilde{h} \ll 1 \\ \# e^{-A(R)/A_0'}, & \tilde{J}/\tilde{h} \gg 1. \end{cases} \tag{114}$$

According to the discussion of §2.1, in a symmetry preserving phase, any candidate order parameter for, say, a row symmetry should decay with the separation of the two points at which local operators are inserted on that row. This may seem at odds with the fact that we found a perimeter (area) law in the decoupled wire (plaquette Ising) phase. The resolution is that we should be studying how the order parameter decays with the width of $R$, assuming it has a large but *fixed* height. Indeed, if the height is fixed, then the perimeter and area laws can both be thought of as "width laws", which is consistent with the general picture of order parameters for subsystem symmetries we explained earlier.

Another natural combination is as follows

$$ {}^{\mathrm{D}}\mathcal{O}_\gamma = \mathbf{Z}_{\gamma_{\mathrm{i}}}^{\mu_{\mathrm{i}}} \left( \prod_{v \in \text{corner}(\gamma^{\circ})} \mathbf{Z}_v^{\mathrm{x}} \mathbf{Z}_v^{\mathrm{y}} \right) \mathbf{Z}_{\gamma_{\mathrm{f}}}^{\mu_{\mathrm{f}}}, \qquad {}^{\mathrm{D}}\mathcal{D}_R = \prod_{v \in R} \mathbf{X}_v^{\mathrm{x}} \mathbf{X}_v^{\mathrm{y}}, \tag{115}$$

where here, $\gamma$ is an open path on the lattice with vertex endpoints $\gamma_{\mathrm{i}}$ and $\gamma_{\mathrm{f}}$, and corner($\gamma^{\circ}$) is defined as the set of vertices in $\gamma$ which are met by two edges of $\gamma$ which form a right angle. Also, $\mu_{\mathrm{i}}$ is the direction of the edge in $\gamma$ which emanates from $\gamma_{\mathrm{i}}$, and similarly for $\mu_{\mathrm{f}}$. Again, when $K_X = 0$, the behavior of these parameters can be computed using trivially decoupled wires,

$$\langle {}^{\mathrm{D}}\mathcal{O}_\gamma \rangle \xrightarrow[\text{large } \gamma]{K_X = 0} \begin{cases} \# e^{-L(\gamma)/\xi'}, & J/h \ll 1, \\ \# e^{-\alpha(|\text{corner}(\gamma^{\circ})|+1)}, & J/h \gg 1, \end{cases} \qquad \langle {}^{\mathrm{D}}\mathcal{D}_R \rangle \xrightarrow[\text{large } R]{K_X = 0} \begin{cases} \# e^{-P(R)/\xi'}, & J/h \ll 1, \\ \# e^{-A(R)/\tilde{A}_0}, & J/h \gg 1. \end{cases} \tag{116}$$

The disorder parameter is a truncated symmetry operator corresponding to the diagonal symmetry ${}^{\mathrm{D}}\mathbb{Z}_2^{(0)}$. We explain why the order parameter is natural below

**Relation to (2+1)D TFIM with dynamical disorder**

The fact that the k-string condensation transition is a spontaneous symmetry breaking transition allows us to obtain an intriguing alternate perspective on its physics as follows. First, we notice that when $h = 0$, the Ising quilt gains an extensive number of conserved charges

$$Q_{i,j}^{\mathbf{Z}} = \mathbf{Z}_{i,j}^{\mathrm{x}} \mathbf{Z}_{i,j}^{\mathrm{y}}, \tag{117}$$

which commute with the Hamiltonian and among themselves. Therefore, we can work in a simultaneous eigenbasis of all these operators. To facilitate this, we can perform the following change of variables (which is similar to one used in Ref. [101]),

$$\hat{\mathbf{Z}}_{i,j} = \mathbf{Z}^{\mathrm{x}}_{i,j}, \qquad \hat{\mathbf{X}}_{i,j} = \mathbf{X}^{\mathrm{x}}_{i,j}\mathbf{X}^{\mathrm{y}}_{i,j},$$
$$Q^{\mathbf{Z}}_{i,j} = \mathbf{Z}^{\mathrm{x}}_{i,j}\mathbf{Z}^{\mathrm{y}}_{i,j}, \qquad Q^{\mathbf{X}}_{i,j} = \mathbf{X}^{\mathrm{y}}_{i,j}. \tag{118}$$

One can verify that these operators have the same algebra as two commuting sets of Pauli operators. If one expresses the Ising quilt in terms of these new variables, one finds

$$H_{\mathrm{IQ}}\big|_{h=0} = -\sum_{i,j}\Big( J(\hat{\mathbf{Z}}_{i,j}\hat{\mathbf{Z}}_{i+1,j} + Q^{\mathbf{Z}}_{i,j}Q^{\mathbf{Z}}_{i,j+1}\hat{\mathbf{Z}}_{i,j}\hat{\mathbf{Z}}_{i,j+1}) + K_X\hat{\mathbf{X}}_{i,j}\Big). \tag{119}$$

More suggestively, if we act on eigenstates of the $Q^{\mathbf{Z}}_{i,j}$ with eigenvalues $q_{i,j}$, then the Hamiltonian within a such a "$q$-sector" is

$$H_{\mathrm{IQ}}(\{q_{i,j}\})\big|_{h=0} = -\sum_{i,j}\big( J(\hat{\mathbf{Z}}_{i,j}\hat{\mathbf{Z}}_{i+1,j} + q_{i,j}q_{i,j+1}\hat{\mathbf{Z}}_{i,j}\hat{\mathbf{Z}}_{i,j+1}) + K_X\hat{\mathbf{X}}_{i,j}\big). \tag{120}$$

This is none other than a (2+1)D transverse field Ising model with antiferromagnetic disorder![11] This gives the qubits $Q^{\mathbf{Z}}_{i,j}$ the interpretation of "dynamical disorder fields" which are very similar to those employed in Ref. [102] to achieve disorder-free localization. Thus, we find that the k-string condensation transition is in the same universality class as a conventional Ising critical point decorated with dynamical disorder. In connection with this, we see that the order/disorder parameters written in Eq. (115) become precisely the order/disorder parameters of an Ising transition.

We hope that this observation will help catalyze future investigations into the relationship between fracton physics and many-body localization [103], excited state phase transitions [104], etc. (see Refs. [105, 106] for some work in this direction).

### 3.1.2 Gauging the diagonal subgroup

We now explore an alternative method for coupling together (1+1)D wires. Instead of directly coupling together TFIM wires through local operators supported at their intersections, we immerse them inside a (2+1)D $\mathbb{Z}_2$ gauge theory that mediates their interactions. This idea is similar to the foliated description of the X-cube model presented in Ref. [52], or more generally the topological defect networks of Ref. [68], however it differs in that we do not require the various strata to be topological. (See also Ref. [107] for a spiritually similar supersymmetric $U(1)$ construction which arises on the world-volumes of branes in string theory.) Our ultimate aim, as we stated earlier, is to study the fate of the subsystem symmetry breaking phase transition of decoupled Ising wires after they are coupled to the bulk topological gauge theory.

**The gauged Ising quilt**

We obtain the desired model by applying the *energetic* gauging prescription (cf. §2.1) to the diagonal subgroup $\mathscr{H} = {}^{\mathrm{D}}\mathbb{Z}^{(0)}_2$ of the global subsystem symmetry group of the Ising quilt in

---

[11]One might be confused about why the defects only seem to appear on vertical links in the lattice. This is essentially because our change of variables in Eq. (118) picked a direction. If we had instead chosen $\hat{\mathbf{Z}}_{i,j} = \mathbf{Z}^{\mathrm{y}}_{i,j}$ and $Q^{\mathbf{X}}_{i,j} = \mathbf{X}^{\mathrm{x}}_{i,j}$ then the dislocations would appear on the horizontal edges. In general, we are free at each site to perform the change of variables differently, in which case the defects appear on some combination of both vertical and horizontal edges.

Eq. (98). One motivation for gauging this particular subgroup is that we are interested in constructions of the plaquette Ising model, and we can observe that this overall ${}^{\mathrm{D}}\mathbb{Z}_2^{(0)}$ is precisely the difference between the global symmetry group of decoupled Ising wires, $\mathbb{Z}_2^{(0,1)}(\mathscr{F}_{\mathrm{x}}^{\parallel}, \mathscr{F}_{\mathrm{y}}^{\parallel})$, and the global symmetry group of the plaquette Ising model, $\mathbb{Z}_2^{(0,1)}(\mathscr{F}_{\mathrm{x}}^{\parallel}, \mathscr{F}_{\mathrm{y}}^{\parallel})\big/{}^{\mathrm{D}}\mathbb{Z}_2^{(0)}$. In other words, the idea is that, in order to eliminate the extra symmetries of decoupled Ising wires that act trivially in the plaquette Ising model, we can simply gauge them. We indeed find below that in a certain limit, the PIM is recovered from this construction.

The resulting *gauged Ising quilt* has Hamiltonian

$$H_{\mathrm{IQ}}\Big|_{E}^{{}^{\mathrm{D}}\mathbb{Z}_2^{(0)}} = H_{\mathrm{CDW}} + H_{\mathrm{C}} + H_{\mathrm{gauge}}$$

$$G_v = \mathbf{X}_v^{\mathrm{x}} \mathbf{X}_v^{\mathrm{y}} \prod_{e \ni v} \mathbf{X}_e = \quad , \tag{121}$$

whose ingredients are defined as follows. First, in addition to the two qubits per lattice site of the Ising quilt (which we alternate between labeling with their coordinates $(i, j)$ or simply by $v$ for "vertex"), we have added a qubit to every link of the lattice (which we label either by e.g. $(i + \frac{1}{2}, j)$, $(i, j + \frac{1}{2})$, or by $e$ for "edge") to play the role of the $\mathbb{Z}_2$ gauge field. The term $G_v$ written above is the Gauss's law constraint for this gauge symmetry. We take the energetics of this gauge field to further be governed by

$$H_{\mathrm{gauge}} = -U \sum_e \mathbf{X}_e - t \sum_p \prod_{e \in p} \mathbf{Z}_e \tag{122}$$

$$= -U \left( \sum_{e \| \mathrm{y}} \quad + \sum_{e \| \mathrm{x}} \quad \right) - t \sum_p \quad .$$

These terms play the role of the "$E^2$" and "$B^2$" terms one would encounter in gauge theories with a continuous gauge group. In the strict gauging procedure, we would have called the second term a "flux term" and imposed it as a constraint on the Hilbert space, however we choose to impose it energetically for now. To give ourselves more knobs to tune, we have also included the first term proportional to $U$; we investigate how it impacts the physics shortly. We demand that the rest of the terms in the gauged Ising quilt be gauge invariant, i.e. that they commute with each $G_v$.

The term $H_{\mathrm{CDW}}$ is obtained by taking the Hamiltonian ${}^1H_{\mathbb{Z}_2}^{(0)}(\mathscr{F}_{\mathrm{x}}^{\parallel}, \mathscr{F}_{\mathrm{y}}^{\parallel})$ for the decoupled Ising wires, Eq. (94), and "covariantizing" the two-body Pauli-$\mathbf{Z}$ terms—e.g. by making substitutions like $\mathbf{Z}_{i,j}^{\mathrm{x}} \mathbf{Z}_{i,j+1}^{\mathrm{x}} \to \mathbf{Z}_{i,j}^{\mathrm{x}} \mathbf{Z}_{i+\frac{1}{2},j} \mathbf{Z}_{i+1,j}^{\mathrm{x}}$—so that they are gauge-invariant, leading to

$$H_{\mathrm{CDW}} = -\sum_{i,j} \left( J \mathbf{Z}_{i,j}^{\mathrm{x}} \mathbf{Z}_{i+\frac{1}{2},j} \mathbf{Z}_{i+1,j}^{\mathrm{x}} + h \mathbf{X}_{i,j}^{\mathrm{x}} + J \mathbf{Z}_{i,j}^{\mathrm{y}} \mathbf{Z}_{i,j+\frac{1}{2}} \mathbf{Z}_{i,j+1}^{\mathrm{y}} + h \mathbf{X}_{i,j}^{\mathrm{y}} \right)$$

$$= -\sum_{i,j} \left( J \quad + h \quad + J \quad + h \quad \right). \tag{123}$$

Finally, for completeness, we also include the coupling terms from the Hamiltonian in Eq. (95),

$$H_{\rm C} = -\sum_{i,j}\left(K_X \mathbf{X}^{\rm x}_{i,j}\mathbf{X}^{\rm y}_{i,j} + K_Z \mathbf{Z}^{\rm x}_{i,j}\mathbf{Z}^{\rm y}_{i,j}\right), \tag{124}$$

since they are gauge invariant without requiring any modification. However, they do not play a crucial role in our story, and for the most part we keep them turned off, in which case the only interactions between the wires are mediated indirectly through the gauge theory.

We note in passing that, when $K_X = K_Z = 0$, many aspects of this model are amenable to exact analysis, simply because the (1+1)D transverse field Ising model is exactly soluble. In particular, any ${}^{\rm D}\mathbb{Z}_2^{(0)}$-even correlator function of the decoupled Ising wire model, ${}^1 H^{(0)}_{\mathbb{Z}_2}(\mathscr{F}^{\|}_{\rm x}, \mathscr{F}^{\|}_{\rm y})$, descends to a correlator of the gauged Ising quilt with the same value.

Before moving on, we perform a local unitary circuit that makes imposing the Gauss's law constraint more straightforward. It is defined as

$$V = \prod_{i,j}\left(C^{\rm x}_{i,j}\mathbf{X}_{i+\frac{1}{2},j}\right)\left(C^{\rm x}_{i+1,j}\mathbf{X}_{i+\frac{1}{2},j}\right)\left(C^{\rm y}_{i,j}\mathbf{X}_{i,j+\frac{1}{2}}\right)\left(C^{\rm y}_{i,j+1}\mathbf{X}_{i,j+\frac{1}{2}}\right)$$

$$= \prod_{i,j} \quad \text{} \quad . \tag{125}$$

Here, $C^{\mu}_{v}\mathbf{X}_e$ (for $\mu = {\rm x, y}$) is a controlled-$\mathbf{X}$ gate. Using the identities from Eq. (14), we find

$$V\left(H_{\rm IQ}\Big/^{{\rm D}}_{E}\mathbb{Z}_2^{(0)}\right)V^{\dagger} = -J\sum_e \mathbf{Z}_e - h\sum_{i,j}\left(\mathbf{X}_{i-\frac{1}{2},j}\mathbf{X}^{\rm x}_{i,j}\mathbf{X}_{i+\frac{1}{2},j} + \mathbf{X}_{i,j-\frac{1}{2}}\mathbf{X}^{\rm y}_{i,j}\mathbf{X}_{i,j+\frac{1}{2}}\right)$$

$$-U\sum_e \mathbf{X}_e - t\sum_p \prod_{e\in p}\mathbf{Z}_e\prod_{v\in p}\mathbf{Z}^{\rm x}_v\mathbf{Z}^{\rm y}_v - \sum_v\left(K_X \mathbf{X}^{\rm x}_v\mathbf{X}^{\rm y}_v\prod_{e\ni v}\mathbf{X}_e + K_Z\mathbf{Z}^{\rm x}_v\mathbf{Z}^{\rm y}_v\right) \tag{126}$$

$$V G_v V^{\dagger} = \mathbf{X}^{\rm x}_v\mathbf{X}^{\rm y}_v.$$

The motivation for considering this unitary circuit $V$ is that, as one can see, it converts the Gauss's law constraint to a product of operators supported on a single site. Solving this constraint can now be accomplished in the same way we computed the $K_X \gg 1$ effective Hamiltonian in §3.1; i.e. we reduce to a single effective qubit per site, whose effective Pauli operators are defined in terms of $\mathbf{X}^{\mu}_v, \mathbf{Z}^{\mu}_v$ as in Eq. (100). Then, within this constrained Hilbert space, the gauged Ising quilt becomes

$$H_{\rm GIQ} := V\left(H_{\rm IQ}\Big/^{{\rm D}}_{E}\mathbb{Z}_2^{(0)}\right)V^{\dagger}\Big|_{V G_v V^{\dagger}=1}$$

$$= -\sum_e (J\mathbf{Z}_e + U\mathbf{X}_e) - h\sum_{i,j}\left(\mathbf{X}_{i-\frac{1}{2},j}\mathbf{X}_{i,j}\mathbf{X}_{i+\frac{1}{2},j} + \mathbf{X}_{i,j-\frac{1}{2}}\mathbf{X}_{i,j}\mathbf{X}_{i,j+\frac{1}{2}}\right) \tag{127}$$

$$- t\sum_p \prod_{e\in p}\mathbf{Z}_e\prod_{v\in p}\mathbf{Z}_v - \sum_v\left(K_X\prod_{e\ni v}\mathbf{X}_e + K_Z\mathbf{Z}_v\right)$$

$$
\begin{aligned}
= -J\left(\sum_{e\|y} \boxed{\mathbf{Z}} + \sum_{e\|x} \boxed{\mathbf{Z}}\right) - U\left(\sum_{e\|y}\boxed{\mathbf{X}} + \sum_{e\|x}\boxed{\mathbf{X}}\right) \\
-h\sum_v\left(\boxed{\mathbf{X}\,\mathbf{X}\,\mathbf{X}} + \boxed{\mathbf{X}}\right) - t\sum_p \boxed{\begin{matrix}\mathbf{Z}\,\mathbf{Z}\,\mathbf{Z}\\ \mathbf{Z}\ p\ \mathbf{Z}\\ \mathbf{Z}\,\mathbf{Z}\,\mathbf{Z}\end{matrix}} \\
-\sum_v\left(K_X\,\boxed{\begin{matrix}\mathbf{X}\\ \mathbf{X}\,\mathbf{X}\\ \mathbf{X}\end{matrix}} + K_Z\,\boxed{\mathbf{Z}}\right).
\end{aligned}
\tag{128}
$$

This is the model we work with. We note that if we were to instead use the "strict" gauging procedure, then we would obtain almost the same model, except with $U = 0$ and with the term proportional to $t$ imposed as a constraint. The strict version of the theory has an emergent quantum $^{D}\widehat{\mathbb{Z}}_2^{(1)}$ one-form symmetry (cf. Expectation 1 of §2.1), whose symmetry operators take the form

$$
\widehat{U}_\gamma = \prod_{e\in\gamma}\mathbf{Z}_e \prod_{v\in\text{corner}(\gamma)}\mathbf{Z}_v\,,
\tag{129}
$$

where $v$ is considered a "corner" of $\gamma$ if two edges of $\gamma$ meet $v$ at a right angle. In addition, both the strict and energetic versions of the model have a subsystem symmetry group of the form $\mathbb{Z}_2^{(0,1)}(\mathscr{F}_x^\|,\mathscr{F}_y^\|)\big/^{D}\mathbb{Z}_2^{(0)}$ that are generated by the operators

$$
U_j^x = \prod_i \mathbf{X}_{i,j}\,, \qquad U_i^y = \prod_j \mathbf{X}_{i,j}\,.
\tag{130}
$$

Interestingly, by virtue of this construction, the strict version of the gauged Ising quilt has a sector whose correlation functions are well-described by decoupled Ising wires, and so are exactly soluble. This appears similar to the idea of "dimensional reduction" (see e.g. Ref. [97, 108]).

**Phases and excitations**

To gain some intuition for the phase diagram of this Hamiltonian, we study it in various extreme limits of its parameters. We are primarily interested in what happens as one tunes the competition between $J$ and $h$ when $K_X = K_Z = 0$, which before gauging corresponds to the subsystem symmetry breaking phase transition of decoupled Ising wires. After gauging, Expectation 2 of §2.1 suggests that it should correspond to a simultaneous breaking of the $\mathbb{Z}_2^{(0,1)}(\mathscr{F}_x^\|,\mathscr{F}_y^\|)\big/^{D}\mathbb{Z}_2^{(0)}$ subsystem symmetry group and an unbreaking of the emergent $^{D}\widehat{\mathbb{Z}}_2^{(1)}$ one-form symmetry (cf. Figure 5). We find that this expectation is fulfilled below.

*Plaquette Ising model at $K_X = K_Z = 0$ and $J \gg 1$*

First, let us turn off the couplings $K_X$ and $K_Z$ and study what happens when $J$ is taken to be very large. Before gauging, this is the subsystem symmetry breaking phase of decoupled Ising wires. After gauging, one is still in the subsystem symmetry breaking phase. Since $^{D}\mathbb{Z}_2^{(0)}$ was spontaneously broken before gauging, the emergent quantum $^{D}\widehat{\mathbb{Z}}_2^{(1)}$ one-form symmetry should act trivially on the low energy theory deep in this phase after gauging.

Let us check that this expectation is borne out. The effect of taking $J$ large is to freeze out the degrees of freedom on the links so that they are all in the spin up state. We can then

compute the effective Hamiltonian within this subspace using perturbation theory, in the same way we have been in previous sections. We find that if one goes to third order in perturbation theory, the low energy effective Hamiltonian is

$$H_{\text{GIQ}} \xrightarrow{\text{energetic}} -\tilde{t} \sum_p \prod_{v \in p} \mathbf{Z}_v - \tilde{h} \sum_v \mathbf{X}_v \,, \tag{131}$$

which is precisely the plaquette Ising model. The term proportional to $U$ in $H_{\text{GIQ}}$ is crucial here for the purposes of generating the transverse field term proportional to $\tilde{h}$, which is why we have included it.

In the "strict" version of the gauging procedure, introduced in §2.1, the term proportional to $U$ is not present, the flux term is imposed as a hard constraint on the Hilbert space, rather than energetically. In this case, the Hamiltonian is essentially zero, and the entire Hilbert space by definition consists of the ground states of the plaquette Ising model with the transverse field term turned off, i.e.

$$\begin{aligned} H_{\text{GIQ}} &\xrightarrow{\text{strict}} 0 \,, \\ F_p &= \prod_{v \in p} \mathbf{Z}_p = 1 \,, \quad \forall p \,. \end{aligned} \tag{132}$$

Fractonic excitations (i.e. violations of the $F_p$) correspond to inserting antiferromagnetic defects in the Ising wires before gauging; this is completely analogous to the fact that electric particles in $\mathbb{Z}_2$ lattice gauge theory correspond to anti-ferromagnetic defect networks of the transverse field Ising model, as we have described in §2.2.2.

In total, we confirm our expectations that the subsystem symmetry breaking phase of decoupled Ising wires maps, after gauging, to the subsystem symmetry breaking phase of the plaquette Ising model, and that the ${}^{\text{D}}\widehat{\mathbb{Z}}_2^{(1)}$ is realized trivially on the low-energy Hamiltonian (i.e. is unbroken).

*Deconfined $\mathbb{Z}_2$ lattice gauge theory at $K_X = K_Z = 0$ and $h \gg 1$*

On the other hand, we can consider the limit in which $h$ becomes very large. Before gauging, this corresponds to the subsystem symmetry preserving phase of the decoupled Ising wires, and so the subsystem symmetry group remains unbroken after gauging the diagonal subgroup. Moreover, after gauging, the emergent quantum ${}^{\text{D}}\widehat{\mathbb{Z}}_2^{(1)}$ symmetry is spontaneously broken. We see that this corresponds to a deconfined $\mathbb{Z}_2$ gauge theory phase (i.e. a toric code phase) below.

It is actually easier to work with the Hamiltonian in Eq. (121) rather than $H_{\text{GIQ}}$ (note that the two are unitarily equivalent). If we take $h \gg 1$, then the two qubits at each site of the lattice are energetically forced into $+1$ eigenstates of Pauli-$\mathbf{X}$. This has the effect of freezing out all the matter degrees of freedom, leaving behind only the gauge qubits, which are governed by the Hamiltonian

$$\begin{aligned} H_{\text{IQ}}\Big|_E^{{}^{\text{D}}\mathbb{Z}_2^{(0)}} &\xrightarrow{h \gg 1} -U \sum_e \mathbf{X}_e - t \sum_p \prod_{e \in p} \mathbf{Z}_e \,, \\ G_v &= \prod_{e \ni v} \mathbf{X}_e \,. \end{aligned} \tag{133}$$

This is simply pure $\mathbb{Z}_2$ lattice gauge theory. In the strict gauging procedure, the Hamiltonian is a constant, and in addition to the Gauss's law constraint, $G_v = 1$, we also impose the term proportional to $t$ as a constraint. In this case, one sees that the emergent quantum ${}^{\text{D}}\widehat{\mathbb{Z}}_2^{(1)}$ one-form symmetry is identified with one of the one-form symmetries of the toric code, which allows us to verify that it is indeed spontaneously broken. The model evidently also grows another one-form symmetry in this limit as well.

Having identified this phase with toric code, we offer an interesting picture of its ground state. To achieve this, we make use of the unitary circuit

$$\widetilde{V} = \prod_{i,j}\left(C_{i,j}\mathbf{X}_{i+\frac{1}{2},j}\right)\left(C_{i+1,j}\mathbf{X}_{i+\frac{1}{2},j}\right) = \prod_{i,j} \boxed{\,\,_{(i,j)}\,\, \text{}\,\,}\,. \tag{134}$$

Then the gauged Ising quilt becomes

$$\widetilde{V}H_{\mathrm{GIQ}}\widetilde{V}^{\dagger} = -J\sum_{i,j}\left(\mathbf{Z}_{i,j+\frac{1}{2}} + \mathbf{Z}_{i,j}\mathbf{Z}_{i+\frac{1}{2},j}\mathbf{Z}_{i+1,j}\right) - U\sum_e \mathbf{X}_e$$
$$-h\sum_v\left(\mathbf{X}_v + \mathbf{X}_v\prod_{e\ni v}\mathbf{X}_e\right) - t\sum_p\prod_{e\in p}\mathbf{Z}_e. \tag{135}$$

Consider $K_X = K_Z = U = J = 0$ in $\widetilde{V}H_{\mathrm{GIQ}}\widetilde{V}^{\dagger}$. We recognize that the ground state wavefunction in the Pauli-$\mathbf{X}$ basis corresponds to an equal weight superposition of closed strings (similar to the conventional toric code groundstate) decorated with points where the strings turn corners. This model has interesting symmetry properties under the gauged zero-form linear subsystem symmetry [85].

*Phase transition at $K_X = K_Z = U = 0$ and $J = h$*

Let us trace how excitations of the Ising wires descend to excitations of the gauged Ising quilt. We set $K_X = K_Z = U = 0$.

When $h = 0$, a domain wall on an Ising wire maps to an excitation of the single body Pauli-$\mathbf{Z}$ term proportional to $J$. A pair of such excitations can be created e.g. by the operator

$$(\text{pair of gauged domain walls}) \sim \mathbf{X}_{i_0-\frac{1}{2},j_0}\left(\prod_{\substack{i\\i_0\leq i\leq i_1}}\mathbf{X}_{i,j_0}\right)\mathbf{X}_{i_1+\frac{1}{2},j_0}|\Omega\rangle\,. \tag{136}$$

Such excitations are locally equivalent to violations of the plaquette terms proportional to $t$. For example, multiplying the above creation operator by $\mathbf{X}_{i_0-\frac{1}{2},j_0}$ converts one of the gauged domain walls to a pair of plaquette excitations. Such plaquette excitations coincide with fractons of the plaquette Ising model.

On the other hand, when $J = 0$, a spinon on an Ising wire is mapped to a violation of the three-body Pauli-$\mathbf{X}$ term proportional to $h$. In the deconfined $\mathbb{Z}_2$ gauge theory phase, they coincide with gauge charges/flux excitations.

With these identifications in place, we see that the image of the decoupled (1+1)D Ising symmetry breaking phase transitions on wires maps after gauging to a phase transition from the toric code to the ferromagnetic phase of the plaquette Ising model induced by the condensation of gauge charges along lines. We point out that the independent condensation of the gauge charges along vertical and horizontal lines is due to the presence of two different kinds of terms with coefficient $h$, whose excitations lie in the same superselction sector but differ via some local dressing that causes one type to be condensed along horizontal lines, and the other vertical. If one instead considers the phase transition point passing from the ferromagnetic plaquette Ising model to the toric code, one finds pairs of adjacent corner kinks are condensed along horizontal or vertical lines, orthogonal to their displacement vector.

**Order and disorder parameters**

As we've emphasized, the $K_X = K_Z = 0$ line of the phase diagram, before gauging, corresponds to the subsystem symmetry breaking transition of decoupled (1+1)D Ising wires. After gauging

Ising quilt ($K_X = K_Z = 0$ decoupled wires): ${}^1H_{\mathbb{Z}_2}^{(0)}(\mathscr{F}_{\mathrm{x}}^{\|}, \mathscr{F}_{\mathrm{y}}^{\|})$

$\mathbb{Z}_2^{(0,1)}(\mathscr{F}_{\mathrm{x}}^{\|}, \mathscr{F}_{\mathrm{y}}^{\|})$      $\mathbb{Z}_2^{(0,1)}(\mathscr{F}_{\mathrm{x}}^{\|}, \mathscr{F}_{\mathrm{y}}^{\|})$
subsystem symmetry      subsystem symmetry
preserving phase      breaking phase
$\longrightarrow J/h$

$\downarrow$ gauging ${}^{\mathrm{D}}\mathbb{Z}_2^{(0)}$

Gauged Ising quilt ($K_X = K_Z = 0$): ${}^1H_{\mathbb{Z}_2}^{(0)}(\mathscr{F}_{\mathrm{x}}^{\|}, \mathscr{F}_{\mathrm{y}}^{\|})\big/{}^{\mathrm{D}}\mathbb{Z}_2^{(0)}$

$\mathbb{Z}_2^{(0,1)}(\mathscr{F}_{\mathrm{x}}^{\|}, \mathscr{F}_{\mathrm{y}}^{\|})\big/{}^{\mathrm{D}}\mathbb{Z}_2^{(0)}$ preserved    $\mathbb{Z}_2^{(0,1)}(\mathscr{F}_{\mathrm{x}}^{\|}, \mathscr{F}_{\mathrm{y}}^{\|})\big/{}^{\mathrm{D}}\mathbb{Z}_2^{(0)}$ broken
${}^{\mathrm{D}}\widehat{\mathbb{Z}}_2^{(1)}$ broken      ${}^{\mathrm{D}}\widehat{\mathbb{Z}}_2^{(1)}$ preserved
(toric code phase)      (PIM phase)
$\longrightarrow J/h$

Figure 5: Phase diagram of the Ising quilt, before and after gauging the diagonal ${}^{\mathrm{D}}\mathbb{Z}_2^{(0)}$ subgroup of its symmetry group.

the diagonal ${}^{\mathrm{D}}\mathbb{Z}_2^{(0)}$, we have seen how this transition is mapped to one that spontaneously breaks $\mathbb{Z}_2^{(0,1)}(\mathscr{F}_{\mathrm{x}}^{\|}, \mathscr{F}_{\mathrm{y}}^{\|})/{}^{\mathrm{D}}\mathbb{Z}_2^{(0)}$ while unbreaking ${}^{\mathrm{D}}\widehat{\mathbb{Z}}_2^{(1)}$. Indeed, the symmetry preserving phase of the decoupled Ising wires maps after gauging to the toric code phase, where the ${}^{\mathrm{D}}\widehat{\mathbb{Z}}_2^{(1)}$ one-form symmetry is broken but there are no non-trivial subsystem symmetries. On the other hand, the symmetry breaking phase of the decoupled Ising wires maps at low energies to a plaquette Ising model phase, where the subsystem symmetry is spontaneously broken but the ${}^{\mathrm{D}}\widehat{\mathbb{Z}}_2^{(1)}$ symmetry acts trivially. See Figure 5. In between, there is a critical point, which our results suggest is described by a grid of (1+1)D Ising conformal field theories coupled to a (2+1)D $\mathbb{Z}_2$ gauge field. It would be interesting to try to explore this critical point further in the continuum using techniques of (1+1)D CFT.

We can obtain order/disorder parameters for this symmetry breaking transition by mapping over the order/disorder parameters of the Ising quilt (described in the previous subsection) via gauging. In the strict version of the Ising quilt, these order/disorder parameters have the same ground state expectation values as in the ungauged model. Carrying this out for the most basic parameters described in Eq. (109), we find for the order parameters that

$$
V\left(\mathscr{O}_{(i,j),r}^{\mathrm{x}}\Big/{}^{\mathrm{D}}\mathbb{Z}_2^{(0)}\right)V^{\dagger} \xrightarrow{V G_v V^{\dagger}=1} \prod_{i<k+\frac{1}{2}<i+r} \mathbf{Z}_{k+\frac{1}{2},j}\,,
$$
$$
V\left(\mathscr{O}_{(i,j),r}^{\mathrm{y}}\Big/{}^{\mathrm{D}}\mathbb{Z}_2^{(0)}\right)V^{\dagger} \xrightarrow{V G_v V^{\dagger}=1} \prod_{j<k+\frac{1}{2}<j+r} \mathbf{Z}_{i,k+\frac{1}{2}}\,,
\tag{137}
$$

and for the disorder parameters that

$$
V\left(\mathscr{D}_{(i,j),r}^{\mathrm{x}}\Big/{}^{\mathrm{D}}\mathbb{Z}_2^{(0)}\right)V^{\dagger} \xrightarrow{V G_v V^{\dagger}=1} \mathbf{X}_{i-\frac{1}{2},j}\left(\prod_{i\le k\le i+r}\mathbf{X}_{k,j}\right)\mathbf{X}_{i+r+\frac{1}{2},j}\,,
$$
$$
V\left(\mathscr{D}_{(i,j),r}^{\mathrm{y}}\Big/{}^{\mathrm{D}}\mathbb{Z}_2^{(0)}\right)V^{\dagger} \xrightarrow{V G_v V^{\dagger}=1} \mathbf{X}_{i,j-\frac{1}{2}}\left(\prod_{j\le k\le j+r}\mathbf{X}_{i,k}\right)\mathbf{X}_{i,j+r+\frac{1}{2}}\,.
\tag{138}
$$

As one can see, the disorder parameters $\mathscr{D}_{(i,j),r}^{\mu}$ are mapped again to truncated subsystem symmetry operators of the gauged Ising quilt, decorated by local operators at their endpoints.

These can therefore be thought of again as disorder parameters for the subsystem symmetry in Eq. (130). On the other hand, the order parameters $\mathcal{O}_{(i,j),r}^{\mu}$ also appear to be mapped to truncated symmetry operators, except this time they are associated to the emergent quantum one-form symmetry $^{\mathrm{D}}\widehat{\mathbb{Z}}_2^{(1)}$ from Eq. (129). Since the symmetry operators are oriented only strictly in the $\mu$ direction, they could also perhaps be thought of more specifically as disorder parameters for the zero-form subsystem subgroup $^{\mathrm{D}}\widehat{\mathbb{Z}}_2^{(0,1)}(\mathscr{F}_{\mathrm{x}}^{\parallel}, \mathscr{F}_{\mathrm{y}}^{\parallel})$ of $^{\mathrm{D}}\widehat{\mathbb{Z}}_2^{(1)}$. (Cf. §2.1 for a description of subgroups obtained by refinement of foliations.) The vacuum expectation values inherited by these operators is consistent with the claimed pattern of symmetry breaking/unbreaking.

For completeness, we also write how the order/disorder parameters from Eq. (111) map,

$$V\left(^{\mathrm{SS}}\mathcal{O}_R \middle| ^{\mathrm{D}}\mathbb{Z}_2^{(0)}\right) V^{\dagger} \xrightarrow{VG_{\nu}V^{\dagger}=1} \prod_{v\in\mathrm{corner}(R)} \mathbf{Z}_v\,,$$

$$V\left(^{\mathrm{SS}}\mathscr{D}_R^{\mathrm{x}} \middle| ^{\mathrm{D}}\mathbb{Z}_2^{(0)}\right) V^{\dagger} \xrightarrow{VG_{\nu}V^{\dagger}=1} \prod_{j_0\leq j\leq j_1} \mathbf{X}_{i_0-\frac{1}{2},j}\left(\prod_{i_0\leq i\leq i_1} \mathbf{X}_{i,j}\right)\mathbf{X}_{i_1+\frac{1}{2},j}\,, \qquad (139)$$

$$V\left(^{\mathrm{SS}}\mathscr{D}_R^{\mathrm{y}} \middle| ^{\mathrm{D}}\mathbb{Z}_2^{(0)}\right) V^{\dagger} \xrightarrow{VG_{\nu}V^{\dagger}=1} \prod_{i_0\leq i\leq i_1} \mathbf{X}_{i,j_0-\frac{1}{2}}\left(\prod_{j_0\leq j\leq j_1} \mathbf{X}_{i,j}\right)\mathbf{X}_{i,j_1+\frac{1}{2}}\,,$$

as well as the parameters from Eq. (115),

$$V\left(^{\mathrm{D}}\mathcal{O}_{\gamma} \middle| ^{\mathrm{D}}\mathbb{Z}_2^{(0)}\right) V^{\dagger} \xrightarrow{VG_{\nu}V^{\dagger}=1} \prod_{e\in\gamma} \mathbf{Z}_e\,,$$

$$V\left(^{\mathrm{D}}\mathscr{D}_R \middle| ^{\mathrm{D}}\mathbb{Z}_2^{(0)}\right) V^{\dagger} \xrightarrow{VG_{\nu}V^{\dagger}=1} \prod_{e\in\partial R} \mathbf{X}_e\,. \qquad (140)$$

Here, $\gamma$ is an open path on the lattice, the rectangle $R$ has corners $(i_0, j_0)$, $(i_0, j_1)$, $(i_1, j_0)$, and $(i_1, j_1)$, and $\partial R$ is the closed path on the dual lattice that surrounds all the vertices of $R$. Note that, for example, $^{\mathrm{SS}}\mathcal{O}_R$ is mapped to an order parameter for the spontaneous breaking of the subsystem symmetry group $\mathbb{Z}_2^{(0,1)}(\mathscr{F}_{\mathrm{x}}^{\parallel}, \mathscr{F}_{\mathrm{y}}^{\parallel}) \middle| ^{\mathrm{D}}\mathbb{Z}_2^{(0)}$. Also, it is clear that $^{\mathrm{D}}\mathcal{O}_{\gamma}$ gets mapped to a truncated symmetry operator for $^{\mathrm{D}}\widehat{\mathbb{Z}}_2^{(1)}$ (with its corners stripped of their local operators) and so may be thought of as a disorder operator for the spontaneous breaking of the emergent quantum one-form symmetry. On the other hand, $^{\mathrm{D}}\mathscr{D}_R$ is mapped to a closed line operator that is charged under the emergent quantum one-form symmetry, and so can naturally be thought of as an order parameter for its spontaneous symmetry breaking. Again, the vacuum expectation values of all these operators, supplemented with their interpretations as order/disorder parameters in the gauged theory, are consistent with the claimed pattern of symmetry breaking.

Most of this discussion generalizes straightforwardly to the case of $\mathbb{Z}_N$ where, following the analysis of Ref. [74], we expect this phase transition to become stable for sufficiently large $N$. A natural further generalization would be to study arrays of more general (1+1)D RCFTs with global symmetry coupled to bulk discrete (2+1)D gauge fields.

### 3.1.3 Dualizing the leaves

It is interesting to ask how the coupled wire construction of the previous section plays with performing a Kramers-Wannier duality transformation on each of the wires. Because of the close connection between KW duality and gauging (cf. §2.2.1), the discussion of this section could alternatively be rephrased in terms of gauging the linear subsystem symmetry of the Ising quilt. We circle back to this perspective in §3.2.

We briefly recall how KW duality acts on the (1+1)D transverse field Ising model. If we place dual qubits on the edges of the lattice (which we label by half integer coordinates) then the KW map is implemented at the level of the operator algebra by

$$
\mathbf{X}_i = \widetilde{\mathbf{Z}}_{i-\frac{1}{2}}\widetilde{\mathbf{Z}}_{i+\frac{1}{2}},
$$
$$
\mathbf{Z}_i \mathbf{Z}_{i+1} = \widetilde{\mathbf{X}}_{i+\frac{1}{2}} \implies \mathbf{Z}_i = \prod_{i' \geq i} \widetilde{\mathbf{X}}_{i'+\frac{1}{2}}. \tag{141}
$$

The TFIM then transforms as[12]

$$
-\sum_i (J\mathbf{Z}_i \mathbf{Z}_{i+1} + h\mathbf{X}_i) \longleftrightarrow -\sum_i \left( J\widetilde{\mathbf{X}}_{i+\frac{1}{2}} + h\widetilde{\mathbf{Z}}_{i-\frac{1}{2}}\widetilde{\mathbf{Z}}_{i+\frac{1}{2}} \right). \tag{142}
$$

We can consider KW dualizing, say, the TFIM wires which run along columns before coupling them together.

Similarly, the plaquette Ising model enjoys a duality with the 90° compass model,

$$
H_{\mathrm{QC}} = -\sum_{i,j} \left( \tilde{J}\widetilde{\mathbf{X}}_{i,j+\frac{1}{2}}\widetilde{\mathbf{X}}_{i+1,j+\frac{1}{2}} + \tilde{h}\widetilde{\mathbf{Z}}_{i,j-\frac{1}{2}}\widetilde{\mathbf{Z}}_{i,j+\frac{1}{2}} \right)
$$

$$
= -\sum_{i,j} \left( \tilde{J} \quad \boxed{\begin{array}{c} \widetilde{X}\bullet \quad \widetilde{X}\bullet \\ {\scriptstyle (i,j)} \end{array}} \quad + \tilde{h} \quad \boxed{\begin{array}{c} \bullet\,\widetilde{Z} \\ {\scriptstyle (i,j)} \\ \bullet\,\widetilde{Z} \end{array}} \right) \tag{143}
$$

see e.g. Refs. [98, 108, 109]. The above Hamiltonian is precisely the one obtained by applying the KW map column-by-column to the plaquette Ising model.

The XX coupling remains local under the KW map, this suggests that we should be able to obtain a coupled-wire construction of the quantum compass model as well. This intuition is correct; we can consider the columnwise KW dual of the Ising quilt,

$$
H = -\sum_{i,j} \left( J\mathbf{Z}^{\mathrm{x}}_{i,j}\mathbf{Z}^{\mathrm{x}}_{i+1,j} + h\mathbf{X}^{\mathrm{x}}_{i,j} + J\widetilde{\mathbf{X}}^{\mathrm{y}}_{i,j+\frac{1}{2}} + h\widetilde{\mathbf{Z}}^{\mathrm{y}}_{i,j-\frac{1}{2}}\widetilde{\mathbf{Z}}^{\mathrm{y}}_{i,j+\frac{1}{2}} + K_X \widetilde{\mathbf{Z}}^{\mathrm{y}}_{i,j-\frac{1}{2}}\mathbf{X}^{\mathrm{x}}_{i,j}\widetilde{\mathbf{Z}}^{\mathrm{y}}_{i,j+\frac{1}{2}} \right) \tag{144}
$$

$$
= -\sum_{i,j} \left( J \boxed{\begin{array}{c} \bullet \quad \bullet \\ \widehat{Z}\,{\scriptstyle(i,j)}\,\widehat{Z} \\ \bullet \quad \bullet \end{array}} + h \boxed{\begin{array}{c} \bullet \\ \!—\widehat{X}\!— \\ {\scriptstyle(i,j)} \\ \bullet \end{array}} + J \boxed{\begin{array}{c} \!—\!\!— \\ \widehat{X} \\ {\scriptstyle(i,j)} \end{array}} + h \boxed{\begin{array}{c} \widehat{Z} \\ \!—\bullet\!— \\ {\scriptstyle(i,j)} \\ \widehat{Z} \end{array}} + K_X \boxed{\begin{array}{c} \widehat{Z} \\ \!—\widehat{X}\!— \\ {\scriptstyle(i,j)} \\ \widehat{Z} \end{array}} \right),
$$

and proceed by computing the low energy effective Hamiltonian of this model at large $K_X$, following the same procedure discussed in the previous section.

Now, the low energy effective Hilbert space coincides with the ground space of the term proportional to $K_X$, which can be thought of as follows. If we write the full Hilbert space as

$$
\mathcal{H} = \bigotimes_{i,j} \left( \mathcal{H}^{\mathrm{x}}_{i,j} \otimes \mathcal{H}^{\mathrm{y}}_{i,j+\frac{1}{2}} \right), \qquad \left( \mathcal{H}^{\mathrm{x}}_{i,j} \cong \mathcal{H}^{\mathrm{y}}_{i,j+\frac{1}{2}} \cong \mathbb{C}^2 \right), \tag{145}
$$

and consider tensor product states in $\mathcal{H}_{\mathrm{gs}}$, once one fixes the qubits in the spaces $\mathcal{H}^{\mathrm{y}}_{i,j+\frac{1}{2}}$ to be in eigenstates of $\widetilde{\mathbf{Z}}^{\mathrm{y}}_{i,j+\frac{1}{2}}$, the states of the qubits in $\mathcal{H}^{\mathrm{x}}_{i,j}$ are completely determined: namely, we

---

[12]Here and in the rest of this section we ignore global issues. For example, the TFIM is technically not self-dual under Kramers-Wannier duality, but rather dual to a TFIM coupled to a $\mathbb{Z}_2$ gauge theory; since discrete gauge theories in (1+1)D are topological, this often neglected subtlety does not rear its head if one restricts their attention to certain local aspects of the physics, including symmetry-even correlation functions of local operators. See Ref. [86] for a careful treatment of Kramers-Wannier duality.



Figure 6: A typical state in the low energy subspace of the Hamiltonian in Eq. (144) in the $K_X \to \infty$ limit. Arrows pointing right/left indicate an eigenstate of Pauli-$\mathbf{X}$ with eigenvalue $+1/-1$. Arrows pointing up/down indicate an eigenstate of Pauli-$\mathbf{Z}$ with eigenvalue $+1/-1$. The effective qubits are simply the yellow ones.

place $\mathcal{H}^{\mathrm{x}}_{i,j}$ in an eigenstate of $\mathbf{X}^{\mathrm{x}}_{i,j}$ with eigenvalue $-1$ if there is a domain wall between the edges $(i, j - \frac{1}{2})$ and $(i, j + \frac{1}{2})$, and in a $+1$ eigenstate otherwise. Thus, we can think of the effective qubits of the low energy space as living on the vertical edges of the lattice labeled by $(i, j + \frac{1}{2})$, see Figure 6. The Pauli operators on these effective qubits can be expressed in multiple ways, e.g. as

$$
\begin{aligned}
\widetilde{\mathbf{Z}}_{i,j+\frac{1}{2}} &= \widetilde{\mathbf{Z}}^{\mathrm{y}}_{i,j+\frac{1}{2}} \\
&= \mathbf{X}^{\mathrm{x}}_{i,j} \widetilde{\mathbf{Z}}^{\mathrm{y}}_{i,j-\frac{1}{2}} = \mathbf{X}^{\mathrm{x}}_{i,j} \mathbf{X}^{\mathrm{x}}_{i,j-1} \widetilde{\mathbf{Z}}^{\mathrm{y}}_{i,j-\frac{3}{2}} = \cdots = \left( \prod_{k \le j' \le j} \mathbf{X}^{\mathrm{x}}_{i,j'} \right) \widetilde{\mathbf{Z}}^{\mathrm{y}}_{i,k-\frac{1}{2}} = \cdots \\
&= \mathbf{X}^{\mathrm{x}}_{i,j+1} \widetilde{\mathbf{Z}}^{\mathrm{y}}_{i,j+\frac{3}{2}} = \mathbf{X}^{\mathrm{x}}_{i,j+1} \mathbf{X}^{\mathrm{x}}_{i,j+2} \widetilde{\mathbf{Z}}^{\mathrm{y}}_{i,j+\frac{5}{2}} = \cdots = \left( \prod_{k \ge j' > j} \mathbf{X}^{\mathrm{x}}_{i,j'} \right) \widetilde{\mathbf{Z}}^{\mathrm{y}}_{i,k+\frac{1}{2}} = \cdots, \\
\widetilde{\mathbf{Z}}_{i,j-\frac{1}{2}} \widetilde{\mathbf{Z}}_{i,j+\frac{1}{2}} &= \widetilde{\mathbf{Z}}^{\mathrm{y}}_{i,j-\frac{1}{2}} \widetilde{\mathbf{Z}}^{\mathrm{y}}_{i,j+\frac{1}{2}} = \mathbf{X}^{\mathrm{x}}_{i,j}, \\
\widetilde{\mathbf{X}}_{i,j+\frac{1}{2}} &= \mathbf{Z}^{\mathrm{x}}_{i,j} \widetilde{\mathbf{X}}^{\mathrm{y}}_{i,j+\frac{1}{2}} \mathbf{Z}^{\mathrm{x}}_{i,j+1}.
\end{aligned}
\tag{146}
$$

If one then carries out perturbation theory to fourth order and expresses the resulting effective Hamiltonian in terms of the effective operators above, one recovers the quantum compass model in Eq. (143). It would be interesting to understand the "dimensional reduction" property of the quantum compass model [108] from the perspective of this coupled wire construction.

If one now dualizes the Ising quilt both along columns and along rows, one obtains the Hamiltonian

$$
H = -\sum_{i,j} \left( J \widetilde{\mathbf{X}}^{\mathrm{x}}_{i+\frac{1}{2},j} + h \widetilde{\mathbf{Z}}^{\mathrm{x}}_{i-\frac{1}{2},j} \widetilde{\mathbf{Z}}^{\mathrm{x}}_{i+\frac{1}{2},j} + J \widetilde{\mathbf{X}}^{\mathrm{y}}_{i,j+\frac{1}{2}} + h \widetilde{\mathbf{Z}}^{\mathrm{y}}_{i,j-\frac{1}{2}} \widetilde{\mathbf{Z}}^{\mathrm{y}}_{i,j+\frac{1}{2}} + K_X \widetilde{\mathbf{Z}}^{\mathrm{x}}_{i-\frac{1}{2},j} \widetilde{\mathbf{Z}}^{\mathrm{x}}_{i+\frac{1}{2},j} \widetilde{\mathbf{Z}}^{\mathrm{y}}_{i,j-\frac{1}{2}} \widetilde{\mathbf{Z}}^{\mathrm{y}}_{i,j+\frac{1}{2}} \right) \tag{147}
$$

$$
= -\sum_{i,j} \left( J \begin{array}{c} \includegraphics \end{array} + h \begin{array}{c} \includegraphics \end{array} + J \begin{array}{c} \includegraphics \end{array} + h \begin{array}{c} \includegraphics \end{array} + K_X \begin{array}{c} \includegraphics \end{array} \right),
$$

where we can now think of the qubits as living on the links of the lattice. The low energy subspace is again the ground space of the term on the second line. This ground space is

spanned by "closed string states". I.e. if we work in the basis of eigenstates of all the Pauli-$\mathbf{Z}$ operators acting on the links, and color links that are spin down blue, while leaving links that are spin up uncolored, the ground space consists of states for which the blue links form closed loops with no endpoints. For simplicity, let us consider the sector of Hilbert space consisting of closed string states that can be consistently thought of as domain walls.[13] Then, up to global issues, we can think of the low energy subspace as having the same degrees of freedom as a Hilbert space consisting of an effective qubit on every plaquette (whose Pauli operators we write $\widetilde{\mathbf{X}}_{i+\frac{1}{2},j+\frac{1}{2}}$, $\widetilde{\mathbf{Z}}_{i+\frac{1}{2},j+\frac{1}{2}}$), with the colored links forming domain walls separating qubits with eigenvalue $+1$ from those with eigenvalue $-1$ with respect to Pauli-$\mathbf{X}$ (see Figure 7). The logical operators of this ground space can then be expressed in terms of the original Paulis as

$$
\begin{aligned}
&\widetilde{\mathbf{Z}}_{i+\frac{1}{2},j+\frac{1}{2}} = \widetilde{\mathbf{X}}^{\mathrm{x}}_{i+\frac{1}{2},j} \widetilde{\mathbf{X}}^{\mathrm{x}}_{i+\frac{1}{2},j+1} \widetilde{\mathbf{X}}^{\mathrm{y}}_{i,j+\frac{1}{2}} \widetilde{\mathbf{X}}^{\mathrm{y}}_{i+1,j+\frac{1}{2}}\,, \\
&\widetilde{\mathbf{X}}_{i-\frac{1}{2},j-\frac{1}{2}} \widetilde{\mathbf{X}}_{i+\frac{1}{2},j-\frac{1}{2}} \widetilde{\mathbf{X}}_{i-\frac{1}{2},j+\frac{1}{2}} \widetilde{\mathbf{X}}_{i+\frac{1}{2},j+\frac{1}{2}} = \widetilde{\mathbf{Z}}^{\mathrm{x}}_{i-\frac{1}{2},j} \widetilde{\mathbf{Z}}^{\mathrm{x}}_{i+\frac{1}{2},j} = \widetilde{\mathbf{Z}}^{\mathrm{y}}_{i,j-\frac{1}{2}} \widetilde{\mathbf{Z}}^{\mathrm{y}}_{i,j+\frac{1}{2}}\,.
\end{aligned}
\tag{148}
$$

Again going to fourth order in perturbation theory recovers the following "nexus" Hamiltonian [110],

$$
\begin{aligned}
H_{\mathrm{nexus}} &= -\sum_{i,j}\left(\tilde{J}\widetilde{\mathbf{Z}}_{i+\frac{1}{2},j+\frac{1}{2}} + \tilde{h}\widetilde{\mathbf{X}}_{i-\frac{1}{2},j-\frac{1}{2}}\widetilde{\mathbf{X}}_{i+\frac{1}{2},j-\frac{1}{2}}\widetilde{\mathbf{X}}_{i-\frac{1}{2},j+\frac{1}{2}}\widetilde{\mathbf{X}}_{i+\frac{1}{2},j+\frac{1}{2}}\right) \\
&= -\sum_{i,j}\left(\tilde{J}\ \boxed{\substack{\widetilde{\mathbf{Z}}}} + \tilde{h}\ \boxed{\substack{\widetilde{\mathbf{X}}\ \widetilde{\mathbf{X}} \\ \widetilde{\mathbf{X}}\ \widetilde{\mathbf{X}}}}\right),
\end{aligned}
\tag{149}
$$

which is precisely the Hamiltonian obtained by KW dualizing the plaquette Ising model along columns and rows.

This entire discussion can be summarized by the "commuting diagram" in Figure 8.

## 3.2 Gauged subsystem symmetry enriched anyon models

In the previous section, we considered a somewhat unconventional matter content which trivially realized subsystem symmetries—i.e. decoupled wires, each with an ordinary global symmetry—and subjected it to a conventional gauging procedure to bring it into a more interesting phase. In this section, we show that the reverse is also effective: that is, we start with a conventional matter theory and couple it to an unconventional gauge theory.

More specifically, our matter theory is a (2+1)D theory with a one-form symmetry group $G^{(1)}$. By definition, such a theory admits, for each group element $g \in G$ and path $\gamma$ through the system, a symmetry operator $U_\gamma(g)$ that commutes with the Hamiltonian. By restricting attention to the symmetry operators associated to rows and columns of the system, the one-form symmetry can be thought of as having a linear subsystem symmetry subgroup $G^{(0,1)}(\mathscr{F}^{\parallel}_{\mathrm{x}}, \mathscr{F}^{\parallel}_{\mathrm{y}})/\sim$. We consider the theory to be enriched by this subsystem symmetry subgroup, so that we allow ourselves to deform the Hamiltonian by terms which commute with $G^{(0,1)}(\mathscr{F}^{\parallel}_{\mathrm{x}}, \mathscr{F}^{\parallel}_{\mathrm{y}})/\sim$.

We may then consider gauging this subsystem symmetry. As reviewed in §2, just as gauging an ordinary Abelian symmetry $G^{(0)}$ in (1+1)D leads to an emergent quantum symmetry $\widehat{G}^{(0)}$ with $\widehat{G} \cong G$ (see e.g. Ref. [78]), gauging a discrete, Abelian, linear subsystem symmetry $G^{(0,1)}/\sim$ in (2+1)D leads to an emergent quantum subsystem symmetry $\widehat{G}^{(0,1)}/\sim$, where again

---

[13]Note that e.g. on a torus, not every string state corresponds to a domain wall configuration: for example, a state with a string stretching along one of the cycles of the torus.

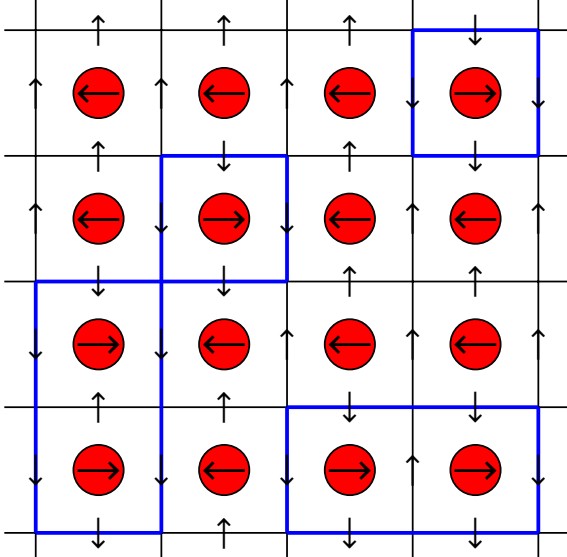

Figure 7: A typical state in the low energy subspace of the Hamiltonian in Eq. (147) in the $K_X \to \infty$ limit. Arrows pointing right/left indicate an eigenstate of Pauli-**X** with eigenvalue $+1/-1$. Arrows pointing up/down indicate an eigenstate of Pauli-**Z** with eigenvalue $+1/-1$. The effective qubits are the red ones. Unlike in Figure 6, the effective qubits do not coincide with any physical qubits of the Hamiltonian in Eq. (147).

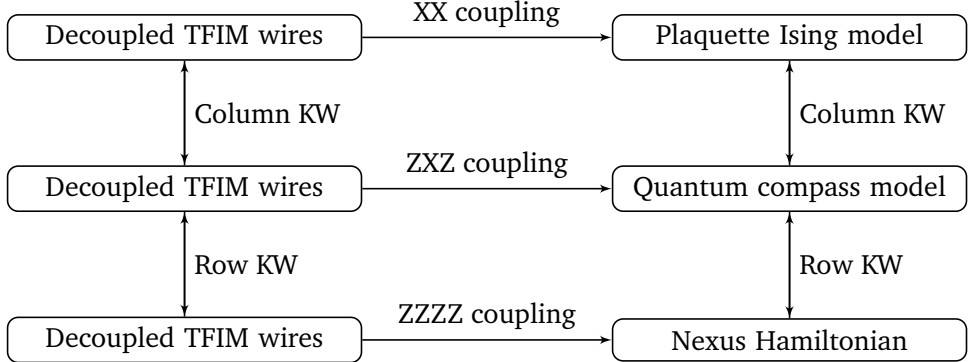

Figure 8: A commuting diagram summarizing how our coupled wire constructions play with dualities.

$G \cong \widehat{G}$. Therefore, our subsystem gauge theory, together with the one-form symmetric matter to which it is coupled, enjoys a quantum subsystem symmetry group $\widehat{G}^{(0,1)}/\sim$; if the theory before gauging possesses a non-trivial phase diagram, so too does the theory after gauging, and so we may hope to obtain a non-trivial theory in this way.

### 3.2.1 A subsystem symmetry enriched (2+1)D toric code

As a first pass, let us try testing this idea in the particular case of $G^{(1)} = \mathbb{Z}_2^{(1)}$ by taking the matter theory to be (2+1)D lattice gauge theory,

$$
\begin{aligned}
{}^2H_{\mathbb{Z}_2}^{(1)} &= -U \sum_e \mathbf{X}_e - t \sum_p \prod_{e \in p} \mathbf{Z}_e \,, \\
G_v &= \prod_{e \ni v} \mathbf{X}_e \,,
\end{aligned}
\tag{150}
$$

where above, $G_v = 1$ is the Gauss's law constraint. As reviewed in §2.2.2, this model has a $\mathbb{Z}_2^{(1)}$ one-form symmetry group that is generated by string operators of the form

$$
U_{\tilde{\gamma}} = \prod_{e \in \tilde{\gamma}} \mathbf{X}_e \,,
\tag{151}
$$

where $\tilde{\gamma}$ is a closed loop on the dual lattice, and the product is over edges which are perpendicularly bisected by $\tilde{\gamma}$. Following the discussion of the previous paragraph, these symmetry generators admit a zero-form linear subsystem symmetry subgroup $\mathbb{Z}_2^{(0,1)}(\mathscr{F}_{\mathrm{x}}^{\parallel}, \mathscr{F}_{\mathrm{y}}^{\parallel})$ which we are interested in gauging, generated by

$$
U_{j+\frac{1}{2}}^{\mathrm{x}} = \prod_i \mathbf{X}_{i,j+\frac{1}{2}} \,, \qquad U_{i+\frac{1}{2}}^{\mathrm{y}} = \prod_j \mathbf{X}_{i+\frac{1}{2},j} \,.
\tag{152}
$$

Since we are gauging just this subsystem subgroup, we could in principle allow ourselves to deform the model by terms that respect $\mathbb{Z}_2^{(0,1)}(\mathscr{F}_{\mathrm{x}}^{\parallel}, \mathscr{F}_{\mathrm{y}}^{\parallel})$, but not the full $\mathbb{Z}_2^{(1)}$. However, the one-form symmetry is a consequence of Gauss's law: so long as we are imposing Gauss's law as a constraint on the Hilbert space, the Hamiltonian can only include terms that commute with it, and it is therefore not possible to add any terms which break the one-form symmetry.

Our solution is to simply work in the extended Hilbert space and drop the Gauss's law constraint. We could choose to impose it energetically, however this is not necessary for our goals. Instead, to avoid a proliferation of terms, we simply perturb by the lowest order term that breaks the one-form symmetry down to its linear subsystem subgroup. The model this leads to is

$$
H_{\mathrm{PTC}} = -U \sum_e \mathbf{X}_e - t \sum_p \prod_{e \in p} \mathbf{Z}_e - h \sum_{i,j} \left( \mathbf{Z}_{i,j+\frac{1}{2}} \mathbf{Z}_{i+1,j+\frac{1}{2}} + \mathbf{Z}_{i+\frac{1}{2},j} \mathbf{Z}_{i+\frac{1}{2},j+1} \right) \,,
\tag{153}
$$

where above, we are alternating between labeling edges by $e$ or by $(i + \frac{1}{2}, j)/(i, j + \frac{1}{2})$ as convenient. We refer to this as a "perturbed toric code". We think of it as being an example of a more general class of "deformed anyon models", by which we mean theories of anyons perturbed by terms that break their one-form symmetries (associated to abelian anyons) down to linear subsystem subgroups.

**Phases and excitations**

We note that if one switches perspectives to the dual lattice, then the Hamiltonian Eq. (153) becomes precisely the model in Eq. (147), i.e. the theory one obtains by performing Kramers-Wannier duality along all the rows and columns of the Ising quilt. We see shortly that this is not an accident. An overview of the phase diagram is then easy to infer. At large $t$, the model is in a nexus Hamiltonian phase. When $t$ is small, it is in a phase that is essentially described by decoupled Ising wires.

**Order and disorder parameters**

Let us focus on the model along the $h = 0$ locus of parameter space. In this case, the phase transition as one tunes the competition between $U$ and $t$ is more or less the confinement/deconfinement phase transition of $\mathbb{Z}_2$ lattice gauge theory (though a version where the Gauss's law constraint is not imposed). Our focus is on this transition. A natural set of order/disorder parameters that diagnose this transition were reviewed in §2.2.2. They are

$$\mathcal{O}_\gamma = \prod_{e \in \gamma} \mathbf{Z}_e \,, \qquad \mathcal{D}_{\tilde{\gamma}(p_i, p_f)} = \prod_{e \in \tilde{\gamma}} \mathbf{X}_e \,, \tag{154}$$

where $\tilde{\gamma}(p_i, p_f)$ is a path on the dual lattice whose endpoints are the plaquettes $p_i$ and $p_f$.

In the present context, we note that $\mathcal{D}_{\tilde{\gamma}(p_i, p_f)}$, if $\tilde{\gamma}(p_i, p_f)$ is oriented just along a single row/column, can be thought of as a truncated symmetry operator for the subsystem symmetry, and can therefore serve as a disorder parameter for this symmetry. Its vacuum expectation value, which transitions from being long-ranged to decaying with the separation of $p_i$ and $p_f$, is consistent with the subsystem symmetry being broken as $U$ is lowered. Consider instead the product of these disorder parameters over two neighboring rows. Here, in contrast to the case of a single row, it is expected that the vacuum expectation value is always long-ranged, even in the toric code phase. This suggests that the symmetry elements corresponding to the product of subsystem symmetry generators over pairs of rows are unbroken rather than broken. Indeed, this is sensible because, in the toric code phase, it is known that wrapping a string operator around a non-trivial cycle acts non-trivially on the ground state, while wrapping two string operators acts trivially.

### 3.2.2 Gauging the linear zero-form subsystem subgroup

We can then gauge the subsystem symmetry of this perturbed toric code as follows. We introduce two qubits on each plaquette of the lattice $\mathbf{X}_p^\mu$, $\mathbf{Z}_p^\mu$ for $\mu = \mathrm{x}, \mathrm{y}$; these are the subsystem gauge field degrees of freedom. The local gauge transformation (i.e. Gauss's law constraint) is implemented by the operators

$$G_{i, j+\frac{1}{2}} = \mathbf{X}_{i-\frac{1}{2}, j+\frac{1}{2}}^\mathrm{x} \mathbf{X}_{i, j+\frac{1}{2}} \mathbf{X}_{i+\frac{1}{2}, j+\frac{1}{2}}^\mathrm{x} \,, \qquad G_{i+\frac{1}{2}, j} = \mathbf{X}_{i+\frac{1}{2}, j-\frac{1}{2}}^\mathrm{y} \mathbf{X}_{i+\frac{1}{2}, j} \mathbf{X}_{i+\frac{1}{2}, j+\frac{1}{2}}^\mathrm{y} \,, \tag{155}$$

where we are labeling plaquettes by tuples of half integers. We can then suitably "covariantize" the terms of $H_{\mathrm{PTC}}$ to obtain a "subsystem gauged toric code"

$$\begin{aligned}
H_{\mathrm{PTC}}\Big/ \mathbb{Z}_2^{(0,1)}(\mathscr{F}_\mathrm{x}^\parallel, \mathscr{F}_\mathrm{y}^\parallel) = &-U \sum_e \mathbf{X}_e - t \sum_p \mathbf{Z}_p^\mathrm{x} \mathbf{Z}_p^\mathrm{y} \prod_{e \in p} \mathbf{Z}_e \\
&- h \sum_{i,j} \left( \mathbf{Z}_{i, j+\frac{1}{2}} \mathbf{Z}_{i+\frac{1}{2}, j+\frac{1}{2}}^\mathrm{x} \mathbf{Z}_{i+1, j+\frac{1}{2}} + \mathbf{Z}_{i+\frac{1}{2}, j} \mathbf{Z}_{i+\frac{1}{2}, j+\frac{1}{2}}^\mathrm{y} \mathbf{Z}_{i+\frac{1}{2}, j+1} \right) \,.
\end{aligned} \tag{156}$$

Note that there are simply no non-trivial flux terms to consider adding into the mix.

As we have done in previous sections, we now simplify this model by solving the Gauss's law constraint. To make this easier, we first transform the model using the following local unitary circuit,

$$V = H^{\otimes N} \prod_{i,j} \left( C_{i+1, j+\frac{1}{2}} \mathbf{X}_{i+\frac{1}{2}, j+\frac{1}{2}}^\mathrm{x} \right) \left( C_{i, j+\frac{1}{2}} \mathbf{X}_{i+\frac{1}{2}, j+\frac{1}{2}}^\mathrm{x} \right) \left( C_{i+\frac{1}{2}, j+1} \mathbf{X}_{i+\frac{1}{2}, j+\frac{1}{2}}^\mathrm{y} \right) \left( C_{i+\frac{1}{2}, j} \mathbf{X}_{i+\frac{1}{2}, j+\frac{1}{2}}^\mathrm{y} \right) \,, \tag{157}$$

where $H^{\otimes N}$ is a Hadamard which rotates all Pauli-$\mathbf{X}$ operators into Pauli-$\mathbf{Z}$ operators, and $C_e \mathbf{X}_p^\mu$ is a CNOT gate.

After applying this to the Gauss's law constraint, it simply becomes $G_e = \mathbf{Z}_e$ so that the qubits on the edges are frozen to $+1$ Pauli-$\mathbf{Z}$ eigenstates. Therefore, after switching perspectives to the dual lattice, the Hamiltonian we're left with is

$$
\begin{aligned}
V\left(H_{\text{PTC}}\middle/\mathbb{Z}_2^{(0,1)}(\mathscr{F}_x^{\parallel}, \mathscr{F}_y^{\parallel})\right)V^{\dagger} &\xrightarrow{VG_eV^{\dagger}=1} \\
&-\sum_{\tilde{i},\tilde{j}}\left(U(\mathbf{Z}_{\tilde{i},\tilde{j}}^x\mathbf{Z}_{\tilde{i}+1,\tilde{j}}^x + \mathbf{Z}_{\tilde{i},\tilde{j}}^y\mathbf{Z}_{\tilde{i},\tilde{j}+1}^y) + h(\mathbf{X}_{\tilde{i},\tilde{j}}^x + \mathbf{X}_{\tilde{i},\tilde{j}}^y) + t\mathbf{X}_{\tilde{i},\tilde{j}}^x\mathbf{X}_{\tilde{i},\tilde{j}}^y\right) \cong \widetilde{H}_{\text{IQ}},
\end{aligned}
\tag{158}
$$

where here, $(\tilde{i}, \tilde{j})$ are coordinatizing sites of the dual lattice. Thus, we precisely find that the gauged perturbed toric code is unitarily equivalent to the Ising quilt, Eq. (93), on the dual lattice! Indeed, the subsystem symmetry group of the Ising quilt arises here as the emergent quantum symmetry group $\widehat{\mathbb{Z}}_2^{(0,1)}(\mathscr{F}_x^{\parallel}, \mathscr{F}_y^{\parallel})$ guaranteed by the gauging procedure. When $h = 0$, the symmetry group of $H_{\text{PTC}}$ enhances from a subsystem symmetry to a one-form symmetry, and one can ask what remains of this enhanced symmetry in the gauged model. To answer this, one can simply study how the symmetry operators from Eq. (151) map under the gauging, where one finds

$$
VU_{\tilde{\gamma}}V^{\dagger} \xrightarrow{VG_eV^{\dagger}=1} \prod_{\tilde{v}\in\text{corner}(\tilde{\gamma})} \mathbf{Z}_{\tilde{v}}^x\mathbf{Z}_{\tilde{v}}^y.
\tag{159}
$$

In a local Hamiltonian, being symmetric under $VU_{\tilde{\gamma}}V^{\dagger}$ for any $\tilde{\gamma}$ implies in fact that the Hamiltonian commutes with the operators $\mathbf{Z}_{\tilde{v}}^x\mathbf{Z}_{\tilde{v}}^y$ for any $\tilde{v}$. These are precisely the local conservation laws that were leveraged to make contact with the disordered transverse field Ising model at the very end of §3.1.1. One could perhaps think of this as a point-like subsystem symmetry $\mathbb{Z}_2^{(0,2)}$. See e.g. Ref. [53] for a $U(1)$ example of a point-like subsystem symmetry in a continuum setting.

Recall that $H_{\text{PTC}}$ was identical (after switching to the dual lattice) to the Hamiltonian one obtains after performing Kramers–Wannier duality on all the rows and columns of the Ising quilt. Here, we have found that gauging the $\mathbb{Z}_2^{(0,1)}(\mathscr{F}_x^{\parallel}, \mathscr{F}_y^{\parallel})$ subsystem symmetry has lead back to the Ising quilt. We could have anticipated this beforehand because for this class of models, subsystem gauging implements row-wise/column-wise KW duality, much in the same way as we found that gauging the global $\mathbb{Z}_2^{(0)}$ of the (1+1)D TFIM implemented KW duality in §2.2.1. Thus, the manipulations of this section are more or less the same as those of §3.1.3, only described in different words.

One intriguing upshot of this is that if we set $h = 0$, then we find that what is essentially the confinement/deconfinement phase transition of the toric code (though without imposing Gauss's law) maps after gauging $\mathbb{Z}_2^{(0,1)}$ to the k-string condensation transition that we discussed in §3.1. Furthermore, the order/disorder parameters from Eq. (154) map under gauging the subsystem symmetry to the disorder/order parameters from Eq. (115), respectively.

## 3.3 The point-string-net model

We now unify the models of the previous two subsections into a single parent model, which we call the *point-string-net model*. It is defined on a Hilbert space consisting of two qubits per site and one qubit per edge of a square lattice

$$
\mathcal{H} = \bigotimes_{i,j}\left(\mathcal{H}_{i,j}^x \otimes \mathcal{H}_{i,j}^y \otimes \mathcal{H}_{i+\frac{1}{2},j} \otimes \mathcal{H}_{i,j+\frac{1}{2}}\right),
\tag{160}
$$

where

$$
\mathcal{H}_{i,j}^x \cong \mathcal{H}_{i,j}^y \cong \mathcal{H}_{i+\frac{1}{2},j} \cong \mathcal{H}_{i,j+\frac{1}{2}} \cong \mathbb{C}^2.
\tag{161}
$$

The perturbed point-string-net Hamiltonian is

$$
\begin{aligned}
H_{\mathrm{PPSN}} = {} & -\Delta \sum_v \mathbf{X}_v^{\mathrm{x}} \mathbf{X}_v^{\mathrm{y}} \prod_{e \ni v} \mathbf{X}_e - \lambda \sum_e \mathbf{Z}_e \\
& - \gamma \sum_{i,j} \Big( \mathbf{X}_{i-\frac{1}{2},j} \mathbf{X}_{i,j}^{\mathrm{x}} \mathbf{X}_{i+\frac{1}{2},j} + \mathbf{X}_{i,j-\frac{1}{2}} \mathbf{X}_{i,j}^{\mathrm{y}} \mathbf{X}_{i,j+\frac{1}{2}} \Big) \\
& - \Delta' \sum_{i,j} \Big( \mathbf{Z}_{i,j}^{\mathrm{x}} \mathbf{Z}_{i+\frac{1}{2},j} \mathbf{Z}_{i+1,j}^{\mathrm{x}} + \mathbf{Z}_{i,j}^{\mathrm{y}} \mathbf{Z}_{i,j+\frac{1}{2}} \mathbf{Z}_{i,j+1}^{\mathrm{y}} \Big) \\
& - \sum_v \Big( \lambda' \mathbf{X}_v^{\mathrm{x}} \mathbf{X}_v^{\mathrm{y}} + \kappa' \mathbf{Z}_v^{\mathrm{x}} \mathbf{Z}_v^{\mathrm{y}} + \gamma' \big( \mathbf{X}_v^{\mathrm{x}} + \mathbf{X}_v^{\mathrm{y}} \big) \Big) .
\end{aligned}
\tag{162}
$$

This Hamiltonian respects a $\mathbb{Z}_2^{(0,1)}(\mathscr{F}_{\mathrm{x}}^{\parallel}, \mathscr{F}_{\mathrm{y}}^{\parallel})$ linear zero-form subsystem symmetry generated by $\mathbf{Z}_e$ operators along straight lines through the lattice,

$$
U_j^{\mathrm{x}} = \prod_i \mathbf{Z}_{i+\frac{1}{2},j}, \qquad U_i^{\mathrm{y}} = \prod_j \mathbf{Z}_{i,j+\frac{1}{2}} .
\tag{163}
$$

When the perturbation strength $\gamma$ is set to 0, this $\mathbb{Z}_2^{(0,1)}(\mathscr{F}_{\mathrm{x}}^{\parallel}, \mathscr{F}_{\mathrm{y}}^{\parallel})$ is enhanced to a more conventional $\mathbb{Z}_2^{(1)}$ one-form symmetry group whose symmetry operators are supported on arbitrary loops $\gamma$ through the lattice,

$$
U_\gamma^{(1)} = \prod_{e \in \gamma} \mathbf{Z}_e ,
\tag{164}
$$

as opposed to just on straight lines. (Cf. the discussion about subgroups obtained by refining foliations in §2.1.)

The perturbed point-string-net model also respects an ordinary global $\widetilde{\mathbb{Z}}_2^{(0)}$ symmetry obtained by flipping all vertex spins,

$$
\widetilde{U} = \prod_v \mathbf{X}_v^{\mathrm{x}} \mathbf{X}_v^{\mathrm{y}} .
\tag{165}
$$

When $\kappa' = 0$, this symmetry enlarges to another $\widetilde{\mathbb{Z}}_2^{(0,1)}(\mathscr{F}_{\mathrm{x}}^{\parallel}, \mathscr{F}_{\mathrm{y}}^{\parallel})$ linear subsystem symmetry group whose generators are

$$
\widetilde{U}_j^{\mathrm{x}} = \prod_i \mathbf{X}_{i,j}^{\mathrm{x}}, \qquad \widetilde{U}_i^{\mathrm{y}} = \prod_j \mathbf{X}_{i,j}^{\mathrm{y}} .
\tag{166}
$$

(Cf. the discussion about subgroups obtained by coarsening foliations in §2.1.)

There is also a local unitary duality that acts on the model by exchanging $\Delta \leftrightarrow \lambda'$, $\Delta' \leftrightarrow \lambda$, and $\gamma \leftrightarrow \gamma'$, which is given by the circuit

$$
\widetilde{V} = \prod_{i,j} C_{i,j}^{\mathrm{x}} \mathbf{X}_{i-\frac{1}{2},j} \, C_{i,j}^{\mathrm{x}} \mathbf{X}_{i+\frac{1}{2},j} \, C_{i,j}^{\mathrm{y}} \mathbf{X}_{i,j-\frac{1}{2}} \, C_{i,j}^{\mathrm{y}} \mathbf{X}_{i,j+\frac{1}{2}} .
\tag{167}
$$

This circuit is essentially the "entangler" for the $(1 + 1)$D symmetry-protected cluster phase [111] along each row and column.

As we mentioned above, the purpose of introducing the point-string-net Hamiltonian is to collect the constructions explored in §3.1-§3.2 into a single model. The limits in which these previous constructions are recovered are the following.

- In the $\lambda \to \infty$ limit the edge degrees of freedom are pinned into the $|\uparrow\rangle$ state of Pauli-$\mathbf{Z}$, and the $\Delta, \gamma$ terms in Eq. (162) are projected out (because they do not commute with

the term proportional to $\lambda$). The remaining terms reduce to the Ising quilt, which was explored in §3.1.1,

$$H_{\text{PPSN}} \xrightarrow{\lambda \to \infty} H_{\text{IQ}}, \tag{168}$$

(cf. Eq. (93)). This is equivalent to the limit $\Delta' \to \infty$ under $\widetilde{V}$, which corresponds to a perturbed coupled cluster chain phase, and reproduces the cluster chain construction of the toric code from Ref. [112] as $\lambda', \kappa' \to \infty$.

- In the limit that $\gamma \to 0$ and $\Delta \to \infty$ with $\lambda$ small, the Hamiltonian reduces to the gauged Ising quilt, Eq. (121). In this limit, the term proportional to $\Delta$ enforces the Gauss's law constraint, and the plaquette flux terms $\prod_{e \in p} \mathbf{Z}_e$ of the gauged Ising quilt are generated at leading order in perturbation theory in $\lambda$. Thus,

$$H_{\text{PPSN}} \xrightarrow[\substack{\gamma \to 0 \\ \Delta \to \infty \\ \lambda \text{ small}}]{} H_{\text{IQ}}\Big|_E^{\text{D}}\mathbb{Z}_2. \tag{169}$$

- In the $\gamma' \to \infty$ limit the vertex degrees of freedom are pinned into the $|+\rangle$ state of Pauli-$\mathbf{X}$ and the $\Delta', \kappa'$ terms are projected out. After switching perspectives to the dual lattice and performing a Hadamard rotation $H^{\otimes N}$ which swaps all Pauli-$\mathbf{X}$ operators with Pauli-$\mathbf{Z}$ operator, the remaining Hamiltonian reduces to the deformed toric code considered in §3.2.1 (cf. Eq. (153)),

$$H_{\text{PPSN}} \xrightarrow[\gamma' \to \infty]{H^{\otimes N}} H_{\text{PTC}}. \tag{170}$$

This is equivalent to the limit $\gamma \to \infty$ under $\widetilde{V}$, which coincides with a perturbed subsystem symmetry-enriched toric code phase [85] after additionally taking $\lambda' \to \infty$.

- For $\lambda', \gamma', \kappa' \to 0$, after applying a Hadamard rotation $H^{\otimes N}$ and switching perspectives to the dual lattice, the Hamiltonian reduces to a deformed toric code model with its zero-form linear subsystem symmetry gauged, as we considered in §3.2.2,

$$H_{\text{PPSN}} \xrightarrow[\substack{\Delta' \to \infty \\ \lambda', \gamma', \kappa' \to 0}]{H^{\otimes N}} H_{\text{PTC}}\Big/\mathbb{Z}_2^{(0,1)}\left(\mathscr{F}_{\text{x}}^{\parallel}, \mathscr{F}_{\text{y}}^{\parallel}\right) \tag{171}$$

(cf. Eq. (156)). Here, the term proportional to $\Delta'$ corresponds to Gauss's law (cf. Eq. (155)), which becomes a strict constraint on the Hilbert space in the $\Delta' \to \infty$ limit.

The zero correlation length point-string-net model is recovered in the limit $\Delta, \Delta' \to \infty$ and $\kappa' \to 0$, which yields

$$H_{\text{PSN}} = -\sum_v \mathcal{A}_v - \sum_p \mathcal{B}_p - \sum_e \mathcal{C}_e, \tag{172}$$

where we have defined

$$\mathcal{A}_v = \mathbf{X}_v^{\text{x}} \mathbf{X}_v^{\text{y}} \prod_{e \ni v} \mathbf{X}_e, \qquad \mathcal{B}_p = \prod_{e \in p} \mathbf{Z}_e,$$

$$\mathcal{C}_{i+\frac{1}{2},j} = \mathbf{Z}_{i,j}^{\text{x}} \mathbf{Z}_{i+\frac{1}{2},j} \mathbf{Z}_{i+1,j}^{\text{x}}, \qquad \mathcal{C}_{i,j+\frac{1}{2}} = \mathbf{Z}_{i,j}^{\text{y}} \mathbf{Z}_{i,j+\frac{1}{2}} \mathbf{Z}_{i,j+1}^{\text{y}}. \tag{173}$$

For simplicity of presentation we have kept only the leading order plaquette terms generated by products of $\mathbf{Z}_{i+\frac{1}{2},j}$ and $\mathbf{Z}_{i,j+\frac{1}{2}}$ terms from the above Hamiltonian. In addition, we have

rescaled the coupling strengths to 1. Both of these redefinitions preserve the phase of matter as the Hamiltonian consists of commuting projector terms (up to an overall rescaling of the energy).

This model can equally be viewed as ferromagnetic Ising wires with their diagonal global symmetry gauged, or as toric code model with its linear subsystem symmetry gauged. Excitations of the plaquette term are equivalent to corner domain walls of the ferromagnetic plaquette Ising model, as shown below. An excitation of an edge term is equivalent to a pair of plaquette excitations, as all three are created by an $\mathbf{X}_e$ operator. Excitations of the star term are seen to be trivial below.

The point-string-net model is equivalent to the ferromagnetic phase of the plaquette Ising (Xu-Moore) model [96, 97]. Hence the full perturbed point-string net model above can be understood as exploring the phase diagram proximate to the ferromagentic plaquette Ising model, including the limit of decoupled Ising wires and the toric code. To demonstrate the equivalence we apply the local unitary circuit $\widetilde{V}$ introduced in Eq. (167), followed by the on site transformation

$$V = \prod_v C_v^x \mathbf{X}_v^y, \tag{174}$$

to the point-string-net Hamiltonian to find

$$V\widetilde{V}H_{\mathrm{PSN}}\widetilde{V}^\dagger V^\dagger = -\sum_v \mathbf{X}_v^x - \sum_p \prod_{v\in p} \mathbf{Z}_v^y \prod_{e\in p} \mathbf{Z}_e - \sum_e \mathbf{Z}_e. \tag{175}$$

Each plaquette term can be further modified to $-\prod_{v\in p}\mathbf{Z}_v^y$ by multiplication with single body $\mathbf{Z}_e$ terms from the Hamiltonian without effecting the phase of matter. Finally, the qubits on the $\mathcal{H}_v^x$ vertices and edges are decoupled in $|+\rangle$ and $|\uparrow\rangle$ states, respectively, resulting in the ferromagnetic plaquette Ising model

$$V\widetilde{V}H_{\mathrm{PSN}}\widetilde{V}^\dagger V^\dagger \sim -\sum_p \prod_{v\in p} \mathbf{Z}_v. \tag{176}$$

The point-sting-net picture for $H_{\mathrm{PSN}}$ is obtained by interpreting $|+\rangle$ states as empty, $|-\rangle$ states on vertices as points (red if $\mathbf{X}_v^x = -\mathbf{X}_v^y = -1$, blue if $\mathbf{X}_v^y = -\mathbf{X}_v^x = -1$, and purple if $\mathbf{X}_v^x = \mathbf{X}_v^y = -1$), and $|-\rangle$ states on edges as black string segments. The term $\mathcal{A}_v$ then energetically enforce a $\mathbb{Z}_2$ parity constraint on the total number of points and strings incident at vertex $v$. The term $\mathcal{B}_p$ fuses a closed $\mathbb{Z}_2$ string bordering $p$ into the lattice, while $\mathcal{C}_e$ fuses an open $\mathbb{Z}_2$ string segment, ending on a pair of points, into the lattice. The ground states of the point-string-net model can then be understood as point-string-net wavefunctions that satisfy the $\mathbb{Z}_2$ parity constraint at each vertex, and that involve a uniform sum over all ways of fusing in closed strings or horizontal/vertical open string segments into a fixed reference state that satisfies the $\mathbb{Z}_2$ parity constraint at each vertex. For example, one such ground state is

$$|\Psi\rangle \propto \prod_e (1 + \mathcal{C}_e) \prod_p (1 + \mathcal{B}_p) |\Psi_0\rangle$$

$$\propto \left|\begin{array}{c}\end{array}\right\rangle + \left|\begin{array}{c}\end{array}\right\rangle + \left|\begin{array}{c}\end{array}\right\rangle + \left|\begin{array}{c}\end{array}\right\rangle + \cdots \tag{177}$$

where here the reference state is

$$|\Psi_0\rangle = \mathbf{Z}_v^x \mathbf{Z}_v^y |\Omega\rangle, \tag{178}$$

with $|\Omega\rangle$ the completely empty state, and $v$ the vertex colored purple in the first term in Eq. (177). Note that this state is different than the ground state one would obtain by choosing the reference state to be $|\Omega\rangle$ since there is no way to obtain $|\Psi_0\rangle$ from $|\Omega\rangle$ by fusing in closed/open string segments.

# 4 (3+1)D Cubic Ising Transitions

In this section, we focus on constructions of phase diagrams involving the cubic Ising model. In §4.1, we demonstrate how coupled wire constructions are well-suited to this task. In §4.2, we combine the various theories considered in §4.1 as well as the X-cube model into a single parent Hamiltonian that we call the *point-cage-net model*.

As in the previous section, we expect that all the constructions we present here can be generalized to $\mathbb{Z}_N$ and beyond.

## 4.1 Coupled wire constructions

Consider 3 orthogonal foliations of 3-dimensional space—$\mathscr{F}_x^{\|}$, $\mathscr{F}_y^{\|}$, and $\mathscr{F}_z^{\|}$—each consisting of one-dimensional leaves. The wires of $\mathscr{F}_x^{\|}$ stretch in the x direction and are located at fixed positions in the y and z directions, and similarly for $\mathscr{F}_y^{\|}$ and $\mathscr{F}_z^{\|}$. We then imagine placing (1+1)D transverse field Ising models on the leaves of each of these foliations to produce a Hamiltonian ${}^1H_{\mathbb{Z}_2}^{(0)}(\mathscr{F}_x^{\|}, \mathscr{F}_y^{\|}, \mathscr{F}_z^{\|})$. Such decoupled Ising models enjoy a linear zero-form sub-system symmetry, i.e. a $\mathbb{Z}_2^{(0,2)}(\mathscr{F}_x^{\|}, \mathscr{F}_y^{\|}, \mathscr{F}_z^{\|})$ symmetry structure, which we would like to identify with the linear subsystem symmetries of the cubic Ising model

$$ {}^3H_{\mathbb{Z}_2}^{(0,2)} = -h\sum_v \mathbf{X}_v - J\sum_c \prod_{v\in c} \mathbf{Z}_v\,, \tag{179} $$

which are generated by the operators

$$ U_{j,k}^x = \prod_i \mathbf{X}_{i,j,k}\,, \quad U_{i,k}^y = \prod_j \mathbf{X}_{i,j,k}\,, \quad U_{i,j}^z = \prod_k \mathbf{X}_{i,j,k}\,. \tag{180} $$

However, the symmetries of the cubic Ising model enjoy certain relations that are not satisfied by the symmetries of decoupled wires. In particular, there is a planar zero-form subsystem symmetry subgroup

$$ \mathscr{H} = \mathbb{Z}_2^{(0,1)}(\mathscr{F}_{xy}^{\|}, \mathscr{F}_{xz}^{\|}, \mathscr{F}_{yz}^{\|})/\sim = \left\langle \prod_i U_{i,k}^y \prod_j U_{j,k}^x, \ \prod_k U_{i,k}^y \prod_j U_{i,j}^z, \ \prod_k U_{j,k}^x \prod_i U_{i,j}^z \right\rangle, \tag{181} $$

which acts trivially in the cubic Ising model, but faithfully in the decoupled wires. (Here, $\mathscr{F}_{xy}^{\|}$ is the foliation of 3d space by planes which span the xy directions, and similarly for $\mathscr{F}_{xz}^{\|}$ and $\mathscr{F}_{yz}^{\|}$.)

As in §3.1, there are two ways to fix this discrepancy. In §4.1.1, we show that it is possible to directly couple the wires together strongly enough that they are driven to a phase where the planar subgroup $\mathscr{H}$ is unbroken; we then find that the low energy effective Hamiltonian in this phase is the cubic Ising model. In §4.1.2, we proceed instead by simply gauging the the planar subgroup, in which case we force it to act trivially by construction.

### 4.1.1 Coupling a grid of transverse field Ising wires

A discretized version of the setup described above can be achieved by considering an $N_x \times N_y \times N_z$ 3d cubic lattice, and placing 3 qubits on each site $v = (i,j,k)$, each qubit belonging to a leaf of one of the three foliations. We use $\mathbf{X}_v^x, \mathbf{Z}_v^x$ to denote the Pauli operators acting on the qubit at vertex $v$ belonging to the foliation $\mathscr{F}_x^{\|}$, and similarly for the other two foliations. The Hamil-

tonian of a decoupled grid of transverse field Ising wires is then

$${}^{1}H^{(0)}_{\mathbb{Z}_2}(\mathscr{F}^{\parallel}_{\mathrm{x}},\mathscr{F}^{\parallel}_{\mathrm{y}},\mathscr{F}^{\parallel}_{\mathrm{z}}) = -\sum_{\mu=\mathrm{x,y,z}}\sum_{\nu}\left(J\mathbf{Z}^{\mu}_{\nu}\mathbf{Z}^{\mu}_{\nu+\hat{\mu}} + h\mathbf{X}^{\mu}_{\nu}\right), \tag{182}$$

where $\nu + \hat{\mu}$ denotes the lattice site which is one over from $\nu$ in the $\mu$ direction. Although our primary focus remains on this decoupled model, it is also useful at times to consider the slightly more general *Ising grid* model,

$$H_{\mathrm{IG}} = {}^{1}H^{(0)}_{\mathbb{Z}_2}\left(\mathscr{F}^{\parallel}_{\mathrm{x}},\mathscr{F}^{\parallel}_{\mathrm{y}},\mathscr{F}^{\parallel}_{\mathrm{z}}\right) + H_{\mathrm{C}}, \tag{183}$$

which has its wires coupled together by the inter-wire coupling terms

$$H_{\mathrm{C}} = -K_X\sum_{\nu}(\mathbf{X}^{\mathrm{x}}_{\nu}\mathbf{X}^{\mathrm{y}}_{\nu} + \mathbf{X}^{\mathrm{x}}_{\nu}\mathbf{X}^{\mathrm{z}}_{\nu} + \mathbf{X}^{\mathrm{y}}_{\nu}\mathbf{X}^{\mathrm{z}}_{\nu}) - K_Z\sum_{\nu}\mathbf{Z}^{\mathrm{x}}_{\nu}\mathbf{Z}^{\mathrm{y}}_{\nu}\mathbf{Z}^{\mathrm{z}}_{\nu}. \tag{184}$$

Now, when $K_Z = 0$, this model has a relation-free linear subsystem symmetry group, $\mathbb{Z}^{(0,2)}_2(\mathscr{F}^{\parallel}_{\mathrm{x}},\mathscr{F}^{\parallel}_{\mathrm{y}},\mathscr{F}^{\parallel}_{\mathrm{z}})$. The operators that generate these symmetries are

$$U^{\mathrm{x}}_{j,k} = \prod_{i=1}^{L}\mathbf{X}^{\mathrm{x}}_{i,j,k}, \quad U^{\mathrm{y}}_{i,k} = \prod_{j=1}^{L}\mathbf{X}^{\mathrm{y}}_{i,j,k}, \quad U^{\mathrm{z}}_{i,j} = \prod_{k=1}^{L}\mathbf{X}^{\mathrm{z}}_{i,j,k}. \tag{185}$$

Note that, at least when $K_X = K_Z = 0$, as the competition between $J$ and $h$ is varied, each of the wires undergoes a phase transition related to the spontaneous breaking of their global $\mathbb{Z}^{(0)}_2$ symmetry; from the perspective of the full Ising grid, we may think of this as a spontaneous breaking of the $\mathbb{Z}^{(0,2)}_2(\mathscr{F}^{\parallel}_{\mathrm{x}},\mathscr{F}^{\parallel}_{\mathrm{y}},\mathscr{F}^{\parallel}_{\mathrm{z}})$ linear subsystem symmetries.

When $K_Z$ is turned on, the $\mathbb{Z}^{(0,2)}_2(\mathscr{F}^{\parallel}_{\mathrm{x}},\mathscr{F}^{\parallel}_{\mathrm{y}},\mathscr{F}^{\parallel}_{\mathrm{z}})$ is explicitly broken down to a *planar* subsystem symmetry subgroup. To describe this, we note for example that, in the language of §2.1, the foliations $\mathscr{F}^{\parallel}_{\mathrm{x}}$ and $\mathscr{F}^{\parallel}_{\mathrm{y}}$ admit a common *coarsening* $\mathscr{F}^{\parallel}_{\mathrm{xy}}$: namely, the foliation of 3d space by planes that span the xy directions. Associated to this coarsening is a subgroup $\mathbb{Z}^{(0,1)}_2(\mathscr{F}^{\parallel}_{\mathrm{xy}},\mathscr{F}^{\parallel}_{\mathrm{yz}},\mathscr{F}^{\parallel}_{\mathrm{xz}})\big/\!\sim$ whose symmetry generators are

$$U^{\mathrm{xy}}_k = \prod_{i,j=1}^{L}\mathbf{X}^{\mathrm{x}}_{i,j,k}\mathbf{X}^{\mathrm{y}}_{i,j,k}, \quad U^{\mathrm{yz}}_i = \prod_{j,k=1}^{L}\mathbf{X}^{\mathrm{y}}_{i,j,k}\mathbf{X}^{\mathrm{z}}_{i,j,k}, \quad U^{\mathrm{xz}}_j = \prod_{i,k=1}^{L}\mathbf{X}^{\mathrm{x}}_{i,j,k}\mathbf{X}^{\mathrm{z}}_{i,j,k}, \tag{186}$$

(cf. Eq. (20)). The quotient by $\sim$ is due to the fact that not all of these symmetries are independent. In particular, they enjoy a relation of the form

$$\prod_{k=1}^{L}U^{\mathrm{xy}}_k\prod_{j=1}^{L}U^{\mathrm{xz}}_j\prod_{i=1}^{L}U^{\mathrm{yz}}_i = 1. \tag{187}$$

The left-hand side is the generator of the diagonal subgroup ${}^{\mathrm{D}}\mathbb{Z}^{(0)}_2$ obtained by mutually coarsening the three foliations $\mathscr{F}^{\parallel}_{\mathrm{xy}},\mathscr{F}^{\parallel}_{\mathrm{yz}},\mathscr{F}^{\parallel}_{\mathrm{xz}}$ to the trivial foliation $\mathscr{F}^{\parallel}_{\mathrm{xyz}}$. Thus, in total, we find that the planar subgroup can be described as

$$\mathscr{H} = \mathbb{Z}^{(0,1)}_2\left(\mathscr{F}^{\parallel}_{\mathrm{xy}},\mathscr{F}^{\parallel}_{\mathrm{yz}},\mathscr{F}^{\parallel}_{\mathrm{xz}}\right)\Big/{}^{\mathrm{D}}\mathbb{Z}^{(0)}_2. \tag{188}$$

In the next section, we show that gauging $\mathscr{H}$ leads to an interesting phase diagram featuring the cubic Ising model and the X-cube model. Before we do this, we now (relatedly) argue that the Ising grid undergoes a phase transition when $K_Z = 0$ as $K_X$ is varied from $\infty$ to 0, which corresponds to the spontaneous breaking of $\mathscr{H}$.

**Phases and excitations**

We now explore the strong coupling phases.

*Cubic Ising model at $K_Z = 0$ and $K_X \gg 1$*

The strong $K_X$ coupling limit of the Ising grid can be taken similarly as we did in §3.1.1 for the Ising quilt. Namely, we compute the ground state of the $K_X$ term, and perturbatively compute a low energy effective Hamiltonian $H_{\text{eff}}$ that acts on this ground state and accounts for the splittings of the energy levels induced by ${}^1H_{\mathbb{Z}_2}^{(0)}(\mathscr{F}_x^{\parallel}, \mathscr{F}_y^{\parallel}, \mathscr{F}_z^{\parallel})$, thought of as a perturbation.

The ground state simply consists of an effective qubit on each lattice site, $|\pm\rangle_{\text{eff}} := |\pm\pm\pm\rangle$. The effective Hamiltonian can be constructed out of operators which preserve this ground space; these are

$$\mathbf{Z}_\nu := \mathbf{Z}_\nu^x \mathbf{Z}_\nu^y \mathbf{Z}_\nu^z, \quad \mathbf{X}_\nu := \mathbf{X}_\nu^x = \mathbf{X}_\nu^y = \mathbf{X}_\nu^z, \tag{189}$$

which act in the standard way as Pauli operators on the effective qubit. In terms of these, the effective Hamiltonian (if computed to 12th order in perturbation theory) takes the form

$$H_{\text{eff}} \sim -\tilde{h} \sum_\nu \mathbf{X}_\nu - \tilde{J} \sum_c \prod_{\nu \in c} \mathbf{Z}_\nu, \tag{190}$$

up to an overall constant, which is precisely the cubic Ising model. When $\tilde{h} \gg \tilde{J}$, the excitations of the cubic Ising model are simply spinons

$$|\nu\rangle \sim \mathbf{Z}_\nu |\Omega\rangle + \cdots \tag{191}$$

On the other hand, when $\tilde{J} \gg \tilde{h}$, the excitations are supported on the fundamental cubes. They must be created 8 at a time, and the corresponding states take the form

$$|c_1, \ldots, c_8\rangle \sim \prod_{\nu \in C} \mathbf{X}_\nu |\Omega\rangle + \cdots, \tag{192}$$

where $c_1, \ldots, c_8$ are the eight fundamental cubes at the corners of the cuboid shape $C$, thought of as a collection of vertices.

We can note immediately that the planar subsystem symmetry subgroup $\mathscr{H}$ (Eq. (186)) acts trivially on this low energy Hilbert space, corresponding to the fact that the Ising grid leaves it unbroken in this phase. Because the symmetry is clearly broken when $K_X = 0$ (at least when $h \ll J$) we find that there must be at least one phase transition as one interpolates between the weakly and strongly coupled regimes. We offer a characterization of this phase transition in terms of "planar k-ribbon condensation" next.

We can elucidate the phase transition to the cubic Ising phase by analyzing the patterns of condensation. For simplicity, we turn off the transverse fields on the wires, $h = 0$. Just as in §3.1.1 (see Figure 4), when $K_X = 0$, an operator of the form $\mathbf{X}_\nu^{\mu_1} \mathbf{X}_\nu^{\mu_2}$ creates a k-string in the $\mu_1 \mu_2$ plane. It is actually more natural to puff this k-string up in the direction orthogonal to the $\mu_1 \mu_2$ plane, so that it turns into a "planar k-ribbon" formed out of plaquettes of the dual lattice. By taking a product of such operators over multiple vertices $\nu$ in this $\mu_1 \mu_2$ plane, we create a larger and larger planar k-ribbon. Note that k-ribbons on different planes fuse into each other and disappear on the dual plaquettes where they intersect. A network of planar k-ribbons (thought of as a network of dual plaquettes) has the property that every fundamental cube of the lattice is touched by an even number of corners of the dual plaquettes. As $K_X$ is driven to become large, such planar k-ribbons proliferate throughout the system and eventually condense.

Now, we consider what happens when a single-body Pauli-X, i.e. $\mathbf{X}_\nu^\mu$, is applied. This produces two parallel open ribbons. In contrast to closed k-ribbons, the eight cubes that have $\nu$

as a vertex are touched by an odd number of corners of dual plaquette kinks. We consider these cubes to be supporting their own excitations. Any application of operators of the form $\mathbf{X}_\nu^{\mu_1}\mathbf{X}_\nu^{\mu_2}$ does *not* affect the excitations living on these 8 cubes, it only changes the "bulk" planar k-ribbon configuration while leaving its corner cubes invariant. Therefore, as we drive $K_X$ to infinity the planar k-ribbons fluctuate at all length scales and become unphysical. The only remnant of the condensed k-ribbons is their corner cubes, surviving as excitations of the model deep in the $K_X \gg 1$ phase which we established above to be the cubic Ising model.

*Two coupled plaquette Ising layers at $K_Z \gg 1$ and $K_X = 0$*

In the large $K_Z$ limit, we focus on a low energy Hilbert space consisting of two effective qubits per site $\nu$, (which we label $\mathcal{H}_\nu^{\text{xz}} \otimes \mathcal{H}_\nu^{\text{yz}}$), spanned by

$$|\uparrow\uparrow\rangle_{\text{eff}} = |\uparrow\uparrow\uparrow\rangle, \qquad |\downarrow\uparrow\rangle_{\text{eff}} = |\uparrow\downarrow\downarrow\rangle, \qquad |\uparrow\downarrow\rangle_{\text{eff}} = |\downarrow\uparrow\downarrow\rangle, \qquad |\downarrow\downarrow\rangle_{\text{eff}} = |\downarrow\downarrow\uparrow\rangle. \tag{193}$$

The effective Pauli operators that act on these two qubits are

$$\mathbf{Z}_\nu^{\text{xz}} = \mathbf{Z}_\nu^{\text{x}}\mathbf{Z}_\nu^{\text{z}}, \qquad \mathbf{Z}_\nu^{\text{yz}} = \mathbf{Z}_\nu^{\text{y}}\mathbf{Z}_\nu^{\text{z}}, \qquad \mathbf{X}_\nu^{\text{xz}} = \mathbf{X}_\nu^{\text{y}}\mathbf{X}_\nu^{\text{z}}, \qquad \mathbf{X}_\nu^{\text{yz}} = \mathbf{X}_\nu^{\text{x}}\mathbf{X}_\nu^{\text{z}}. \tag{194}$$

In terms of these operators, one can show that the low energy effective Hamiltonian that governs the $K_Z \to \infty$ limit (to fourth order in perturbation theory) takes the form

$$H_{\text{eff}} \sim \sum_{\mu_1\mu_2=\text{xz,yz}} \left( -\tilde{h}\sum_\nu \mathbf{X}_\nu^{\mu_1\mu_2} - \tilde{J}\sum_{p\|\mu_1\mu_2}\prod_{\nu\in p}\mathbf{Z}_\nu^{\mu_1\mu_2} \right) \\ - \tilde{K}_X \sum_\nu \mathbf{X}_\nu^{\text{xz}}\mathbf{X}_\nu^{\text{yz}} - \tilde{K}_Z \sum_{p\|\text{xy}}\prod_{\nu\in p}\mathbf{Z}_\nu^{\text{xz}}\mathbf{Z}_\nu^{\text{yz}}. \tag{195}$$

The first line has the interpretation of two decoupled stacks of plaquette Ising model layers, while the terms on the second line couple these layers together.

**Order and disorder parameters**

We are mainly interested in the $\mathbb{Z}_2^{(0,2)}(\mathscr{F}_{\text{x}}^\|, \mathscr{F}_{\text{y}}^\|, \mathscr{F}_{\text{z}}^\|)$ subsystem symmetry breaking phase transition at $K_X = K_Z = 0$ since, as shown below, it maps under gauging the planar subgroup to a phase transition between the cubic Ising model and the X-cube model. There are several choices of order/disorder parameters that diagnose this phase transition. The obvious disorder parameters to consider are truncated symmetry operators of the linear subsystem symmetry and its planar subgroup, e.g.

$$\mathscr{D}_{L_{\text{x}}}^{\text{x}} = \prod_{\nu\in L_{\text{x}}}\mathbf{X}_\nu^{\text{x}}, \qquad \mathscr{D}_{R_{\text{xy}}}^{\text{xy}} = \prod_{\nu\in R_{\text{xy}}}\mathbf{X}_\nu^{\text{x}}\mathbf{X}_\nu^{\text{y}}. \tag{196}$$

The disorder parameters $\mathscr{D}_{L_{\text{y}}}^{\text{y}}$, $\mathscr{D}_{L_{\text{z}}}^{\text{z}}$ and $\mathscr{D}_{R_{\text{xz}}}^{\text{xz}}$, $\mathscr{D}_{R_{\text{yz}}}^{\text{yz}}$ are defined similarly. Above, $L_{\text{x}}$ is an open line segment that points in the x direction and $R_{\text{xy}}$ is a rectangle that spans the xy directions, both thought of as a set of vertices. The behavior of both of these disorder parameters is easily inferred from properties of the (1+1)D TFIM,

$$\langle\mathscr{D}_{L_{\text{x}}}^{\text{x}}\rangle \xrightarrow[K_X=K_Z=0]{|L_{\text{x}}|\to\infty} \begin{cases} \text{const.}, & J/h \ll 1, \\ \#e^{-|L_{\text{x}}|/\xi}, & J/h \gg 1, \end{cases}$$

$$\langle\mathscr{D}_{R_{\text{xy}}}^{\text{xy}}\rangle \xrightarrow[K_X=K_Z=0]{|R_{\text{xy}}|\to\infty} \begin{cases} \#e^{-P(R_{\text{xy}})/\tilde{\xi}}, & J/h \ll 1, \\ \#e^{-A(R_{\text{xy}})/A_0}, & J/h \gg 1. \end{cases} \tag{197}$$

Above, where we have written $|L_x| \to \infty$ and $|R_{xy}| \to \infty$, we mean that all dimensions of $L_x$ and $R_{xy}$ should be taken large. We take $|L_x|$ to be the length of $L_x$, $P(R_{xy})$ the perimeter of $R_{xy}$, and $A(R_{xy})$ the area of $R_{xy}$.

As far as order parameters go, we can consider the operator

$$^{SS}\mathcal{O}_C = \prod_{v \in \text{corner}(C)} Z_v^x Z_v^y Z_v^z, \tag{198}$$

where $C$ is a cuboid subset of the lattice, thought of as a set of vertices. This is the natural generalization of the order parameter of the Ising quilt from Eq. (111), and behaves as

$$\langle {}^{SS}\mathcal{O}_C \rangle \xrightarrow[K_X=K_Z=0]{|C|\to\infty} \begin{cases} \# e^{-(|C|_x+|C|_y+|C|_z)/\xi'}, & J/h \ll 1, \\ \text{const.}, & J/h \gg 1, \end{cases} \tag{199}$$

where above, $|C|_\mu$ is the size of the cuboid $C$ in the $\mu$ direction, and again we write $|C| \to \infty$ to mean that $|C|_x, |C|_y, |C|_z$ should all be taken large. One sees that this order parameter diagnoses the spontaneous breaking of the linear subsystem symmetry.

Another choice is

$$^{P}\mathcal{O}_\gamma^{xy} = Z_{\gamma_i}^{\mu_i} Z_{\gamma_i}^z \left( \prod_{v \in \text{corner}(\gamma^\circ)} Z_v^x Z_v^y Z_v^z \right) Z_{\gamma_f}^{\mu_f} Z_{\gamma_f}^z$$

$$Z_{\gamma_i+\hat{z}}^{\mu_i} Z_{\gamma_i+\hat{z}}^z \left( \prod_{v \in \text{corner}(\gamma^\circ)} Z_{v+\hat{z}}^x Z_{v+\hat{z}}^y Z_{v+\hat{z}}^z \right) Z_{\gamma_f+\hat{z}}^{\mu_f} Z_{\gamma_f+\hat{z}}^z, \tag{200}$$

which is a choice of generalization of the order parameter of Eq. (115). Here, $\gamma$ is a path on the lattice which is restricted to lie parallel to the xy directions, $v + \hat{z}$ is the nearest neighbor of $v$ in the $+\hat{z}$ direction, $\gamma_i/\gamma_f$ are the initial/final vertices of the path, and $\mu_i/\mu_f$ are the initial/final directions of the path. Also, corner($\gamma^\circ$) is the set of vertices of $\gamma$ at which two edges of $\gamma$ meet at a right angle. The order parameters $^{P}\mathcal{O}_\gamma^{xz}$ and $^{P}\mathcal{O}_\gamma^{yz}$ are defined similarly. Again, the behavior of both of these order parameters can be inferred from properties of the (1+1)D TFIM.

### 4.1.2 Gauging the planar zero-form subsystem subgroup

We now gauge the planar subgroup $\mathbb{Z}_2^{(0,1)}(\mathscr{F}_{xy}^\parallel, \mathscr{F}_{yz}^\parallel, \mathscr{F}_{xz}^\parallel)/^D\mathbb{Z}_2^{(0)}$. To do this, we add a gauge qubit to each edge $e$ of the cubic lattice, and take the Gauss's law terms that implement the local versions of the symmetries to be

$$G_v^{\mu_1\mu_2} = X_v^{\mu_1} X_v^{\mu_2} \prod_{\substack{e \ni v \\ e \| \mu_1\mu_2}} X_e. \tag{201}$$

Notice that these Gauss's law operators satisfy the relation

$$G_v^{xy} G_v^{yz} G_v^{xz} = 1. \tag{202}$$

This is necessary because the symmetry we are gauging satisfies an analogous relation, Eq. (187), which is already apparent from the action of the symmetries at a single vertex.[14] The minimal flux operators one can define that commute with these Gauss's law operators are

---

[14]It is insightful to briefly compare this gauging protocol to what one would need to do in order to gauge the $\mathbb{Z}_2^{(0,1)}(\mathscr{F}_{xy}^\parallel, \mathscr{F}_{yz}^\parallel, \mathscr{F}_{xz}^\parallel)$ symmetry of (2+1)D TFIM layers, i.e. the model $^2H_{\mathbb{Z}_2}^{(0)}(\mathscr{F}_{xy}^\parallel, \mathscr{F}_{yz}^\parallel, \mathscr{F}_{xz}^\parallel)$. In this latter case, the global planar subsystem symmetry does not satisfy any relations, and when it is gauged, the Gauss's law operators should not either. This fact requires one to introduce two gauge qubits per edge as opposed to the single gauge qubit we have used here.

$$F_c = \prod_{e \in c} \mathbf{Z}_e \,. \tag{203}$$

We impose $G_v^{\mu_1 \mu_2} = 1$ and $F_c = 1$ as constraints, and take the gauged Hamiltonian to be

$$H_{\mathrm{IG}} \Big/ \mathscr{H} = - \sum_{\mu=\mathrm{x,y,z}} \sum_v \Big( J \mathbf{Z}_v^\mu \mathbf{Z}_{\langle v, v+\hat{\mu} \rangle} \mathbf{Z}_{v+\hat{\mu}}^\mu + h \mathbf{X}_v^\mu \Big) + H_{\mathrm{C}} \,, \tag{204}$$

where $\langle v, v + \hat{\mu} \rangle$ is the edge which stretches between $v$ and $v + \hat{\mu}$, and $\mathscr{H}$ is as in Eq. (188). Note that $H_{\mathrm{C}}$ simply comes along for the ride, without requiring any modification, because every term in it already commutes with the Gauss's law generators.

To solve the Gauss's law constraint, we transform the model by a local unitary circuit,

$$V = \prod_{\mu=\mathrm{x,y,z}} \prod_v \big( C_v^\mu \mathbf{X}_{\langle v, v+\hat{\mu} \rangle} \big) \big( C_v^\mu \mathbf{X}_{\langle v, v-\hat{\mu} \rangle} \big) \,. \tag{205}$$

This leads to

$$V \Big( H_{\mathrm{IG}} \Big/ \mathscr{H} \Big) V^\dagger = - \sum_{\mu=\mathrm{x,y,z}} \sum_v \big( J \mathbf{Z}_{\langle v, v+\hat{\mu} \rangle} + h \mathbf{X}_{\langle v, v-\hat{\mu} \rangle} \mathbf{X}_v^\mu \mathbf{X}_{\langle v, v+\hat{\mu} \rangle} \big)$$
$$- K_X \sum_{\mu_1 \mu_2 = \mathrm{xy,yz,xz}} \sum_v \mathbf{X}_v^{\mu_1} \mathbf{X}_v^{\mu_2} \prod_{\substack{e \ni v \\ e \| \mu_1 \mu_2}} \mathbf{X}_e - K_Z \sum_v \mathbf{Z}_v^\mathrm{x} \mathbf{Z}_v^\mathrm{y} \mathbf{Z}_v^\mathrm{z}$$
$$V G_v^{\mu_1 \mu_2} V^\dagger = \mathbf{X}_v^{\mu_1} \mathbf{X}_v^{\mu_2} \,, \qquad V F_c V^\dagger = \prod_{e \in c} \mathbf{Z}_e \prod_{v \in c} \mathbf{Z}_v^\mathrm{x} \mathbf{Z}_v^\mathrm{y} \mathbf{Z}_v^\mathrm{z} \,. \tag{206}$$

Imposing that $G_v^{\mu_1 \mu_2} = 1$ for all $v$ and for $\mu_1 \mu_2 = \mathrm{xy, yz, xz}$ reduces the local Hilbert space at the sites of the lattice from 3 qubits to 1 effective qubit. In terms of the 3 qubits, the two states of this effective qubit are $|+\rangle_{\mathrm{eff}} := |{+}{+}{+}\rangle$ and $|-\rangle_{\mathrm{eff}} = |{-}{-}{-}\rangle$, where $|\pm\rangle$ denotes the eigenstate of Pauli-$\mathbf{X}$ with eigenvalue $\pm 1$. It is straightforward to see that, when restricted to the subspace spanned by this effective qubit, the operators $\mathbf{Z}_v := \mathbf{Z}_v^\mathrm{x} \mathbf{Z}_v^\mathrm{y} \mathbf{Z}_v^\mathrm{z}$ and $\mathbf{X}_v := \mathbf{X}_v^\mathrm{x} = \mathbf{X}_v^\mathrm{y} = \mathbf{X}_v^\mathrm{z}$ act in the standard way as Pauli operators, i.e.

$$\mathbf{Z}_v |\pm\rangle_{\mathrm{eff}} = |\mp\rangle_{\mathrm{eff}} \,, \qquad \mathbf{X}_v |\pm\rangle_{\mathrm{eff}} = \pm |\pm\rangle_{\mathrm{eff}} \,. \tag{207}$$

Thus, in terms of these operators, the gauged model becomes

$$V \Big( H_{\mathrm{IG}} \Big/ \mathscr{H} \Big) V^\dagger = - \sum_{\mu=\mathrm{x,y,z}} \sum_v ( J \mathbf{Z}_{\langle v, v+\hat{\mu} \rangle} + h \mathbf{X}_{\langle v, v-\hat{\mu} \rangle} \mathbf{X}_v \mathbf{X}_{\langle v, v+\hat{\mu} \rangle} )$$
$$- K_X \sum_{\mu_1 \mu_2 = \mathrm{xy,yz,xz}} \sum_v \prod_{\substack{e \ni v \\ e \| \mu_1 \mu_2}} \mathbf{X}_e - K_Z \sum_v \mathbf{Z}_v$$
$$V F_c V^\dagger = \prod_{e \in c} \mathbf{Z}_e \prod_{v \in c} \mathbf{Z}_v \,. \tag{208}$$

We note that, when $K_Z = 0$, this gauged model inherits the linear subsystem symmetries of the original Ising grid, which now act as

$$U_{j,k}^\mathrm{x} = \prod_{i=1}^L \mathbf{X}_{i,j,k} \,, \qquad U_{i,k}^\mathrm{y} = \prod_{j=1}^L \mathbf{X}_{i,j,k} \,, \qquad U_{i,j}^\mathrm{z} = \prod_{k=1}^L \mathbf{X}_{i,j,k} \,. \tag{209}$$

The main difference is that because we have gauged the planar subsystem symmetry subgroup $\mathscr{H}$, these symmetries are now subject to certain relations, e.g.

$$\prod_{j=1}^L U_{j,k}^\mathrm{x} \prod_{i=1}^L U_{i,k}^\mathrm{y} = 1 \,. \tag{210}$$

and so the group is $\mathbb{Z}_2^{(0,2)}(\mathscr{F}_{\mathrm{x}}^{\|}, \mathscr{F}_{\mathrm{y}}^{\|}, \mathscr{F}_{\mathrm{z}}^{\|})/\mathscr{H}$.

On the other hand, the model also gains emergent quantum planar one-form subsystem symmetry groups

$$\widehat{\mathscr{H}} = \widehat{\mathbb{Z}}_2^{(1,1)}\left(\mathscr{F}_{\mathrm{xy}}^{\|}, \mathscr{F}_{\mathrm{yz}}^{\|}, \mathscr{F}_{\mathrm{xz}}^{\|}\right)/\sim, \tag{211}$$

which are generated by the flux terms. More precisely, if $\tilde{\gamma}$ is a path on the dual lattice which is restricted to lie in the xy plane at some fixed location in the z direction, then we have symmetry string operators

$$\widehat{U}_{\tilde{\gamma}}^{\mathrm{xy}} = \prod_{c \in \tilde{\gamma}} V F_c V^{\dagger}, \tag{212}$$

which, together with analogous operators $\widehat{U}_{\tilde{\gamma}}^{\mathrm{xz}}, \widehat{U}_{\tilde{\gamma}}^{\mathrm{yz}}$, generate the higher-form subsystem symmetry group $\widehat{\mathscr{H}}$.

**Phases and excitations**

Let us study the fate of these symmetries as the value of the coupling $J/h$ in the gauged Hamiltonian is tuned. We find that the subsystem symmetry breaking phase transition before gauging maps to a simultaneous breaking of $\mathbb{Z}_2^{(0,2)}(\mathscr{F}_{\mathrm{x}}^{\|}, \mathscr{F}_{\mathrm{y}}^{\|}, \mathscr{F}_{\mathrm{z}}^{\|})/\mathscr{H}$ and an unbreaking of $\widehat{\mathscr{H}}$, as summarized in Figure 9.

*Cubic Ising model at $K_X = K_Z = 0$ and $J \gg h$*

If we take $J$ to be very large, then the qubits that live on the edges are frozen to be in $+1$ eigenstates of Pauli-$\mathbf{Z}$ operators. The term proportional to $h$ essentially decouples in this limit, so that the Hamiltonian is a constant. The flux term constrains the low energy Hilbert space to be in eigenstates of

$$\prod_{v \in c} \mathbf{Z}_v = 1, \tag{213}$$

with eigenvalue $+1$. Thus, the effective Hilbert space in this limit agrees with the ground state of the cubic Ising model deep in its ferromagnetic phase. In this phase, the linear subsystem symmetry group $\mathbb{Z}_2^{(0,2)}(\mathscr{F}_{\mathrm{x}}^{\|}, \mathscr{F}_{\mathrm{y}}^{\|}, \mathscr{F}_{\mathrm{z}}^{\|})/\mathscr{H}$ is spontaneously broken, while the emergent planar one-form symmetry group $\widehat{\mathscr{H}}$ is unbroken, which is consistent with Expectation 2 from §2.1.

*X-cube model at $K_X = K_Z = 0$ and $h \gg J$*

Take on the other hand the opposite limit, $h \gg J$. To make the analysis of the Hamiltonian simpler, let us perform one more unitary circuit,

$$\widetilde{V} = \prod_v \left(C_v \mathbf{X}_{\langle v, v+\hat{\mathbf{x}}\rangle}\right)\left(C_v \mathbf{X}_{\langle v, v-\hat{\mathbf{x}}\rangle}\right). \tag{214}$$

Then, dropping the term proportional to $J$ which is irrelevant in this limit, we find

$$\widetilde{V} V \left({}^1 H_{\mathbb{Z}_2}^{(0)}(\mathscr{F}_{\mathrm{x}}^{\|}, \mathscr{F}_{\mathrm{y}}^{\|}, \mathscr{F}_{\mathrm{z}}^{\|})/\mathscr{H}\right) V^{\dagger} \widetilde{V}^{\dagger} = -h \sum_v \left(\mathbf{X}_v + \mathbf{X}_v \prod_{\substack{e \ni v \\ e \perp \mathrm{y}}} \mathbf{X}_e + \mathbf{X}_v \prod_{\substack{e \ni v \\ e \perp \mathrm{z}}} \mathbf{X}_e\right)$$

$$\widetilde{V} V F_c V^{\dagger} \widetilde{V}^{\dagger} = \prod_{e \in c} \mathbf{Z}_e. \tag{215}$$

The ground state of this Hamiltonian is such that the qubits on the sites are frozen in $\mathbf{X}_v = +1$ eigenstates. The qubits on the edges are then constrained to satisfy the vertex terms of the

Ising grid ($K_X = K_Z = 0$ decoupled wires): ${}^1H_{\mathbb{Z}_2}^{(0)}(\mathscr{F}_{\mathrm{x}}^{\parallel}, \mathscr{F}_{\mathrm{y}}^{\parallel}, \mathscr{F}_{\mathrm{z}}^{\parallel})$

$\mathbb{Z}_2^{(0,2)}(\mathscr{F}_{\mathrm{x}}^{\parallel}, \mathscr{F}_{\mathrm{y}}^{\parallel}, \mathscr{F}_{\mathrm{z}}^{\parallel})$ | $\mathbb{Z}_2^{(0,2)}(\mathscr{F}_{\mathrm{x}}^{\parallel}, \mathscr{F}_{\mathrm{y}}^{\parallel}, \mathscr{F}_{\mathrm{z}}^{\parallel})$
subsystem symmetry | subsystem symmetry
preserving phase | breaking phase

$\longrightarrow J/h$

gauging $\mathscr{H}$

Gauged Ising grid ($K_X = K_Z = 0$): ${}^1H_{\mathbb{Z}_2}^{(0)}(\mathscr{F}_{\mathrm{x}}^{\parallel}, \mathscr{F}_{\mathrm{y}}^{\parallel}, \mathscr{F}_{\mathrm{z}}^{\parallel})\big/\mathscr{H}$

$\mathbb{Z}_2^{(0,2)}(\mathscr{F}_{\mathrm{x}}^{\parallel}, \mathscr{F}_{\mathrm{y}}^{\parallel}, \mathscr{F}_{\mathrm{z}}^{\parallel})\big/\mathscr{H}$ preserved | $\mathbb{Z}_2^{(0,2)}(\mathscr{F}_{\mathrm{x}}^{\parallel}, \mathscr{F}_{\mathrm{y}}^{\parallel}, \mathscr{F}_{\mathrm{z}}^{\parallel})\big/\mathscr{H}$ broken
$\widehat{\mathscr{H}}$ broken | $\widehat{\mathscr{H}}$ preserved
(X-cube phase) | (CIM phase)

$\longrightarrow J/h$

Figure 9: Phase diagram of the Ising grid, before and after gauging the planar subgroup $\mathcal{H} = \mathbb{Z}_2^{(0,1)}(\mathscr{F}_{\mathrm{xy}}^{\parallel}, \mathscr{F}_{\mathrm{xz}}^{\parallel}, \mathscr{F}_{\mathrm{yz}}^{\parallel})\big/{}^{\mathrm{D}}\mathbb{Z}_2^{(0)}$ of its symmetry group.

X-cube model, while the flux term further constrains them to satisfy the cube term of the X-cube model. Thus in this phase, we find that the low energy Hamiltonian is essentially a constant, with the low energy effective Hilbert space agreeing with the ground state of the X-cube Hamiltonian. In this phase, the emergent planar one-form symmetry group $\widehat{\mathscr{H}}$ is spontaneously broken, while the linear subsystem symmetry $\mathbb{Z}_2^{(0,2)}(\mathscr{F}_{\mathrm{x}}^{\parallel}, \mathscr{F}_{\mathrm{y}}^{\parallel}, \mathscr{F}_{\mathrm{z}}^{\parallel})\big/\mathscr{H}$ is unbroken. This is again consistent with Expectation 2 from §2.1.

*Excitations and phase transition at $K_X = K_Z = 0$ and $J = h$*

Under gauging the planar subsystem symmetry, kink excitations on the wires in their ferromagnetic phase are mapped to quadrupoles of cube excitations in the ferromagnetic cubic Ising model phase. To see this we note that an excitation of a single edge $J$ term in the gauged Hamiltonian is locally equivalent to a quadrupole of flux term $F_c$ excitations, which become excitations of the cube term in the cubic Ising model phase. This follows by considering the trivial local cluster of excitations created by a single $\mathbf{X}_e$ term, which precisely corresponds to a single $J$ edge term excitation and a quadrupole of $F_c$ cube excitations.

Hence the phase transition from the ferromagnetic cubic Ising model to the X-cube model obtained by gauging the planar subsytstem symmetry of decoupled critical Ising wires can be viewed as the condensation of quadrupoles of cube excitations along lines. On the other hand, the phase transition from the X-cube to the ferromagnetic cubic Ising model is clearly driven by single body $\mathbf{Z}_e$ operators, which induce the condensation of lineon excitations.

**Order and disorder parameters**

To obtain order/disorder parameters that diagnose the transition from the cubic Ising model to the X-cube model, we can simply map over the order/disorder parameters of the ungauged model. The simplest to study is the disorder parameter for the linear subsystem symmetry,

$$V\left(\mathscr{D}_{L_x}^{\mathrm{x}}\big/\mathscr{H}\right)V^{\dagger} \xrightarrow{V G_v^{\mu_1\mu_2}V^{\dagger}=1} \prod_{e\in\partial L_x}\mathbf{X}_e \prod_{v\in L_x}\mathbf{X}_v, \tag{216}$$

where above, the product over $\partial L_{\mathrm{x}}$ is a product over the two edges at the boundary of the line segment $L_{\mathrm{x}}$. Thus, we see that in the gauged model, this disorder parameter is mapped

to a truncated symmetry operator, with additional insertions decorating its boundary, and so it retains its interpretation as a disorder parameter of the $\mathbb{Z}_2^{(0,2)}(\mathscr{F}_x^{\parallel}, \mathscr{F}_y^{\parallel}, \mathscr{F}_z^{\parallel})\big/\mathscr{H}$ symmetry.

On the other hand, the disorder parameter for the planar subsystem symmetry in the ungauged model maps as

$$V\left(\mathscr{D}_{R_{xy}}^{xy}\Big/\mathscr{H}\right)V^{\dagger} \xrightarrow{VG_{\nu}^{\mu_1\mu_2}V^{\dagger}=1} \prod_{e\in\partial R_{xy}} \mathbf{X}_e\,. \tag{217}$$

As one expects, since we are gauging this symmetry, the bulk of the symmetry operator disappears, and we are left just with a line operator supported on its boundary, which we can think of as an order parameter for the emergent planar one-form subsystem symmetry $\widehat{\mathscr{H}}$.

Now, if we map over the order parameter for the linear subsystem symmetry, Eq. (198),

$$V\left({}^{\mathrm{SS}}\mathscr{O}_C\Big/\mathscr{H}\right)V^{\dagger} \xrightarrow{VG_{\nu}^{\mu_1\mu_2}V^{\dagger}=1} \prod_{v\in\mathrm{corner}(C)} \mathbf{Z}_{\nu}\,, \tag{218}$$

we find that it retains its interpretation as an order parameter in the gauged model. If we map over the order parameter for the planar subsystem symmetry, Eq. (200), we find a ribbon operator of the form

$$
\begin{aligned}
&V\left({}^{\mathrm{P}}\mathscr{O}_{\gamma}^{xy}\Big/\mathscr{H}\right)V^{\dagger} \\
&\xrightarrow{VG_{\nu}^{\mu_1\mu_2}V^{\dagger}=1} \mathbf{Z}_{\langle\gamma_i,\gamma_i+\hat{z}\rangle}\left(\prod_{e\in\gamma}\mathbf{Z}_e\mathbf{Z}_{e+\hat{z}}\right)\left(\prod_{v\in\mathrm{corner}(\gamma^\circ)}\mathbf{Z}_{\langle v,v+\hat{z}\rangle}\right)\mathbf{Z}_{\langle\gamma_f,\gamma_f+\hat{z}\rangle}\,,
\end{aligned} \tag{219}
$$

where we are using the notation $\langle v, w\rangle$ to denote the edge which stretches between $v$ and $w$, and $e+\hat{z}$ is the edge one obtains by displacing $e$ by one unit in the $\hat{z}$ direction. As one can see by comparing to the symmetry operators for the emergent quantum planar one-form symmetry in Eq. (212), this has the form of a truncated symmetry operator (with the operators which decorate its corners stripped off) and so can be thought of as a disorder parameter for this emergent planar one-form symmetry.

In total, we find that order/disorder parameters for the linear subsystem symmetry $\mathbb{Z}_2^{(0,2)}(\mathscr{F}_x^{\parallel}, \mathscr{F}_y^{\parallel}, \mathscr{F}_z^{\parallel})$ of the ungauged model remain order/disorder parameters for the linear subsystem symmetry $\mathbb{Z}_2^{(0,2)}(\mathscr{F}_x^{\parallel}, \mathscr{F}_y^{\parallel}, \mathscr{F}_z^{\parallel})\big/\mathscr{H}$ in the gauged model, while order/disorder parameters for the planar subgroup $\mathscr{H}$ are mapped to disorder/order parameters respectively of the emergent planar one-form symmetry group $\widehat{\mathscr{H}}$.

### 4.1.3 Dualizing the leaves

Before closing this subsection, we remark briefly on what happens if one sets $K_Z = 0$ and gauges the full $\mathbb{Z}_2^{(0,2)}(\mathscr{F}_x^{\parallel}, \mathscr{F}_y^{\parallel}, \mathscr{F}_z^{\parallel})$ subsystem symmetry group (or equivalently, performs Kramers–Wannier duality on each of the wires). One finds a model with a qubit on each edge whose Hamiltonian is unitarily equivalent to

$$H_{\mathrm{IG}}\Big/\mathbb{Z}_2^{(0,2)}(\mathscr{F}_x^{\parallel}, \mathscr{F}_y^{\parallel}, \mathscr{F}_z^{\parallel}) \simeq -J\sum_e \mathbf{Z}_e - \sum_v \sum_{\mu=\mathrm{x,y,z}}\left(h\prod_{e\ni v}^{\parallel\mu}\mathbf{X}_e + K_X\prod_{e\ni v}^{\perp\mu}\mathbf{X}_e\right). \tag{220}$$

Note that, consistent with Expectation 1 from §2.1, this model enjoys an emergent quantum $\widehat{\mathbb{Z}}_2^{(0,2)}(\mathscr{F}_x^{\parallel}, \mathscr{F}_y^{\parallel}, \mathscr{F}_z^{\parallel})$ linear subsystem symmetry

$$\widehat{U}_{j,k}^{x} = \prod_i \mathbf{Z}_{i+\frac{1}{2},j,k}\,, \quad \widehat{U}_{i,k}^{y} = \prod_j \mathbf{Z}_{i,j+\frac{1}{2},k}\,, \quad \widehat{U}_{i,j}^{z} = \prod_k \mathbf{Z}_{i,j,k+\frac{1}{2}}\,. \tag{221}$$

The low-energy effective Hamiltonian around $K_X \to \infty$ is precisely an X-cube model perturbed by the lowest order non-trivial term which respects this linear subsystem symmetry. This is analogous to the fact that gauging the linear subsystem symmetry of the Ising quilt (or equivalently, performing row and column-wise Kramers–Wannier duality) produces a toric code perturbed by operators which respect a linear subsystem symmetry subgroup of its global symmetry group (cf. §3.2). In principle, we could dedicate another subsection to studying this subsystem symmetry-enriched X-cube model, mirroring §3.2, but we instead choose to leave this to future work.

## 4.2 The point-cage-net model

We now introduce the *point-cage-net-model*, which is designed to combine the coupled Ising wires, the X-cube model, and their planar and linear subsystem symmetry gauged variants, respectively. This model is defined on a Hilbert space with three qubits per site and one qubit per edge of a cubic lattice

$$\mathcal{H} = \bigotimes_{i,j,k} \left( \mathcal{H}^{\mathrm{x}}_{i,j,k} \otimes \mathcal{H}^{\mathrm{y}}_{i,j,k} \otimes \mathcal{H}^{\mathrm{z}}_{i,j,k} \otimes \mathcal{H}_{i+\frac{1}{2},j,k} \otimes \mathcal{H}_{i,j+\frac{1}{2},k} \otimes \mathcal{H}_{i,j,k+\frac{1}{2}} \right), \tag{222}$$

where

$$\mathcal{H}^{\mathrm{x}}_{i,j,k} \cong \mathcal{H}^{\mathrm{y}}_{i,j,k} \cong \mathcal{H}^{\mathrm{z}}_{i,j,k} \cong \mathcal{H}_{i+\frac{1}{2},j,k} \cong \mathcal{H}_{i,j+\frac{1}{2},k} \cong \mathcal{H}_{i,j,k+\frac{1}{2}} \cong \mathbb{C}^2. \tag{223}$$

The perturbed point-cage-net Hamiltonian is then

$$
\begin{aligned}
H_{\mathrm{PPCN}} = &- \Delta \sum_{v} \left( \mathbf{X}^{\mathrm{x}}_{v} \mathbf{X}^{\mathrm{y}}_{v} \prod_{e \ni v}^{\perp \mathrm{z}} \mathbf{X}_{e} + \mathbf{X}^{\mathrm{x}}_{v} \mathbf{X}^{\mathrm{z}}_{v} \prod_{e \ni v}^{\perp \mathrm{y}} \mathbf{X}_{e} + \mathbf{X}^{\mathrm{y}}_{v} \mathbf{X}^{\mathrm{z}}_{v} \prod_{e \ni v}^{\perp \mathrm{x}} \mathbf{X}_{e} \right) - \lambda \sum_{e} \mathbf{Z}_{e} \\
&- \gamma \sum_{i,j,k} (\mathbf{X}_{i-\frac{1}{2},j,k} \mathbf{X}^{\mathrm{x}}_{i,j,k} \mathbf{X}_{i+\frac{1}{2},j,k} + \mathbf{X}_{i,j-\frac{1}{2},k} \mathbf{X}^{\mathrm{y}}_{i,j,k} \mathbf{X}_{i,j+\frac{1}{2},k} + \mathbf{X}_{i,j,k-\frac{1}{2}} \mathbf{X}^{\mathrm{z}}_{i,j,k} \mathbf{X}_{i,j,k+\frac{1}{2}}) \\
&- \Delta' \sum_{i,j,k} \left( \mathbf{Z}^{\mathrm{x}}_{i,j,k} \mathbf{Z}_{i+\frac{1}{2},j,k} \mathbf{Z}^{\mathrm{x}}_{i+1,j,k} + \mathbf{Z}^{\mathrm{y}}_{i,j,k} \mathbf{Z}_{i,j+\frac{1}{2},k} \mathbf{Z}^{\mathrm{y}}_{i,j+1,k} + \mathbf{Z}^{\mathrm{z}}_{i,j,k} \mathbf{Z}_{i,j,k+\frac{1}{2}} \mathbf{Z}^{\mathrm{z}}_{i,j,k+1} \right) \\
&- \lambda' \sum_{v} (\mathbf{X}^{\mathrm{x}}_{v} \mathbf{X}^{\mathrm{y}}_{v} + \mathbf{X}^{\mathrm{y}}_{v} \mathbf{X}^{\mathrm{z}}_{v} + \mathbf{X}^{\mathrm{x}}_{v} \mathbf{X}^{\mathrm{z}}_{v}) - \gamma' \sum_{v} \left( \mathbf{X}^{\mathrm{x}}_{v} + \mathbf{X}^{\mathrm{y}}_{v} + \mathbf{X}^{\mathrm{z}}_{v} \right) \\
&- \kappa' \sum_{v} \mathbf{Z}^{\mathrm{x}}_{v} \mathbf{Z}^{\mathrm{y}}_{v} \mathbf{Z}^{\mathrm{z}}_{v}.
\end{aligned} \tag{224}
$$

This model respects a $\widetilde{\mathbb{Z}}^{(0,2)}_2(\mathscr{F}^{\parallel}_{\mathrm{x}}, \mathscr{F}^{\parallel}_{\mathrm{y}}, \mathscr{F}^{\parallel}_{\mathrm{z}})$ linear zero-form subsystem symmetry generated by Pauli-$\mathbf{Z}$ operators acting on the edges of 1d sublattices which are parallel to the x, y, and z directions, i.e.

$$\widetilde{U}^{\mathrm{x}}_{j,k} = \prod_{i} \mathbf{Z}_{i+\frac{1}{2},j,k}, \qquad \widetilde{U}^{\mathrm{y}}_{i,k} = \prod_{j} \mathbf{Z}_{i,j+\frac{1}{2},k}, \qquad \widetilde{U}^{\mathrm{z}}_{j,k} = \prod_{k} \mathbf{Z}_{i,j,k+\frac{1}{2}}. \tag{225}$$

This symmetry can be understood as a subgroup of a larger subsystem symmetry generated by "cages" of $\mathbf{Z}_e$ operators acting on the edges of a cube which is recovered when the perturbation strength $\gamma$ is set to zero.

The model also respects a $\mathbb{Z}^{(0,1)}_2(\mathscr{F}^{\parallel}_{\mathrm{xy}}, \mathscr{F}^{\parallel}_{\mathrm{xz}}, \mathscr{F}^{\parallel}_{\mathrm{yz}}) / ^{\mathrm{D}}\mathbb{Z}^{(0)}_2$ zero-form planar subsystem symmetry group generated by spin flips of the form

$$\widetilde{U}^{\mathrm{xy}}_{k} = \prod_{i,j} \mathbf{X}^{\mathrm{x}}_{i,j,k} \mathbf{X}^{\mathrm{y}}_{i,j,k}, \qquad \widetilde{U}^{\mathrm{xz}}_{j} = \prod_{i,k} \mathbf{X}^{\mathrm{x}}_{i,j,k} \mathbf{X}^{\mathrm{z}}_{i,j,k}, \qquad \widetilde{U}^{\mathrm{yz}}_{i} = \prod_{j,k} \mathbf{X}^{\mathrm{y}}_{i,j,k} \mathbf{X}^{\mathrm{z}}_{i,j,k}. \tag{226}$$

When $\kappa' = 0$, this symmetry is enlarged to a $\mathbb{Z}_2^{(0,2)}(\mathscr{F}_x^{\parallel}, \mathscr{F}_y^{\parallel}, \mathscr{F}_z^{\parallel})$ linear zero-form subsystem symmetry (distinct from the $\widetilde{\mathbb{Z}}_2^{(0,2)}(\mathscr{F}_x^{\parallel}, \mathscr{F}_y^{\parallel}, \mathscr{F}_z^{\parallel})$ symmetry above),

$$U_{j,k}^{\mathrm{x}} = \prod_i \mathbf{X}_{i,j,k}^x, \quad U_{i,k}^{\mathrm{y}} = \prod_j \mathbf{X}_{i,j,k}^y, \quad U_{i,j}^{\mathrm{z}} = \prod_k \mathbf{X}_{i,j,k}^z. \tag{227}$$

There is also a global $\mathbb{Z}_2$ duality that acts on the phase diagram by swapping $\Delta \leftrightarrow \lambda'$, $\Delta' \leftrightarrow \lambda$, and $\gamma \leftrightarrow \gamma'$; it is implemented by the circuit

$$\widetilde{V} = \prod_{i,j,k} C_{i,j,k}^{\mathrm{x}} \mathbf{X}_{i-\frac{1}{2},j,k} \, C_{i,j,k}^{\mathrm{x}} \mathbf{X}_{i+\frac{1}{2},j,k} \, C_{i,j,k}^{\mathrm{y}} \mathbf{X}_{i,j-\frac{1}{2},k} \, C_{i,j}^{\mathrm{y}} \mathbf{X}_{i,j+\frac{1}{2},k} \, C_{i,j,k}^{\mathrm{z}} \mathbf{X}_{i,j,k-\frac{1}{2}} \, C_{i,j}^{\mathrm{z}} \mathbf{X}_{i,j,k+\frac{1}{2}}. \tag{228}$$

The point-cage-net Hamiltonian above combines the coupled wire example explored in §4.1 with a deformed X-cube model. These models are recovered in various limits, as described below.

- As $\lambda \to \infty$ the edge qubits are pinned into the $|\uparrow\rangle$ state, and the $\Delta, \gamma$ terms are projected out, resulting in the Ising grid Hamiltonian explored in §4.1,

$$H_{\mathrm{PPCN}} \xrightarrow{\lambda \to \infty} H_{\mathrm{IG}} \tag{229}$$

  (cf. Eq. (183)). This is equivalent to the limit $\Delta' \to \infty$ under $\widetilde{V}$, which corresponds to a coupled cluster chain model that has an X-cube phase in the limit $\gamma', \kappa' \to \infty$ [112].

- As $\lambda, \gamma \to 0$, we recover the Hamiltonian of the Ising grid with its planar subsystem symmetry gauged, Eq. (204), up to the term proportional to $\Delta$, which can be thought of as imposing the Gauss's law constraint Eq. (201) energetically. The piece that is absent in the PPCN Hamiltonian in the strict $\lambda \to 0$ limit is the flux terms which arise in the gauging of the Ising grid, Eq. (203). These can be incorporated energetically by taking $\lambda$ to be small but non-zero, in which case the flux terms are generated in perturbation theory. In total, we have that

$$H_{\mathrm{PPCN}} \xrightarrow[\substack{\gamma \to 0 \\ \Delta \to \infty \\ \lambda \text{ small}}]{} H_{\mathrm{IG}}\Big|_E \mathscr{H}. \tag{230}$$

- As $\gamma' \to \infty$, the vertex qubits are pinned into the $|+\rangle$ state and the $\Delta', \kappa'$ terms are projected out. Backing away slightly from the $\gamma' \to \infty$ limit, we can perturbatively incorporate the effects of the $\Delta', \kappa'$ terms: if one calculates the effective Hamiltonian to 8th order in $\kappa'$ and 12th order in $\Delta'$, then one finds that these terms conspire to produce the cage term of the X-cube model. Altogether, we find

$$\begin{aligned}
H_{\mathrm{PPCN}} \xrightarrow{\gamma' \text{ large}} &-\Delta \sum_v \left( \prod_{e \ni v}^{\perp z} \mathbf{X}_e + \prod_{e \ni v}^{\perp y} \mathbf{X}_e + \prod_{e \ni v}^{\perp x} \mathbf{X}_e \right) - \tilde{t} \sum_c \prod_{e \in c} \mathbf{Z}_e \\
&- \lambda \sum_e \mathbf{Z}_e - \gamma \sum_{i,j,k} \left( \mathbf{X}_{i-\frac{1}{2},j,k} \mathbf{X}_{i+\frac{1}{2},j,k} + \mathbf{X}_{i,j-\frac{1}{2},k} \mathbf{X}_{i,j+\frac{1}{2},k} + \mathbf{X}_{i,j,k-\frac{1}{2}} \mathbf{X}_{i,j,k+\frac{1}{2}} \right).
\end{aligned} \tag{231}$$

The first two terms are those of the X-cube model, while the last two terms are perturbations that respect the linear subsystem symmetry group, Eq. (225). Thus, in this limit, the point-cage-net model is equivalent to a linear subsystem symmetry-enriched X-cube phase, a model whose study we leave to future work.

- Consider the limit $\lambda', \gamma', \kappa' \to 0$. If we also take $\Delta$ to be large but non-infinite and compute the effects of the term proportional to $\lambda$ in perturbation theory, we find that it generates the cage term of the X-cube model. Thus, in this limit, we find

$$
\begin{aligned}
H_{\mathrm{PPCN}} &\xrightarrow[\lambda',\gamma',\kappa' \to 0]{\substack{\Delta \text{ large}\\ \lambda \text{ small}}} \\
&-\Delta \sum_v \left( \mathbf{X}_v^x \mathbf{X}_v^y \prod_{e \ni v}^{\perp z} \mathbf{X}_e + \mathbf{X}_v^x \mathbf{X}_v^z \prod_{e \ni v}^{\perp y} \mathbf{X}_e + \mathbf{X}_v^y \mathbf{X}_v^z \prod_{e \ni v}^{\perp x} \mathbf{X}_e \right) - t' \sum_c \prod_{e \in c} \mathbf{Z}_e \\
&- \gamma \sum_{i,j,k} (\mathbf{X}_{i-\frac{1}{2},j,k} \mathbf{X}_{i,j,k}^x \mathbf{X}_{i+\frac{1}{2},j,k} + \mathbf{X}_{i,j-\frac{1}{2},k} \mathbf{X}_{i,j,k}^y \mathbf{X}_{i,j+\frac{1}{2},k} + \mathbf{X}_{i,j,k-\frac{1}{2}} \mathbf{X}_{i,j,k}^z \mathbf{X}_{i,j,k+\frac{1}{2}}) \\
&- \Delta' \sum_{i,j,k} \left( \mathbf{Z}_{i,j,k}^x \mathbf{Z}_{i+\frac{1}{2},j,k} \mathbf{Z}_{i+1,j,k}^x + \mathbf{Z}_{i,j,k}^y \mathbf{Z}_{i,j+\frac{1}{2},k} \mathbf{Z}_{i,j+1,k}^y + \mathbf{Z}_{i,j,k}^z \mathbf{Z}_{i,j,k+\frac{1}{2}} \mathbf{Z}_{i,j,k+1}^z \right).
\end{aligned}
\tag{232}
$$

This has the interpretation of a deformed (due to the term proportional to $\gamma$) X-cube model with its linear subsystem symmetry gauged. Here, the $\Delta'$ term energetically enforces the Gauss's law constraint associated with this subsystem gauging.

The zero correlation length point-cage-net model is recovered in the limit $\Delta, \Delta' \to \infty$ and $\kappa' \to 0$,

$$
H_{\mathrm{PCN}} = -\sum_v (\mathcal{A}_v^x + \mathcal{A}_v^y + \mathcal{A}_v^z) - \sum_c \mathcal{B}_c - \sum_e \mathcal{C}_e,
\tag{233}
$$

where we define

$$
\begin{aligned}
\mathcal{A}_v^x = \mathbf{X}_v^y \mathbf{X}_v^z \prod_{e \ni v}^{\perp x} \mathbf{X}_e, \qquad &\mathcal{A}_v^y = \mathbf{X}_v^x \mathbf{X}_v^z \prod_{e \ni v}^{\perp y} \mathbf{X}_e, \qquad \mathcal{A}_v^z = \mathbf{X}_v^x \mathbf{X}_v^y \prod_{e \ni v}^{\perp z} \mathbf{X}_e, \\
\mathcal{B}_c = \prod_{e \in c} \mathbf{Z}_e, \qquad &\mathcal{C}_{i+\frac{1}{2},j,k} = \mathbf{Z}_{i,j,k}^x \mathbf{Z}_{i+\frac{1}{2},j,k} \mathbf{Z}_{i+1,j,k}^x \\
\mathcal{C}_{i,j+\frac{1}{2},k} = \mathbf{Z}_{i,j,k}^y \mathbf{Z}_{i,j+\frac{1}{2},k} \mathbf{Z}_{i,j+1,k}^y, \qquad &\mathcal{C}_{i,j,k+\frac{1}{2}} = \mathbf{Z}_{i,j,k}^z \mathbf{Z}_{i,j,k+\frac{1}{2}} \mathbf{Z}_{i,j,k+1}^z.
\end{aligned}
\tag{234}
$$

For ease of presentation we have included the leading order cube terms generated by products of $\mathbf{Z}_e$ interactions in the Hamiltonian above, and rescaled the energies; both operations preserve the zero temperature phase of matter of this commuting Hamiltonian.

The point-cage-net model can be viewed either as a stack of ferromagnetic Ising wires with their planar subsystem symmetry gauged, or as the X-cube model with its linear subsystem symmetry gauged. Excitations of the cube terms are equivalent to corner domain wall excitations of the cubic Ising model, see below. Excitations of the edge terms are equivalent to quadrupoles of cube excitations, as a single $\mathbf{X}_e$ operator excites an edge term along with the four adjacent cubes. We find below that excitations of the star terms are trivial.

The point-cage-net is equivalent to the ferromagnetic phase of the cubic Ising model, and hence the perturbed phase diagram can be understood as exploring neighboring phases and phase transitions, including decoupled Ising wires and the X-cube model. The equivalence is implemented by the circuit $\widetilde{V}$ followed by the unitary

$$
V = \prod_v C_v^x \mathbf{X}_v^z C_v^y \mathbf{X}_v^z
\tag{235}
$$

resulting in

$$
V \widetilde{V} H_{\mathrm{PCN}} \widetilde{V}^\dagger V^\dagger = -\sum_v (\mathbf{X}_v^x \mathbf{X}_v^y + \mathbf{X}_v^x + \mathbf{X}_v^y) - \sum_c \prod_{v \in c} \mathbf{Z}_v^z \prod_{e \in c} \mathbf{Z}_e - \sum_e \mathbf{Z}_e.
\tag{236}
$$

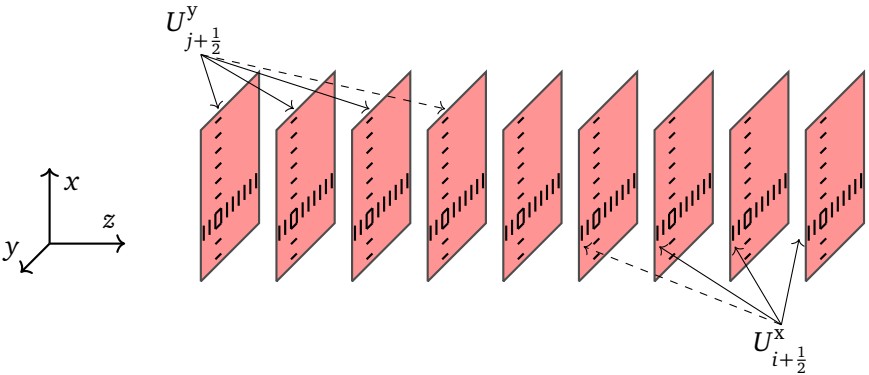

Figure 10: A stack of lattice gauge theory layers. The planar zero-form subsystem symmetry groups are generated by the product of line operators of the gauge theory layers over the leaves of the stack.

Each cube term is equivalent to $\prod_{v \in c} \mathbf{Z}_v^z$ up to multiplication with $\mathbf{Z}_e$ terms from the Hamiltonian. This leaves the subsystem of (3) vertex spins decoupled from a trivial product ground-state on the rest of the system, which can be removed while preserving the phase of matter

$$V \widetilde{V} H_{\mathrm{PCN}} \widetilde{V}^\dagger V^\dagger \sim -\sum_c \prod_{v \in c} \mathbf{Z}_v \,. \tag{237}$$

The point-cage-net picture for $H_{\mathrm{PCN}}$ is obtained similarly as in §3.3 by interpreting the cube term $\mathcal{B}_c$ as fusing a $\mathbb{Z}_2$ cage into the edges of the cube, the edge term $\mathcal{C}_e$ as fusing in a single link ending on a pair of points. The vertex terms $\mathcal{A}_v^\mu$ enforce $\mathbb{Z}_2$ parity constraints at $v$ which say that edges must appear as part of closed cage terms or end on points. The ground states are again given by picking a reference state which satisfies the $\mathbb{Z}_2$ parity constraints, and fusing in all possible cages and line segments into this reference state.

# 5 (3+1)D Anisotropic X-Cube Transitions

In this section, we study anisotropic layer constructions. In §5.1, we realize the X-cube model by gauging a certain planar zero-form subsystem symmetry group of a single stack of $\mathbb{Z}_2$ gauge theory layers, and in §5.2, we do something similar, but utilizing two orthogonal stacks instead of one. This culminates in §5.3, where we combine these two constructions into a single *string-string-net* parent model.

## 5.1 Single layer construction

We start by briefly revisiting (and slightly extending) the single-layer construction that first appeared in Ref. [64].

### 5.1.1 A gauge theory stack

Our starting model is obtained by filling space with gauge theory layers that live on the leaves of a foliation $\mathscr{F}_{\mathrm{xy}}^{\parallel}$ of space by planes parallel to the xy directions. To implement this system, we imagine first placing a qubit on each edge of a 3d cubic lattice, except for those edges that point in the z direction. (Thus, we place qubits on edges whose coordinates are either of the

form $e = (i + \frac{1}{2}, j, k)$ or $e = (i, j + \frac{1}{2}, k)$.) The Hamiltonian of such a decoupled stack is

$$
{}^2 H_{\mathbb{Z}_2}^{(1)}\left(\mathscr{F}_{\mathrm{xy}}^{\parallel}\right) = -U \sum_{e \parallel \mathrm{xy}} \mathbf{X}_e - t \sum_{p \parallel \mathrm{xy}} \prod_{e \in p} \mathbf{Z}_e \,,
$$

$$
G_v = \prod_{\substack{e \ni v \\ e \parallel \mathrm{xy}}} \mathbf{X}_e \,,
\tag{238}
$$

where $e \parallel$ xy and $p \parallel$ xy indicate that the edge $e$ or plaquette $p$ should be parallel to the xy plane. Such a system has a

$$
\mathscr{G} := \mathbb{Z}_2^{(1,1)}\left(\mathscr{F}_{\mathrm{xy}}^{\parallel}\right),
\tag{239}
$$

planar one-form subsystem symmetry group associated with the one-form symmetries on each of the gauge theory layers. The generators of this symmetry are

$$
U_{\tilde{\gamma}, k} = \prod_{(\alpha, \beta) \in \tilde{\gamma}} \mathbf{X}_{\alpha, \beta, k} \,.
\tag{240}
$$

Here, $k$ is an index that labels the layer that the symmetry generator acts on, and $\tilde{\gamma} = \{(\alpha, \beta)\}$ is a path on the dual lattice of a 2d square lattice, thought of as the set of edges $e = (\alpha, \beta)$ on the original 2d lattice that it intersects.

Let $\mathscr{F}_{\mathrm{x}}^{\perp}$ and $\mathscr{F}_{\mathrm{y}}^{\perp}$ be the foliations whose leaves consist of planes perpendicular to the x and y directions, respectively. The $\mathscr{G}$ symmetry group admits a planar zero-form subsystem subgroup,

$$
\mathscr{H} := \mathbb{Z}_2^{(0,1)}\left(\mathscr{F}_{\mathrm{x}}^{\perp}, \mathscr{F}_{\mathrm{y}}^{\perp}\right),
\tag{241}
$$

which can be defined through the following generators,

$$
U_{i+\frac{1}{2}}^{\mathrm{x}} = \prod_{j,k} \mathbf{X}_{i+\frac{1}{2}, j, k} \,, \qquad U_{j+\frac{1}{2}}^{\mathrm{y}} = \prod_{i,k} \mathbf{X}_{i, j+\frac{1}{2}, k} \,.
\tag{242}
$$

That they commute with the Hamiltonian follows from the fact that they are suitable products of the topological line operators of the $\mathbb{Z}_2$ gauge theory constituents, Eq. (240), over the layers of the foliation. See Figure 10 for an illustration.

In the language of §2.1, we could describe this planar zero-form subsystem subgroup using the notions of refinement and coarsening. Specifically, the foliation $\mathscr{F}_{\mathrm{xy}}^{\parallel}$ admits a refinement $\mathscr{F}_{\mathrm{x}}^{\parallel}$ whose leaves are lines that point in the x direction; associated with this refinement, there is a $\mathbb{Z}_2^{(0,2)}(\mathscr{F}_{\mathrm{x}}^{\parallel}) < \mathbb{Z}_2^{(1,1)}(\mathscr{F}_{\mathrm{xy}}^{\parallel})$ subgroup. We can then coarsen the foliation $\mathscr{F}_{\mathrm{x}}^{\parallel}$ to another foliation $\mathscr{F}_{\mathrm{y}}^{\perp}$ whose leaves are simply all the planes which are perpendicular to the y direction; this in turn leads to a $\mathbb{Z}_2^{(0,1)}(\mathscr{F}_{\mathrm{y}}^{\perp}) < \mathbb{Z}_2^{(0,2)}(\mathscr{F}_{\mathrm{x}}^{\parallel})$ subgroup, whose generators are the unitaries in Eq. (242). This process is described in Figure 11.

Although we do not have much to say about it here, we comment for completeness that it is natural to introduce an inter-layer coupling of the form

$$
H_{\text{1-stack}} = {}^2 H_{\mathbb{Z}_2}^{(1)}(\mathscr{F}_{\mathrm{xy}}^{\parallel}) + H_{\mathrm{C}} \,,
$$

$$
H_{\mathrm{C}} = -K_Z \sum_{(\alpha, \beta, k) \parallel \mathrm{xy}} \mathbf{Z}_{\alpha, \beta, k} \mathbf{Z}_{\alpha, \beta, k+1} \,,
\tag{243}
$$

which preserves the $\mathscr{H}$ symmetry, where above the sum over $(\alpha, \beta, k)$ is a sum over edges parallel to the xy directions. We demonstrate how this coupling arises in the string-string-net model of §5.3.

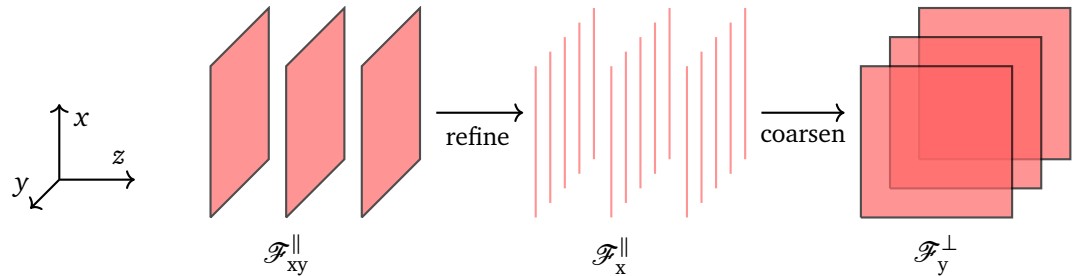

Figure 11: The foliations $\mathscr{F}_{\mathrm{xy}}^{\parallel}$, $\mathscr{F}_{\mathrm{x}}^{\parallel}$, and $\mathscr{F}_{\mathrm{y}}^{\perp}$ associated to the sequence of subgroups $\mathbb{Z}_2^{(0,1)}(\mathscr{F}_{\mathrm{y}}^{\perp}) < \mathbb{Z}_2^{(0,2)}(\mathscr{F}_{\mathrm{x}}^{\parallel}) < \mathbb{Z}_2^{(1,1)}(\mathscr{F}_{\mathrm{xy}}^{\parallel})$.

**Phases and excitations**

Set $K_Z = 0$ for simplicity. As described in §2.2.2, the individual $\mathbb{Z}_2$ gauge theory layers of $^2H_{\mathbb{Z}_2}^{(1)}(\mathscr{F}_{\mathrm{xy}}^{\parallel})$ undergo a spontaneous one-form symmetry breaking transition as the competition between $t$ and $U$ is varied; one can think of this as a planar one-form subsystem symmetry breaking transition of the layered system. In the present case, we are interested in the behavior of the $\mathscr{H}$ subgroup as the model crosses its phase transition point. Here too, this group is spontaneously broken, but only partially. More precisely, any single operator $U_{j+\frac{1}{2}}^{\mathrm{y}}$ or $U_{i+\frac{1}{2}}^{\mathrm{x}}$ acts non-trivially on the ground state space, but a pair of such operators $U_{j+\frac{1}{2}}^{\mathrm{y}} U_{j'+\frac{1}{2}}^{\mathrm{y}}$ or $U_{i+\frac{1}{2}}^{\mathrm{x}} U_{i'+\frac{1}{2}}^{\mathrm{x}}$ acts trivially, essentially due to the topological property of line operators which generate a one-form symmetry in (2+1)D. So the model's $\mathscr{H}$ planar subsystem symmetry group is spontaneously broken down to the subgroup generated by pairs of generators. Since this system simply consists of decoupled layers, the excitations can be determined from the discussion of §2.2.2.

**Order and disorder parameters**

As in previous sections, there are several choices of order and disorder parameters. We start with the disorder parameters. One option is to take a truncated version of the symmetry operators for the planar zero-form subsystem symmetries $\mathscr{H}$ from Eq. (242),

$$\mathscr{D}_{\tilde{R}_{\mathrm{yz}}}^{\mathrm{x}} = \prod_{e \in \tilde{R}_{\mathrm{yz}}} \mathbf{X}_e, \quad \mathscr{D}_{\tilde{R}_{\mathrm{xz}}}^{\mathrm{y}} = \prod_{e \in \tilde{R}_{\mathrm{xz}}} \mathbf{X}_e, \tag{244}$$

where $\tilde{R}_{\mathrm{xz}}$ and $\tilde{R}_{\mathrm{yz}}$ are rectangular open membranes on the dual lattice which are parallel to the xz and the yz directions, respectively. These are products of disorder parameters over the individual $\mathbb{Z}_2$ lattice gauge theory layers, and so their behavior in the different phases is straightforwardly determined,

$$\langle \mathscr{D}_{\tilde{R}_{\mathrm{yz}}}^{\mathrm{x}} \rangle \xrightarrow{|\tilde{R}_{\mathrm{yz}}| \to \infty} \begin{cases} \# e^{-|\tilde{R}_{\mathrm{yz}}|_z / \xi}, & U/t \gg 1 \\ \# e^{-A(\tilde{R}_{\mathrm{yz}})/A_0}, & U/t \ll 1. \end{cases} \tag{245}$$

In the above, $|\tilde{R}_{\mathrm{yz}}|_z$ is the length of $\tilde{R}_{\mathrm{yz}}$ in the z direction, and $A(\tilde{R}_{\mathrm{yz}})$ is its area. We write $|\tilde{R}_{\mathrm{yz}}| \to \infty$ to indicate that all dimensions of $\tilde{R}_{\mathrm{yz}}$ should be taken large.

Another option is to consider a truncated symmetry operator for the planar one-form symmetry $\mathscr{G}$ from Eq. (240), i.e.

$$\mathscr{D}_{\tilde{\gamma}(p,p'),k} = U_{\tilde{\gamma}(p,p'),k}, \tag{246}$$

where here, $\tilde{\gamma}(p, p')$ is an *open* path on the dual lattice of a 2d square lattice with plaquette endpoints $p$ and $p'$. This behaves as

$$\langle \mathscr{D}_{\tilde{\gamma}(p,p'),k} \rangle \xrightarrow{|p-p'|\to\infty} = \begin{cases} \text{const.}, & U/t \gg 1, \\ \# e^{-|p-p'|/\tilde{\xi}}, & U/t \ll 1. \end{cases} \tag{247}$$

For an order parameter, we take a product of two Wegner-Wilson loops that are supported on distantly separated layers,

$$\mathscr{O}_{\gamma,k,k'} = \prod_{(\alpha,\beta)\in\gamma} \mathbf{Z}_{\alpha,\beta,k} \mathbf{Z}_{\alpha,\beta,k'}, \tag{248}$$

where $\gamma = \{(\alpha, \beta)\}$ is a path on a generic 2d square lattice whose edges have coordinates $e = (\alpha, \beta)$. This behaves as

$$\langle \mathscr{O}_{\gamma,k,k'} \rangle \xrightarrow{|\gamma|\to\infty} \begin{cases} \# e^{-A(\gamma)/\tilde{A}_0}, & U/t \gg 1, \\ \# e^{-P(\gamma)/\xi'}, & U/t \ll 1. \end{cases} \tag{249}$$

### 5.1.2 Gauging a planar zero-form subsystem subgroup

Now we may consider gauging the subgroup $\mathscr{H} = \mathbb{Z}_2^{(0,1)}(\mathscr{F}_{\mathrm{x}}^{\perp}, \mathscr{F}_{\mathrm{y}}^{\perp})$. Since $\mathscr{H}$ is free of relations, we can achieve this by gauging $\mathbb{Z}_2^{(0,1)}(\mathscr{F}_{\mathrm{x}}^{\perp})$ in one step and $\mathbb{Z}_2^{(0,1)}(\mathscr{F}_{\mathrm{y}}^{\perp})$ in the next. To gauge $\mathbb{Z}_2^{(0,1)}(\mathscr{F}_{\mathrm{y}}^{\perp})$, we introduce a qubit to each plaquette $p$ of the cubic lattice which is parallel either to the xy plane or the yz plane; we label the Pauli operators which act on these qubits $\mathbf{X}_p^{\mathrm{y}}, \mathbf{Z}_p^{\mathrm{y}}$. Similarly, to gauge $\mathbb{Z}_2^{(0,1)}(\mathscr{F}_{\mathrm{x}}^{\perp})$, we place a qubit on each plaquette which is parallel to either the xy plane or the xz plane; we label Pauli operators that act on these qubits as $\mathbf{X}_p^{\mathrm{x}}, \mathbf{Z}_p^{\mathrm{x}}$. Thus in total, if we gauge the combination of these two symmetries, we should place *two* qubits on each plaquette parallel to the xy plane, and one qubit on each plaquette parallel to the xz plane or yz plane.

Then, the gauged Hamiltonian (using the strict gauging procedure) takes the form

$$H_{1\text{-stack}} \big/ \mathbb{Z}_2^{(0,1)}(\mathscr{F}_{\mathrm{x}}^{\perp}, \mathscr{F}_{\mathrm{y}}^{\perp}) = -U \sum_{e\|\mathrm{xy}} \mathbf{X}_e - t \sum_{p\|\mathrm{xy}} \mathbf{Z}_p^{\mathrm{x}} \mathbf{Z}_p^{\mathrm{y}} \prod_{e\in p} \mathbf{Z}_e$$

$$- K_Z \sum_{i,j,k} \left( \mathbf{Z}_{i+\frac{1}{2},j,k} \mathbf{Z}_{i+\frac{1}{2},j,k+\frac{1}{2}}^{\mathrm{x}} \mathbf{Z}_{i+\frac{1}{2},j,k+1} + \mathbf{Z}_{i,j+\frac{1}{2},k} \mathbf{Z}_{i,j+\frac{1}{2},k+\frac{1}{2}}^{\mathrm{y}} \mathbf{Z}_{i,j+\frac{1}{2},k+1} \right),$$

$$G_v = \prod_{\substack{e\ni v \\ e\|\mathrm{xy}}} \mathbf{X}_e, \quad \text{for all } v, \qquad \widetilde{G}_e = \begin{cases} \mathbf{X}_e \prod_{p\ni e} \mathbf{X}_p^{\mathrm{x}}, & \text{if } e \parallel \mathrm{x}, \\ \mathbf{X}_e \prod_{p\ni e} \mathbf{X}_p^{\mathrm{y}}, & \text{if } e \parallel \mathrm{y}, \end{cases} \tag{250}$$

$$F_c^{\mu} = \prod_{\substack{p\in c \\ p\|\mu}} \mathbf{Z}_p^{\mu}, \quad \text{for} \quad \mu = \mathrm{x}, \mathrm{y}.$$

The operators $\widetilde{G}_e$ are the Gauss's law constraints associated to this gauging, and the $F_c^{\mu}$ are local flux operators associated to the fundamental cubes $c$ of the cubic lattice on which the model is defined.

As in previous sections, we can perform a series of manipulations to make the physics of this model more manifest. We start by applying a local unitary circuit of the form

$$V = H^{\otimes N} \prod_{e\|\mathrm{x}} \prod_{p\ni e} C_e \mathbf{X}_p^{\mathrm{x}} \prod_{e\|\mathrm{y}} \prod_{p\ni e} C_e \mathbf{X}_p^{\mathrm{y}}, \tag{251}$$

where $N$ is the total number of qubits in the gauged model. This acts on the gauged Hamiltonian as

$$
V\left(H_{1\text{-stack}}\Big/\mathbb{Z}_2^{(0,1)}(\mathscr{F}_{\mathrm{x}}^\perp,\mathscr{F}_{\mathrm{y}}^\perp)\right)V^\dagger =
$$

$$
-U\left(\sum_{e\|\mathrm{x}}\mathbf{Z}_e\prod_{p\ni e}\mathbf{Z}_p^{\mathrm{x}}+\sum_{e\|\mathrm{y}}\mathbf{Z}_e\prod_{p\ni e}\mathbf{Z}_p^{\mathrm{y}}\right)-t\sum_{p\|\mathrm{xy}}\mathbf{X}_p^{\mathrm{x}}\mathbf{X}_p^{\mathrm{y}}-K_Z\left(\sum_{p\|\mathrm{yz}}\mathbf{X}_p^{\mathrm{y}}+\sum_{p\|\mathrm{xz}}\mathbf{X}_p^{\mathrm{x}}\right)
$$

$$
VG_v V^\dagger=\prod_{\substack{e\ni v\\ e\|\mathrm{xy}}}\mathbf{Z}_e\prod_{\substack{p\ni v\\ p\|\mathrm{xy,xz}}}\mathbf{Z}_p^{\mathrm{x}}\prod_{\substack{p\ni v\\ p\|\mathrm{xy,yz}}}\mathbf{Z}_p^{\mathrm{y}},\qquad V\widetilde{G}_e V^\dagger=\mathbf{Z}_e\,, \tag{252}
$$

$$
VF_c^\mu V^\dagger=\prod_{\substack{p\in c\\ p\|\mu}}\mathbf{X}_p^\mu\ \text{for}\ \mu=\mathrm{x,y}.
$$

The Gauss's law constraint then simply freezes the qubits on the edges so that they are in $+1$ eigenstates of Pauli-$\mathbf{Z}$. Therefore, after switching perspectives to the dual lattice and solving the Gauss's law constraint, the model becomes

$$
V\left(H_{1\text{-stack}}\Big/\mathbb{Z}_2^{(0,1)}(\mathscr{F}_{\mathrm{x}}^\perp,\mathscr{F}_{\mathrm{y}}^\perp)\right)V^\dagger\xrightarrow{V\widetilde{G}_{\tilde p}V^\dagger=1}
$$

$$
-U\left(\sum_{\tilde p\|\mathrm{xz}}\prod_{\tilde e\in\tilde p}\mathbf{Z}_{\tilde e}^{\mathrm{y}}+\sum_{\tilde p\|\mathrm{yz}}\prod_{\tilde e\in\tilde p}\mathbf{Z}_{\tilde e}^{\mathrm{x}}\right)-t\sum_{\tilde e\|\mathrm{z}}\mathbf{X}_{\tilde e}^{\mathrm{x}}\mathbf{X}_{\tilde e}^{\mathrm{y}}-K_Z\left(\sum_{\tilde e\|\mathrm{x}}\mathbf{X}_{\tilde e}^{\mathrm{y}}+\sum_{\tilde e\|\mathrm{y}}\mathbf{X}_{\tilde e}^{\mathrm{x}}\right) \tag{253}
$$

$$
VG_{\tilde c}V^\dagger\xrightarrow{V\widetilde{G}_{\tilde p}V^\dagger=1}\prod_{\substack{\tilde e\in\tilde c\\ \tilde e\|\mathrm{x}}}\mathbf{Z}_{\tilde e}^{\mathrm{x}}\prod_{\substack{\tilde e\in\tilde c\\ \tilde e\|\mathrm{y}}}\mathbf{Z}_{\tilde e}^{\mathrm{y}}\prod_{\substack{\tilde e\in\tilde c\\ \tilde e\|\mathrm{z}}}\mathbf{Z}_{\tilde e}^{\mathrm{x}}\mathbf{Z}_{\tilde e}^{\mathrm{y}},\quad VF_{\tilde v}^\mu V^\dagger\xrightarrow{V\widetilde{G}_{\tilde p}V^\dagger=1}\prod_{\substack{\tilde e\ni\tilde v\\ \tilde e\perp\mu}}\mathbf{X}_{\tilde e}^\mu\,.
$$

We notice that the symmetry group $\mathscr{H}$ which was present before gauging now disappears in the gauged model; an emergent quantum global symmetry group

$$
\widehat{\mathscr{H}}=\widehat{\mathbb{Z}}_2^{(1,1)}\left(\mathscr{F}_{\mathrm{x}}^\perp,\mathscr{F}_{\mathrm{y}}^\perp\right), \tag{254}
$$

which is generated by the flux terms $VF_{\tilde v}^\mu V^\dagger$, takes its place. The original planar one-form subsystem symmetry group $\mathscr{G}$ associated with the gauge theory layers persists in the gauged model, and is generated by the Gauss's law operators $VG_{\tilde c}V^\dagger$. Because it has a subgroup that is gauged, it is more accurate to refer to the symmetry group of the gauged model as

$$
\mathscr{G}\Big/\mathscr{H}=\mathbb{Z}_2^{(1,1)}\left(\mathscr{F}_{\mathrm{xy}}^\|\right)\Big/\mathbb{Z}_2^{(0,1)}\left(\mathscr{F}_{\mathrm{x}}^\perp,\mathscr{F}_{\mathrm{y}}^\perp\right). \tag{255}
$$

Indeed, one can straightforwardly check that forming the analogs of the operators from Eq. (242) in the gauged model simply produces the identity operator.

**Phases and excitations**

Now we consider the phase diagram of this model. When $U\ll t$, since $\mathscr{G}$ is unbroken before gauging, we expect $\mathscr{G}\big/\mathscr{H}$ to be unbroken after gauging as well. On the other hand, Expectation 2 from §2.1 suggests that $\widehat{\mathscr{H}}$ should be completely broken in the gauged theory since $\mathscr{H}$ is completely unbroken in the ungauged theory. When $t\gg U$, the group $\mathscr{G}$ is broken in the ungauged model, and we should correspondingly find that the group $\mathscr{G}\big/\mathscr{H}$ is broken in the gauged model as well. On the other hand, $\mathscr{H}$ is partially broken in the ungauged model, so we expect that $\widehat{\mathscr{H}}$ is also partially broken. Let us check these expectations explicitly.

*X-cube model at $K_X=K_Z=0$ and $t\gg U$*

When $t \gg U$, the low energy Hilbert space becomes the ground state of the term proportional to $t$, which consists of one effective qubit per dual edge $\tilde{e}$ (so we can remove the superscript on the Pauli operators). Within this low energy Hilbert space, the gauged Hamiltonian becomes 0, and the constraints become

$$V G_{\tilde{c}} V^{\dagger} \xrightarrow[V \widetilde{G}_{\tilde{p}} V^{\dagger}=1]{\overset{t \gg U}{}} \prod_{\tilde{e} \in \tilde{c}} \mathbf{Z}_{\tilde{e}} \,, \qquad V F_{\tilde{v}}^{\mu} V^{\dagger} \xrightarrow[V \widetilde{G}_{\tilde{p}} V^{\dagger}=1]{\overset{t \gg U}{}} \prod_{\substack{\tilde{e} \ni \tilde{v} \\ \tilde{e} \perp \mu}} \mathbf{X}_{\tilde{e}} \,, \qquad \text{for} \quad \mu = \mathrm{x, y}. \tag{256}$$

The Gauss's law constraint of the original gauge theory layers becomes the cube term of the X-cube model, i.e. in this limit, the group $\mathcal{G}/\mathcal{H}$ goes over to the group generated by the cube terms of the X-cube model, which is completely spontaneously broken.

The flux terms associated to the planar zero-form subsystem gauging become two of the three vertex terms of the X-cube model. Since the third vertex term is the product of the two appearing in Eq. (256), it is satisfied automatically, and thus the (constrained) low energy Hilbert space of ${}^{2}H_{\mathbb{Z}_2}^{(1)}(\mathscr{F}_{\mathrm{xy}}^{\parallel})/\mathbb{Z}_2^{(0,1)}(\mathscr{F}_{\mathrm{x}}^{\perp}, \mathscr{F}_{\mathrm{y}}^{\perp})$ in the large $t$ limit can be identified with the ground state space of the X-cube model. In this limit the emergent quantum symmetry group $\widehat{\mathcal{H}}$ goes over to the usual planar one-form subsystem symmetry group of the X-cube model generated by its vertex terms; the relations of the X-cube model are obeyed as a consequence of the fact that $\widehat{\mathcal{H}}$ is only partially broken (cf. Expectation 2 from §2.1).

*Two stacks of gauge theory layers at $K_X = K_Z = 0$ and $U \gg t$*

If we take $U \gg t$, then the low energy Hilbert space becomes the ground state of the term proportional to $U$. This term resembles the plaquette operators of two stacks of toric codes layers spanning the xz and yz directions. In this low energy Hilbert space, the operators $V G_{\tilde{c}} V^{\dagger}$ all act trivially, and so the constraint associated to them is automatically satisfied. Related to this, the group $\mathcal{G}/\mathcal{H}$ is unbroken.

The flux terms $V F_{\tilde{v}}^{\mu} V^{\dagger}$ act as the vertex terms of two stacks of toric code layers. Thus in total we see that in this phase the model at low energies simply resembles a foliation of space by two decoupled stacks of toric codes. The emergent symmetry $\widehat{\mathcal{H}}$ coincides with the relation-free planar one-form subsystem symmetry of these two stacks. It is completely spontaneously broken.

*Excitations and phase transition at $K_X = K_Z = 0$ between $t \ll U$ and $t \gg U$*

Excitations of the $\mathbf{X}_e$ terms in the ungauged Hamiltonian (in its trivial phase) are mapped after gauging $\mathcal{H}$ to gauge charges on one of the decoupled layers of $\mathbb{Z}_2$ gauge theory. As $t$ increases, composite loop excitations consisting of these gauge charges are created and fluctuated within xy planes by the plaquette terms. Hence the phase transition from two decoupled stacks of $\mathbb{Z}_2$ gauge theory to the X-cube phase is induced by xy-planar p-string condensation of gauge charges from the $\mathbb{Z}_2$ gauge theory layers.

Similarly, the gauge flux excitations on the layers of the ungauged Hamiltonian (in the topological phase) are mapped after gauging to composite planon excitations formed by dipoles of lineons in the X-cube phase. This equivalence follows by considering the local charge pattern created by a $\mathbf{Z}_p^{\mathrm{x/y}}$ operator, which excites the gauged xy-plaquette term along with a pair of adjacent flux terms $F_c^{\mathrm{x/y}}$. Hence as $U$ increases, the phase transition described above from the X-cube phase to two decoupled stacks of $\mathbb{Z}_2$ gauge theory is induced by the fluctuation and condensation of composite xy-planon excitations that consist of lineon dipoles.

**Order and disorder parameters**
To obtain order and disorder parameters that diagnose the phase transition from two decou-

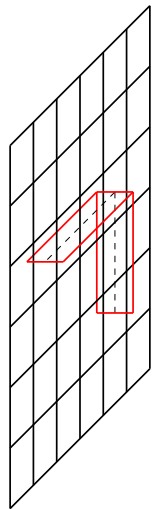

Figure 12: One xy slice of the 3d cubic lattice. The dashed line is the path $\tilde{\gamma}(p, p')$, and the red lines are edges of the dual lattice of the 3d cubic lattice which form the ribbon $\Gamma(p, p')$.

pled toric code layers to the X-cube model, we can study how the parameters from the ungauged model are mapped through the gauging map.

Starting with the disorder parameters for the subsystem symmetry from Eq. (244), we find that it maps to

$$V\left(\mathscr{D}^{\mathrm{x}}_{\tilde{R}_{\mathrm{yz}}}\Big/\mathbb{Z}_2^{(0,1)}(\mathscr{F}_{\mathrm{x}}^\perp, \mathscr{F}_{\mathrm{y}}^\perp)\right)V^\dagger \xrightarrow{V\widetilde{G}_{\tilde{p}}V^\dagger = 1} \prod_{\tilde{e}\in\partial\tilde{R}_{\mathrm{yz}}} \mathbf{Z}_e^{\mathrm{x}}, \tag{257}$$

where $\partial\tilde{R}_{\mathrm{yz}}$ is the path on the dual lattice which borders $\tilde{R}_{\mathrm{yz}}$. This is a Wegner-Wilson loop of the gauged model that is charged under (and thus can serve as an order parameter for) the emergent quantum planar one-form symmetry $\widehat{\mathbb{Z}}_2^{(1,1)}(\mathscr{F}_{\mathrm{x}}^\perp, \mathscr{F}_{\mathrm{y}}^\perp)$.

On the other hand, consider the disorder parameter $\mathscr{D}_{\tilde{\gamma}(p,p'),k}$ for the planar one-form subsystem symmetry from Eq. (246). Recall that $\tilde{\gamma}(p, p')$ is a path on the dual lattice of a generic 2d square lattice. Thicken it in the z direction to a ribbon $\Gamma(p, p')$ as in Figure 12. Then the claim is that

$$V\left(\mathscr{D}_{\tilde{\gamma}(p,p'),k}\Big/\mathbb{Z}_2^{(0,1)}(\mathscr{F}_{\mathrm{x}}^\perp, \mathscr{F}_{\mathrm{y}}^\perp)\right)V^\dagger \xrightarrow{V\widetilde{G}_{\tilde{p}}V^\dagger = 1}$$

$$\mathbf{Z}_{\tilde{e}_{\mathrm{i}}}^{\mu_{\mathrm{i}}}\left(\prod_{\substack{\tilde{e}\in\Gamma(p,p')\\ \tilde{e}\|\mathrm{x}}} \mathbf{Z}_{\tilde{e}}^{\mathrm{x}} \prod_{\substack{\tilde{e}\in\Gamma(p,p')\\ \tilde{e}\|\mathrm{y}}} \mathbf{Z}_{\tilde{e}}^{\mathrm{y}} \prod_{\substack{\tilde{e}\in\Gamma(p,p')\\ \tilde{e}\|\mathrm{z},\,\tilde{e}\neq\tilde{e}_{\mathrm{i}},\tilde{e}_{\mathrm{f}}}} \mathbf{Z}_{\tilde{e}}^{\mathrm{x}}\mathbf{Z}_{\tilde{e}}^{\mathrm{y}}\right)\mathbf{Z}_{\tilde{e}_{\mathrm{f}}}^{\mu_{\mathrm{f}}}. \tag{258}$$

In the above expression, $\tilde{e}_{\mathrm{i}}$ and $\tilde{e}_{\mathrm{f}}$ are the edges on the dual lattice that correspond to the plaquettes $p, p'$ on the original lattice, and $\mu_{\mathrm{i}}$ and $\mu_{\mathrm{f}}$ are the initial and final directions of the path $\tilde{\gamma}$. By comparing this to the operators $VG_{\tilde{c}}V^\dagger$ in Eq. (253), one finds that this is a truncated symmetry operator for the $\mathbb{Z}_2^{(1,1)}(\mathscr{F}_{\mathrm{xy}}^\|)\Big/\mathbb{Z}_2^{(0,1)}(\mathscr{F}_{\mathrm{x}}^\perp, \mathscr{F}_{\mathrm{y}}^\perp)$ symmetry of the gauged theory, and therefore $\mathscr{D}_{\tilde{\gamma}(p,p'),k}$ retains its interpretation as a disorder parameter.

And finally, the order parameter from Eq. (248) is mapped as

$$V\left(\mathscr{O}_{\gamma,k,k'}\Big/\mathbb{Z}_2^{(0,1)}(\mathscr{F}_{\mathrm{x}}^\perp, \mathscr{F}_{\mathrm{y}}^\perp)\right)V^\dagger \xrightarrow{V\widetilde{G}_{\tilde{p}}V^\dagger = 1} \prod_{(\alpha,\beta)\in\gamma} \prod_{\substack{k''\\ k\leq k''<k'}} \mathbf{X}_{\alpha,\beta,k+\frac{1}{2}}, \tag{259}$$

which is a product of truncated symmetry operators for the emergent $\widehat{\mathcal{H}}$ symmetry. Hence, this can serve as a disorder parameter.

### 5.2 Double layer construction

In this section, we treat a construction of the X-cube model by gauging a planar zero-form subsystem symmetry group of two orthogonal stacks of $\mathbb{Z}_2$ lattice gauge theory layers. This construction is based on Ref. [65].

#### 5.2.1 Two gauge theory stacks

Consider two orthogonal foliations $\mathcal{F}_y^\perp$ and $\mathcal{F}_x^\perp$ of 3d space by 2d planes, the first set of planes stretching in the xz directions and the second set stretching in the yz directions. We imagine placing $\mathbb{Z}_2$ lattice gauge theories on the leaves of these foliations. On the cubic lattice, this can be accomplished by placing two qubits on each of the edges that point in the z direction, and placing one qubit on each of the edges that point in the remaining x or y directions. Pauli operators acting on the qubits of the gauge theory layers that are parallel to the xz plane are denoted $\mathbf{X}_e^y, \mathbf{Z}_e^y$, while the qubits of the layers that are parallel to the yz plane are denoted $\mathbf{X}_e^x, \mathbf{Z}_e^x$. The Hamiltonian of these two decoupled stacks is then

$$
\begin{aligned}
{}^2H_{\mathbb{Z}_2}^{(1)}(\mathcal{F}_x^\perp, \mathcal{F}_y^\perp) &= -\sum_{\mu=x,y}\left( U\sum_{e\perp\mu}\mathbf{X}_e^\mu + t\sum_{p\perp\mu}\prod_{e\in p}\mathbf{Z}_e^\mu \right), \\
G_\nu^\mu &= \prod_{\substack{e\ni\nu \\ e\perp\mu}}\mathbf{X}_e^\mu, \quad \text{for all } \nu, \quad \text{for} \quad \mu=x,y.
\end{aligned}
\tag{260}
$$

This model admits a planar one-form subsystem symmetry group

$$
\mathcal{G} = \mathbb{Z}_2^{(1,1)}\left( \mathcal{F}_x^\perp, \mathcal{F}_y^\perp \right),
\tag{261}
$$

corresponding to the two foliations of space by gauge theory layers. This symmetry is spontaneously broken as the coupling $t/U$ is tuned from 0 to $\infty$. For the purposes of our construction, we can focus on a particular

$$
\mathcal{H} = \mathbb{Z}_2^{(0,1)}\left( \mathcal{F}_{xy}^\| \right),
\tag{262}
$$

subgroup of $\mathcal{G}$, where $\mathcal{F}_{xy}^\|$ is the foliation of space by planes which stretch in the xy directions. Its symmetry generators are

$$
U_{k+\frac{1}{2}}^{xy} = \prod_{i,j}\mathbf{X}_{i,j,k+\frac{1}{2}}^x \mathbf{X}_{i,j,k+\frac{1}{2}}^y.
\tag{263}
$$

The reason $\mathcal{H}$ is a subgroup of the planar one-form subsystem symmetry group is that these operators can be expressed as suitable products of the line operators on the gauge theory layers which generate their one-form symmetries.

It is possible to couple these two stacks while preserving the $\mathcal{H}$ subgroup,

$$
H_{\text{2-stack}} = {}^2H_{\mathbb{Z}_2}^{(1)}\left( \mathcal{F}_x^\perp, \mathcal{F}_y^\perp \right) + H_C,
\tag{264}
$$

by adding the term

$$
H_C = -\sum_{e\|z}\left( K_X\mathbf{X}_e^x\mathbf{X}_e^y + K_Z\mathbf{Z}_e^x\mathbf{Z}_e^y \right),
\tag{265}
$$

to the Hamiltonian. In fact, when $K_Z = 0$, the full $\mathcal{G}$ symmetry is preserved.

**Phases and excitations**

For simplicity, we just study the $K_X \to \infty$ limit, keeping $K_Z = 0$.

*X-cube model/$\mathbb{Z}_2$ tensor gauge theory at $K_X \to \infty$ and $K_Z = 0$*

In the $K_X \to \infty$ limit with $K_Z = 0$, the two qubits on the edges that point in the z direction reduce to a single effective qubit, and the effective theory which governs this low energy subspace (computed to fourth order in perturbation theory) is essentially (a deformation of) the X-cube model,

$$
\begin{aligned}
H_{\text{eff}} &\sim -\tilde{U} \sum_{e\|\text{x,y}} \mathbf{X}_e - \tilde{U}_z \sum_{e\|\text{z}} \mathbf{X}_e - \tilde{t} \sum_c \prod_{e\in c} \mathbf{Z}_e, \\
{}^{\text{eff}}G_\nu^\mu &= \prod_{\substack{e\in \nu \\ e\perp\mu}} \mathbf{X}_e, \quad \text{for all} \quad \nu, \quad \text{for} \quad \mu = \text{x,y},
\end{aligned}
\tag{266}
$$

where above, we have stripped the superscripts from the Pauli operators because there is now only a single qubit per edge. The $\mathcal{H}$ symmetry, Eq. (263), acts trivially on the low energy effective Hilbert space on which $H_{\text{eff}}$ acts, and so as $K_X$ is tuned from $\infty$ to 0, the model undergoes spontaneous symmetry breaking phase transition corresponding to the breaking of $\mathcal{H}$. In order for the model to transition into an X-cube phase, this must be the case, as the planar one-form subsystem symmetry of the X-cube model satisfies the relation that its $\mathbb{Z}_2^{(0,1)}(\mathscr{F}_{\text{xy}}^\|)$ subgroup is trivially realized.

This construction of the X-cube model is similar in spirit to the isotropic coupled layer construction of Refs. [61, 62], which we revisit from a different point of view in §6.1.1. In particular, the transition from anistropic decoupled $\mathbb{Z}_2$ gauge theory layers to the X-cube model in the present case is also mediated by a p-string condensation mechanism, though here, the strings that are being condensed lie strictly parallel to the xy planes. Since the mechanism in the anisotropic case is otherwise identical to the isotropic case, we omit the details here, and refer readers to Refs. [64, 65] (see also Ref. [113]).

**Order and disorder parameters**

We now discuss choices of order and disorder parameters that are able to diagnose the planar confinement/deconfinement phase transition of the decoupled layers at $K_X = K_Z = 0$ (we anticipate that they are strong enough to probe the broader phase diagram away from $K_X = K_Z = 0$ as well).

We begin with disorder parameters. One natural option is to consider a truncated symmetry operator for the $\mathcal{H}$ subgroup,

$$
\mathscr{D}_{\tilde{R}_{\text{xy}}}^{\text{xy}} = \prod_{e\in \tilde{R}_{\text{xy}}} \mathbf{X}_e^{\text{x}} \mathbf{X}_e^{\text{y}},
\tag{267}
$$

where here, $\tilde{R}_{\text{xy}}$ is an open rectangular membrane on the dual lattice which is parallel to the xy directions (and so only intersects edges which point in the z direction).

Another option is a truncated symmetry operator for the larger $\mathcal{G}$ symmetry, which takes the form

$$
\mathscr{D}_{\tilde{\gamma}(p,p'),i}^{\text{x}} = \prod_{(\alpha,\beta)\in\tilde{\gamma}(p,p')} \mathbf{X}_{i,\alpha,\beta}^{\text{x}}, \qquad \mathscr{D}_{\tilde{\gamma}(p,p'),j}^{\text{y}} = \prod_{(\alpha,\beta)\in\tilde{\gamma}(p,p')} \mathbf{X}_{\alpha,j,\beta}^{\text{y}},
\tag{268}
$$

where here, $\tilde{\gamma}(p,p')$ is an open path on the dual lattice of a generic 2d square lattice with endpoint plaquettes $p$ and $p'$.

For an order parameter, we will consider

$$\mathcal{O}_C = \prod_{\substack{e\in\text{outline}(C)\\ e\perp \text{x}}} \mathbf{Z}_e^{\text{x}} \prod_{\substack{e\in\text{outline}(C)\\ e\perp \text{y}}} \mathbf{Z}_e^{\text{y}}, \tag{269}$$

where $C$ is a cuboid, and outline($C$) is the set of edges which outline it. This order parameter can be thought of as a product of four Wilson loop operators, two associated to gauge theory layers which are perpendicular to the x direction and two associated to gauge theory layers which are perpendicular to the y direction. Therefore, it is a sort of order parameter for $\mathcal{G}$.

### 5.2.2 Gauging a planar zero-form subsystem subgroup

We now gauge the subgroup $\mathcal{H}$. To promote this global symmetry to a local one, we place qubits on the plaquettes of the lattice which are either parallel to the xz plane or the yz plane (but we do note place qubits on the plaquettes that are parallel to the xy plane). Then the gauged Hamiltonian takes the form

$$H_{\text{2-stack}}\Big/\mathbb{Z}_2^{(0,1)}(\mathscr{F}_{\text{xy}}^{\|}) = -\sum_{\mu=\text{x,y}}\left(U\sum_{e\perp\mu}\mathbf{X}_e^{\mu} + t\sum_{p\perp\mu}\mathbf{Z}_p\prod_{e\in p}\mathbf{Z}_e^{\mu}\right) + H_C$$

$$G_v^{\mu} = \prod_{\substack{e\ni v\\ e\perp\mu}}\mathbf{X}_e^{\mu} \text{ for all } v, \quad \text{for} \quad \mu=\text{x,y}, \quad \widetilde{G}_e = \mathbf{X}_e^{\text{x}}\mathbf{X}_e^{\text{y}}\prod_{p\ni e}\mathbf{X}_p, \quad \text{for} \quad e\parallel\text{z}, \tag{270}$$

$$F_c = \prod_{\substack{p\in c\\ p\parallel\text{xz,yz}}}\mathbf{Z}_p.$$

In the above, $\widetilde{G}_e$ is the Gauss's law constraint for the gauging of $\mathbb{Z}_2^{(0,1)}(\mathscr{F}_{\text{xy}}^{\|})$ and $F_c$ are local flux operators. We can solve Gauss's law by performing the following unitary circuit,

$$V = \prod_{e\parallel\text{z}}\prod_{\substack{p\ni e\\ p\perp\text{x}}}C_e^{\text{x}}\mathbf{X}_p\prod_{\substack{p\ni e\\ p\perp\text{y}}}C_e^{\text{y}}\mathbf{X}_p. \tag{271}$$

This acts on the gauged model as

$$V\left(H_{\text{2-stack}}\Big/\mathbb{Z}_2^{(0,1)}(\mathscr{F}_{\text{xy}}^{\|})\right)V^{\dagger} = -t\sum_{\mu=\text{x,y}}\sum_{p\perp\mu}\mathbf{Z}_p\prod_{\substack{e\in p\\ e\perp\text{z}}}\mathbf{Z}_e^{\mu}$$

$$-\sum_{e\parallel\text{z}}\left(K_X\mathbf{X}_e^{\text{x}}\mathbf{X}_e^{\text{y}}\prod_{p\ni e}\mathbf{X}_p + K_Z\mathbf{Z}_e^{\text{x}}\mathbf{Z}_e^{\text{y}}\right) - U\left(\sum_{e\parallel\text{x}}\mathbf{X}_e^{\text{y}} + \sum_{e\parallel\text{y}}\mathbf{X}_e^{\text{x}} + \sum_{e\parallel\text{z}}\sum_{\mu=\text{x,y}}\mathbf{X}_e^{\mu}\prod_{\substack{p\ni e\\ p\perp\mu}}\mathbf{X}_p\right) \tag{272}$$

$$VG_v^{\mu}V^{\dagger} = \prod_{\substack{e\ni v\\ e\perp\mu}}\mathbf{X}_e^{\mu}\prod_{\substack{p\ni v\\ p\perp\mu}}\mathbf{X}_p, \quad V\widetilde{G}_eV^{\dagger} = \mathbf{X}_e^{\text{x}}\mathbf{X}_e^{\text{y}}, \quad VF_cV^{\dagger} = \prod_{\substack{e\in c\\ e\parallel\text{z}}}\mathbf{Z}_e^{\text{x}}\mathbf{Z}_e^{\text{y}}\prod_{\substack{p\in c\\ p\parallel\text{xz,yz}}}\mathbf{Z}_p.$$

Thus, one can impose $\widetilde{G}_e = 1$ for all edges $e$ which point in the z direction by reducing from two qubits per such edges $e$ to a single effective qubit. In total there is only a single qubit on every edge of the cubic lattice, and so we can erase the superscripts from the Pauli operators.

The model becomes

$$V\left(H_{\text{2-stack}}\Big/\mathbb{Z}_2^{(0,1)}(\mathscr{F}_{\text{xy}}^\parallel)\right)V^\dagger \xrightarrow{V\widetilde{G}_e V^\dagger=1} -t\sum_{\substack{p\parallel\text{xz,yz}}}\mathbf{Z}_p\prod_{\substack{e\in p\\e\perp\text{z}}}\mathbf{Z}_e$$

$$-\sum_{e\parallel\text{z}}\left(K_X\prod_{p\ni e}\mathbf{X}_p+K_Z\mathbf{Z}_e\right)-U\left(\sum_{e\parallel\text{x,y}}\mathbf{X}_e+\sum_{e\parallel\text{z}}\sum_{\mu=\text{x,y}}\mathbf{X}_e\prod_{\substack{p\ni e\\p\perp\mu}}\mathbf{X}_p\right) \tag{273}$$

$$V G_\nu^\mu V^\dagger \xrightarrow{V\widetilde{G}_e V^\dagger=1} \prod_{\substack{e\ni\nu\\e\perp\mu}}\mathbf{X}_e\prod_{\substack{p\ni\nu\\p\perp\mu}}\mathbf{X}_p\,,\qquad V F_c V^\dagger \xrightarrow{V\widetilde{G}_e V^\dagger=1}\prod_{\substack{e\in c\\e\parallel\text{z}}}\mathbf{Z}_e\prod_{\substack{p\in c\\p\parallel\text{xz,yz}}}\mathbf{Z}_p\,.$$

We note that the original $\mathscr{G}$ symmetry persists in the gauged model in the form $\mathscr{G}\big/\mathscr{H}$, and is generated by the operators $V G_\nu^\mu V^\dagger$. On the other hand, the model gains another emergent planar one-form subsystem symmetry group from the gauging

$$\widehat{\mathscr{H}}=\widehat{\mathbb{Z}}_2^{(1,1)}\left(\mathscr{F}_{\text{xy}}^\parallel\right)\,, \tag{274}$$

which is generated by the flux terms, $V F_c V^\dagger$.

**Phases and excitations**
We now analyze the phase diagram.

*X-cube model at $K_X=K_Z=0$ and $t\gg U$*
   In order to make the physics more manifest in the $t\gg U$ limit, we perform one more local unitary circuit,

$$\widetilde{V}_1=\prod_{e\parallel\text{x,y}}\prod_{\substack{p\ni e\\p\perp\text{xy}}}C_e\mathbf{X}_p\,. \tag{275}$$

This brings the model to the form

$$\widetilde{V}_1 V\left({}^2H_{\mathbb{Z}_2}^{(1)}\left(\mathscr{F}_{\text{x}}^\perp,\mathscr{F}_{\text{y}}^\perp\right)\Big/\mathbb{Z}_2^{(0,1)}(\mathscr{F}_{\text{xy}}^\parallel)\right)V^\dagger\widetilde{V}_1^\dagger \xrightarrow{\widetilde{V}_1 V\widetilde{G}_e V^\dagger\widetilde{V}_1^\dagger=1} -t\sum_{p\perp\text{xy}}\mathbf{Z}_p$$

$$-U\left(\sum_{e\parallel\text{x,y}}\mathbf{X}_e\prod_{\substack{p\ni e\\p\perp\text{xy}}}\mathbf{X}_p+\sum_{e\parallel\text{z}}\sum_{\mu=\text{x,y}}\mathbf{X}_e\prod_{\substack{p\ni e\\p\perp\mu}}\mathbf{X}_p\right) \tag{276}$$

$$\widetilde{V}_1 V G_\nu^\mu V^\dagger\widetilde{V}_1^\dagger \xrightarrow{\widetilde{V}_1 V\widetilde{G}_e V^\dagger\widetilde{V}_1^\dagger=1} \prod_{\substack{e\ni\nu\\e\perp\mu}}\mathbf{X}_e\,,\quad\text{for all}\quad\nu\,,\quad\text{for}\quad\mu=\text{x,y,}$$

$$\widetilde{V}_1 V F_c V^\dagger\widetilde{V}_1^\dagger \xrightarrow{\widetilde{V}_1 V\widetilde{G}_e V^\dagger\widetilde{V}_1^\dagger=1} \prod_{e\in c}\mathbf{Z}_e\prod_{\substack{p\in c\\p\parallel\text{xz,yz}}}\mathbf{Z}_p\,.$$

Thus, in the large $t$ limit, the plaquette degrees of freedom are frozen to $\mathbf{Z}_p=1$ eigenstates. The Hamiltonian becomes a constant, the Gauss's law constraint $G_\nu^\mu$ goes over to (two of the three) vertex terms of the X-cube model, and the flux term $F_c$ goes over to the usual cube term of the X-cube model.

In the $t \gg U$ limit before gauging, the model is well-described by two stacks of toric code layers, and hence the original symmetry group $\mathscr{G}$ is spontaneously broken. Upon gauging, the $\mathscr{H}$ symmetry acts trivially which is consistent with the fact that the X-cube model enjoys a $\mathscr{G}/\mathscr{H}$ symmetry, with $\mathscr{H}$ describing the relations. This $\mathscr{G}/\mathscr{H}$ symmetry is completely spontaneously broken.

The subgroup $\mathscr{H}$ being gauged is also partially spontaneously broken, in a somewhat subtle way, as pairs of planar symmetries are preserved, similar to the construction in the previous subsection. The emergent symmetry group $\widehat{\mathscr{H}}$ is accordingly also partially spontaneously broken, as follows. Symmetry operators supported on individual planes are broken as the associated string operators create topologically nontrivial planons consisting of a dipole of X-cube fractons. At the same time, there is a subgroup generated by the product of an identical symmetry operator on each plane which remains unbroken as the truncation of such an operator creates no topological excitation.

*Single stack of toric code layers at $K_X = K_Z = 0$ and $U \gg t$*

Let us now analyze the opposite extreme limit, $U \gg t$. This time we simplify the physics with a slightly different local unitary circuit,

$$\widetilde{V}_2 = \prod_{e\|z} \prod_{\substack{p \ni e \\ p\|xz}} C_e \mathbf{X}_p. \tag{277}$$

In this case, the gauged Hamiltonian becomes

$$\widetilde{V}_2 V \left( {}^2 H^{(1)}_{\mathbb{Z}_2}(\mathscr{F}^\perp_x, \mathscr{F}^\perp_y) \Big/ \mathbb{Z}^{(0,1)}_2(\mathscr{F}^\|_{xy}) \right) V^\dagger \widetilde{V}_2^\dagger \xrightarrow{\widetilde{V}_2 V \widetilde{G}_e V^\dagger \widetilde{V}_2^\dagger = 1} -t \left( \sum_{p\|xz} \prod_{e\in p} \mathbf{Z}_e + \sum_{p\|yz} \prod_{\substack{e\in p \\ e\perp z}} \mathbf{Z}_e \right)$$

$$-U\left( \sum_{e\|x,y} \mathbf{X}_e + \sum_{e\|z} \mathbf{X}_e \left( 1 + \prod_{p\ni e} \mathbf{X}_p \right) \right)$$

$$\widetilde{V}_2 V G^x_\nu V^\dagger \widetilde{V}_2^\dagger \xrightarrow{\widetilde{V}_2 V \widetilde{G}_e V^\dagger \widetilde{V}_2^\dagger = 1} \prod_{\substack{e\ni\nu \\ e\perp x}} \mathbf{X}_e \prod_{p\ni\nu} \mathbf{X}_p, \qquad \widetilde{V}_2 V G^y_\nu V^\dagger \widetilde{V}_2^\dagger \xrightarrow{\widetilde{V}_2 V \widetilde{G}_e V^\dagger \widetilde{V}_2^\dagger = 1} \prod_{\substack{e\ni\nu \\ e\perp y}} \mathbf{X}_e,$$

$$\widetilde{V}_2 V F_c V^\dagger \widetilde{V}_2^\dagger \xrightarrow{\widetilde{V}_2 V \widetilde{G}_e V^\dagger \widetilde{V}_2^\dagger = 1} \prod_{\substack{p\in c \\ p\|xz,yz}} \mathbf{Z}_p. \tag{278}$$

In the $U \to \infty$ limit, the edge degrees of freedom are frozen into $\mathbf{X}_e = +1$ eigenstates, and $A_e := \prod_{p\ni e} \mathbf{X}_p = 1$ for all edges $e$ that point in the z direction. In this low energy subspace, the Gauss's law constraints $G^\mu_\nu = 1$ are automatically satisfied because the $G^\mu_\nu$ act trivially. (Accordingly, the inherited $\mathscr{G}/\mathscr{H}$ symmetry is unbroken in this phase.) The only remaining operators that we must impose are the flux operators. If one imagines the edges that are parallel to the z direction are the sites of a stack of 2d square lattices which span the xy directions, and the plaquettes that emanate from them are the edges of each layer, then the operators $A_e$ behave as the vertex terms of a stack of toric codes. Likewise the flux operators $\widetilde{V}_2 V F_c V^\dagger \widetilde{V}_2^\dagger$ serve as plaquette terms. Thus, deep in this phase, the model has a low energy effective Hamiltonian equal to zero, and a Hilbert space equal to the ground state of a stack of toric code layers. This is consistent with the emergent symmetry $\widehat{\mathscr{H}}$ being fully spontaneously broken, as the original symmetry group $\mathscr{H}$ before gauging is fully respected by the initial trivial phase.

*Excitations and phase transition at $K_X = K_Z = 0$ between $t \ll U$ and $t \gg U$*

For $t \ll U$, gauging the planar subsystem symmetry maps excitations of the $\mathbf{X}_e$ terms, in the trivial phase, to flux excitations on the layers of toric code. Increasing $t$ creates and fluctuates composite string excitations, formed by the flux excitations, over the planes that have been gauged. Hence the phase transition from a single stack of toric code layers to the X-cube phase, obtained by gauging a planar zero-form symmetry on two stacks of $\mathbb{Z}_2$ gauge theory layers undergoing a confinement/deconfinement transition, is driven by planar p-string condensation of composite string excitations.

For $t \gg U$ gauging the planar subsystem symmetry maps the magnetic flux excitations of the plaquette terms in the stacks of toric code layers to composite planon excitations. These planons are equivalent to a pair of fracton excitations in the X-cube phase, as a single $\mathbf{X}_p$ operator creates a local cluster consisting of a pair of fractonic excitations of adjacent cube terms $F_c$ along with a single gauged plaquette excitaiton. Increasing $U$ drives these xz and yz planons to fluctutate and condense resulting in a phase transition from the X-cube phase to a single stack of toric code layers.

**Order and disorder parameters**

We now compute how the order and disorder parameters of the ungauged model map under gauging. First, we compute that the disorder parameter associated to the $\mathscr{H}$ symmetry of the ungauged models maps as

$$V\left(\mathscr{D}^{\mathrm{xy}}_{\tilde{R}_{\mathrm{xy}}} \Big/ \mathbb{Z}_2^{(0,1)}(\mathscr{F}^{\|}_{\mathrm{xy}})\right) V^{\dagger} \xrightarrow{V\tilde{G}_e V^{\dagger}=1} \prod_{p \in \partial \tilde{R}_{\mathrm{xy}}} \mathbf{X}_p , \tag{279}$$

where here, we are thinking of $\partial \tilde{R}_{\mathrm{xy}}$ as a path on the dual lattice. We note that this operator is charged under the emergent quantum $\widehat{\mathbb{Z}}_2^{(1,1)}(\mathscr{F}^{\|}_{\mathrm{xy}})$ symmetry, and thus serves as an order parameter which diagnoses its spontaneous symmetry breaking.

On the other hand, the disorder parameter associated to the $\mathscr{G}$ symmetry of the ungauged theory maps as

$$V\left(\mathscr{D}^{\mathrm{x}}_{\tilde{\gamma}(p,p'),i} \Big/ \mathbb{Z}_2^{(0,1)}(\mathscr{F}^{\|}_{\mathrm{xy}})\right) V^{\dagger} \xrightarrow{V\tilde{G}_e V^{\dagger}=1} \mathbf{X}_p \left(\prod_{(\alpha,\beta) \in \tilde{\gamma}(p,p')} \mathbf{X}_{i,\alpha,\beta}\right) \mathbf{X}_{p'}. \tag{280}$$

Thus, it is mapped again to a truncated symmetry operator, this time with operators decorating its boundary. We interpret this as a disorder parameter for the global symmetry of the gauged model, $\mathscr{G}\big/\mathscr{H}$.

And finally, the order parameter we defined maps as

$$V\left(\mathscr{O}_C \Big/ \mathbb{Z}_2^{(0,1)}(\mathscr{F}^{\|}_{\mathrm{xy}})\right) V^{\dagger} \xrightarrow{V\tilde{G}_e V^{\dagger}=1} \prod_{e \in \mathrm{outline}(C)} \mathbf{Z}_e. \tag{281}$$

This retains its interpretation as an order parameter for the $\mathscr{G}\big/\mathscr{H}$ global symmetry.

## 5.3 The string-string-net model

In this final subsection, we move on to consider the *string-string-net* model for the anisotropic layer constructions presented in §5.1 and §5.2. This model combines layers of $\mathbb{Z}_2$ gauge theory on xz and yz planes with layers of $\mathbb{Z}_2$ gauge theory on dual xy planes in a nontrivial way.

The Hilbert space for this model is constructed on a cubic lattice with two qubits per z edge, one qubit per x and y edge, and one qubit per xz and yz plaquette,

$$\mathcal{H} = \bigotimes_{e \perp \mathrm{x}} \mathcal{H}^{\mathrm{x}}_e \bigotimes_{e \perp \mathrm{y}} \mathcal{H}^{\mathrm{y}}_e \bigotimes_{p \| \mathrm{z}} \mathcal{H}_p , \tag{282}$$

where

$$\mathcal{H}_e^{\mathrm{x}} \cong \mathcal{H}_e^{\mathrm{y}} \cong \mathcal{H}_p \cong \mathbb{C}^2 \,. \tag{283}$$

This can equivalently be viewed on the dual cubic lattice with two qubits per dual xy plaquette, one qubit per dual xz and yz plaquette, and one qubit per dual x and y edge,

$$\mathcal{H} = \bigotimes_{\tilde{p}\|\mathrm{x}} \mathcal{H}_{\tilde{p}}^{\mathrm{x}} \bigotimes_{\tilde{p}\|\mathrm{y}} \mathcal{H}_{\tilde{p}}^{\mathrm{y}} \bigotimes_{\tilde{e}\perp\mathrm{z}} \mathcal{H}_{\tilde{e}} \,, \tag{284}$$

where

$$\mathcal{H}_{\tilde{p}}^{\mathrm{x}} \cong \mathcal{H}_{\tilde{p}}^{\mathrm{y}} \cong \mathcal{H}_{\tilde{e}} \cong \mathbb{C}^2 \,. \tag{285}$$

A useful picture for the model is in terms of square lattice systems which have one qubit per edge. In particular, the degrees of freedom $\mathcal{H}_e^\mu$ ($\mathcal{H}_{\tilde{p}}^\mu$) are the same as those of two stacks of square lattice systems on the xz and yz planes of the cubic lattice, while the degrees of freedom $\mathcal{H}_{\tilde{e}}$ ($\mathcal{H}_p$) are the same as those of a single stack of square lattice systems on the xy planes of the dual lattice. The choice of stacking on either primal or dual planes depending on the direction is responsible for the anisotropy of the model.

The perturbed string-string-net Hamiltonian is

$$\begin{aligned}
H_{\mathrm{PSSN}} = &-\Delta \sum_{e\|\mathrm{z}} \mathbf{X}_e^{\mathrm{x}} \mathbf{X}_e^{\mathrm{y}} \prod_{p\ni e} \mathbf{X}_p - \lambda \sum_{p\|\mathrm{z}} \mathbf{Z}_p - \gamma \left( \sum_{e\perp\mathrm{x}} \mathbf{X}_e^{\mathrm{x}} \prod_{p\ni e}^{\perp\mathrm{x}} \mathbf{X}_p + \sum_{e\perp\mathrm{y}} \mathbf{X}_e^{\mathrm{y}} \prod_{p\ni e}^{\perp\mathrm{y}} \mathbf{X}_p \right) \\
&- \Delta' \left( \sum_{p\perp\mathrm{x}} \mathbf{Z}_p \prod_{e\in p} \mathbf{Z}_e^{\mathrm{x}} + \sum_{p\perp\mathrm{y}} \mathbf{Z}_p \prod_{e\in p} \mathbf{Z}_e^{\mathrm{y}} \right) - \lambda' \sum_{e\|\mathrm{z}} \mathbf{X}_e^{\mathrm{x}} \mathbf{X}_e^{\mathrm{y}} \\
&- \gamma' \left( \sum_{e\perp\mathrm{x}} \mathbf{X}_e^{\mathrm{x}} + \sum_{e\perp\mathrm{y}} \mathbf{X}_e^{\mathrm{y}} \right) - \kappa' \sum_{e\|\mathrm{z}} \mathbf{Z}_e^{\mathrm{x}} \mathbf{Z}_e^{\mathrm{y}} - \varepsilon' \left( \sum_{e\|\mathrm{x}} \mathbf{Z}_e^{\mathrm{y}} + \sum_{e\|\mathrm{y}} \mathbf{Z}_e^{\mathrm{x}} \right) \,.
\end{aligned} \tag{286}$$

This model has a $\widetilde{\mathbb{Z}}_2^{(0,1)}(\mathscr{F}_{\mathrm{x}}^{\perp}, \mathscr{F}_{\mathrm{y}}^{\perp})$ planar subsystem zero-form symmetry generated by $\mathbf{Z}_p$ operators on xz and yz planes, or equivalently

$$\widetilde{U}_{\tilde{i}+\frac{1}{2}}^{\mathrm{x}} = \prod_{j,k} \mathbf{Z}_{\tilde{i}+\frac{1}{2},\tilde{j},\tilde{k}} \,, \qquad \widetilde{U}_{\tilde{j}+\frac{1}{2}}^{\mathrm{y}} = \prod_{i,k} \mathbf{Z}_{\tilde{i},\tilde{j}+\frac{1}{2},\tilde{k}} \,, \tag{287}$$

where in the above, we are thinking of $(\tilde{i}+\frac{1}{2}, \tilde{j}, \tilde{k})$ and $(\tilde{i}, \tilde{j}+\frac{1}{2}, \tilde{k})$ as coordinates for edges $\tilde{e}$ on the dual lattice. This symmetry enhances when $\gamma = 0$ to a larger $\widetilde{\mathbb{Z}}_2^{(1,1)}(\mathscr{F}_{\mathrm{xy}}^{\|})$ planar subsystem one-form symmetry generated by operators

$$\widetilde{U}_{\gamma,\tilde{k}} = \prod_{(\tilde{\alpha},\tilde{\beta})\in\gamma} \mathbf{Z}_{\tilde{\alpha},\tilde{\beta},\tilde{k}} \,, \tag{288}$$

where here, $\gamma = \{(\tilde{\alpha}, \tilde{\beta})\}$ is a path on the dual lattice of a generic 2d square lattice. The edges of the 2d square lattice that it intersects have coordinates $(\tilde{\alpha}, \tilde{\beta})$ such that $\tilde{\alpha} + \tilde{\beta} \in \mathbb{Z} + \frac{1}{2}$, and we think of $\tilde{e} = (\tilde{\alpha}, \tilde{\beta}, \tilde{k})$ as specifying an edge of the dual lattice of the 3d cubic lattice on which the PSSN model is defined. These symmetries should be compared to those of the model in §5.1, in particular the symmetry generators presented in Eq. (242) and Eq. (240).

The model also has a $\mathbb{Z}_2^{(0,1)}(\mathscr{F}_{\mathrm{xy}}^{\|})$ symmetry generated by

$$U_{k+\frac{1}{2}}^{\mathrm{xy}} = \prod_{i,j} \mathbf{X}_{i,j,k+\frac{1}{2}}^{\mathrm{x}} \mathbf{X}_{i,j,k+\frac{1}{2}}^{\mathrm{y}} \,. \tag{289}$$

When $\kappa' = \varepsilon' = 0$, this enhances to a larger $\mathbb{Z}_2^{(1,1)}(\mathscr{F}_x^\perp, \mathscr{F}_y^\perp)$ symmetry generated by

$$U_{\tilde{\gamma},i}^x = \prod_{(\alpha,\beta) \in \tilde{\gamma}} \mathbf{X}_{i,\alpha,\beta}^x, \qquad U_{\tilde{\gamma},j}^y = \prod_{(\alpha,\beta) \in \tilde{\gamma}} \mathbf{X}_{\alpha,j,\beta}^y, \tag{290}$$

where here, $\tilde{\gamma} = \{(\alpha,\beta)\}$ is a path on the dual lattice of a generic 2d square lattice. These symmetries should be compared to those of the model in §5.2.

There is a $\mathbb{Z}_2$ local unitary circuit

$$\widetilde{V} = \prod_{p \perp x} \prod_{e \in p} C_e^x \mathbf{X}_p \prod_{p \perp y} \prod_{e \in p} C_e^y \mathbf{X}_p, \tag{291}$$

that acts on the phase diagram as $\Delta \leftrightarrow \lambda'$, $\Delta' \leftrightarrow \lambda$, and $\gamma \leftrightarrow \gamma'$.

The string-string-net Hamiltonian reduces to an anisotropic single stack or two coupled stacks of $\mathbb{Z}_2$ gauge theory in certain limits, as explained below.

- As $\lambda$ is taken large, the plaquette qubits are pinned into the $|{\uparrow}\rangle$ state, and the effects of the $\Delta, \gamma$ terms can be computed in perturbation theory, leading to star terms

$$\prod_{e \ni v}^{\perp x} \mathbf{X}_e^x, \qquad \prod_{e \ni v}^{\perp y} \mathbf{X}_e^y, \tag{292}$$

that appear at 4th order in perturbation theory in $\gamma$. Their product appears at 4th order in $\gamma$ and 2nd order in $\Delta$. The resulting anisotropic model corresponds to two stacks of $\mathbb{Z}_2$ lattice gauge theory with couplings, Eq. (264),

$$H_{\text{PSSN}} \xrightarrow{\lambda \text{ large}} H_{\text{2-stack}}. \tag{293}$$

- As $\lambda, \gamma \to 0$ we find the model from above, but with its $\mathbb{Z}_2^{(0,1)}(\mathscr{F}_{xy}^\parallel)$ subsystem symmetry gauged (cf. Eq. (270)), where Gauss's law is energetically enforced by the $\Delta$ term, becoming strict in the limit $\Delta \to \infty$. The difference is that the flux term constraints from Eq. (270) are absent when $\lambda = 0$. These flux terms can be incorporated energetically by backing off from the strict $\lambda = 0$ limit, and taking into account small $\lambda$ effects to 4th order in perturbation theory. We therefore find that

$$H_{\text{PSSN}} \xrightarrow[\substack{\gamma \to 0 \\ \Delta \to \infty \\ \lambda \text{ small}}]{} H_{\text{2-stack}}\Big/_E \mathbb{Z}_2^{(0,1)}\left(\mathscr{F}_{xy}^\parallel\right), \tag{294}$$

where we emphasize that we have used the "energetic" gauging prescription described in §2.1 because the flux terms are imposed energetically as opposed to as strict constraints on the Hilbert space.

- As $\gamma' \to \infty$, the edge qubits are pinned into the $|+\rangle$ state, the $\lambda'$ term acts trivially, and the $\Delta', \kappa', \varepsilon'$ terms are projected out. If we take $\gamma'$ to be large but not strictly infinite, and compute an effective Hamiltonian perturbatively in $\Delta', \kappa'$, and $\varepsilon'$, then one finds that at 4th order in both $\Delta'$ and $\kappa'$, and simultaneously eighth order in $\varepsilon'$, terms of the form

$$\prod_{p \in c}^{p \parallel z} \mathbf{Z}_p, \tag{295}$$

are generated. If one applies a Hadamard gate $H^{\otimes N}$ to swap all Pauli-$\mathbf{X}$ operators with Pauli-$\mathbf{Z}$ operators, and switches perspectives to the dual lattice, then one finds that the

PSSN model goes over to the stack of $\mathbb{Z}_2$ gauge theories with inter-layer couplings (cf. Eq. (243)),

$$H^{\otimes N} H_{\text{PSSN}} H^{\otimes N} \simeq H_{\text{1-stack}}, \quad \text{for} \quad \gamma' \text{ large}, \tag{296}$$

where we have used the symbol $\simeq$ to emphasize that the Gauss's law operators of the gauge theory layers in $H_{\text{1-stack}}$ are imposed as constraints on the Hilbert space, whereas they are imposed energetically in $H_{\text{PSSN}}$ by Eq. (295).

- Taking $\lambda' = \varepsilon' = 0$, $\Delta' \to \infty$, and $\gamma', \kappa'$ small, applying a Hadamard $H^{\otimes N}$ gate, and switching perspectives to the dual lattice, the PSSN model reduces to the model from §5.1.2—a single stack of coupled gauge theory layers with its $\mathbb{Z}_2^{(0,1)}(\mathscr{F}_{\text{x}}^{\perp}, \mathscr{F}_{\text{y}}^{\perp})$ subsystem symmetry gauged—as follows. The Gauss's law constraints of the subsystem gauging, i.e. the operators $\widetilde{G}_e$ from Eq. (250) up to a change of basis, are energetically enforced by the $\Delta'$ term, and become strict in the $\Delta' \to \infty$ limit. Computing a low energy effective Hamiltonian perturbatively in $\gamma'$ produces terms

$$\prod_{e\ni v}^{\perp\text{x}} \mathbf{X}_e^{\text{x}}, \qquad \prod_{e\ni v}^{\perp\text{y}} \mathbf{X}_e^{\text{y}}, \tag{297}$$

which play the role of the flux operators $F_c^{\mu}$ from Eq. (250), while finite $\kappa'$ contributions produce the operators from Eq. (295) which again play the role of the Gauss's law operators of the gauge theory layers (i.e. the operators $G_v$ from Eq. (250) up to a change of basis). In total,

$$H^{\otimes N} H_{\text{PSSN}} H^{\otimes N} \simeq H_{\text{1-stack}}\Big|_E \mathbb{Z}_2^{(0,1)}\left(\mathscr{F}_{\text{x}}^{\perp}, \mathscr{F}_{\text{y}}^{\perp}\right)$$
$$\text{as } \lambda' = \epsilon' = 0, \ \Delta' \to \infty, \ \gamma', \kappa' \text{ small}. \tag{298}$$

The zero correlation length string-string-net model emerges in the limit $\Delta, \Delta' \to \infty$

$$H_{\text{SSN}} = -\sum_{e\|\text{z}} \mathcal{A}_e - \sum_{p\perp\text{x}} \mathcal{B}_p^{\text{x}} - \sum_{p\perp\text{y}} \mathcal{B}_p^{\text{y}} - \sum_c \mathcal{C}_c - \sum_v (\mathcal{D}_v^{\text{x}} + \mathcal{D}_v^{\text{y}}), \tag{299}$$

where we have defined

$$\mathcal{A}_e = \mathbf{X}_e^{\text{x}} \mathbf{X}_e^{\text{y}} \prod_{p\ni e} \mathbf{X}_p, \qquad \mathcal{B}_p^{\mu} = \mathbf{Z}_p \prod_{e\in p} \mathbf{Z}_e^{\mu}, \qquad \mathcal{C}_c = \prod_{p\in c}^{p\|\text{z}} \mathbf{Z}_p, \qquad \mathcal{D}_v^{\mu} = \prod_{e\ni v}^{\perp\mu} \mathbf{X}_e^{\mu}. \tag{300}$$

We have included the leading order cube and vertex star terms, and rescaled the energies for simplicity of presentation; both of these operations preserve the zero temperature phase of matter of the commuting Hamiltonian.

The string-string-net model can equally be viewed as xy-planar subsystem gauged stacks of toric code along xz and yz planes, or as an xz- and yz-planar subsystem gauged stack of toric codes along the xy planes. Excitations of the edge star term are equivalent to planon composites formed by pairs of x- or y- lineons separated along z. Excitations of the x- and y-plaquette terms are equivalent to planon composites formed by pairs of fractons separated along x and y respectively. Excitations of the cube term correspond to X-cube fractons. Excitations of the x and y vertex star terms correspond to y- and x-lineons, respectively.

To justify our identification of the excitations above, we show how the string-string-net model is equivalent to the X-cube model. This further implies that the perturbed model can be interpreted as exploring neighboring phases to the X-cube model, including one or two stacks

of (2+1)D toric code, and the relevant anisotropic lineon or fracton dipole condensation phase transitions. The equivalence follows by applying the circuit $\widetilde{V}$ and then an additional circuit

$$V = \prod_{e\|\mathrm{z}} C_e^{\mathrm{x}} \mathbf{X}_e^{\mathrm{y}}, \tag{301}$$

to achieve

$$V\widetilde{V}H_{\mathrm{SSN}}\widetilde{V}^{\dagger}V^{\dagger} = -\sum_{e\|\mathrm{z}}\mathbf{X}_e^{\mathrm{x}} - \sum_{p\|\mathrm{z}}\mathbf{Z}_p - \sum_c \prod_{p\in c}^{p\|\mathrm{z}}\mathbf{Z}_p \prod_{e\in c}^{e\|\mathrm{z}}\mathbf{Z}_e^{\mathrm{x}}\mathbf{Z}_e^{\mathrm{y}}\prod_{e\in c}^{e\perp\mathrm{z}}\mathbf{Z}_e$$
$$- \sum_v \left( \prod_{e\ni v}^{e\|\mathrm{y}}\mathbf{X}_e^{\mathrm{x}} \prod_{e\ni v}^{e\|\mathrm{z}}\mathbf{X}_e^{\mathrm{x}}\mathbf{X}_e^{\mathrm{y}} + \prod_{e\ni v}^{e\perp\mathrm{y}}\mathbf{X}_e^{\mathrm{y}} \right). \tag{302}$$

The cube term is equivalent to the X-cube term (up to a change of basis) $\prod_{e\in c}\mathbf{Z}_e$ after multiplying with four $\mathbf{Z}_p$ terms, and similarly the x vertex star term is equivalent to

$$\prod_{e\ni v}^{e\|\mathrm{y}}\mathbf{X}_e^{\mathrm{x}} \prod_{e\ni v}^{e\|\mathrm{z}}\mathbf{X}_e^{\mathrm{y}}, \tag{303}$$

after multiplying with two z edge terms. This redefinition of Hamiltonian terms, which is a topological phase equivalence of the commuting Hamiltonian, decouples the plaquette qubits and z edge qubits with superscript y with a trivial Hamiltonian, and they can hence be removed while preserving the phase. This results in the X-cube Hamiltonian (up to a change of basis)

$$V\widetilde{V}H_{\mathrm{SSN}}\widetilde{V}^{\dagger}V^{\dagger} \sim -\sum_c \prod_{e\in c}\mathbf{Z}_e - \sum_v \left( \prod_{e\ni v}^{e\perp\mathrm{x}}\mathbf{X}_e + \prod_{e\ni v}^{e\perp\mathrm{y}}\mathbf{X}_e \right), \tag{304}$$

where we have dropped the superscript on the remaining qubits, as there is one per edge.

The string-string-net picture for the Hamiltonian $H_{\mathrm{SSN}}$ follows, similarly as in previous sections, by interpreting the cube term $\mathcal{C}_c$ as fusing a closed $\mathbb{Z}_2$-loop into the edges of the dual lattice xy planes, and similarly interpreting $\mathcal{D}_v^{\mathrm{x}}$ and $\mathcal{D}_v^{\mathrm{y}}$ as fusing a $\mathbb{Z}_2$-loop into the lattice yz and xz planes, respectively. The edge term $\mathcal{A}_e$ enforces a $\mathbb{Z}_2$ parity constraint that the dual lattice loops must be $\mathbb{Z}_2$-closed, or terminate on an odd number of edge strings, while the plaquette terms $\mathcal{B}_p^{\mu}$ similarly enforce that the edge strings must come in $\mathbb{Z}_2$-closed loops, or terminate on a dual string through a plaquette in the same plane. The ground states are then given by an equal weight superposition over all string-string-net configurations built on top of a reference state that satisfies the $\mathbb{Z}_2$ parity constraints.

## 6 (3+1)D Isotropic X-Cube Transitions

In this section, we explore constructions of the X-cube model that are unified within (an extension of) the *string-membrane-net model* of Ref. [52]. We start in §6.1 by reviewing the coupled layer constructions of Refs. [52, 61, 62]. We then move on in §6.2 to consider another construction which proceeds by coupling topological phases to subsystem gauge theories [52, 64]. Finally, we demonstrate in §6.3 that both of these constructions can be recovered by taking certain limits of the string-membrane-net parent model.

This section is structurally similar to §3, and so we are briefer here. One of the novelties of this section over previous sections is that we comment on boundaries, and in fact show that in some instances, the models of §3 govern the boundary dynamics of the models we consider here.

## 6.1 Isotropic layer constructions

The (deformed) X-cube model ${}^3H_{\mathbb{Z}_2}^{(1,1)}$ has a $\mathbb{Z}_2^{(1,1)}(\mathscr{F}_{xy}^{\parallel}, \mathscr{F}_{xz}^{\parallel}, \mathscr{F}_{yz}^{\parallel})\big/{}^{D}\mathbb{Z}_2^{(1)}$ planar one-form subsystem symmetry group, where $\mathscr{F}_{\mu_1\mu_2}^{\parallel}$ is the natural foliation of 3d space by planes that span the $\mu_1\mu_2$ directions. A natural attempt at modeling these symmetries is via the system ${}^2H_{\mathbb{Z}_2}^{(1)}(\mathscr{F}_{xy}^{\parallel}, \mathscr{F}_{xz}^{\parallel}, \mathscr{F}_{yz}^{\parallel})$ of three isotropic decoupled stacks of $\mathbb{Z}_2$ lattice gauge theories; since each individual layer has an ordinary one-form symmetry, the combined system has a planar one-form subsystem symmetry group. However, there is an important difference between the symmetries of ${}^3H_{\mathbb{Z}_2}^{(1,1)}$ and ${}^2H_{\mathbb{Z}_2}^{(1)}(\mathscr{F}_{xy}^{\parallel}, \mathscr{F}_{xz}^{\parallel}, \mathscr{F}_{yz}^{\parallel})$: in the latter Hamiltonian, the symmetries on the different stacks are completely decoupled from one another, whereas in the case of the X-cube model, there is a global relation between them. Specifically, the diagonal subgroup ${}^{D}\mathbb{Z}_2^{(1)}$ (cf. Figure 1) acts trivially in the X-cube model, whereas it acts faithfully in the decoupled stacks.

There are two ways to fix this discrepancy. First, we may introduce interactions between the decoupled layers,

$$
{}^2H_{\mathbb{Z}_2}^{(1)}\left(\mathscr{F}_{xy}^{\parallel}, \mathscr{F}_{xz}^{\parallel}, \mathscr{F}_{yz}^{\parallel}\right) \to {}^2H_{\mathbb{Z}_2}^{(1)}\left(\mathscr{F}_{xy}^{\parallel}, \mathscr{F}_{xz}^{\parallel}, \mathscr{F}_{yz}^{\parallel}\right) + H_{C}, \tag{305}
$$

which are strong enough to induce a transition to a phase where the diagonal subgroup is unbroken. It turns out that this is precisely what the coupled layer construction of the X-cube model from Refs. [61, 62] achieves, and we review it from this perspective of spontaneous symmetry breaking in §6.1.1.

A second way to proceed is to impose the relation that the diagonal subgroup is trivially realized by gauging it [52]. More specifically, one considers the theory

$$
{}^2H_{\mathbb{Z}_2}^{(1)}\left(\mathscr{F}_{xy}^{\parallel}, \mathscr{F}_{xz}^{\parallel}, \mathscr{F}_{yz}^{\parallel}\right) \to {}^2H_{\mathbb{Z}_2}^{(1)}\left(\mathscr{F}_{xy}^{\parallel}, \mathscr{F}_{xz}^{\parallel}, \mathscr{F}_{yz}^{\parallel}\right)\Big/{}^{D}\mathbb{Z}_2^{(1)} \tag{306}
$$

and argues that, in a certain limit, it reproduces the X-cube model. We go down this route in §6.1.2 and, as an added bonus, show that when this gauged model is formulated on a manifold with boundary, the boundary dynamics are described by the analogous coupled wire construction of the plaquette Ising model treated in §3.1.2.

### 6.1.1 Coupling stacks of gauge theory layers

In this section, following Refs. [61, 62] (see also Ref. [99]), we model $\mathbb{Z}_2^{(1,1)}(\mathscr{F}_{xy}^{\parallel}, \mathscr{F}_{yz}^{\parallel}, \mathscr{F}_{xz}^{\parallel})\big/{\sim}$ one-form planar subsystem symmetries in (3+1)D by directly coupling together stacks of (2+1)D theories with ordinary $\mathbb{Z}_2^{(1)}$ one-form symmetries. This section may be thought of as a generalization of §3.1.1, where an analogous coupled wire construction was carried out to model $\mathbb{Z}_2^{(0,1)}(\mathscr{F}_{x}^{\parallel}, \mathscr{F}_{y}^{\parallel})\big/{\sim}$ zero-form linear subsystem symmetries in (2+1)D. Because this case has already been treated extensively in the literature, we are brief; our main goal is to emphasize how this coupled layer construction fits into our broader perspective.

We start by taking three orthogonal stacks of decoupled $\mathbb{Z}_2$ lattice gauge theory layers to produce a (3+1)D model whose Hilbert space consists of two qubits on each edge of a cubic lattice, one qubit from each of the two layers that intersect the edge. We label operators by the edge they act on and the plane to which the qubit being acted on belongs, e.g. $\mathbf{Z}_e^{xy}$ denotes a Pauli-$\mathbf{Z}$ operator acting on the qubit at edge $e$ which resides in the xy plane. With these conventions, the Hamiltonian of decoupled gauge theory layers can be written as

$$
{}^2H_{\mathbb{Z}_2}^{(1)}\left(\mathscr{F}_{xy}^{\parallel}, \mathscr{F}_{yz}^{\parallel}, \mathscr{F}_{xz}^{\parallel}\right) = -U \sum_e \sum_{\mu_1\mu_2 \parallel e} \mathbf{X}_e^{\mu_1\mu_2} - t \sum_{\mu_1\mu_2 = xy,yz,xz} \sum_{p \parallel \mu_1\mu_2} \prod_{e \in p} \mathbf{Z}_e^{\mu_1\mu_2}, \tag{307}
$$

where the sum over $\mu_1\mu_2 \parallel e$ is over the two planes $\mu_1\mu_2$ which are parallel to the edge $e$. We accompany this with a Gauss's law constraint that is implemented by operators

$$G_v^{\mu_1\mu_2} = \prod_{\substack{e\ni v \\ e\parallel\mu_1\mu_2}} \mathbf{X}_e^{\mu_1\mu_2}, \quad \text{for all vertices } v, \quad \text{and for} \quad \mu_1\mu_2 = \text{xy, yz, xz}. \tag{308}$$

From here, we can couple these layers together with terms of the form

$$H_{\text{3-stack}} = {}^2H_{\mathbb{Z}_2}^{(1)}\left(\mathscr{F}_{\text{xy}}^{\parallel}, \mathscr{F}_{\text{yz}}^{\parallel}, \mathscr{F}_{\text{xz}}^{\parallel}\right) + H_{\text{C}},$$

$$H_{\text{C}} = -\sum_e \left( K_X \prod_{\mu_1\mu_2\parallel e} \mathbf{X}_e^{\mu_1\mu_2} + K_Z \prod_{\mu_1\mu_2\parallel e} \mathbf{Z}_e^{\mu_1\mu_2} \right). \tag{309}$$

When $K_Z = 0$, the model enjoys a planar one-form subsystem symmetry group,

$$\mathscr{G} = \mathbb{Z}_2^{(1,1)}\left(\mathscr{F}_{\text{xy}}^{\parallel}, \mathscr{F}_{\text{yz}}^{\parallel}, \mathscr{F}_{\text{xz}}^{\parallel}\right), \tag{310}$$

whose corresponding symmetry operators are

$$U_{\tilde{\gamma},k}^{\text{xy}} = \prod_{(\alpha,\beta)\in\tilde{\gamma}} \mathbf{X}_{\alpha,\beta,k}^{\text{xy}}, \quad U_{\tilde{\gamma},j}^{\text{xz}} = \prod_{(\alpha,\beta)\in\tilde{\gamma}} \mathbf{X}_{\alpha,j,\beta}^{\text{xz}}, \quad U_{\tilde{\gamma},i}^{\text{yz}} = \prod_{(\alpha,\beta)\in\tilde{\gamma}} \mathbf{X}_{i,\alpha,\beta}^{\text{yz}}, \tag{311}$$

where e.g. in the operator $U_{\tilde{\gamma},k}^{\text{xy}}$, the object $\tilde{\gamma} = \{(\alpha,\beta)\}$ is a path on the dual lattice of the 2d square sub-lattice at the location $\text{z} = k$. We think of $\tilde{\gamma}$ as the set of edges that it intersects on the original 2d square sub-lattice, which we specify with 2d coordinates $(\alpha,\beta)$. These symmetry operators simply act in the same way as do the one-form symmetries of the individual $\mathbb{Z}_2$ lattice gauge theory layers. When $K_Z \neq 0$, this $\mathbb{Z}_2^{(1,1)}(\mathscr{F}_{\text{xy}}^{\parallel}, \mathscr{F}_{\text{xz}}^{\parallel}, \mathscr{F}_{\text{yz}}^{\parallel})$ is broken down to its diagonal subgroup,

$$\mathscr{H} = {}^{\text{D}}\mathbb{Z}_2^{(1)}, \tag{312}$$

whose symmetry operators take the form

$$^{\text{D}}U_{\tilde{m}} = \prod_{e\in\tilde{m}} \prod_{\mu_1\mu_2\parallel e} \mathbf{X}_e^{\mu_1\mu_2} = \prod_{\mu_1\mu_2=\text{xy,yz,xz}} \prod_{L\in\mathscr{F}_{\mu_1\mu_2}^{\parallel}} U_{L\cap\tilde{m}}^{\mu_1\mu_2}. \tag{313}$$

In the above, $\tilde{m}$ is a membrane on the dual of the 3d cubic lattice on which the model is defined, $L$ is a leaf of the foliation $\mathscr{F}_{\mu_1\mu_2}^{\parallel}$, and by $L\cap\tilde{m}$ we mean $(\tilde{\gamma},\ell)$, where $\tilde{\gamma}$ is a path on the dual of $L$ (thought of as a 2d lattice) obtained by intersecting the membrane $\tilde{m}$ with the leaf $L$, and $\ell$ is the coordinate of the leaf $L$ in the direction orthogonal to $\mu_1\mu_2$. See Figure 1 for a visualization in the case of one foliation.

**Phases and excitations**

We now study some of the phases that occur at extreme limits of the coupling strengths in the coupled gauge theory layer model above.

*Decoupled gauge theory layers at $K_X = K_Z = 0$*

When $K_X = K_Z = 0$, the model is in a "decoupled layers" phase. As one varies the competition between $U$ and $t$, the decoupled gauge theory layers each undergo a confinement/deconfinement phase transition. We interpret this as a spontaneous breaking of the $\mathbb{Z}_2^{(1,1)}(\mathscr{F}_{\text{xy}}^{\parallel}, \mathscr{F}_{\text{xz}}^{\parallel}, \mathscr{F}_{\text{yz}}^{\parallel})$ one-form planar subsystem symmetry of the full (3+1)D model. In the next section, we demonstrate that this phase transition maps, upon gauging the diagonal $^{\text{D}}\mathbb{Z}_2^{(1)}$

in Eq. (313), to a transition between the (3+1)D toric code and the X-cube model.

Next we briefly recall some of the main results of Refs. [61, 62].

*X-cube model at $K_Z = 0$ and $K_X \gg 1$*

When $K_Z = 0$ and $K_X \gg 1$, one can compute the low energy effective Hamiltonian which describes $H_{\text{3-stack}}$ deep in this phase. Using techniques similar to those used in §3.1 for the Ising quilt, this leads to a lattice model with one qubit per edge, and with the low energy effective operators expressed in terms of the "UV operators" as

$$\mathbf{Z}_e = \mathbf{Z}_e^{\mu_1\mu_2}\mathbf{Z}_e^{\mu_1'\mu_2'}, \quad \mathbf{X}_e = \mathbf{X}_e^{\mu_1\mu_2} = \mathbf{X}_e^{\mu_1'\mu_2'}, \tag{314}$$

where $\mu_1\mu_2$ and $\mu_1'\mu_2'$ are the two planes parallel to the edge $e$. Going to sixth order in perturbation theory, one finds

$$H_{\text{eff}} \sim -\tilde{U}\sum_e \mathbf{X}_e - \tilde{t}\sum_c \prod_{e \in c} \mathbf{Z}_e,$$
$$G_v^{\mu_1\mu_2} = \prod_{\substack{e \ni v \\ e \| \mu_1\mu_2}} \mathbf{X}_e. \tag{315}$$

If one were to set $\tilde{U} = 0$ and impose Gauss's law $G_v^{\mu_1\mu_2} = 1$ energetically as opposed to as a constraint, then this would correspond to the X-cube model. Instead, we have chosen to present the model more in the style of the generalized (tensor) gauge theory $^3H_{\mathbb{Z}_2}^{(1,1)}$.

This model inherits a $\mathbb{Z}_2^{(1,1)}(\mathscr{F}_{\text{xy}}^\|, \mathscr{F}_{\text{xz}}^\|, \mathscr{F}_{\text{yz}}^\|)/\sim$ one-form planar subsystem symmetry structure from its toric code constituents, which is furnished by the symmetry operators

$$U_{\tilde{\gamma},k}^{\text{xy}} = \prod_{(\alpha,\beta)\in\tilde{\gamma}} \mathbf{X}_{\alpha,\beta,k}, \quad U_{\tilde{\gamma},j}^{\text{xz}} = \prod_{(\alpha,\beta)\in\tilde{\gamma}} \mathbf{X}_{\alpha,j,\beta}, \quad U_{\tilde{\gamma},i}^{\text{yz}} = \prod_{(\alpha,\beta)\in\tilde{\gamma}} \mathbf{X}_{i,\alpha,\beta}. \tag{316}$$

The gauge theory enjoys a phase transition associated with the spontaneous breaking of this symmetry as one dials $\tilde{U}/\tilde{t}$. On the other hand, we notice that the diagonal $^D\mathbb{Z}_2^{(1)}$ subgroup from Eq. (313) is trivially realized on the low energy subspace throughout this $K_X \gg 1$ phase, i.e. $^DU_{\tilde{m}} = 1$ regardless of the values of $\tilde{U}, \tilde{t}$. Since this diagonal subgroup acts non-trivially in the decoupled layer phase, the phase transition as one tunes from $K_X = \infty$ down to $K_X = 0$ is a $^D\mathbb{Z}_2^{(1)}$ spontaneous symmetry breaking phase transition. See Ref. [61] for a description of this phase transition in terms of p-string condensation.

*$\mathbb{Z}_2$ lattice gauge theory at $K_X = 0$ and $K_Z \gg 1$*

One can perform a similar computation to determine the low energy effective Hamiltonian in the $K_Z \gg 1$ phase. Again, one finds a lattice model with one qubit per edge, this time with the low energy operators defined as

$$\mathbf{X}_e = \mathbf{X}_e^{\mu_1\mu_2}\mathbf{X}_e^{\mu_1'\mu_2'}, \quad \mathbf{Z}_e = \mathbf{Z}_e^{\mu_1\mu_2} = \mathbf{Z}_e^{\mu_1'\mu_2'}. \tag{317}$$

The effective Hamiltonian is then precisely that of $\mathbb{Z}_2$ lattice gauge theory

$$H_{\text{eff}} = -\hat{U}\sum_e \mathbf{X}_e - \hat{t}\sum_p \prod_{e \in p} \mathbf{Z}_e, \tag{318}$$

with Gauss's law constraint

$$G_v = \prod_{e \ni v} \mathbf{X}_e. \tag{319}$$

The diagonal $^D\mathbb{Z}_2^{(1)}$ one-form symmetry group from Eq. (313) goes over to the usual one-form symmetry of $\mathbb{Z}_2$ lattice gauge theory, and tuning $\hat{U}/\hat{t}$ takes one through the usual confinement/deconfinement phase transition. See §2.2.3 for a brief review of this model.

**Order and disorder parameters**

Let us consider the natural disorder parameters one could write down to diagnose the phase diagram of $H_{\text{3-stack}}$. One option is a truncated symmetry operator for the diagonal one-form symmetry $^D\mathbb{Z}_2^{(1)}$,

$$\mathscr{D}^D_{\tilde{m}(\tilde{\gamma})} = \prod_{e\in\tilde{m}} \prod_{\mu_1\mu_2\|e} \mathbf{X}_e^{\mu_1\mu_2} \,, \tag{320}$$

where here, $\tilde{m}$ is an open membrane on the dual lattice with boundary path $\tilde{\gamma}$. Another option is to consider disorder operators associated to the individual gauge theory layers, e.g.

$$\mathscr{D}^{\text{xy}}_{\tilde{\gamma},k} = \prod_{(\alpha,\beta)\in\tilde{\gamma}} \mathbf{X}^{\text{xy}}_{\alpha,\beta,k} \,. \tag{321}$$

A natural order parameter can be obtained as

$$\mathscr{O}_C = \prod_{e\in\text{outline}(C)} \prod_{\mu_1\mu_2\|e} \mathbf{Z}_e^{\mu_1\mu_2}. \tag{322}$$

Alternatively, one can simply consider a single Wegner–Wilson loop on one of the gauge theory layers.

The behavior of the vacuum expectation values of these parameters can be determined easily, at least when $K_X = K_Z = 0$, from the corresponding behavior of order/disorder parameters of (2+1)D $\mathbb{Z}_2$ lattice gauge theory.

### 6.1.2 Gauging the diagonal subgroup

In this section, extending slightly the work of Ref. [52], we study the model that one obtains upon gauging the diagonal one-form symmetry subgroup of decoupled gauge theory layers, Eq. (313). We find a phase diagram that interpolates between a (3+1)D toric code phase and an X-cube phase (see also Ref. [75] for related work). One novelty of this section over others is that we formulate the theory on a manifold with boundary (see also Ref. [114] for related work in this direction) which we use to make closer contact with the results of §3.1.2.

Actually, it is simpler to phrase the discussion in terms of perturbed toric code layers, rather than gauge theory layers, the difference being that in the former we impose Gauss's law energetically, whereas in the latter we impose it as a constraint on the Hilbert space. We place these perturbed toric code layers on $M = T^2 \times (-\infty, 0]$ with a rough $T^2$ boundary (i.e. the obvious generalization of the 1d rough boundary in Figure 2) which we take to lie in the xy plane. In the bulk, the Hamiltonian is simply[15]

$$H_{\text{bulk}} = \sum_{\mu_1\mu_2=\text{xy,xz,yz}} \left( -t \sum_{p\|\mu_1\mu_2} \prod_{e\in p} \mathbf{Z}_e^{\mu_1\mu_2} - \Delta \sum_v \prod_{\substack{e\ni v \\ e\|\mu_1\mu_2}} \mathbf{X}_e^{\mu_1\mu_2} - U \sum_{e\|\mu_1\mu_2} \mathbf{X}_e^{\mu_1\mu_2} \right), \tag{323}$$

where we understand the sums to be over bulk plaquettes $p$, bulk vertices $v$, and bulk edges $e$. On the boundary, we include interactions of the form

$$H_\partial = -h \sum_{v_\partial} (A^{\text{xz}}_{v_\partial} + A^{\text{yz}}_{v_\partial}) - J \sum_{p_\partial} B_{p_\partial} \,, \tag{324}$$

---

[15]We could also include the XX and ZZ coupling terms considered in the previous section, as was done for the Ising quilt. We have chosen to omit them for clarity.

where

$$A^{\mu_1\mu_2}_{v_\partial} = \prod_{\substack{e \ni v_\partial \\ e \parallel \mu_1\mu_2}} \mathbf{X}_e, \quad B_{p_\partial} = \prod_{e \in p_\partial} \mathbf{Z}^{\mu_1\mu_2(p_\partial)}_e. \tag{325}$$

The notation $v_\partial$ and $p_\partial$ denotes boundary vertices and boundary plaquettes, i.e. vertices of the lattice that have only one edge emanating from them, and plaquettes of the lattice that have three edges bordering them. The operator $A^{\mu_1\mu_2}_{v_\partial}$ is simply the Pauli operator $\mathbf{X}^{\mu_1\mu_2}_e$, where $e$ is the single edge that meets the boundary vertex $v_\partial$. The operator $B_{p_\partial}$ is a three-body Pauli-$\mathbf{Z}$ operator, where $\mu_1\mu_2(p_\partial)$ is the plane to which the boundary plaquette $p_\partial$ is parallel.

Our total "toric layer" Hamiltonian is the sum of these two,

$$H^{\mathrm{E}}_{\text{3-stack}} = H_{\text{bulk}} + H_\partial, \tag{326}$$

where the E in the superscript is to emphasize that Gauss's law is imposed energetically in this model. For the moment, let us set $U = 0$ for simplicity. Straightforwardly generalizing the discussion of §2.2.2, the effective edge Hamiltonian of the above model is described by decoupled transverse field Ising wires,

$$\left(H^{\mathrm{E}}_{\text{3-stack}}\big|_{U=0}\right)_{\text{edge}} \simeq -\sum_{v_\partial}\left(J\widetilde{\mathbf{Z}}^{\mathrm{xz}}_{v_\partial}\widetilde{\mathbf{Z}}^{\mathrm{xz}}_{v_\partial+\hat{x}} + h\widetilde{\mathbf{X}}^{\mathrm{xz}}_{v_\partial} + J\widetilde{\mathbf{Z}}^{\mathrm{yz}}_{v_\partial}\widetilde{\mathbf{Z}}^{\mathrm{yz}}_{v_\partial+\hat{y}} + h\widetilde{\mathbf{X}}^{\mathrm{yz}}_{v_\partial}\right), \tag{327}$$

which is the same as the Hamiltonian $^1H^{(0)}_{\mathbb{Z}_2}(\mathscr{F}^{\parallel}_{\mathrm{x}}, \mathscr{F}^{\parallel}_{\mathrm{y}})$.

Now, in analogy with the construction of the gauged Ising quilt, we can consider gauging the diagonal one-form symmetry subgroup $^{\mathrm{D}}\mathbb{Z}^{(1)}_2$ by coupling the model to a $\mathbb{Z}_2$ two-form gauge theory, and analyzing what effect this has on the edge. Following the standard minimal prescription for gauging one-form symmetries on the lattice (cf. §2.2.3), the Hamiltonian takes the form

$$H^{\mathrm{E}}_{\text{3-stack}}\Big/_E\,^{\mathrm{D}}\mathbb{Z}^{(1)}_2 = H_{\text{c-bulk}} + H_{\text{gauge}} + H_{\text{c-}\partial},$$
$$G_e = \prod_{\mu_1\mu_2 \parallel e} \mathbf{X}^{\mu_1\mu_2}_e\left(\prod_{p \ni e}\mathbf{X}_p\right), \tag{328}$$

where $G_e$ is the Gauss's law constraint term, defined for both bulk and edge vertices. The rest of the pieces are defined as follows. The term $H_{\text{gauge}}$ contains energetics for the two-form gauge field,

$$H_{\text{gauge}} = -U_g\sum_p \mathbf{X}_p - t_g\sum_c\prod_{p \in c}\mathbf{Z}_p. \tag{329}$$

The term $H_{\text{c-bulk}}$ covariantly couples the bulk part of the toric code layers to the gauge field so that they enjoy local gauge symmetry (i.e. so that they commute with each $G_e$),

$$H_{\text{c-bulk}} = \sum_{\mu_1\mu_2=\mathrm{xy,xz,yz}}\left(-t\sum_{p \parallel \mu_1\mu_2}\mathbf{Z}_p\prod_{e \in p}\mathbf{Z}^{\mu_1\mu_2}_e - \Delta\sum_v\prod_{\substack{e \ni v \\ e \parallel \mu_1\mu_2}}\mathbf{X}^{\mu_1\mu_2}_e - U\sum_{e \parallel \mu_1\mu_2}\mathbf{X}^{\mu_1\mu_2}_e\right). \tag{330}$$

Finally, $H_{\text{c-}\partial}$ encodes the boundary terms

$$H_{\text{c-}\partial} = -U^\partial_g\sum_{p_\partial}\mathbf{X}_{p_\partial} - t^\partial_g\sum_{c_\partial}\prod_{p \in c_\partial}\mathbf{Z}_p$$
$$-h\sum_{v_\partial}(A^{\mathrm{xz}}_{v_\partial} + A^{\mathrm{yz}}_{v_\partial}) - J\sum_{p_\partial}\mathbf{Z}_{p_\partial}B_{p_\partial}. \tag{331}$$

In the above, a boundary cube is one which has 7 plaquettes forming its boundary.

Now, the upshot is that the boundary Hamiltonian can be thought of as acting on an effective/virtual 2d lattice: we identify the boundary plaquettes $p_\partial$ with edges $\underline{e}$ of this effective lattice, and boundary edges $e_\partial$ with vertices $\underline{v}$ of the effective lattice. Boundary cubes $c_\partial$ are then naturally associated with plaquettes $\underline{p}$, and boundary vertices $v_\partial$ with vertices $\underline{v}$. Correspondingly, we place a qubit on each edge and two qubits on each vertex of the effective lattice. We can then map the various operators which appear in the boundary Hamiltonian to operators which act on these effective qubits, as in Table 7. Indeed, the algebra furnished by the operators in the last two columns is the same as that furnished by the operators in the first two columns. Making these substitutions, we find that the boundary Hamiltonian of this gauged theory is isomorphic to

$$\left(H^{\mathrm{E}}_{\text{3-stack}}\Big/{}^{\mathrm{D}}\mathbb{Z}^{(1)}_2\Big|_{U=0}\right)_{\text{edge}} \simeq {}^{1}H^{(0)}_{\mathbb{Z}_2}\left(\mathscr{F}^{\parallel}_{\mathrm{x}},\mathscr{F}^{\parallel}_{\mathrm{y}}\right)\Big/{}^{\mathrm{D}}\mathbb{Z}^{(0)}_2, \tag{332}$$

(cf. Eq. (121)), i.e. it is the same as the model obtained by gauging the diagonal subgroup ${}^{\mathrm{D}}\mathbb{Z}^{(0)}_2$ (Eq. (98)) of the subsystem symmetry group of decoupled transverse field Ising models. Evidently, gauging the diagonal one-form symmetry ${}^{\mathrm{D}}\mathbb{Z}^{(1)}_2$ of the bulk corresponds on the boundary to gauging the corresponding diagonal zero-form symmetry ${}^{\mathrm{D}}\mathbb{Z}^{(0)}_2$, i.e. the symmetry obtained by pushing the topological surface operators that implement the one-form symmetry in the bulk to the boundary.

Now, in Ref. [52], the authors applied a local unitary circuit to this theory (on a manifold without boundary) in order to exhibit its equivalence to the X-cube model. We can apply the same circuit here, extended in the obvious way to our space with a boundary,

$$V = \left(\prod_{p\parallel\mathrm{xy}}\prod_{e\in p} C_p \mathbf{X}^{\mathrm{xy}}_e\right)\left(\prod_{p\parallel\mathrm{xz}}\prod_{e\in p} C_p \mathbf{X}^{\mathrm{xz}}_e\right)\left(\prod_{p\parallel\mathrm{yz}}\prod_{e\in p} C_p \mathbf{X}^{\mathrm{yz}}_e\right), \tag{333}$$

and then solve the resulting Gauss's law constraint $VG_eV^\dagger = 1$

$$H_{\mathrm{GTL}} := V\left(H^{\mathrm{E}}_{\text{3-stack}}\Big/{}^{\mathrm{D}}\mathbb{Z}^{(1)}_2\right)V^\dagger\Big|_{VG_eV^\dagger=1,\ \Delta\to\infty}. \tag{334}$$

In the above, we understand that if $p$ is a boundary plaquette, then the product over $e \in p$ is a product over 3 edges (see Figure 7 of Ref. [52] for a visualization of this circuit in the case that there is no boundary). If one follows how this circuit acts at the level of the effective boundary Hamiltonian, one finds that it acts in the same way as the circuit in Eq. (125) does on ${}^{1}H^{(0)}_{\mathbb{Z}_2}(\mathscr{F}^{\parallel}_{\mathrm{x}},\mathscr{F}^{\parallel}_{\mathrm{y}})\Big/{}^{\mathrm{D}}\mathbb{Z}^{(0)}_2$. Accordingly, the boundary theory of $H_{\mathrm{GTL}}$ becomes exactly the gauged Ising quilt appearing in Eq. (127), i.e.

$$\left(H_{\mathrm{GTL}}\right)_{\text{edge}} \sim H_{\mathrm{GIQ}}. \tag{335}$$

The Hamiltonian $H_{\mathrm{GTL}}$ inherits a planar one-form subsystem symmetry group

$$\mathscr{G}\big/\mathscr{H} = \mathbb{Z}^{(1,1)}_2\left(\mathscr{F}^{\parallel}_{\mathrm{xy}},\mathscr{F}^{\parallel}_{\mathrm{yz}},\mathscr{F}^{\parallel}_{\mathrm{xz}}\right)\big/{}^{\mathrm{D}}\mathbb{Z}^{(1)}_2, \tag{336}$$

from the ungauged toric code layers, and also grows an emergent symmetry group,

$$\widehat{\mathscr{H}} = {}^{\mathrm{D}}\widehat{\mathbb{Z}}^{(1)}_2. \tag{337}$$

Table 7: The first two columns contain operators which appear in the boundary Hamiltonian in Eq. (331). To each such operator, we associate a corresponding "effective operator" which acts on an effective 2d lattice which describes the boundary. These operators are the same as those appearing in the Hamiltonian of the gauged Ising quilt, Eq. (121). The algebra furnished by the boundary operators is the same as that furnished by their corresponding effective operators.

| Boundary operator | | Effective operator | |
|---|---|---|---|
| $\mathbf{X}_{p_\partial}$ | | $\underline{X}_{\underline{e}}$ | |
| $\prod_{p \in c_\partial} \mathbf{Z}_p$ | | $\prod_{\underline{e} \in \underline{p}} \underline{Z}_{\underline{e}}$ | |
| $\prod_{\mu_1\mu_2\|e_\partial} \mathbf{X}_{e_\partial}^{\mu_1\mu_2}\left(\prod_{p_\partial \in e_\partial} \mathbf{X}_{p_\partial}\right)$ | | $\underline{X}_{\underline{v}}^{\mathrm{x}}\underline{X}_{\underline{v}}^{\mathrm{y}}\prod_{\underline{e}\in\underline{v}}\underline{X}_{\underline{e}}$ | |
| $A_{v_\partial}^{\mathrm{xz}}, A_{v_\partial}^{\mathrm{yz}}$ | | $\underline{X}_{\underline{v}}^{\mathrm{x}}, \underline{X}_{\underline{v}}^{\mathrm{y}}$ | |
| $\mathbf{Z}_{p_\partial}B_{p_\partial} \ (p_\partial\|\mathrm{xz})$ | | $\underline{Z}_{\underline{e}}\prod_{\underline{v}\in\underline{e}}\underline{Z}_{\underline{v}}^{\mathrm{x}}$ | |
| $\mathbf{Z}_{p_\partial}B_{p_\partial} \ (p_\partial\|\mathrm{yz})$ | | $\underline{Z}_{\underline{e}}\prod_{\underline{v}\in\underline{e}}\underline{Z}_{\underline{v}}^{\mathrm{y}}$ | |

**Phases and excitations**

As one varies $U/t$ in the ungauged model, one passes through a planar one-form subsystem symmetry breaking phase transition. Let us analyze how this phase diagram maps after gauging ${}^{\mathrm{D}}\mathbb{Z}_2^{(1)}$. We find that (at least when the strict version of the gauging procedure is used) the corresponding phase transition of the gauged model is a simultaneous breaking of $\mathscr{G}\big/\mathscr{H}$ and an unbreaking of $\widehat{\mathscr{H}}$, in line with Expectation 2 of §2.1. We are telegraphic here because the logic and calculations are all very similar to §3.1.2.

*X-cube model at $t/U \gg 1$*

The $t/U \to \infty$ limit freezes out the qubits that live on the plaquettes, and the bulk low energy effective theory in this limit is simply the X-cube model,

$$H_{\mathrm{GTL}} \xrightarrow{t/U \to \infty} {}^3 H_{\mathbb{Z}_2}^{(1,1)}. \tag{338}$$

Intriguingly, using the results of §3.1.2 and additionally taking the boundary coupling $J/h \to \infty$ recovers the plaquette Ising model (Eq. (102)) as the edge Hamiltonian of the X-cube model. This is a sort of subsystem symmetric analog of the fact that the (1+1)D Ising model can arise on the boundary of the toric code/$\mathbb{Z}_2$ gauge theory (cf. §2.2.2).

Using the strict gauging procedure, the Hamiltonian is a constant, and the Hilbert space precisely coincides with the ground state of the X-cube model. Thus, $\mathscr{G}\big/\mathscr{H}$ is spontaneously broken. On the other hand, $\widehat{\mathscr{H}}$ is unbroken in this phase, and does not act on the low energy effective Hamiltonian.

*$\mathbb{Z}_2$ gauge theory at $U/t \gg 1$*

Let us now consider only the bulk phase, ignoring effects of the boundary. In the other limit, one can show that when $U/t \gg 1$, the edge degrees of freedom are all frozen out, and the remaining low energy effective Hamiltonian is unitarily equivalent to $\mathbb{Z}_2$ gauge theory on the dual lattice,

$$H_{\mathrm{GTL}} \xrightarrow{U/t \to \infty} {}^3 \widetilde{H}_{\mathbb{Z}_2}^{(1)}. \tag{339}$$

When the strict gauging procedure is used, the low energy Hamiltonian is a constant, and the low energy Hilbert space coincides with the ground state of the (3+1)D toric code. In this case, one sees that the planar one-form subsystem symmetries $\mathscr{G}\big/\mathscr{H}$ are unbroken, while the emergent ordinary one-form symmetry $\widehat{\mathscr{H}}$ is spontaneously broken.

The statements we have made so far are summarized in Figure 13.

*Excitations and phase transition between $t \ll U$ and $U \gg t$*

Gauging the diagonal one-form subsystem symmetry in the trivial phase $U \gg t$ maps excitations of single body $\mathbf{X}_e^{\mu_1 \mu_2}$ terms to topological loop excitations. The two Pauli-$\mathbf{X}$ operators on each edge give rise to a pair of representatives for a segment of topological loop excitation, associated to the planes passing through that edge, that are equivalent under local operations. Increasing $t$ causes the different representative loop excitations to fluctuate and condense within their associated planes. This is an instance of planar string condensation, however unlike the previous examples in this work, here the string excitations that condense are not composite. This drives the phase transition from the (3+1)D toric code phase to the X-cube phase, obtained by gauging an isotropic stack of $\mathbb{Z}_2$ gauge theory layers that are undergoing confinement/deconfinement phase transitions.

For $t \gg U$ the layers are in the toric code phase before gauging, and the magnetic flux excitations of the plaquette terms are mapped after gauging to composite planon excitations,

Decoupled gauge theory layers: $^2H^{(1)}_{\mathbb{Z}_2}(\mathscr{F}^{\|}_{xy}, \mathscr{F}^{\|}_{yz}, \mathscr{F}^{\|}_{xz})$

$\mathbb{Z}^{(1,1)}_2(\mathscr{F}^{\|}_{xy}, \mathscr{F}^{\|}_{yz}, \mathscr{F}^{\|}_{xz})$ | $\mathbb{Z}^{(1,1)}_2(\mathscr{F}^{\|}_{xy}, \mathscr{F}^{\|}_{yz}, \mathscr{F}^{\|}_{xz})$
subsystem symmetry | subsystem symmetry
preserving phase | breaking phase

$\longrightarrow t/U$

$\downarrow$ gauging $^D\mathbb{Z}^{(1)}_2$

Gauged layers: $^2H^{(1)}_{\mathbb{Z}_2}(\mathscr{F}^{\|}_{xy}, \mathscr{F}^{\|}_{yz}, \mathscr{F}^{\|}_{xz})\big/^D\mathbb{Z}^{(1)}_2$

$\mathbb{Z}^{(1,1)}_2(\mathscr{F}^{\|}_{xy}, \mathscr{F}^{\|}_{yz}, \mathscr{F}^{\|}_{xz})\big/^D\mathbb{Z}^{(1)}_2$ preserved | $\mathbb{Z}^{(1,1)}_2(\mathscr{F}^{\|}_{xy}, \mathscr{F}^{\|}_{yz}, \mathscr{F}^{\|}_{xz})\big/^D\mathbb{Z}^{(1)}_2$ broken
$^D\widehat{\mathbb{Z}}^{(1)}_2$ broken | $^D\widehat{\mathbb{Z}}^{(1)}_2$ preserved
((3+1)D toric code phase) | (X-cube phase)

$\longrightarrow t/U$

Figure 13: A summary of the phase diagram obtained after gauging the diagonal one-form symmetry of decoupled toric code/$\mathbb{Z}_2$ gauge theory layers.

consisting of a pair of X-cube fractons. This follows by observing that a single $\mathbf{X}_p$ operator creates a local charge cluster consisting of a single gauged plaquette excitation along with a pair of fracton excitations on adjacent cubes. As $t$ increases these planons are condensed within their respective planes of mobility, resulting in the phase transition to the (3+1)D toric code phase via the same critical point as in the above paragraph.

**Order and disorder parameters**

Order/disorder parameters for this phase transition can be obtained by mapping the order/disorder parameters of the ungauged toric code layers across the gauging map. For example,

$$
\begin{aligned}
V(\mathscr{O}_C\big/^D\mathbb{Z}^{(1)}_2)V^{\dagger} &\xrightarrow{VG_eV^{\dagger}=1} \prod_{e\in\text{outline}(C)} \mathbf{Z}_e \,, \\
V(^D\mathscr{D}_{\tilde{m}(\tilde{\gamma})}\big/^D\mathbb{Z}^{(1)}_2)V^{\dagger} &\xrightarrow{VG_eV^{\dagger}=1} \prod_{p\in\tilde{\gamma}} \mathbf{X}_p \,.
\end{aligned}
\tag{340}
$$

The first of these can be interpreted as an order parameter for the $\mathscr{G}\big/\mathscr{H}$ subsystem symmetry, while the second of these can be interpreted as an order parameter for the emergent $\widehat{\mathscr{H}}$ symmetry. The other order/disorder parameters can be treated similarly.

### 6.1.3 Dualizing the leaves

Just as in §3.1.3, we can ask what happens if we perform a Wegner duality on each of the gauge theory layers to transform them into transverse field Ising models. Because Wegner duality is equivalent to gauging the one-form symmetry of $\mathbb{Z}_2$ lattice gauge theory (in the sense described in §2.2.2), we could equivalently say that we are gauging the $\mathbb{Z}^{(1,1)}_2(\mathscr{F}^{\|}_{xy}, \mathscr{F}^{\|}_{xz}, \mathscr{F}^{\|}_{yz})$ planar one-form subsystem symmetry of the coupled gauge theory layers model, Eq. (309). In either case,

we end up with a model with a qubit on each plaquette, and Hamiltonian

$$H_{\text{IL}} = -t \sum_p \mathbf{X}_p - U \sum_{\langle p, p' \rangle} \mathbf{Z}_p \mathbf{Z}_{p'} - K_X \sum_e \prod_{p \ni e} \mathbf{Z}_p \,, \tag{341}$$

where the sum over nearest neighbor plaquettes is a sum over pairs of plaquettes which share an edge and lie in the same plane.

We now compute the low energy effective Hamiltonian in the strongly coupled $K_X \gg 1$ regime. The low energy Hilbert space is the ground space of the term proportional to $K_X$. If we think of $\mathbf{Z}_p = -1$ as a colored plaquette, and $\mathbf{Z}_p = 1$ as a plaquette without color, then the ground space is spanned by states for which the colored plaquettes form closed membranes (cf. §3.1.3 where the ground space was the space of "closed string states"). We restrict attention to closed membranes which can be consistently thought of as forming domain walls separating regions of effective spins in $+1$ eigenstates of Pauli-$\mathbf{Z}$ from regions of effective spins in $-1$ eigenstates of Pauli-$\mathbf{Z}$. In other words, up to an overall global qubit, these states can be parametrized by an effective qubit on each fundamental cube $c$. The logical operators which act on this effective qubit can be expressed in terms of the original operators as

$$\mathbf{X}_c = \prod_{p \in c} \mathbf{X}_p \,, \qquad \prod_{c \ni e} \mathbf{Z}_c = \prod_{\substack{p \ni e \\ p \| \mu_1}} \mathbf{Z}_p = \prod_{\substack{p \ni e \\ p \| \mu_2}} \mathbf{Z}_p \,, \tag{342}$$

where in the above, the product over $c \ni e$ is a product over the four cubes which have $e$ as an edge, and the product over $p \ni e$ such that $p \| \mu_i$ is a product over the two plaquettes which have $e$ as an edge such that both plaquettes are parallel with the $\mu_i$ plane, where $\mu_1, \mu_2$ are the two planes parallel to the edge $e$.

If ones carries out the computation of the low energy effective Hamiltonian, treating the terms proportional to $t$ and $U$ as small perturbations, then symmetry considerations alone imply that to leading order it should take the shape

$$H_{\text{eff}} \sim \tilde{h} \sum_{\tilde{v}} \mathbf{X}_{\tilde{v}} - \tilde{J} \sum_{\tilde{p}} \prod_{\tilde{v} \in \tilde{p}} \mathbf{Z}_{\tilde{p}} \,, \tag{343}$$

where we have switched perspectives to the dual lattice, i.e. $\tilde{v}$ and $\tilde{p}$ are dual vertices and plaquettes corresponding to cubes and edges on the original lattice, respectively. In other words, we recover the (3+1)D plaquette Ising model on the dual lattice. This is consistent with the fact that the plaquette Ising model in (3+1)D is essentially "Kramers–Wannier dual" to the X-cube model [33] (up to global issues that we are neglecting).

## 6.2 Gauged subsystem symmetry enriched topological phases

In the previous section, we explored coupled layer constructions (and their gauging) as a means of recovering the X-cube model. In this section, we describe another method for obtaining the X-cube model. Our starting point is a (3+1)D model with a one-form symmetry $G^{(1)}$: such a model admits a planar zero-form subsystem symmetry subgroup $\mathscr{H} := G^{(0,1)}(\mathscr{F}_x^\perp, \mathscr{F}_y^\perp, \mathscr{F}_z^\perp)/\sim$, as we have described in §2.1. Gauging this subsystem subgroup results in an emergent, quantum $\widehat{\mathscr{H}} = \widehat{G}^{(1,1)}(\mathscr{F}_x^\perp, \mathscr{F}_y^\perp, \mathscr{F}_z^\perp)/\sim$ planar one-form subsystem symmetry group. Taking $G = \mathbb{Z}_2$ then leads to the symmetry structure of the X-cube model. This section can be thought of as a higher-dimensional generalization of the results of §3.2.

### 6.2.1 A subsystem symmetry enriched (3+1)D gauge theory

As reviewed in §2.2.3, one example of a model with a one-form symmetry is (3+1)D $\mathbb{Z}_2$ lattice gauge theory,

$$^3H_{\mathbb{Z}_2}^{(1)} = -U\sum_e \mathbf{X}_e - t\sum_p \prod_{e\in p}\mathbf{Z}_e\,, \tag{344}$$

with Gauss's law

$$G_v = \prod_{e\ni v}\mathbf{X}_e\,. \tag{345}$$

The one-form symmetry is implemented by operators supported on membranes $\tilde{m}$ of the dual lattice,

$$U_{\tilde{m}} = \prod_{e\in\tilde{m}}\mathbf{X}_e. \tag{346}$$

As in §3.2.1, we would like to deform this model by terms which break the one-form symmetry $\mathbb{Z}_2^{(1)}$ down to its $\mathscr{H} := \mathbb{Z}_2^{(0,1)}(\mathscr{F}_{\mathrm{x}}^\perp,\mathscr{F}_{\mathrm{y}}^\perp,\mathscr{F}_{\mathrm{z}}^\perp)$ subsystem subgroup. However, imposing Gauss's law as a constraint forbids the addition of any terms that violate $\mathbb{Z}_2^{(1)}$. We therefore discard Gauss's law (we could equally well choose to impose it energetically, but this is not needed in what follows) and add the lowest order term which breaks $\mathbb{Z}_2^{(1)}$ but respects $\mathscr{H}$. This leads us to a Hamiltonian for a deformed topological phase,

$$H_{\mathrm{DTP}} = -U\sum_e \mathbf{X}_e - t\sum_p \prod_{e\in p}\mathbf{Z}_e - h\sum_{\langle e,e'\rangle}\mathbf{Z}_e\mathbf{Z}_{e'}\,, \tag{347}$$

where the sum over $\langle e,e'\rangle$ is a sum over nearest-neighbor edges, by which we mean pairs of parallel edges which differ by one unit of translation in a direction perpendicular to the direction they stretch in. Note that, after switching perspectives to the dual lattice, this is precisely equivalent to the Ising layer model of Eq. (341); here we have re-obtained it by taking subsystem symmetry enriched topological phases as our starting point. The computations of the previous section elucidate the rough structure of the phase diagram. At large $t$, one is driven into a plaquette Ising phase. As the competition between $U$ and $h$ is varied, one undergoes a partial subsystem symmetry breaking phase transition.

### 6.2.2 Gauging the planar zero-form subsystem subgroup

We can now gauge the $\mathscr{H}$ subsystem symmetry of this deformed topological phase, following [115]. On each plaquette of the lattice, we place two qubits. If the plaquette lies in the xy plane, we label operators acting on these two qubits $\mathbf{X}_p^{\mathrm{x}},\mathbf{Z}_p^{\mathrm{x}}$ and $\mathbf{X}_p^{\mathrm{y}},\mathbf{Z}_p^{\mathrm{y}}$, and similarly for plaquettes lying in the yz and xy planes. The gauged deformed topological phase is then

$$H_{\mathrm{DTP}}\big/\mathscr{H} = -U\sum_e \mathbf{X}_e - t\sum_p\prod_{\mu\|p}\mathbf{Z}_p^\mu\prod_{e\in p}\mathbf{Z}_e - h\sum_{\langle e,e'\rangle}\mathbf{Z}_e\mathbf{Z}_{\langle e,e'\rangle}^{\mu(e)}\mathbf{Z}_{e'}\,,$$
$$G_e = \mathbf{X}_e\prod_{p\ni e}\mathbf{X}_p^{\mu(e)}\,,\qquad F_c^\mu = \prod_{\substack{p\in c\\p\|\mu}}\mathbf{Z}_p^\mu\,. \tag{348}$$

On the first line, in the operator $\mathbf{Z}_{\langle e,e'\rangle}^{\mu(e)}$, we are thinking of $\langle e,e'\rangle$ as the plaquette between the edges $e, e'$, and $\mu(e)$ is the direction that $e$ points in. On the second line, $G_e$ is a Gauss's law operator. Also, $F_c^\mu$ is a flux operator, $\mu$ is any direction (either x, y, or z), and $p\parallel\mu$ means that

we restrict the product to be over plaquettes bounding the cube $c$ which are parallel to the $\mu$ direction.

We can solve the Gauss's law constraint by first applying a local unitary operator of the form

$$V = H^{\otimes N} \prod_e \prod_{p \ni e} C_e \mathbf{X}_p^{\mu(e)},\tag{349}$$

which converts Gauss's law to the simple form $V G_e V^\dagger = \mathbf{Z}_e$. Thus, after freezing out the qubits on the edges and switching perspectives to the dual lattice, the gauged Hamiltonian becomes

$$V(H_{\mathrm{DTP}}\big/\mathcal{H})V^\dagger = -U \sum_{\tilde{p}} \prod_{\tilde{e} \in \tilde{p}} \mathbf{Z}_{\tilde{e}}^{\mu(\tilde{p})} - t \sum_{\tilde{e}} \prod_{\mu \| \tilde{e}} \mathbf{X}_{\tilde{e}}^{\mu} - h \sum_{\tilde{e}} \sum_{\mu \| \tilde{e}} \mathbf{X}_{\tilde{e}}^{\mu}$$
$$V F_c^{\mu} V^\dagger = \prod_{\substack{\tilde{e} \ni \tilde{v} \\ \tilde{e} \| \mu}} \mathbf{X}_{\tilde{e}}^{\mu} =: \widetilde{G}_{\tilde{v}}^{\mu},\tag{350}$$

where now, instead of thinking of $\mu$ as a direction, we are thinking of it as the plane orthogonal to that direction (xy, yz, or zx). This is precisely the coupled gauge theory layers from Eq. (309). Putting together the computations of the last few sections, we have found that performing Wegner duality on the leaves, or gauging planar subsystem symmetries (zero-form or one-form) passes one back and forth between the coupled Ising layers model of Eq. (341) and the coupled gauge theory layers model of Eq. (309).

*Excitations and phase transition at $h = 0$ between $t \ll U$ and $t \gg U$*

Excitations of the $\mathbf{X}_e$ terms in the trivial phase, $t \ll U$, are mapped under gauging planar subsystem symmetry to planon gauge charges. As $t$ is increased composite string excitations made up of these gauge charges, in the layers they pass through, fluctuate and condense throughout three dimensional space. This drives a p-string condensation phase transition from an isotropic stack of toric code layers to the X-cube phase [62], which is obtained by gauging the familiar confinement/deconfinement phase transition of $\mathbb{Z}_2$ lattice gauge theory in (3+1)D.

The gauge flux loop excitations in the (3+1)D toric code phase, $t \gg U$, are condensed within layers by gauging the subsystem symmetry, as they correspond to TSEs of the planar symmetry being gauged. However, the gauging procedure introduces two locally equivalent representatives of each segment of loop excitation to play the role of the TSE for the two intersecting planar symmetries. At a junction where loops from intersecting planes meet to run along a common line, forming a topologically trivial excitation there, a lineon excitation is left behind. This follows by noting that an open string of $\mathbf{X}_{\tilde{e}}^{\mu}$ excitations in a plane perpendicular to $\mu$ is equivalent to a pair of $\widetilde{G}_{\tilde{v}}^{\mu}$ excitations at the end points via the application of a string of $\mathbf{Z}_{\tilde{e}}^{\mu}$ operators. Furthemore, a pair of $\widetilde{G}_{\tilde{v}}^{\mathrm{x}}$ and $\widetilde{G}_{\tilde{v}}^{\mathrm{y}}$ excitations is equivalent to a z lineon excitation in the X-cube model (and similarly for permutations of x,y,z, see below). Hence, the junctions where loops of $\mathbf{X}_{\tilde{e}}^{\mathrm{x}}$ and $\mathbf{X}_{\tilde{e}}^{\mathrm{y}}$ excitations join are equivalent to z lineon excitations, and the same holds for permutations of x,y,z.

## 6.3 The string-membrane-net model

Finally, we turn to a parent model that combines the isotropic layer construction with the deformed toric code model considered in the preceding subsections. This model combines layers of (2+1)D toric code on the planes of a cubic lattice with the (3+1)D toric code. The Hilbert space is defined on a cubic lattice with two qubits per edge and one qubit per plaquette

$$\mathcal{H} = \bigotimes_e (\mathcal{H}_e^{\mu_1} \otimes \mathcal{H}_e^{\mu_2}) \bigotimes_p \mathcal{H}_p,\tag{351}$$

where $\mu_1, \mu_2$ are the two directions orthogonal to the edge $e$, and $\mathcal{H}_e^{\mu_i} \cong \mathcal{H}_p \cong \mathbb{C}^2$. One can equally view the plaquette qubits as living on the edges of the dual cubic lattice. A useful picture for the model is in terms of square lattice systems, each with one qubit per edge, stacked on the planes of a primal cubic lattice, which hosts one qubit per plaquette. Unlike the model in the previous section, this model respects the spatial symmetries of the cubic lattice, i.e. it is isotropic.

The perturbed string-membrane-net Hamiltonian is

$$
\begin{aligned}
H_{\text{PSMN}} = & -\Delta \sum_e \prod_{\mu \perp e} \mathbf{X}_e^\mu \prod_{p \ni e} \mathbf{X}_p - \lambda \sum_p \mathbf{Z}_p - \gamma \sum_e \sum_{\mu \perp e} \mathbf{X}_e^\mu \prod_{p \ni e}^{\perp \mu} \mathbf{X}_p \\
& - \Delta' \sum_\mu \sum_{p \perp \mu} \mathbf{Z}_p \prod_{e \in p} \mathbf{Z}_e^\mu - \lambda' \sum_e \prod_{\mu \perp e} \mathbf{X}_e^\mu - \gamma' \sum_e \sum_{\mu \perp e} \mathbf{X}_e^\mu - \kappa' \sum_e \prod_{\mu \perp e} \mathbf{Z}_e^\mu .
\end{aligned}
\tag{352}
$$

This model has a $\mathbb{Z}_2^{(1)}$ global one-form symmetry generated by operators supported on surfaces of the dual lattice,

$$
U_{\tilde{m}} = \prod_{e \in \tilde{m}} \mathbf{X}_e^{\mu_1} \mathbf{X}_e^{\mu_2} .
\tag{353}
$$

This symmetry enhances when $\kappa' = 0$ to a $\mathbb{Z}_2^{(1,1)}(\mathcal{F}_x^\perp, \mathcal{F}_y^\perp, \mathcal{F}_z^\perp)$ planar one-form subsystem symmetry generated by the operators

$$
U_{\tilde{\gamma}, i}^x = \prod_{(\alpha,\beta) \in \tilde{\gamma}} \mathbf{X}_{i,\alpha,\beta}^x , \quad U_{\tilde{\gamma}, j}^y = \prod_{(\alpha,\beta) \in \tilde{\gamma}} \mathbf{X}_{\alpha,j,\beta}^y , \quad U_{\tilde{\gamma}, k}^z = \prod_{(\alpha,\beta) \in \tilde{\gamma}} \mathbf{X}_{\alpha,\beta,k}^z ,
\tag{354}
$$

where e.g. in the first operator, $\tilde{\gamma}$ is a path on the dual lattice of the 2d sublattice in the $yz$ plane at transverse position $i$, and similarly for the second two operators.

The model also has a $\widetilde{\mathbb{Z}}_2^{(0,1)}(\mathcal{F}_x^\perp, \mathcal{F}_y^\perp, \mathcal{F}_z^\perp)$ planar zero-form subsystem symmetry generated by

$$
\widetilde{U}_i^x = \prod_{j,k} \mathbf{Z}_{i,j+\frac{1}{2},k+\frac{1}{2}} , \quad \widetilde{U}_j^y = \prod_{i,k} \mathbf{Z}_{i+\frac{1}{2},j,k+\frac{1}{2}} , \quad \widetilde{U}_k^z = \prod_{i,j} \mathbf{Z}_{i+\frac{1}{2},j+\frac{1}{2},k} .
\tag{355}
$$

This can be understood as a subgroup of a $\widetilde{\mathbb{Z}}_2^{(1)}$ global one-form symmetry

$$
\widetilde{U}_m = \prod_{p \in m} \mathbf{Z}_p ,
\tag{356}
$$

that is restored when $\gamma = 0$.

The following local unitary circuit

$$
\widetilde{V} = \prod_p \prod_{e \in p} C_e^{\hat{p}} \mathbf{X}_p ,
\tag{357}
$$

has a $\mathbb{Z}_2$ action on the phase diagram $\Delta \leftrightarrow \lambda'$, $\Delta' \leftrightarrow \lambda$, $\gamma \leftrightarrow \gamma'$.

The perturbed string-membrane-net Hamiltonian reduces to a stack of coupled (2+1)D toric codes, or a (3+1)D toric code in certain limits:

- If one takes $\lambda$ to be large, the plaquette qubits are pinned in the $|\uparrow\rangle$ state. The $\gamma$ term then contributes star terms

$$
\prod_{e \ni v}^{e \perp \mu} \mathbf{X}_e^\mu ,
\tag{358}
$$

at 4th order in perturbation theory. The product of three of such terms appears at 6th order in $\Delta$ (other products also appear at various orders in $\gamma$ and $\Delta$). The resulting model corresponds to coupled stacks of (2+1)D toric code on the planes of the cubic lattice (cf. Eq. (309)),

$$H_{\text{PSMN}} \xrightarrow{\lambda \text{ large}} H_{\text{3-stack}}^{\text{E}}, \tag{359}$$

where the E in the superscript indicates that the Gauss's law of $H_{\text{3-stack}}$ should be imposed energetically. The $\lambda'$ term induces p-string condensation [61,62].

- As $\gamma \to 0$ with $\lambda$ small, we find coupled stacks of (2+1)D $\mathbb{Z}_2$ gauge theory with the $^{\text{D}}\mathbb{Z}_2^{(1)}$ one-form symmetry gauged.[16] The $\Delta$ term energetically enforces the Gauss's law constraint, which becomes strict in the limit $\Delta \to \infty$. The cube flux term $\prod_{p \in c} \mathbf{Z}_p$ is generated at 6th order in perturbation theory in $\lambda$. In total,

$$H_{\text{PSMN}} \xrightarrow[\substack{\lambda \text{ small}}]{\substack{\gamma \to 0 \\ \Delta \to \infty}} H_{\text{3-stack}}\Big|_E {}^{\text{D}}\mathbb{Z}_2^{(1)}. \tag{360}$$

In this gauged model, the confinement/deconfinement phase transitions of the layers of $H_{\text{3-stack}}$ map to a phase transition from the (3+1)D toric code to the X-cube topological phase. This phase transition is driven by the condensation of fracton-composite planons in the X-cube phase.

- As $\gamma'$ is taken large, the edge qubits are all pinned into the $|+\rangle$ state, and the $\Delta', \kappa'$ terms are projected out, generating cube terms $\prod_{p \in c} \mathbf{Z}_p$ at 6th order in $\Delta'$ and 12th order in $\kappa'$. Switching to the dual lattice and applying a Hadamard gate $H^{\otimes N}$ that swaps all Pauli-$\mathbf{X}$ operators with Pauli-$\mathbf{Z}$ operators, we find the subsystem symmetry enriched toric code from Eq. (347),

$$H_{\text{PSMN}} \xrightarrow[\substack{\gamma' \text{ large}}]{\substack{H^{\otimes N}}} \widetilde{H}_{\text{DTP}}. \tag{361}$$

- As $\lambda', \gamma', \kappa' \to 0$ we find the subsystem symmetry enriched (3+1)D toric code with its $\mathbb{Z}_2^{(0,1)}$ planar subsystem symmetry gauged,

$$H_{\text{PSMN}} \xrightarrow[\substack{\lambda', \gamma', \kappa' \to 0}]{\substack{H^{\otimes N} \\ \Delta' \to \infty}} \widetilde{H}_{\text{DTP}}\Big|_E \mathbb{Z}_2^{(0,1)}. \tag{362}$$

The $\Delta'$ term energetically enforces the Gauss's law constraint, which becomes strict in the limit $\Delta' \to \infty$. Planar star flux terms are generated at 4th order in perturbation theory in $\gamma'$ (their products are generated at various orders in $\gamma'$ and $\lambda'$). In this model, the (3+1)D $\mathbb{Z}_2$ confinement/deconfinement transition of $\widetilde{H}_{\text{DTP}}$ is mapped via the gauging of $\mathbb{Z}_2^{(0,1)}$ to a transition from stacks of (2+1)D $\mathbb{Z}_2$ gauge theory layers to the X-cube topological phase driven by the condensation of p-string excitations.

The zero correlation length string-membrane-net model [52] emerges in the limit $\Delta, \Delta' \to \infty$

$$H_{\text{SMN}} = -\sum_e \prod_{\mu \perp e} \mathbf{X}_e^\mu \prod_{p \ni e} \mathbf{X}_p - \sum_\mu \sum_{p \perp \mu} \mathbf{Z}_p \prod_{e \in p} \mathbf{Z}_e^\mu - \sum_c \prod_{p \in c} \mathbf{Z}_p - \sum_v \sum_\mu \prod_{e \ni v}^{e \perp \mu} \mathbf{X}_e^\mu, \tag{363}$$

---

[16]The vertex terms of these (2+1)D gauge theory layers appear in perturbation theory in small $\gamma'$.

where we have included the leading order cube and vertex star terms, and rescaled the energies, both of which leave the zero temperature phase of matter of the commuting Hamiltonian invariant.

This zero correlation length model can equally be viewed as the (3+1)D toric code with its planar subsystem symmetries gauged, or as layers of (2+1)D toric code with a diagonal one-form symmetry gauged. Excitations of the edge star terms are equivalent to planon composites of X-cube lineons. Excitations of the plaquette terms are equivalent to planon composites of X-cube fractons. Excitations of the cube terms correspond to X-cube fractons. Excitations of the vertex star terms correspond to X-cube lineons.

The model is equivalent to the X-cube model, and hence the perturbed model can be understood as exploring neighboring phases and phase transitions of the X-cube model, including transitions to the (3+1)D toric code, and stacks of (2+1)D toric code. To demonstrate the equivalence we define the unitary

$$V = \prod_e C_e^A \mathbf{X}_e^B,$$ (364)

where the qubit labels $A$ and $B$ are chosen arbitrarily, and conjugate $H_{\text{SMN}}$ by the product $V\widetilde{V}$ to find

$$V\widetilde{V}H_{\text{SMN}}\widetilde{V}^\dagger V^\dagger = -\sum_e \mathbf{X}_e^A - \sum_p \mathbf{Z}_p - \sum_c \prod_{p\in c} \mathbf{Z}_p \prod_{e\in c} \mathbf{Z}_p^B - \sum_v \sum_\mu \prod_{e\ni v}^{e\perp\mu} \mathbf{X}_e^{A/B},$$ (365)

where $\mathbf{X}_e^{A/B}$ stands for $\mathbf{X}_e^A, \mathbf{X}_e^B$, or $\mathbf{X}_e^A \mathbf{X}_e^B$, depending on the arbitrary choice made for $V$ (we demonstrate below that this choice does not effect the final model). The cube terms are equivalent (up to a Hadamard) to the X-cube term $\prod_{e\in c} \mathbf{Z}_e^B$ after multiplying with six $\mathbf{Z}_p$ terms, and similarly the vertex terms are equivalent to $\prod_{e\ni v} \mathbf{X}_e^B$ after multiplying with $\mathbf{X}_e^B$ terms. This redefinition of the Hamiltonian decouples the cube and vertex terms from the edge and plaquette terms, allowing the trivial qubits to be decoupled while preserving the quantum phase of matter. This results in the X-cube Hamiltonian (up to a Hadamard)

$$V\widetilde{V}H_{\text{SMN}}\widetilde{V}^\dagger V^\dagger \sim -\sum_c \prod_{e\in c} \mathbf{Z}_e - \sum_v \sum_\mu \prod_{e\ni v}^{e\perp\mu} \mathbf{X}_e,$$ (366)

where we have dropped the $B$ superscript, as there is now one qubit per edge.

The string-membrane-net picture for the Hamiltonian can be seen by interpreting the cube term as fusing a $\mathbb{Z}_2$ closed membrane into the plaquettes on the boundary of the cube, and the plaquette terms as fusing in an open membrane ending on a $\mathbb{Z}_2$ closed loop into the plaquette and its boundary edges. Here, the vertex terms act as parity constraints that energetically enforce the edge strings to come in $\mathbb{Z}_2$ closed loops, and similarly the edge terms energetically enforce the membranes to be either $\mathbb{Z}_2$ closed, or end on an odd number of strings. The ground state is then given by an equal weight superposition of string-membrane-net configurations that obey these $\mathbb{Z}_2$ parity constraints [52].

# 7 Conclusion and Future Directions

In this work, we have described a number of mixed-dimensional constructions of fracton-adjacent lattice models, focusing on the unifying role played by higher-form subsystem symmetries. We explored an almost exhaustive list of examples based on layering low dimensional Ising models or lattice gauge theories, and then gauging natural higher-form subsystem symmetry subgroups. This allowed us to map stacks of well-known critical models, that occur at

the phase transition points of the layers, onto seemingly unconventional subdimensional critical points, that occur at fracton phase transitions. The mappings revealed generalized order parameters for these phase transitions, inherited from those of the layers, and allowed us to derive the subdimensional condensation mechanisms that drive these phase transitions. We also constructed various parent models, each of which subsumes several of our layered constructions and reveals a fracton analog of the string-net picture for the associated phases and phase transitions. For a summary of our results see Tables 1 & 8. While our focus was on $\mathbb{Z}_2$ degrees of freedom, for simplicity, much of the analysis extends directly to $\mathbb{Z}_N$ degrees of freedom, where interesting stable critical phases may arise, see Ref. [74].

Finally, we summarize a number of directions for future research.

1) It would be fruitful to develop a more systematic theory of higher-form subsystem symmetries. Many of the cherished facts about ordinary higher-form symmetries should have suitable analogs in the subsystem setting. Non-invertible generalizations would of course also be interesting and potentially useful.

2) It would be interesting to carry out our constructions in the continuum. This should in particular yield foliated field theory presentations [52, 58, 116] of several models that were given continuum descriptions in e.g. [54–57, 117–119]. A related problem is to demonstrate the equivalence between the foliated field theory approach and the field theories of op. cit. directly in the continuum.

3) The critical point in the phase diagram of the gauged Ising quilt (see §3.1.2), which separates a toric code phase from a plaquette Ising model phase, should admit a description in terms of an "orbifold" of a quilt of (1+1)D Ising CFT wires. Can techniques from conformal field theory be brought to bear in the analysis of this critical point? Could other RCFTs be used in place of the Ising model to produce interesting foliated conformal field theories?

4) To what extent can the bifurcating entanglement renormalization group flows of gapped fracton phases [51, 120–122] be generalized to their subdimensional critical points, including those studied here? We plan to report some progress in this direction in a forthcoming work [123].

5) Most of the constructions we have presented can be vastly generalized. For example, in (2+1)D, the Ising quilt of §3.1 can be generalized by substituting the transverse field Ising model with any (1+1)D lattice model with a global symmetry group $G$. Similarly, in §3.2, the toric code can be replaced with an arbitrary string-net model, or perhaps even in the continuum with any theory with a one-form symmetry (Yang-Mills, BF theory, Chern-Simons, etc.) And so on and so forth.

6) In this work, up to gauging, we have only considered systems which can be obtained by foliating $D$-dimensional space with $d$-dimensional theories for a single $d \leq D$. For example, in $D = 3$, we only considered coupled wires, or coupled layers, but not both simultaneously. One could imagine foliating space with $d$-dimensional theories for several values of $d \leq D$ simultaneously, and gauging more intricate subgroups of the resulting network model. In particular, sequential gauging of a more complicated group may result in non-Abelian phases [124–126].

7) Similar constructions to those we have used throughout the paper generalize to some fractal type-I fracton phases, in the terminology of Ref. [127], including a subset of Yoshida's fractal spin liquids [29, 31]. These models support lineons along a preferred

direction, and the local operators that create and propagate these lineons have the same commutation relations with Hamiltonian terms along lines as the terms in a stack of Ising wires. One can, in fact, realize such models by gauging a symmetry generated by operators that form fractals in two directions and extend linearly along the wire direction [128]. This produces certain lineon driven phase transitions out of the fracton phase that are related to a stack of decoupled critical Ising wire transitions via gauging, in a similar fashion to the higher-form subsystem symmetries we have considered in this work.

8) In most of our network constructions, we were able to obtain the symmetry-breaking phase of $^{D}H_{\mathbb{Z}_2}^{(q,k)}$ in two ways from a decoupled network model: either by gauging a subgroup $\mathcal{H}$ of the global symmetry of the decoupled network model, or by condensing TSEs of $\mathcal{H}$. This appears to be a version of the familiar relationship between gauging and condensation in the setting of (2+1)D topological phases, but it remains to be fully understood.

**Note added:** *During the course of the work reported here, we became aware of Ref. [74] which contains results that have some overlap with our own. It has also been brought to our attention that the term "fracton" has appeared previously in the physics literature in a different context [129]; we trust that the multiple meanings of this word will not cause confusion.*

# Acknowledgements

BR thanks Hao Geng, Shamit Kachru, Andreas Karch, and Richard Nally for a previous collaboration on fractons which has informed much of his thinking on the subject. BR also thanks Daniel Ranard, Shu-Heng Shao, and Max Zimet for many enlightening discussions. DW thanks David Aasen and Kevin Slagle for useful discussions.

**Funding information:** BR acknowledges support from NSF grant PHY 1720397, and DW acknowledges support from the Simons Foundation.

Table 8: A summary of the constructions covered in this appendix, using the notations and conventions from Table 1 (gen. stands for generalized).

| § | DNM | Subgroup $\mathcal{H}$ | Phase A | Phase B | A→B | B→A |
|---|---|---|---|---|---|---|
| §A.1 | ${}^1H_{\mathbb{Z}_2}^{(0)}(\mathscr{F}_x^{\parallel},\mathscr{F}_y^{\parallel},\mathscr{F}_z^{\parallel})$ Ising wires | ${}^{\mathrm{D}}\mathbb{Z}_2^{(0)}$ diagonal 0-form | deconfined $\mathbb{Z}_2$ gauge theory | ferro. gen. plaq. Ising | e charges (on lines) | 4x plaq. kinks (on lines) |
| §A.1 | ${}^3H_{\mathbb{Z}_2}^{(2)}$ 2-form gauge theory | $\mathbb{Z}_2^{(0,2)}(\mathscr{F}_x^{\parallel},\mathscr{F}_y^{\parallel},\mathscr{F}_z^{\parallel})$ linear subsystem | ferro. Ising wires | ferro. gen. plaq. Ising | k-membranes | – |
| §A.2 | ${}^2H_{\mathbb{Z}_2}^{(0)}(\mathscr{F}_x^{\perp},\mathscr{F}_y^{\perp},\mathscr{F}_z^{\perp})$ Ising planes | ${}^{\mathrm{D}}\mathbb{Z}_2^{(0)}$ diagonal 0-form | deconfined $\mathbb{Z}_2$ gauge theory | ferro. gen. Ising | e charges (in planes) | – |
| §A.2 | ${}^3H_{\mathbb{Z}_2}^{(2)}$ 2-form gauge theory | $\mathbb{Z}_2^{(1,1)}(\mathscr{F}_x^{\perp},\mathscr{F}_y^{\perp},\mathscr{F}_z^{\perp})$ planar 1-form | ferro. Ising planes | ferro. gen. Ising | domain wall membranes | – |

# A (3+1)D Toric Code Transitions

In this section we introduce a pair of models that both realize a (3+1)D toric code phase as well as unconventional phase transitions out of that phase; the first model we consider transitions to stacks of (1+1)D transverse field Ising models, and the second model transitions to stacks of (2+1)D transverse field Ising models.

## A.1 The linear point-string-net model

The first model we consider is a point-string-net model that combines coupled Ising wires with the (3+1)D toric code, and their diagonal zero-form and linear subsystem gauged variants, respectively. This is the (3+1)D analog of the (2+1)D point-string-net model introduced in §3.3 of the main text. The model is defined on a Hilbert space with three qubits per vertex and one qubit per edge

$$\mathcal{H} = \bigotimes_v (\mathcal{H}_v^x \otimes \mathcal{H}_v^y \otimes \mathcal{H}_v^z) \bigotimes_e \mathcal{H}_e \,, \tag{A.1}$$

where

$$\mathcal{H}_v^x \cong \mathcal{H}_v^y \cong \mathcal{H}_v^z \cong \mathcal{H}_e \cong \mathbb{C}^2 \,. \tag{A.2}$$

The perturbed linear point-string-net Hamiltonian is

$$
\begin{aligned}
H_{\mathrm{PLPSN}} = &- \Delta \sum_v \mathbf{X}_v^x \mathbf{X}_v^y \mathbf{X}_v^z \prod_{e \ni v} \mathbf{X}_e - \lambda \sum_e \mathbf{Z}_e \\
&- \gamma \sum_v \left( \mathbf{X}_v^x \prod_{e \ni v}^{e\|x} \mathbf{X}_e + \mathbf{X}_v^y \prod_{e \ni v}^{e\|y} \mathbf{X}_e + \mathbf{X}_v^z \prod_{e \ni v}^{e\|z} \mathbf{X}_e \right) \\
&- \Delta' \sum_{\mu=x,y,z} \sum_{e\|\mu} \mathbf{Z}_e \prod_{v \in e} \mathbf{Z}_v^\mu - \lambda' \sum_v \mathbf{X}_v^x \mathbf{X}_v^y \mathbf{X}_v^z - \gamma' \sum_v (\mathbf{X}_v^x + \mathbf{X}_v^y + \mathbf{X}_v^z) \\
&- \kappa' \sum_v (\mathbf{Z}_v^x \mathbf{Z}_v^y + \mathbf{Z}_v^y \mathbf{Z}_v^z + \mathbf{Z}_v^x \mathbf{Z}_v^z) \,.
\end{aligned} \tag{A.3}
$$

It has an ordinary $\mathbb{Z}_2^{(0)}$ global zero-form symmetry

$$U = \prod_v \mathbf{X}_v^x \mathbf{X}_v^y \mathbf{X}_v^z \,, \tag{A.4}$$

which is a subgroup of a larger ${}^{\text{x}}\mathbb{Z}_2^{(0)} \times {}^{\text{y}}\mathbb{Z}_2^{(0)} \times {}^{\text{z}}\mathbb{Z}_2^{(0)}$ global zero-form group which is respected when $\kappa' = 0$,

$$U^{\text{x}} = \prod_v \mathbf{X}_v^{\text{x}}, \quad U^{\text{y}} = \prod_v \mathbf{X}_v^{\text{y}}, \quad U^{\text{z}} = \prod_v \mathbf{X}_v^{\text{z}}. \tag{A.5}$$

The model also admits a $\widetilde{\mathbb{Z}}_2^{(0,2)}(\mathscr{F}_{\text{x}}^{\parallel}, \mathscr{F}_{\text{y}}^{\parallel}, \mathscr{F}_{\text{z}}^{\parallel})$ linear zero-form subsystem symmetry generated by $\mathbf{Z}_e$ operators along straight lines of the cubic lattice, i.e.

$$\widetilde{U}_{j,k}^{\text{x}} = \prod_i \mathbf{Z}_{i+\frac{1}{2},j,k}, \quad \widetilde{U}_{i,k}^{\text{y}} = \prod_j \mathbf{Z}_{i,j+\frac{1}{2},k}, \quad \widetilde{U}_{i,j}^{\text{z}} = \prod_k \mathbf{Z}_{i,j,k+\frac{1}{2}}. \tag{A.6}$$

When $\gamma = 0$, this linear subsystem symmetry group enhances to a larger $\widetilde{\mathbb{Z}}_2^{(2)}$ global two-form symmetry generated by $\mathbf{Z}_e$ operators along any $\mathbb{Z}_2$-closed paths in the cubic lattice, i.e.

$$\widetilde{U}_\gamma = \prod_{e \in \gamma} \mathbf{Z}_e. \tag{A.7}$$

There is additionally a $\mathbb{Z}_2$ local unitary circuit

$$\widetilde{V} = \prod_v \left( \prod_{e \ni v}^{e \parallel \text{x}} C_v^{\text{x}} \mathbf{X}_e \prod_{e \ni v}^{e \parallel \text{y}} C_v^{\text{y}} \mathbf{X}_e \prod_{e \ni v}^{e \parallel \text{z}} C_v^{\text{z}} \mathbf{X}_e \right), \tag{A.8}$$

that induces the following transformation on the phase diagram: $\Delta \leftrightarrow \lambda'$, $\Delta' \leftrightarrow \lambda$, $\gamma \leftrightarrow \gamma'$.

The linear point-string-net Hamiltonian, introduced above, reduces to a stack of Ising wires, the (3+1)D toric code, and their global zero-form and linear subsystem symmetry gauged variants in certain limits:

- As $\lambda \to \infty$ the edge qubits are pinned into the $|\uparrow\rangle$ state and the $\Delta$, $\gamma$ terms are projected out. The resulting model corresponds to coupled Ising wires stacked on the cubic lattice,

$$H_{\text{PLPSN}} \xrightarrow{\lambda \to \infty} {}^1H_{\mathbb{Z}_2}^{(0)}\left( \mathscr{F}_{\text{x}}^{\parallel}, \mathscr{F}_{\text{y}}^{\parallel}, \mathscr{F}_{\text{z}}^{\parallel} \right) + H_{\text{C}}, \tag{A.9}$$

  where the inter-wire coupling terms are given by

$$H_{\text{C}} = -\lambda' \sum_v \mathbf{X}_v^{\text{x}} \mathbf{X}_v^{\text{y}} \mathbf{X}_v^{\text{z}} - \kappa' \sum_v \left( \mathbf{Z}_v^{\text{x}} \mathbf{Z}_v^{\text{y}} + \mathbf{Z}_v^{\text{y}} \mathbf{Z}_v^{\text{z}} + \mathbf{Z}_v^{\text{x}} \mathbf{Z}_v^{\text{z}} \right). \tag{A.10}$$

- As $\lambda, \gamma \to 0$ the model reduces to the theory one obtains from gauging the diagonal global zero-form $\mathbb{Z}_2^{(0)}$ symmetry of coupled Ising wires, where Gauss's law is enforced energetically by the $\Delta$ term, becoming strict in the $\Delta \to \infty$ limit. A generating set of local flux terms, given by $\prod_{e \in p} \mathbf{Z}_e$, appears at 4th order in perturbation theory in $\lambda$. In total, we find that

$$H_{\text{PLPSN}} \xrightarrow[\lambda\text{ small}]{\substack{\gamma \to 0 \\ \Delta \to \infty}} \left( {}^1H_{\mathbb{Z}_2}^{(0)}\left( \mathscr{F}_{\text{x}}^{\parallel}, \mathscr{F}_{\text{y}}^{\parallel}, \mathscr{F}_{\text{z}}^{\parallel} \right) + H_{\text{C}} \right)\Big/_E \mathbb{Z}_2^{(0)}. \tag{A.11}$$

The linear subsystem symmetry breaking phase transition of the decoupled Ising wires is mapped, upon gauging $\mathbb{Z}_2^{(0)}$, to a transition from the toric code/deconfined gauge theory to a linear subsystem symmetry breaking phase governed by the Hamiltonian

$$H_{\text{LSSB}} = -\sum_v \mathbf{X}_v^{\text{x}} \mathbf{X}_v^{\text{y}} \mathbf{X}_v^{\text{z}} - \sum_p \prod_{v \in p}^{p \perp \text{x}} \mathbf{Z}_v^{\text{y}} \mathbf{Z}_v^{\text{z}} - \sum_p \prod_{v \in p}^{p \perp \text{y}} \mathbf{Z}_v^{\text{x}} \mathbf{Z}_v^{\text{z}} - \sum_p \prod_{v \in p}^{p \perp \text{z}} \mathbf{Z}_v^{\text{x}} \mathbf{Z}_v^{\text{y}}. \tag{A.12}$$

(This linear subsystem symmetry breaking phase is equivalent to the ferromagnetic generalized plaquette Ising model that appears below in Eq. (A.18).) This phase transition is driven by the condensation of pointlike topological charges in the toric code phase; similarly as in §3.1.2, there are three locally different manifestations of such particles which condense along the axial directions of the lattice. Viewing the same phase transition going the other way, we find that groups of four plaquette excitations in the ferromagnetic generalized plaquette Ising model are condensed along lines, again analogous to §3.1.2.

- As $\gamma'$ is taken large, the vertex qubits are pinned into the $|+\rangle$ state and the $\Delta'$, $\kappa'$ terms are projected out, aside from generating plaquette terms $\prod_{e \in p} \mathbf{Z}_e$ at 4th order in perturbation theory in $\Delta'$ and $\kappa'$. The resulting model corresponds to a toric code with linear subsystem symmetry breaking perturbations,

$$
\begin{aligned}
H_{\text{PLPSN}} \xrightarrow{\gamma' \text{ large}} &-\Delta \sum_v \prod_{e \in v} \mathbf{X}_e - \widetilde{\Delta} \sum_p \prod_{e \in p} \mathbf{Z}_e \\
&- \lambda \sum_e \mathbf{Z}_e - \gamma \sum_v \left( \prod_{e \ni v}^{e \| \mathrm{x}} \mathbf{X}_e + \prod_{e \ni v}^{e \| \mathrm{y}} \mathbf{X}_e + \prod_{e \ni v}^{e \| \mathrm{z}} \mathbf{X}_e \right).
\end{aligned}
\tag{A.13}
$$

- As $\lambda', \gamma', \kappa' \to 0$, the model is equivalent to a $\mathbb{Z}_2$ gauge theory (with the plaquette terms being generated in perturbation theory in $\lambda$) in the presence of linear subsystem symmetry breaking perturbations, with the linear subsystem symmetry along each wire being gauged,

$$
\begin{aligned}
H_{\text{PLPSN}} \xrightarrow[\lambda', \gamma', \kappa' \to 0]{\lambda \text{ small}} &-\Delta \sum_v \mathbf{X}_v^{\mathrm{x}} \mathbf{X}_v^{\mathrm{y}} \mathbf{X}_v^{\mathrm{z}} \prod_{e \in v} \mathbf{X}_e - \lambda \sum_e \mathbf{Z}_e \\
&- \gamma \sum_v \left( \mathbf{X}_v^{\mathrm{x}} \prod_{e \ni v}^{e \| \mathrm{x}} \mathbf{X}_e + \mathbf{X}_v^{\mathrm{y}} \prod_{e \ni v}^{e \| \mathrm{y}} \mathbf{X}_e + \mathbf{X}_v^{\mathrm{z}} \prod_{e \ni v}^{e \| \mathrm{z}} \mathbf{X}_e \right) - \Delta' \sum_{\mu = \mathrm{x,y,z}} \sum_{e \| \mu} \mathbf{Z}_e \prod_{v \in e} \mathbf{Z}_v^{\mu}.
\end{aligned}
\tag{A.14}
$$

The Gauss's law within the wires is enforced by the $\Delta'$ term, becoming strict as $\Delta' \to \infty$. The confineent/deconfinement transition of the $\mathbb{Z}_2$ gauge theory before its subsystem symmetr is gauged is mapped to a transition from ferromagnetic decoupled Ising wires to the subsystem symmetry broken phase governed by the Hamiltonian $H_{LSSB}$ introduced in Eq. (A.12) above. This transition is crossed as one varies $\lambda/\Delta$. Similar to §3.2.2, the phase transition is driven by the condensation of k-membranes, i.e. composite membrane excitations formed by kinks on the Ising wires they intersect.

The zero correlation length linear point-string-net model is obtained in the limit $\Delta, \Delta' \to \infty$ and $\kappa' \to 0$,

$$
H_{\text{LPSN}} = \sum_v \mathbf{X}_v^{\mathrm{x}} \mathbf{X}_v^{\mathrm{y}} \mathbf{X}_v^{\mathrm{z}} \prod_{e \ni v} \mathbf{X}_e - \sum_p \prod_{e \in p} \mathbf{Z}_e - \sum_{\mu = \mathrm{x,y,z}} \sum_{e \| \mu} \mathbf{Z}_e \prod_{v \in e} \mathbf{Z}_v^{\mu},
\tag{A.15}
$$

where we have included the leading order plaquette terms and rescaled the energies to 1, both of which preserve the zero temperature phase of matter of the Hamiltonian.

The linear point-string-net Hamiltonian can be viewed as a (3+1)D toric code with the linear subsystem subgroup of its two-form symmetry gauged. It can equally well be viewed as a stack of (1+1)D ferromagnetic Ising models with the diagonal zero-form symmetry gauged. Excitations of the vertex terms are found to be trivial, see below. Excitations of the plaquette term are corner domain walls of a generalized plaquette Ising model with two spins per vertex, shown below. Excitations of the edge terms are equivalent to a group of four plaquette excitations via the application of an $\mathbf{X}_e$ operator.

We now show that the linear point-string-net model is equivalent to a generalized plaquette Ising model with two spins per vertex. Hence the perturbed model above can be understood as exploring the phase diagram in the proximity of the ferromagnetic phase of this model, which includes decoupled Ising wires and the (3+1)D toric code. The equivalence follows from the application of the circuit $\widetilde{V}$ in Eq. (A.8), followed by the on-site unitary

$$V = \prod_v C_v^z \mathbf{X}_v^x C_v^z \mathbf{X}_v^y \,, \tag{A.16}$$

to the point-string-net Hamiltonian above, resulting in

$$
V \widetilde{V} H_{\mathrm{LPSN}} \widetilde{V}^\dagger V^\dagger = -\sum_v \mathbf{X}_v^z - \sum_p \overset{\perp x}{\prod_{e\in p}} \mathbf{Z}_e \prod_{v\in p} \mathbf{Z}_v^y - \sum_p \overset{\perp y}{\prod_{e\in p}} \mathbf{Z}_e \prod_{v\in p} \mathbf{Z}_v^x
$$
$$
- \sum_p \overset{\perp z}{\prod_{e\in p}} \mathbf{Z}_e \prod_{v\in p} \mathbf{Z}_v^x \mathbf{Z}_v^y - \sum_e \mathbf{Z}_e \,. \tag{A.17}
$$

The plaquette terms can be modified to act only on the vertices, up to multiplication with single body $\mathbf{Z}_e$ terms, which preserves the phase of matter. We then decouple the qubits on the $\mathcal{H}^z$ vertices and edges in $|+\rangle$ and $|\uparrow\rangle$ states, respectively, to find the ferromagnetic generalized plaquette Ising model

$$
V \widetilde{V} H_{\mathrm{LPSN}} \widetilde{V}^\dagger V^\dagger \sim -\sum_p \overset{\perp x}{\prod_{v\in p}} \mathbf{Z}_v^y - \sum_p \overset{\perp y}{\prod_{v\in p}} \mathbf{Z}_v^x - \sum_p \overset{\perp z}{\prod_{v\in p}} \mathbf{Z}_v^x \mathbf{Z}_v^y \,. \tag{A.18}
$$

The point-string-net picture for the above Hamiltonian follows by interpreting $|+\rangle$ states as empty, and $|-\rangle$ states on vertices as points, and on edges as string segments. The vertex term then energetically enforces a $\mathbb{Z}_2$ parity constraint that the total number of strings entering a vertex equals the number of points on that vertex, modulo 2. The edge terms fuse $\mathbb{Z}_2$ string segments attached to pairs of points into the lattice, while the plaquette term fuses a closed $\mathbb{Z}_2$ string into the lattice.

The ground state is then given by an equal weight superposition over allowed point-string-net configurations, built on top of a reference state which satisfies the $\mathbb{Z}_2$ parity constraints.

## A.2 The planar point-string-net model

The second model we consider is another point-string-net model that combines the (3+1)D toric code with coupled planes of the (2+1)D transverse field Ising model, and their planar one-form subsystem symmetry and diagonal zero-form symmetry gauged variants. The model is defined on a Hilbert space made up of three qubits per vertex and one per edge

$$\mathcal{H} = \bigotimes_v (\mathcal{H}_v^x \otimes \mathcal{H}_v^y \otimes \mathcal{H}_v^z) \bigotimes_e \mathcal{H}_e \,, \tag{A.19}$$

where

$$\mathcal{H}_v^x \cong \mathcal{H}_v^y \cong \mathcal{H}_v^z \cong \mathcal{H}_e \cong \mathbb{C}^2 \,. \tag{A.20}$$

The perturbed planar point-string-net Hamiltonian is

$$
\begin{aligned}
H_{\text{PPPSN}} = &-\Delta \sum_v \mathbf{X}_v^x \mathbf{X}_v^y \mathbf{X}_v^z \prod_{e \ni v} \mathbf{X}_e - \lambda \sum_e \mathbf{Z}_e \\
&- \varepsilon \sum_v \left( \mathbf{X}_v^x \prod_{e \ni v}^{e \perp x} \mathbf{X}_e + \mathbf{X}_v^y \prod_{e \ni v}^{e \perp y} \mathbf{X}_e + \mathbf{X}_v^z \prod_{e \ni v}^{e \perp z} \mathbf{X}_e \right) \\
&- \Delta' \sum_{\mu = x,y,z} \sum_{e \perp \mu} \mathbf{Z}_e \prod_{v \in e} \mathbf{Z}_v^\mu - \lambda' \sum_v \mathbf{X}_v^x \mathbf{X}_v^y \mathbf{X}_v^z - \varepsilon' \sum_v (\mathbf{X}_v^x + \mathbf{X}_v^y + \mathbf{X}_v^z) \\
&- \kappa' \sum_v \left( \mathbf{Z}_v^x \mathbf{Z}_v^y + \mathbf{Z}_v^y \mathbf{Z}_v^z + \mathbf{Z}_v^x \mathbf{Z}_v^z \right),
\end{aligned} \tag{A.21}
$$

which again has a $\mathbb{Z}_2^{(0)}$ global zero-form symmetry generated by

$$
U = \prod_v \mathbf{X}_v^x \mathbf{X}_v^y \mathbf{X}_v^z. \tag{A.22}
$$

This is the diagonal subgroup of a ${}^x\mathbb{Z}_2^{(0)} \times {}^y\mathbb{Z}_2^{(0)} \times {}^z\mathbb{Z}_2^{(0)}$ global zero-form symmetry,

$$
U^x = \prod_v \mathbf{X}_v^x, \quad U^y = \prod_v \mathbf{X}_v^y, \quad U^z = \prod_v \mathbf{X}_v^z, \tag{A.23}
$$

which is present when $\kappa' = 0$. The model also has a $\widetilde{\mathbb{Z}}_2^{(2)}$ global two-form symmetry,

$$
\widetilde{U}_\gamma = \prod_{e \in \gamma} \mathbf{Z}_e. \tag{A.24}
$$

There is additionally a $\mathbb{Z}_2$ local unitary circuit

$$
\widetilde{V} = \prod_{\mu = x,y,z} \prod_v \prod_{e \ni v}^{e \perp \mu} C_v^\mu \mathbf{X}_e, \tag{A.25}
$$

that acts on the phase diagram with $\lambda = 0$ by swapping $\varepsilon \leftrightarrow \varepsilon'$.

The planar point-string-net Hamiltonian above reduces to a stack of (2+1)D Ising models, the (3+1)D toric code, and their global zero-form and planar one-form subsystem symmetry gauged variants in different limits:

- As $\lambda \to \infty$, the edge qubits are pinned into the $|\uparrow\rangle$ state and the $\Delta, \varepsilon$ terms are projected out. The resulting model corresponds to coupled stacks of the (2+1)D Ising model,

$$
H_{\text{PPPSN}} \xrightarrow{\lambda \to \infty} {}^2H_{\mathbb{Z}_2}^{(0)} \left( \mathscr{F}_x^\perp, \mathscr{F}_y^\perp, \mathscr{F}_z^\perp \right) + H_C, \tag{A.26}
$$

where the inter-plane coupling terms are given by

$$
H_C = -\lambda' \sum_v \mathbf{X}_v^x \mathbf{X}_v^y \mathbf{X}_v^z - \kappa' \sum_v \left( \mathbf{Z}_v^x \mathbf{Z}_v^y + \mathbf{Z}_v^y \mathbf{Z}_v^z + \mathbf{Z}_v^x \mathbf{Z}_v^z \right). \tag{A.27}
$$

- As $\varepsilon \to 0$ with $\lambda$ small, the model approaches the theory that arises from gauging the diagonal zero-form $\mathbb{Z}_2^{(0)}$ symmetry of coupled (2+1)D Ising models, with Gauss's law enforced energetically, becoming strict as $\Delta \to \infty$. A generating set of local flux terms, $\prod_{e \in p} \mathbf{Z}_e$, appear at 4th order in $\lambda$. In total,

$$
H_{\text{PPPSN}} \xrightarrow[\substack{\lambda \text{ small}}]{\substack{\varepsilon \to 0 \\ \Delta \to \infty}} \left( {}^2H_{\mathbb{Z}_2}^{(0)}(\mathscr{F}_x^\perp, \mathscr{F}_y^\perp, \mathscr{F}_z^\perp) + H_C \right)\Big|_E \mathbb{Z}_2^{(0)}. \tag{A.28}
$$

Under the gauging of this $\mathbb{Z}_2^{(0)}$, the spontaneous breaking of the planar zero-form subsystem symmetry of the stacks of (2+1)D Ising models maps to a transition from a (3+1)D toric code phase to a planar subsystem symmetry-breaking phase. Similar to §3.1.2, and the previous subsection, this phase transition is driven by condensations within lattice planes of locally differing excitations that are all equivalent to the pointlike-topological charge.

- As $\varepsilon'$ is taken large, the vertex qubits are pinned into the $|+\rangle$ state and the $\Delta', \kappa'$ terms are projected out, generating plaquette terms $\prod_{e \in p} \mathbf{Z}_e$ at 4th order in $\Delta'$. The resulting model corresponds to the (3+1)D toric code with perturbations,

$$
\begin{aligned}
H_{\text{PPPSN}} \xrightarrow{\varepsilon' \text{ large}} & -\Delta \sum_v \prod_{e \ni v} \mathbf{X}_e - \widetilde{\Delta} \sum_p \prod_{e \in p} \mathbf{Z}_e \\
& -\lambda \sum_e \mathbf{Z}_e - \varepsilon \sum_v \left( \sum_{\mu=\text{x,y,z}} \prod_{e \ni v}^{e \perp \mu} \mathbf{X}_e \right).
\end{aligned}
\tag{A.29}
$$

- As $\lambda', \varepsilon', \varepsilon, \kappa' \to 0$, the model approaches a (3+1)D $\mathbb{Z}_2$ gauge theory[17] with the planar one-form $\mathbb{Z}_2^{(1,1)}$ subgroup of the global two-form symmetry gauged. The Gauss's law within each plane is enforced by the $\Delta'$ term, becoming strict as $\Delta' \to \infty$. In total,

$$
\begin{aligned}
H_{\text{PPPSN}} \xrightarrow{\lambda', \varepsilon, \varepsilon', \kappa' \to 0} & -\Delta \sum_v \mathbf{X}_v^{\text{x}} \mathbf{X}_v^{\text{y}} \mathbf{X}_v^{\text{z}} \prod_{e \ni v} \mathbf{X}_e - \lambda \sum_e \mathbf{Z}_e \\
& -\Delta' \sum_{\mu=\text{x,y,z}} \sum_{e \perp \mu} \mathbf{Z}_e \prod_{v \in e} \mathbf{Z}_v^\mu.
\end{aligned}
\tag{A.30}
$$

The confinement/deconfinement transtion of the $\mathbb{Z}_2$ gauge theory (as one varies $\lambda$) maps after gauging $\mathbb{Z}_2^{(1,1)}$ to a transition from stacks of decoupled (2+1)D Ising ferromagnets to a planar subsystem symmetry broken phase. Similar to §3.2.2, and the previous subsection, this phase transition is driven by the condensation of composite membrane excitations made up of Ising domain walls where the membrane intersects the ferromagnetic Ising layers.

The zero correlation length planar point-string-net model is obtained in the limit $\Delta, \Delta' \to \infty$ and $\kappa' \to 0$

$$
H_{\text{PPSN}} = -\sum_v \mathbf{X}_v^{\text{x}} \mathbf{X}_v^{\text{y}} \mathbf{X}_v^{\text{z}} \prod_{e \ni v} \mathbf{X}_e - \sum_p \prod_{e \in p} \mathbf{Z}_e - \sum_{\mu=\text{x,y,z}} \sum_{e \perp \mu} \mathbf{Z}_e \prod_{v \in e} \mathbf{Z}_v^\mu,
\tag{A.31}
$$

where we have included the leading order plaquette terms and rescaled the energies to 1, both of which preserve the zero temperature phase of matter.

The planar point-string-net Hamiltonian can be viewed as a (3+1)D toric code with planar one-form subsystem symmetries gauged. It can equally be viewed as stacks of (2+1)D Ising ferromagnets with the diagonal zero-form symmetry gauged. Excitations of the vertex terms are found to be trivial below. Excitations of the plaquette terms are found to be redundant with the edge excitations below. The edge excitations are found to be equivalent to those of a generalized Ising model with two spins per site.

The planar point-string-net model is equivalent to a generalized ferromagnetic Ising model with two spins per vertex. That is, the perturbed model above can be understood as exploring

---

[17]The plaquette terms are generated at 4th order in perturbation theory about $\lambda = 0$.

the phase diagram in the proximity of this model, including decoupled (2+1)D Ising models and the (3+1)D toric code. To simplify the model we apply the following unitary

$$V = \prod_v C_v^z \mathbf{X}_v^x C_v^z \mathbf{X}_v^y \prod_v \prod_{e \ni v}^{e \| y} C_v^x \mathbf{X}_e \prod_{e \ni v}^{e \| z} C_v^y \mathbf{X}_e \prod_{e \ni v}^{e \| x} C_v^z \mathbf{X}_e \,, \tag{A.32}$$

to the planar point-string-net Hamiltonian to find

$$VH_{\mathrm{PPSN}}V^\dagger = -\sum_v \mathbf{X}_v^z - \sum_p^{\perp x} \prod_{e \in p} \mathbf{Z}_e \prod_{v \in p} \mathbf{Z}_v^x \mathbf{Z}_v^y - \sum_p^{\perp y} \prod_{e \in p} \mathbf{Z}_e \prod_{v \in p} \mathbf{Z}_v^y - \sum_p^{\perp z} \prod_{e \in p} \mathbf{Z}_e \prod_{v \in p} \mathbf{Z}_v^x$$

$$- \sum_e \mathbf{Z}_e - \sum_e^{\| x} \mathbf{Z}_e \prod_{v \in e} \mathbf{Z}_v^y - \sum_e^{\| y} \mathbf{Z}_e \prod_{v \in e} \mathbf{Z}_v^x - \sum_e^{\| z} \mathbf{Z}_e \prod_{v \in e} \mathbf{Z}_v^x \mathbf{Z}_v^y \,. \tag{A.33}$$

The vertex z spins are now decoupled in the $|+\rangle$ state, and can be removed. The plaquette terms are redundant, as they are products of the single body $\mathbf{Z}_e$ terms, and vertex-edge $\mathbf{Z}$ terms, hence they can be removed without changing the zero temperature phase of matter, only shifting the energetics of excitations. Similarly, the vertex-edge terms can be modified to act solely on the vertices, up to multiplication with $\mathbf{Z}_e$ terms. The edge qubits are then also decoupled in the $|\uparrow\rangle$ state and can be removed, resulting in a ferromagnetic generalized Ising model with two spins per site

$$VH_{\mathrm{PPSN}}V^\dagger \sim -\sum_e^{\| x} \prod_{v \in e} \mathbf{Z}_v^y - \sum_e^{\| y} \prod_{v \in e} \mathbf{Z}_v^x - \sum_e^{\| z} \prod_{v \in e} \mathbf{Z}_v^x \mathbf{Z}_v^y \,. \tag{A.34}$$

The point-string-net picture for the Hamiltonian in Eq. (A.32) follows by viewing $|+\rangle$ states as empty, and $|-\rangle$ states on verties as points, and on edges as segments of string. The vertex term then enforces that the parity of the points on a vertex matches the parity of the incoming string segments on adjacent edges. The edge terms fuse a segment of string attached to a pair of points into an edge and the adjacent vertices, while the plaquette terms fuse a closed loop of string into the edges. The ground state can then be interpreted as an equal weight superposition over the allowed point-string-net configurations under the $\mathbb{Z}_2$ parity constraints described above (we emphasize that these differ from those of the linear point-string-net Hamiltonian).

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
