# Peer review of "Higher-Form Subsystem Symmetry Breaking: Subdimensional Criticality and Fracton Phase Transitions"

_SciPost Physics, doi:SciPost Phys. 15, 017 (2023)_

## Round 1 · Referee Report · Anonymous (Referee 1) · 2023-3-19

Strengths

clear and concise, easy to follow, very comprehensive view of Higher-Form Subsystem Symmetry Breaking.

Weaknesses

need more comments and insights on UV-IR mixing in fracton and higher form symmetry breaking. however, this is an open & long last question so we do not expect it can be solved in a single paper.

Report

Higher-Form Subsystem Symmetry Breaking (HFSSB) is a new concept in condensed matter physics that has emerged in recent years. It can lead to the emergence of exotic phenomena, such as Fracton Phase Transitions.

The author extensively researched various examples by layering low-dimensional Ising models or lattice gauge theories and identifying natural higher-form subsystem symmetry subgroups. They mapped stacks of critical models occurring at phase transition points in the layers to subdimensional critical points that arise during fracton phase transitions, despite being seemingly unconventional. Through these mappings, the author discovered generalized order parameters inherited from the layers for these phase transitions and derived the subdimensional condensation mechanisms driving them.

The paper is clear written and provides various insights on higher form SB in well-known models, and study fracton topological phase transitions by relating them in an exact manner, via gauging, to spontaneous higher-form subsystem symmetry. The idea is illuminating and can be used for exploration of other fracton phase transitions in the future.

However,one concern I have is the method of stack+gauging. For a higher form SB phase, I agree that is a universal method one can apply. For critical points, I am not sure stacking and gauging of canonical models with higher-form symmetry can provide much insight on the universality of the phase transition.

In addition, I am not sure one can follow the procedure to carry out authors' constructions in the continuum. For fracton phase transitions, the UV-cut off can change the universality of the IR behavior. I am not sure if this aspect is reflected in their methodology.
  • validity: high
  • significance: good
  • originality: high
  • clarity: high
  • formatting: excellent
  • grammar: perfect

Author:  Brandon Rayhaun  on 2023-03-25  [id 3510]

(in reply to Report 1 on 2023-03-19)

We thank the referee for their thoughtful feedback.

"However,one concern I have is the method of stack+gauging. For a higher form SB phase, I agree that is a universal method one can apply. For critical points, I am not sure stacking and gauging of canonical models with higher-form symmetry can provide much insight on the universality of the phase transition."

This is a fair point. Extra steps are needed in order to work out whether the phase transitions we find are fine-tuned or capture universal behaviour. This is already studied in the work of Lake and Hermele (which we reference e.g. at the end of Section 3.1.2), who argue for the stability of certain subdimensional phase transitions which are similar to ours using large N techniques. We would be happy to clarify this in a revised version of the manuscript.

"In addition, I am not sure one can follow the procedure to carry out authors' constructions in the continuum. For fracton phase transitions, the UV-cut off can change the universality of the IR behavior. I am not sure if this aspect is reflected in their methodology."

The generalization of our constructions to the continuum is indeed non-trivial. While the idea of breaking up the construction into pieces which individually have a clear continuum limit (such as conventional criticality of spin chains and the gauging of 1-form symmetries) is supposed to serve as a guide for generalizing to the continuum, carrying this out carefully is beyond the scope of our work. We have already flagged this as a future direction in Section 7 of our paper.

"need more comments and insights on UV-IR mixing in fracton and higher form symmetry breaking. however, this is an open & long last question so we do not expect it can be solved in a single paper."

Our constructions suggest that, at least for the simple lattice models we look at, the UV-IR mixing can be reduced to understanding how lower-dimensional conventional models (that do not individually have UV-IR mixing) play with the scale introduced by stacking. Then on top of this we apply standard gauging maps with which we can easily see what UV aspects of the decoupled models the IR properties of the gauge fields couple to. This presents one potentially promising approach to understanding the UV-IR mixing of fracton models in the continuum better, but we by no means claim to have worked out all of the details. We are happy however to add comments along these lines to the paper.

-- Brandon and Dominic

---

## Round 1 · Referee Report · Anonymous (Referee 2) · 2023-5-8

Strengths

This work is thorough, pretty timely, and the ideas and results are presented in a clear and transparent manner. The conceptualization of lattice network models to lay out a more robust analogue of string-net condensation is interesting.

Weaknesses

  1. Could add a more detailed discussion on how this framework can be used to construct a phase diagram for gapless fracton models (just the approach should suffice).
  2. Consider adding a discussion on scaling laws associated with sub-critical points in the examples discussed.

Report

Based on the above strengths and weaknesses / requests for additional discussions to be included, I think this work is of appropriate quality for this journal. I recommend publication after the authors address the above questions.

Requested changes

  1. Add a more detailed discussion on how this framework can be used to construct a phase diagram for gapless fracton models (just the approach should suffice).
  2. Add a discussion on scaling laws associated with sub-critical points in the examples discussed.

  • validity: high
  • significance: good
  • originality: ok
  • clarity: good
  • formatting: good
  • grammar: good

Author:  Brandon Rayhaun  on 2023-05-12  [id 3666]

(in reply to Report 2 on 2023-05-08)

We thank the reviewer for their feedback.

"1. Could add a more detailed discussion on how this framework can be used to construct a phase diagram for gapless fracton models (just the approach should suffice)."

The gauging constructions we outlined in our paper relate (parts of) fracton phase diagrams to phase diagrams of well-known/exactly solvable models in lower dimensions. See e.g. Figure 5 for the simplest example, which relates the order/disorder transition of Ising wires to a transition between a toric code and a plaquette Ising model phase.

"2. Consider adding a discussion on scaling laws associated with sub-critical points in the examples discussed."

Similarly as above, we found that scaling laws in certain (fine-tuned) subdimensional critical points can be obtained from scaling laws in lower dimensional theories via gauging, simply because correlators (of operators which are even with respect to the symmetry being gauged) can be mapped to correlators in the gauged theory. As an example, in Section 3.1.2 we do explain how order/disorder parameters are mapped across the gauging described in the previous paragraph.

-- Brandon and Dom

---

## Round 1 · Referee Report · Anonymous (Referee 3) · 2023-5-9

Strengths

Very thorough, clear, succeeds in piecing together many related in constructing models for fracton phases. Particularly nice is the definition and use of subsystem higher-form symmetry, which unifies many ideas. Coupled wire/layer constructions are physically transparent, while gauging allows the phase transitions to be related to lower dimensional transitions which are understood.

Weaknesses

Use of the term "subsystem symmetry enriched" is slightly misleading, as it leads one to think about e.g. symmetry fractionalization and related phenomena, which does not appear in the paper. It seems that rather than enriching a topological order with a subsystem symmetry, they start with a topological order and break the 1-form symmetries down to subsystem symmetries. It would perhaps be clearer if a different name (like "subsystem symmetric toric code") was chosen, or if justification for the current name was given.

Report

Excellent paper for this journal, recommended for publication after clarifying the point above.

Requested changes

Clarification or modification of name "subsystem symmetry enriched ..." .

  • validity: high
  • significance: high
  • originality: high
  • clarity: high
  • formatting: perfect
  • grammar: perfect

Author:  Brandon Rayhaun  on 2023-05-12  [id 3667]

(in reply to Report 3 on 2023-05-09)

We appreciate the comments of the referee.

"Use of the term "subsystem symmetry enriched" is slightly misleading, as it leads one to think about e.g. symmetry fractionalization and related phenomena, which does not appear in the paper. It seems that rather than enriching a topological order with a subsystem symmetry, they start with a topological order and break the 1-form symmetries down to subsystem symmetries. It would perhaps be clearer if a different name (like "subsystem symmetric toric code") was chosen, or if justification for the current name was given."

We are sympathetic to the critique. We went with the word "enriched" in situations where we imposed a particular symmetry, and only considered phase diagrams obtained by deforming the Hamiltonian by operators which respect that symmetry. We believe this is in the same spirit as other usages of the word, even though we do not discuss symmetry fractionalization as you point out.

-- Brandon and Dom

---

## Editorial Decision

published